# Evidence for deliberate burial of the dead by *Homo naledi*

Lee R Berger[1,2,3]*, Tebogo Vincent Makhubela[2,4]*, Keneiloe Molopyane[5], Ashley Krüger[6], Patrick Randolph-Quinney[2,7]*, Marina Elliott[2,8], Becca Peixotto[2,9], Agustín Fuentes[10], Paul Tafforeau[11], Vincent Beyrand[2,11], Kathleen Dollman[11], Zubair Jinnah[2,12], Angharad Brewer Gillham[13], Kenneth Broad[14], Juliet Brophy[2,15], Gideon Chinamatira[16], Paul HGM Dirks[2,17], Elen Feuerriegel[2,18], Alia Gurtov[19], Nompumelelo Hlophe[20], Lindsay Hunter[21], Rick Hunter[2], Kudakwashe Jakata[16], Corey Jaskolski[22], Hannah Morris[23], Ellie Pryor[24], Maropeng Mpete[2], Eric M Roberts[2,17], Jacqueline S Smilg[2], Mathabela Tsikoane[2], Steven Tucker[2], Dirk Van Rooyen[2], Kerryn Warren[25], Colin D Wren[26], Marc Kissel[27], Penny Spikins[28], John Hawks[2,19]*

[1]The National Geographic Society, Washington DC, United States; [2]Centre for the Exploration of the Deep Human Journey, School of Anatomical Sciences, University of the Witwatersrand, Johannesburg, South Africa; [3]The Carnegie Institution for Science, Washington, DC, United States; [4]Department of Geology, University of Johannesburg, Johannesburg, South Africa; [5]Geography, Archaeology and Environmental Studies, University of the Witwatersrand, Johannesburg, South Africa; [6]Department of Palaeobiology, Swedish Museum of Natural History, Stockholm, Sweden; [7]Department of Archaeology and Ancient History, Uppsala University, Campus Gotland, Uppsala, Sweden; [8]Department of Archaeology, Simon Fraser University, University Drive, Burnaby, Canada; [9]The American University, Washington, D.C, United States; [10]Department of Anthropology, Princeton University, Princeton, United States; [11]European Synchrotron Radiation Facility, Grenoble, France; [12]School of GeoSciences, University of the Witwatersrand, Johannesburg, South Africa; [13]Frontiers Media Limited, London, United Kingdom; [14]Rosenstiel School of Marine, Atmospheric, and Earth Science, Miami, United States; [15]Department of Geography and Anthropology, Louisiana State University, Baton Rouge, United States; [16]Evolutionary Studies Institute, University of the Witwatersrand, Johannesburg, South Africa; [17]Department of Geology and Geological Engineering, Colorado School of Mines, Golden, United States; [18]Primate Evolutionary Biomechanics Laboratory, Department of Anthropology, University of Washington, Seattle, United States; [19]Department of Anthropology, University of Wisconsin-Madison, Madison, United States; [20]Department of Anthropology, Texas A&M University, College Station, United States; [21]Center for Academic Research & Training in Anthropogeny, University of California, San Diego, San Diego, United States; [22]Synthetaic, Delafield, United States; [23]ICON & Warnell School of Forestry, University of Georgia, Athens, United States; [24]School of Earth and Environmental Sciences, Cardiff University, Cardiff, United Kingdom; [25]Human Evolution Research Institute, University of Cape Town, Cape Town, South Africa; [26]Department of Anthropology, University of Colorado Colorado Springs, Colorado Springs, United States; [27]Department of Anthropology, Appalachian State University, Boone, United States; [28]Department of Archaeology, University of York; The King's Manor, York, United Kingdom

*For correspondence:
lrberger@ngs.org (LRB);
tvmakhubela@uj.ac.za (TVM);
patrick.randolph-quinney@
arkeologi.uu.se (PR-Q);
jhawks@wisc.edu (JH)

## eLife Assessment

The authors study the context of the skeletal remains of three individuals and associated sediment samples to conclude that the hominin species *Homo naledi* intentionally buried their dead. Demonstration of the earliest known instance of intentional funerary practice – with a relatively small-brained hominin engaging in a highly complex behavior that has otherwise been observed from *Homo sapiens* and *Homo neanderthalensis* – would represent a **landmark** finding. The authors have revised their manuscript extensively in light of the reviews of their initial submission, with improved illustration, context, discussion, and theoretical frameworks, leading to an improved case supporting their conclusion that *Homo naledi* intentionally buried their dead. One of the reviewers concludes that the findings **convincingly** demonstrate intentional burial practices, while another considers evidence for such an unambiguous conclusion to be **incomplete** given a lack of definitive knowledge around how the hominins got into the chamber. We look forward to seeing the continued development and assessment of this hypothesis. It is worth noting that the detailed reviews (both rounds) and comprehensive author response are commendable and consequential parts of the scientific record of this study. The editors note that the authors' response repeatedly invokes precedent from previous publications to help justify the conclusions in this paper. While doing so is helpful, the editors also note that scientific norms and knowledge are constantly evolving, and that any study has to rest on its own scientific merit.

**Abstract** In this study, we describe new results of excavations in the Dinaledi Subsystem of the Rising Star cave system, South Africa. In two areas within the Hill Antechamber and the Dinaledi Chamber, this work uncovered concentrations of abundant *Homo naledi* fossils including articulated, matrix-supported skeletal regions consistent with rapid covering by sediment prior to the decomposition of soft tissue. We additionally re-examine the spatial positioning of skeletal material and associated sediments within the Puzzle Box area, from which abundant *H. naledi* remains representing a minimum of six individuals were recovered in 2013 and 2014. Multiple lines of evidence exclude the hypothesis that skeletal remains from these three areas come from bodies that decomposed on the floor of the chamber or within a shallow depression prior to burial by sediments. The spatial positioning of skeletal material, the topography of the subsystem, and observations on sediments within and surrounding features exclude the hypothesis that rapid burial by sediment was a result of gravity-driven slumping or spontaneous movement of sediments. We present a minimal hypothesis of hominin cultural burial and test the evidence from all three areas, finding that this hypothesis is most compatible with the pattern of evidence. These results suggest that mortuary behavior, including cultural burial, was part of the repertoire of *Homo naledi*.

## Introduction

Burials have been broadly recognized in the archaeological record as pits within the earth that were intentionally dug to inter the remains of the dead and then covered with sediments or other materials (*Parker Pearson, 1999*; *Pettitt and Anderson, 2020*). Burial is just one of many kinds of mortuary practices, which in their presentation have diverse forms, functions, and meanings across human societies (*Metcalf and Huntington, 1991*; *Robben, 1991*). It is additionally currently accepted that late Pleistocene humans and some Neandertals had varied mortuary practices including burials and that some Neandertals also emplaced remains within natural soil depressions or within rock niches, which they then, sometimes, covered with sediments. Burials of these Pleistocene-aged humans and Neandertal bodies show varied disposition, sometimes buried alone, and sometimes with multiple bodies represented within the same archaeological level or site (*Martinón-Torres et al., 2021*; *Tillier, 2022*; *Jaubert et al., 2022*; *Jaubert et al., 2022*). However, a number of researchers have challenged the idea that burial was a frequent aspect of Neanderthal mortuary behavior (*Gargett et al., 1989*; *Dibble et al., 2015*), and some examples once widely accepted as burials later have been questioned or reinterpreted (*Sandgathe et al., 2011*; *Dibble et al., 2015*).

Researchers thus have diverse opinions about how to test whether ancient hominin remains were intentionally buried by the living of their own kind (*Pettitt, 2013*; *Pomeroy et al., 2020b*). One reason

for continuing uncertainty is that more recent human cultural mortuary practices are physically diverse, expressing many different forms of burial as part of a wider mortuary and funerary repertoire, which includes among many others, processes such as cremation, excarnation, mummification, and curation. Recognition of burial by any formal criteria must consider this diversity. Such mortuary diversity may express through the size, orientation, and depth of burial features, the nature of deliberate infills, grave coverings, and the position or disposition of the body, in addition to factors such as secondary disturbance and reuse of burial features. Although often considered typical, in fact, many (if not most) intentional burials lack artefactual remains or grave goods (*Brownlee, 2021a* and *Brownlee, 2021b*) depending on region, time period, and social and ritual context. In fact, unfurnished burial may be the norm for intentional funerary rites throughout most of human funerary history, excluding practices such as cremation, excarnation, and communal interment in mortuary structures. As such, in burial contexts lacking grave goods or other material cultural remains, archaeologists are forced to rely on spatial and contextual criteria to examine possible intentionality.

Any social interpretation must take into account taphonomic postmortem processes which affect the body and its environment (cadaver decomposition island [CDI]) over time (*Carter et al., 2007*). The buried body and associated sediments undergo diverse patterns of postburial diagenesis, chemical, biological, and physical alteration, and/or disturbance. After burial, the decomposition of organic remains and geochemical and physical alteration of sediments can result in highly varied configurations of skeletal and sedimentary evidence which are only now beginning to be understood as complex systems. Accumulating evidence from archaeothanatology, funerary archaeology, and forensic science indicates that there is enormous variability in the activity, morphology, and postmortem systems experienced by the body in cases of interment and disposal (e.g. *Aspöck, 2008*; *Boulestin and Duday, 2005*; *Boulestin and Duday, 2006*; *Connolly et al., 2005*; *Channing and Randolph-Quinney, 2006*; *Cherryson, 2008*; *Donnelly et al., 1999*; *Finlay, 2000*; *Hunter, 2014*; *Parker Pearson, 1999*). Disruption by later disturbance, intentional or otherwise, can damage bones, disturb articulations, confuse anatomical association, and move bones from original positions. It can also be difficult to compare data from possible burials at different sites due to the varied recording and excavation processes of different archaeological projects done at different times and places, and under different conditions. This includes cases where new excavations have been opened at sites that were excavated in the past, where data from new and original work may be hard to correlate.

For these reasons, there has been much debate about what observations indicate cultural burial of hominin remains as opposed to noncultural processes (*Gargett et al., 1989*; *Gargett, 1999*; *Pettitt, 2013*; *Sandgathe et al., 2011*; *Dibble et al., 2015*; *Rendu et al., 2014*; *Pomeroy et al., 2020b*). Many criteria have been proposed by these and other authors, including ritualized body positioning, a disproportionate preservation of hominin remains relative to other faunal remains, presence of grave goods, retention of anatomical articulations, stratigraphic evidence of a cut separating the immediate context of skeletal remains from surrounding sediments, or geochemical contrasts between the sediments in the burial and surrounding sediments. Additionally, some noncultural (e.g. 'natural') processes may mimic evidence for cultural burial, in particular cases where hominin remains may settle into depressions in the ground surface, later to be covered by natural sedimentation. As a result, debate about burial evidence has revolved around the need to distinguish ancient surfaces with natural depressions or undulations from surfaces that were disrupted by digging (*Sandgathe et al., 2011*; *Rendu et al., 2014*; *Dibble et al., 2015*), to distinguish holes dug by hominins from holes dug by other animals such as cave bears or hyenas (*Camarós et al., 2017*), and to distinguish burial by hominins from other processes. such as gradual subaerial sedimentation, waterborne sedimentation, or gravity-driven sediment slumping (*Sandgathe et al., 2011*; *Rendu et al., 2014*; *Dibble et al., 2015*; *Goldberg et al., 2017*; *Pomeroy et al., 2020b*). Testing the hypothesis of cultural burial ideally requires clear criteria to distinguish cultural from noncultural processes, but such criteria may seldom be articulated outside the sphere of forensic science and the medico-legal application of mortuary archaeology, and some archaeothanatological applications (*Duday, 2009*).

In this study, we report two new contexts of skeletal remains of *Homo naledi* from the Rising Star cave system. We assess the skeletal remains and other material from these new excavations as well as their sedimentary and geoarchaeological context. These new findings are part of a broader context of *H. naledi* remains within the system. We also reanalyze material recovered from previous excavations conducted by us as part of this broader reassessment. To understand the overall patterns

observed, we additionally review the geological and sedimentary context of all *H. naledi* fossils, as well as previous tests of hypotheses for the entrance and deposition of these *H. naledi* individuals into the cave system. The new data enable us to test several hypotheses concerning the disposition and deposition of remains that were not testable based on observations published in previous work.

The new evidence comes from analysis and reanalysis of the stratigraphy, textures, geochemical composition, and granulometry of the sediments around and within the various features, as well as analysis of the spatial configuration and individuation of the skeletal remains. To evaluate this evidence in comparison to other sites with well-accepted evidence of burial, we present a minimal definition of hominin cultural burial against which we test these data and discuss how other burial evidence from Late Pleistocene modern humans and Neandertals relates to this definition.

## The Rising Star cave system and *Homo naledi* localities

The Rising Star cave system (26°1'13″ S; 27°42'43″ E, *Figure 1*) is located within the Cradle of Humankind UNESCO World Heritage Site, Gauteng Province, South Africa, situated geographically within the Blaubank River valley (*Dirks et al., 2015*). The system comprises more than 4 linear km of passages, tunnels, and chambers, beneath an area of approximately 400 m by 200 m of land surface (*Elliott et al., 2021*). The cave system formed within a 15- to 20-m-thick horizon of stromatolitic dolomite, which dips at 17° toward the west. Surface entrances into the system occur where this layer outcrops beneath a 1- to 1.2-m-thick chert layer along an east-northeast-facing hillside (*Dirks et al., 2015*).

Hominin skeletal remains have been discovered at several localities within the cave system. Some of these fossil-bearing localities are located within the Dinaledi Subsystem, a group of interconnected chambers and fissure passages located approximately 30 m beneath the present ground surface and 80 m west-southwest of the closest current cave entrance (*Figure 2*). There is no straight-line access into this subsystem from the ground surface above the cave, and the shortest path is more than 100 m through the adjacent Dragon's Back and Postbox Chambers. The Dinaledi Subsystem is itself extensive with an overall length of more than 40 m from its entrance in the Hill Antechamber to its most distal extent (*Elliott et al., 2021*). The Hill Antechamber leads via narrow (<50 cm wide) passages approximately 7 m long into the Dinaledi Chamber. From this chamber, a fissure network extends a further 25 m or more in both southwest and north directions, with many passages generally less than 25 cm in width and in many places even more narrow. Within the Dinaledi Subsystem, remains of *H. naledi* have been reported from the Dinaledi Chamber, the Hill Antechamber, and several fissure passages designated U.W. 108 to U.W. 111 (*Berger et al., 2015*; *Elliott et al., 2021*; *Brophy et al., 2021*). The hominin-bearing localities within the Dinaledi subsystem are separated by distances of up to 40 m from each other.

The initial 2013–2014 investigations of the Dinaledi Subsystem were limited to survey and collection of fossil material from the floor surface, and one excavation unit within the Dinaledi Chamber of approximately 80 cm by 80 cm (*Berger et al., 2015*). Analysis of the subsurface deposits in the subsystem came from that limited excavation area, in addition to observations of surface outcrops of sedimentary units in the subsystem (*Dirks et al., 2015*; *Dirks et al., 2017*; *Elliott et al., 2021*). Within these earlier excavations, abundant *H. naledi* remains were found within the first 25 cm below the chamber floor surface, forming several spatial clusters (*Kruger et al., 2016*; *Kruger, 2017*). Information about the sediments below this level is limited to a sondage of ~25 cm width that was excavated in the middle of the excavation unit to a depth of 80 cm below surface (*Dirks et al., 2017*). Uranium series-electron spin resonance (US-ESR) direct dating of two *H. naledi* teeth from the Dinaledi Chamber yielded a range of 139–335 ka for these individuals (*Dirks et al., 2017*). A comprehensive uranium-thorium (U/Th) dating of flowstone units in the subsystem was carried out, resulting in a chronology for flowstone formation; the most informative sample was a flowstone unit directly encasing bone. The U/Th age of 244±3 ka for this flowstone (RS18, RS68) is inferred to be a minimum age for the osteological specimen embedded within it, which has been suggested as the minimum age constraint for *H. naledi* skeletal material in the subsystem (*Dirks et al., 2017*; *Robbins et al., 2021*). In these previously described excavations and investigations within the Dinaledi Chamber, no non-hominin macrofaunal material was excavated in direct association with *H. naledi* skeletal material. Microfaunal elements in the Dinaledi Chamber are rare and appear to

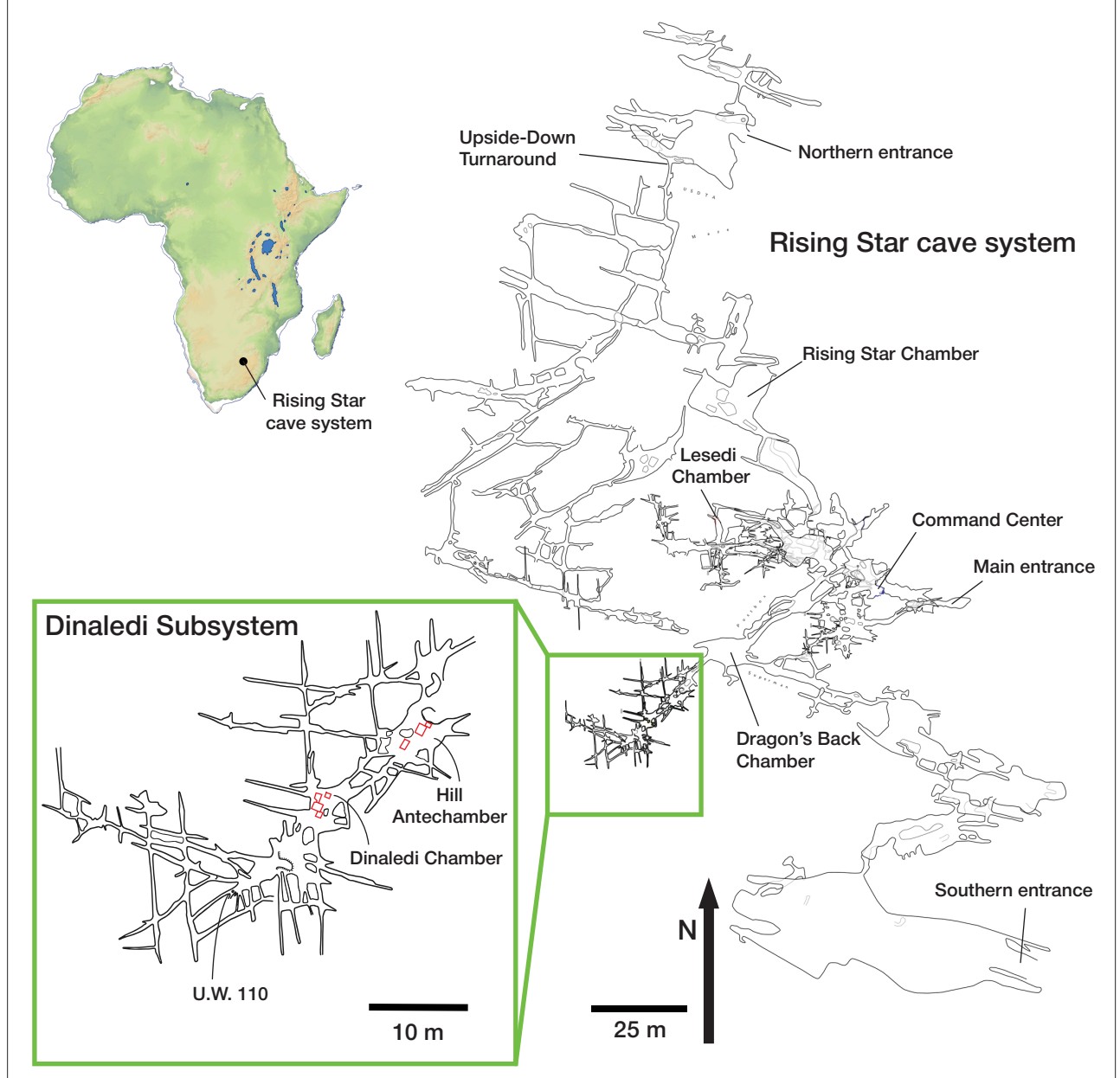

**Figure 1.** Rising Star cave system map. The relative locations of current entrances and underground fossil localities are indicated, including the Dinaledi Subsystem and Lesedi Chamber. The inset is an expanded map showing the Dinaledi Subsystem with the Hill Antechamber, Dinaledi Chamber, and associated fissure passages including the U.W. 110 locality.

be derived from Unit 1 sediments (*Dirks et al., 2015*; *Dirks et al., 2017*), although this hypothesis remains untested. Undated macrofaunal remains include rare fragments of an owl (*Kruger and Badenhorst, 2018*) and a few cercopithecoid elements (*Elliott et al., 2021*) from surface contexts (the latter recovered from a remote passage). A single *Papio* tooth was recovered from near the base of the sondage unit underlying the hominin material at a depth of 55–60 cm beneath the present surface during 2013–2014 excavations (*Dirks et al., 2017*). This tooth produced a US-ESR age of 723±181 ka or 635±148 ka for two uptake models, and this predates the maximum age estimated for *H. naledi* remains that are stratigraphically above this tooth. There is at present no evidence of hominin or other animal activity in the Dinaledi subsystem between c335 ka (the maximum direct

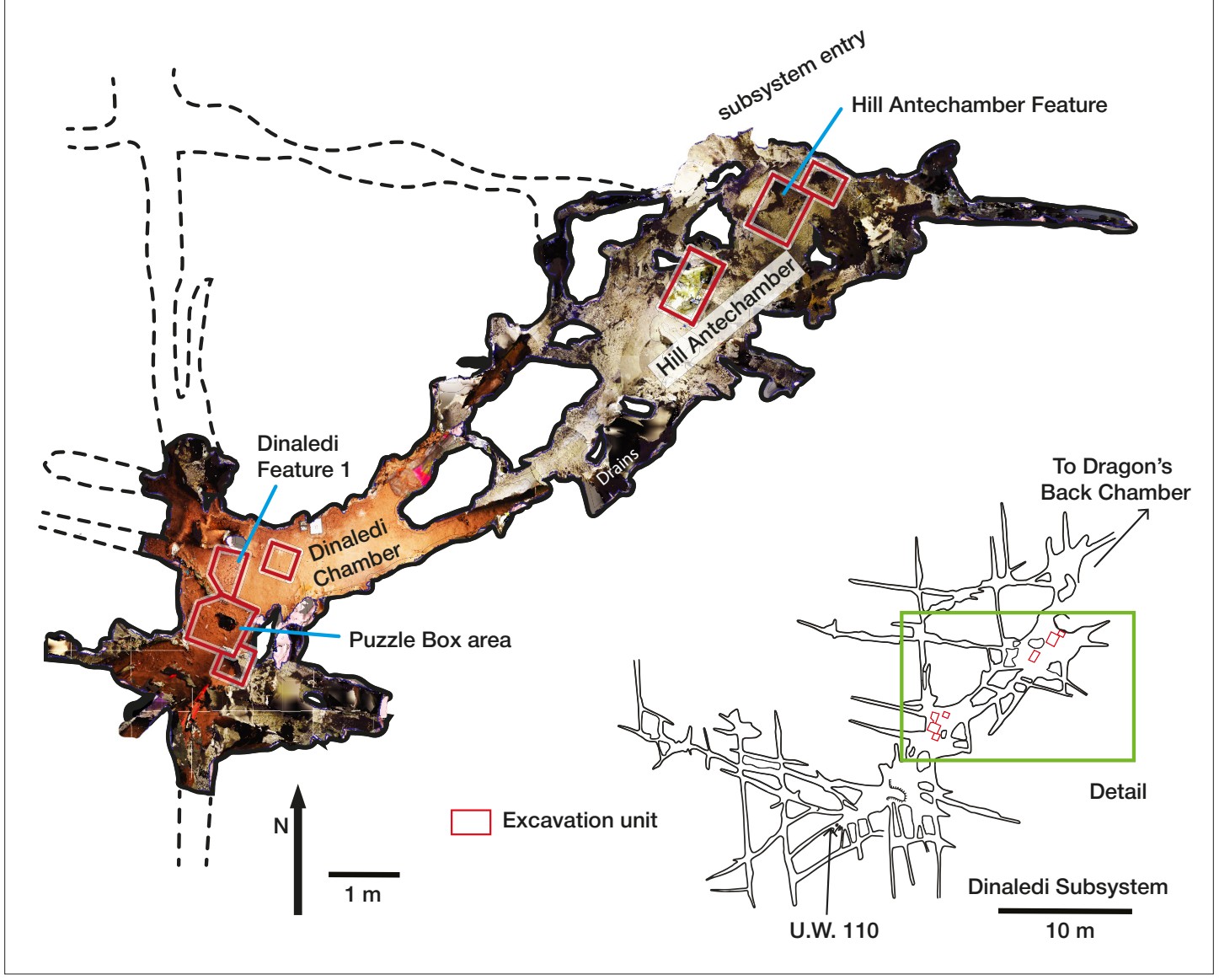

**Figure 2.** Dinaledi Subsystem map and detail showing floor space of the Dinaledi Chamber and Hill Antechamber. The floor visualization is derived from LIDAR and photogrammetric survey. Differences in color in this visualization reflect the lighting used in different parts of the subsystem during data capture and do not accurately depict contrasts in sediment color. Excavation units are indicated.

dates on *H. naledi* remains) and 487 ka (the lower confidence age for the *Papio* tooth recovered from the sondage).

A second space within the cave system with fossils was identified in 2013. This chamber, named the Lesedi Chamber, is located 30 m below surface and 86 m from the closest cave entrance by the shortest route (*Hawks et al., 2017*). The hominin fossils in this chamber were identified as *Homo naledi* after study in comparison with the Dinaledi material. As in the case of the Dinaledi Chamber, hominin remains were first identified on the surface, followed by limited excavation to recover these remains and ascertain whether more were present. Remains of at least three *H. naledi* individuals have been recovered from this chamber, all recovered from three areas of investigation, including the partial skeleton of an adult individual designated as LES1 (*Hawks et al., 2017*). No estimate of geological age for these *H. naledi* remains is yet known. In addition to the *H. naledi* material, a small quantity of microfaunal and small carnivore remains have been recovered from this chamber (*Hawks et al., 2017*), although the temporal association of these faunal remains with the hominin material is

unknown at present. At the present time, the most distant described localities with *H. naledi* remains are the Lesedi Chamber (U.W. 102) and the U.W. 110 locality within the Dinaledi Subsystem, separated by a minimum distance through the cave system of 170 m (*Figure 1*).

## Review of previous work

The original excavation unit in the Dinaledi Chamber, which was the focus of research in 2013 and 2014 was selected for excavation due to the visibility of the cross section of a partial cranium at this location in the sediment surface of the Dinaledi Chamber floor. Excavations to uncover the remainder of this cranium led to the discovery of additional skeletal elements in a dense subsurface deposit. The total extent of excavation was confined to an area of 80 cm by 80 cm and depth of 25 cm, at which depth sediments ceased to yield bone material. A sondage of limited (25 cm) diameter to a total depth of 80 cm provided evidence of the composition of sediments beneath this unit. Despite the limited extent of excavation in the Dinaledi Chamber, the resulting data have provided a great deal of information about the formation processes of the sediment and fossil deposit (*Dirks et al., 2015*; *Hawks et al., 2017*; *Robbins et al., 2021*; *Wiersma et al., 2020*). This information shaped our previous hypotheses about the deposition of *H. naledi* fossils and our development of strategies for further investigation of the site.

The geological context of the Dinaledi Subsystem, including the definitions of sediment units, was initially presented by *Dirks et al., 2015* with subsequent revision (*Dirks et al., 2017*). These papers include geochemical analyses of sediments, stratigraphic, micromorphological, and grain size characterization of sediment units, and identification and dating of flowstone units within the subsystem. A review of these findings is provided in Appendix 1.1 ('Description of the sedimentary deposits of the Dinaledi subsystem'). To summarize, *Dirks et al., 2015* recognized three sediment units in the subsystem that had been produced by distinct formation processes. Unit 1 comprises laminated orange-red mud (LORM) interpreted as being deposited by the precipitation of orange sediment from water into layers that are horizontal or parallel to surfaces where the deposits formed. Unit 1 sediments have formed recently where water percolates onto chert outcrops. Unit 2 and 3 sediments are brown unlithified breccia that contains clasts of the LORM of Unit 1. *Wiersma et al., 2020* provided additional sedimentological and micromorphological data from the Dinaledi Subsystem while providing a model for the autobrecciation of Unit 1 deposits, which led to the formation of Units 2 and 3. *Homo naledi* fossil material occurs within Unit 3 deposits in the subsystem; no *H. naledi* material has been found within Unit 1 deposits (*Dirks et al., 2015*; *Dirks et al., 2017*). For analysis of the sediments in Unit 3, deposits were subdivided into two sub-units: sub-unit 3a that does not include *H. naledi* material and sub-unit 3b that contains *H. naledi* material.

## Formation of sediments associated with *H. naledi*

The process resulting in the autobrecciation of Unit 1 deposits and formation of Units 2 and 3 did not happen in a 'wet' water-saturated environment. Rather, this process happened in 'dry' cave conditions without inundation or saturation of the sediments by water but with evidence of periods of higher and lower humidity (*Wiersma et al., 2020*). The evidence for this conclusion is based on the macromorphology of Unit 2 and Unit 3 sediments, as well as the micromorphology of Unit 3 sediments, including observations made on sediments sampled in direct contact with *H. naledi* skeletal remains (*Dirks et al., 2015*; *Dirks et al., 2017*). The Unit 3 sediments include unlithified clasts of LORM with sharply angled contours around their circumference and with laminations out of horizontal at random orientations. If the breakdown of Unit 1 LORM and incorporation of these clasts into Unit 3 had occurred in saturated conditions or with water flow, these clasts would either have broken down or their edges would be indistinct and would not retain sharply angled contours. These LORM clasts are embedded in matrix which is composed in part of clay mineral fragments which themselves are angular and locally derived without rounding by transport (*Dirks et al., 2015*). The autobrecciation process that gave rise to Unit 3 sediments can be observed within the subsystem as described in *Wiersma et al., 2020*. Some Unit 1 sediments include desiccation fractures inferred to result from periods of lower humidity in the system. Infiltration of these cracks by mineralization and microbial activity proceeded in cycles, causing expansion of the Unit 1 sediments. Weathering of the external mud clast surfaces resulted in the formation of an iron-rich and manganese-rich matrix, leading to the formation of Units 2 and 3 sediments (*Wiersma et al., 2020*). Additional patterns of evidence further exclude the hypothesis

of substantial water flow in the formation of the *H. naledi*-bearing sediments, including the relative absence of horizontal laminations or size-sorting within the Unit 3 sediments, and the random orientations visible micromorphologically, lack of abrasive surface erosion on sediment particles, and lack of evidence of water wear or size-sorting of *H. naledi* material (*Dirks et al., 2015*).

## Articulation of skeletal material

A second set of observations from earlier work involves the preservation of articulated *H. naledi* skeletal elements and spatially associated elements derived from single individuals. Articulated and spatially associated elements have previously been reported from two contexts, including the excavated material from the Dinaledi Chamber (*Dirks et al., 2015*) and within the Lesedi Chamber (*Hawks et al., 2017*). The spatial situation of skeletal remains within the excavated unit in the Dinaledi Chamber was complex. The full assemblage, including surface-collected fossil material and excavated material, represents a minimum of 15 individuals (*Bolter et al., 2018*). At least six of these individuals are represented within the excavation unit. The complex pattern of closely packed skeletal elements in direct contact, at varied angles, led excavators to denote this excavation area as the 'Puzzle Box'. Some of the elements were highly fragmented, but many were found in articulation or close spatial association with anatomically contiguous elements. Examples include articulated hand and wrist (*Dirks et al., 2015*; *Kivell et al., 2015*; *Kruger et al., 2016*), foot and ankle elements (*Dirks et al., 2015*; *Harcourt-Smith et al., 2015*; *Bolter et al., 2020*), mandible and associated temporal (*Dirks et al., 2015*), and vertebrae and ribs (*Berger et al., 2015*; *Williams et al., 2017*). By examining the developmental stage of elements, *Bolter et al., 2020* demonstrated the spatial association of lower and upper limb elements from one subadult individual amid other remains excavated from the Puzzle Box area. The Lesedi Chamber evidence includes the spatial localization of one partial skeleton, including some elements in articulation and others (including the fragmented cranium and mandible) in spatial association (*Hawks et al., 2017*). These observations show that in both the Dinaledi and Lesedi Chamber contexts, body parts were encased with sediment prior to the decomposition of soft tissue.

## Orientations of skeletal elements

Some of the long bone elements recovered from the Puzzle Box excavation unit were situated in subvertical orientations that would have been impossible to maintain unsupported on a horizontal surface or within a shallow depression. As described previously (*Dirks et al., 2015*; *Kruger et al., 2016*; *Kruger, 2017*), these elements must have been supported by sediment in these positions at the time of burial and could not have been covered slowly or gradually over time. Some of these elements with subvertical orientations were in contact with other horizontal or near-horizontal long bone elements and elements oriented at a range of pitch angles (*Dirks et al., 2015*). Some are in close contact with articulated elements including some that they underlie. The resulting arrangements of skeletal elements are inconsistent with deposition upon a surface followed by slow sedimentation.

## Fragmentation and spatial patterning of elements

The *H. naledi* skeletal assemblage is characterized by a high degree of postdepositional fracturing. The representation of elements exhibits underrepresentation of portions of elements that have thin cortical bone and high trabecular composition. No fragments have been found with fractures that are consistent with perimortem or 'green' fractures; long bone fractures are columnar or otherwise consistent with fracturing after loss of mechanical resilience (*Dirks et al., 2015*; *Hawks et al., 2017*). The Dinaledi Chamber assemblage includes many fragments that refit directly with each other. Some of these were recovered in spatial alignment, that is postmortem fracturing occurred within sediment without subsequent displacement or movement of fragments. Many other refitting fragments were recovered out of spatial alignment with each other, meaning that sediments and bone fragments were reworked after deposition and subsequent loss of bone mechanical strength. In some instances, bone fragments bear surface markings or stains, including 'tide marks' of mineral staining, that align with each other after refitting. In these cases, these marks must have been produced within a primary context, followed by reworking of fragments into a secondary context (*Dirks et al., 2015*). The presence of such marks does not mean that flowing water was present, because flowing water is excluded by many other kinds of evidence (*Dirks et al., 2015*; *Dirks et al., 2017*; *Wiersma et al.,*

*2020*). These marks may either relate to the soil-air interface during the postdepositional history of the remains, or to the presence of microbial action during or after decomposition of the remains, or other processes (*Randolph-Quinney et al., 2016*). Many of these reworked fragments were recovered in contact with articulated material, the same kind of juxtaposition seen in the cases of subvertically oriented elements noted above.

## Hypotheses discussed in previous work

The pattern of evidence, including the key observations noted above, constrains interpretations of the formation of the *H. naledi*-bearing fossil deposits. The most salient observations noted in previous work include: (1) the entry of whole bodies into the Dinaledi Subsystem; (2) the presence of some articulated remains in primary context far from the entry point to the subsystem; (3) the lack of any evidence of carnivore activity or marks on the bones; (4) the evidence that Unit 3 sediments in contact with *H. naledi* formed in dry cave conditions; (5) the absence of externally-derived sediment input; (6) the imposition of remains on and near each other, being buried at different times due to bone positions; (7) and the evidence for reworking of the fossil deposit in selected situations. These data clearly and readily rejected some hypotheses for the transport and deposition of the remains. Rejected hypotheses include carnivore accumulation, accidental falls from outside the cave, a catastrophic mass death accumulation, and accumulation by water flow or mud flow of remains into depressions or drains. The evidence addressing these hypotheses is discussed further in Appendix ('Hypotheses tested in previous work').

*Dirks et al., 2015* discussed three hypotheses that they felt could not be rejected with the data from the Dinaledi Subsystem at that time. The first was a death trap scenario in which individuals entered the subsystem at varied times and died there. The second was a deliberate deposition scenario in which bodies were dropped into the subsystem via the entry Chute, forming a deposit at the base of that entry, followed by slumping of fleshed or semi-fleshed remains deeper into the subsystem. The third was a deliberate deposition scenario in which *H. naledi* individuals carried bodies directly into the subsystem and placed them there. While this was not explicitly discussed by *Dirks et al., 2015*, the three scenarios shared one important feature: there was no role for *H. naledi* as an agent operating within the Dinaledi Subsystem. None of these scenarios envisioned any role for *H. naledi* in the formation or reworking of the sediment deposits, in burying bodies, or in manipulating remains after deposition.

Nonetheless, each of these scenarios had obvious difficulty explaining some aspects of the data. None provided a completely satisfactory explanation for the evidence of reworking of skeletal remains within the Puzzle Box area, in which some remains are in primary context while others show evidence of fragmentation and reworking into a secondary context, including vertically or subvertically emplaced elements. The scenario of dumping bodies into the subsystem could not account for the presence of articulated remains in primary context more than 15 meters from the entry point. These challenges were noted by *Dirks et al., 2015*.

## Reevaluation of the Dinaledi Subsystem

In addition to these difficulties in explaining the distribution of remains, other aspects of the Dinaledi Subsystem that were not obvious in earlier work *Dirks et al., 2015* have emerged through further exploration, mapping, and characterization of sediment deposits (*Dirks et al., 2017*; *Elliott et al., 2021*). Briefly, these aspects are as follows:

1. The Hill Antechamber and Dinaledi Chamber are separate spaces that are connected through two narrow (<50 cm) passages of approximately 7 meters in length (*Figure 2*; *Figure 3*; *Elliott et al., 2021*). The floor topography differs substantially in these spaces: The Hill Antechamber slopes steeply downward toward the Dinaledi Chamber, while the Dinaledi Chamber slopes slightly in reverse, measured at 4.4 degrees approximately west to east and between 9 and 11 degrees from north to south. Large amounts of sediment carrying bodies or bones could not have moved through these narrow passages without choking them and obstructing further movement into the Dinaledi Chamber.
2. *Dirks et al., 2015*; *Dirks et al., 2017* discussed floor drains within the subsystem. These are passages too small to accommodate exploration, but that may accommodate some slumping or gravity-driven shifting of sediment. One area with floor drains is within a small side passage

from the south end of the Hill Antechamber. Here, sediment can move by gravity into lower cave passages that do not accommodate human explorers. Sediments, bones, bodies, or water moving down the sloping Hill Antechamber floor would encounter these drains prior to the passages leading to the Dinaledi Chamber. A second area of floor drains was characterized within a fissure passage leading north from the Dinaledi Chamber. Upon further exploration, it is clear that the locations with these drains are not adjacent to the fossil deposits described in the current study.

3. The small sondage unit that formed an essential part of the geochronological work (*Dirks et al., 2017*) by characterizing the sedimentary deposit below the fossil material revealed that this area had no downward movement of sediment or other subsurface features that might have caused localized reworking. Reanalysis of the skeletal element assemblage in this study further tests the hypothesis that subsurface reworking was a result of slumping or subsidence of sediments in the excavated areas.

4. An extensive fissure network of very narrow (<25 cm) fissures and passages extends past the Dinaledi Chamber to the west and southwest. Within this network were recovered additional hominin remains from several localities including the U.W. 110 locality, which is located 40 m from the entry point to the subsystem (*Figure 1*; *Elliott et al., 2021*; *Brophy et al., 2021*).

Each of these observations poses additional challenges to the explanations considered by *Dirks et al., 2015*. The reworking of deposits by gravitational movement or slumping did occur in some specific areas of the subsystem. For example, gravitational slumping and reworking of sediments and skeletal remains within the Hill Antechamber, with its steep floor slope, occurred at some times during the formation of the sediments. But the displacement of bodies through narrow constrained passage-ways for a distance of more than 10 meters into the Dinaledi Chamber did not happen by gravitational slumping or water flow. Not only the slope but also the presence of floor drains near the base of the Hill Antechamber precludes passive transport into the Dinaledi Chamber. The Puzzle Box area, which was the source of the majority of *H. naledi* fossil remains described in our previous work, is not located in a position of high surface complexity or slope. The sondage below this area did not reveal subsurface movement or downward slumping, instead showing that beneath the *H. naledi* material is a deep long-term deposit of Unit 3 sediment that was deposited before 487 ka (the minimum confidence limit of the dated *Papio* tooth). This evidence leaves no passive explanation for the vertically emplaced elements, fractured and displaced elements, in direct contact with articulated material in primary context. As described by *Dirks et al., 2015*, some kind of highly selective reworking is necessary to explain these observations.

## The place of new work

Our previous work left several unresolved questions concerning the formation and reworking of deposits containing *H. naledi* remains in the Dinaledi Subsystem and the broader Rising Star cave system. First, the distribution and abundance of fossil material in the sedimentary deposits throughout the subsystem was unknown. The observation and collection of material from surfaces throughout the subsystem suggested that *H. naledi* remains might be present widely in subsurface contexts, and *Dirks et al., 2015* illustrated the entire floor of the subsystem as possibly composed of fossil-bearing sediments. However, only further excavation could determine the subsurface disposition of skeletal remains. Second, our earlier work provided only very minimal information about the Hill Antechamber. The difficult constraints associated with entering the subsystem suggested that this antechamber, just at the subsystem entrance, might provide more evidence of hominin material or possible cultural evidence if these were present anywhere.

We therefore designed new research with several aims. Excavations would provide more information about the sedimentary context of *H. naledi* remains and the formation processes of the sediments. These excavations would test whether *H. naledi* remains were present in subsurface deposits beyond the Puzzle Box area in similar disposition and abundance, or whether the Puzzle Box situation was uncharacteristic of the broader Dinaledi Subsystem. Excavations would also test whether any cultural remains occur near the subsystem entrance point. Whatever the outcome of these excavations, the new results would provide evidence relevant to understanding the deposition of the *H. naledi* remains.

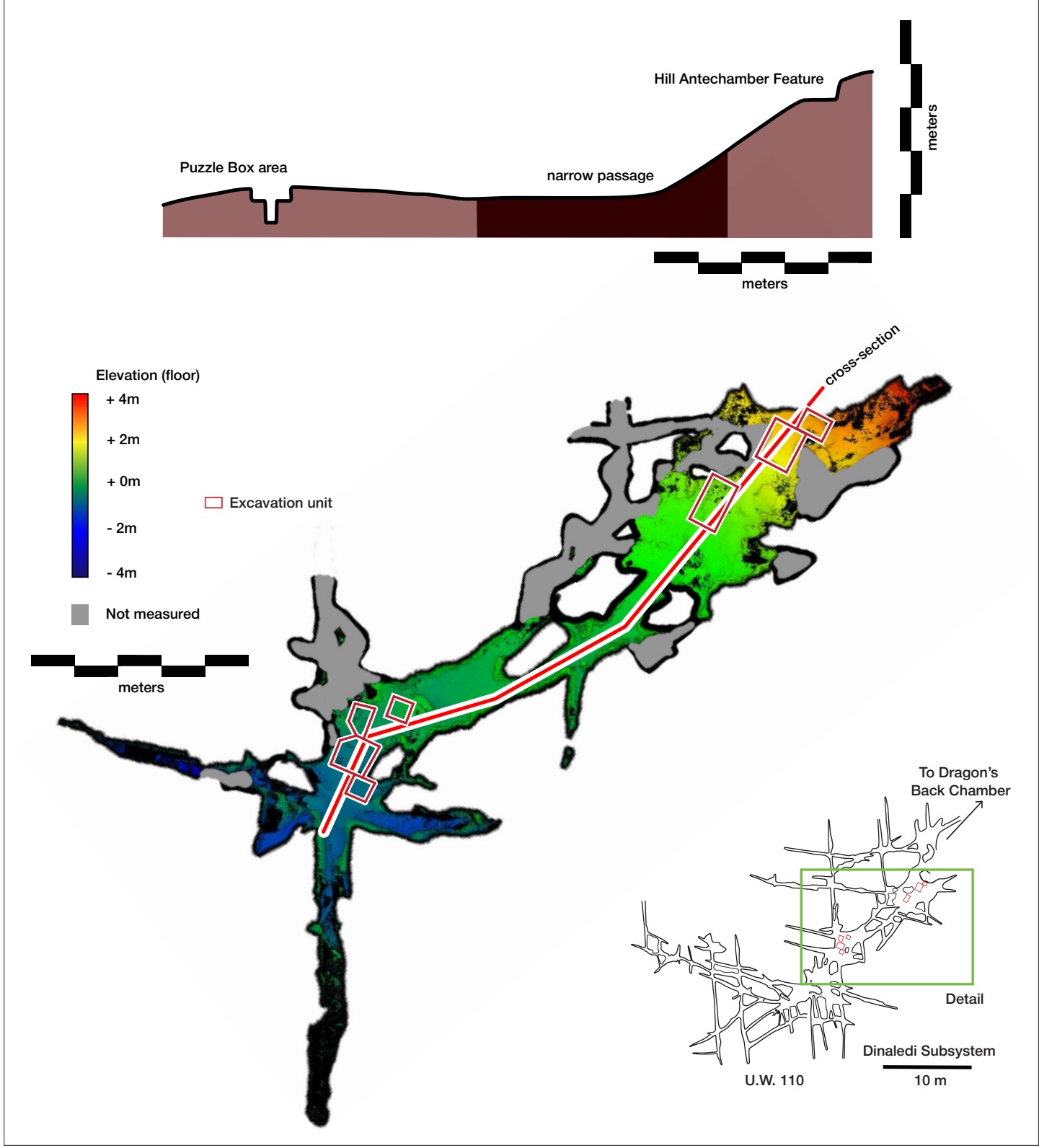

**Figure 3.** Dinaledi Subsystem plan view and cross-sectional view. In the plan view (bottom), floor elevation acquired from FARO scanning is indicated with the color ramp. The location of the cross section is indicated with the red line running from the Hill Antechamber via one of the interconnecting passages into the Dinaledi Chamber. The cross-section, indicated with a red line, includes both the Hill Antechamber Feature and Puzzle Box excavation areas. The cross-section view (top) indicates the relative elevation of these areas and the intervening floor surfaces.

In this study, we discuss the results of these excavations from new areas within the Dinaledi Chamber and the Hill Antechamber (*Elliott et al., 2018*). The overall spatial context of the new and previous excavation areas in the Dinaledi subsystem is presented in *Figure 2*. These new excavations have revealed *H. naledi* skeletal remains and surrounding sedimentary context in settings with substantially less depositional complexity compared to our initial excavation work in this subsystem. The new evidence points to the possibility that other *H. naledi* remains might have resulted from hominin mortuary practices. We therefore examined the evidence from each previously described situation in the Rising Star cave system with *H. naledi* remains in order to test the same hypothesis. For each of these situations, we first considered alternative hypotheses, which vary due to the particular evidence each situation presents.

## Results

### Description of excavation results

Excavations were carried out in 2017 and 2018 in the Hill Antechamber and the Dinaledi Chamber. We identified appropriate excavation units within a 50 cm grid to sample the subsurface composition of different parts of the floor space of these two spaces (*Figure 2*). The Hill Antechamber excavation focused on three units extending from the northeast wall to midway down the sloping floor. We additionally excavated a thin (<10 cm) layer of sediment coating a flowstone shelf that forms the floor of the entry Chute; no fossils or other material were found in this location. The Dinaledi Chamber work involved three excavation areas: one to the immediate south and one to the immediate north of the original 80 cm by 80 cm Puzzle Box excavation area, and one in a nearby open area of the chamber floor with the limited purpose of testing ground-penetrating radar (GPR) analysis in this location.

Excavation revealed that the different areas had a varied subsurface composition of sediments and varied presence of *H. naledi* fossil material. In two areas of excavation, one in the Hill Antechamber and one in the Dinaledi Chamber, we discovered substantial concentrations of skeletal and dental remains of *H. naledi*. These concentrations are made up of *H. naledi* skeletal elements and fragments in close contact with each other, with a thickness extending more than 10 cm, separated from surrounding sediment in which skeletal fragments are rare or absent. After recognizing these concentrations, we defined them as *features*. Our use of this term denotes a spatial association of evidence without any assumptions about formation processes or history. One of these was located within the Hill Antechamber in an area near the subsystem entrance, which we designated as the Hill Antechamber Feature. The other area with similar concentrations of fossil material was located in the Dinaledi Chamber, within the new excavation area just north of the previous Puzzle Box excavation unit. In this area, we exposed and investigated one concentration which we designated as Dinaledi Feature 1. At the edge of the same excavation unit, we exposed a second concentration of fossil material that extends into the wall of the unit, and therefore, its further extent is unknown. This concentration we designated as Dinaledi Feature 2. We have not investigated this second feature further, and aside from noting its presence and separation from Dinaledi Feature 1, it is not part of our analyses. We have left this feature intact to make our hypotheses repeatable and testable for future work.

The Hill Antechamber Feature and Dinaledi Feature 1 provide the majority of new skeletal and contextual evidence relevant to the deposition of *H. naledi* material. To better understand the formation and subsequent taphonomic history of each feature, we carried out additional analyses of the fossil material and sedimentary context. We report the results of these analyses for each feature taken separately.

Other excavation areas in both the Hill Antechamber and the Dinaledi Chamber held much less *H. naledi* skeletal material or other subsurface content. Consequently, the analysis of each unit does not provide the same density of information concerning the formation of the *H. naledi* deposits. While each of these tells only a small part of the story, collectively the results from all areas are important to testing some of the hypotheses considered below. For this reason, we provide a brief description of results from each of these areas, followed by detailed results for the two features. After these descriptions, we provide two additional sections of results. One section includes the results of reanalysis of the Puzzle Box excavation area, which we now recognize as a feature; the other includes results of analysis of the spatial relationships of all fossil material from the subsystem.

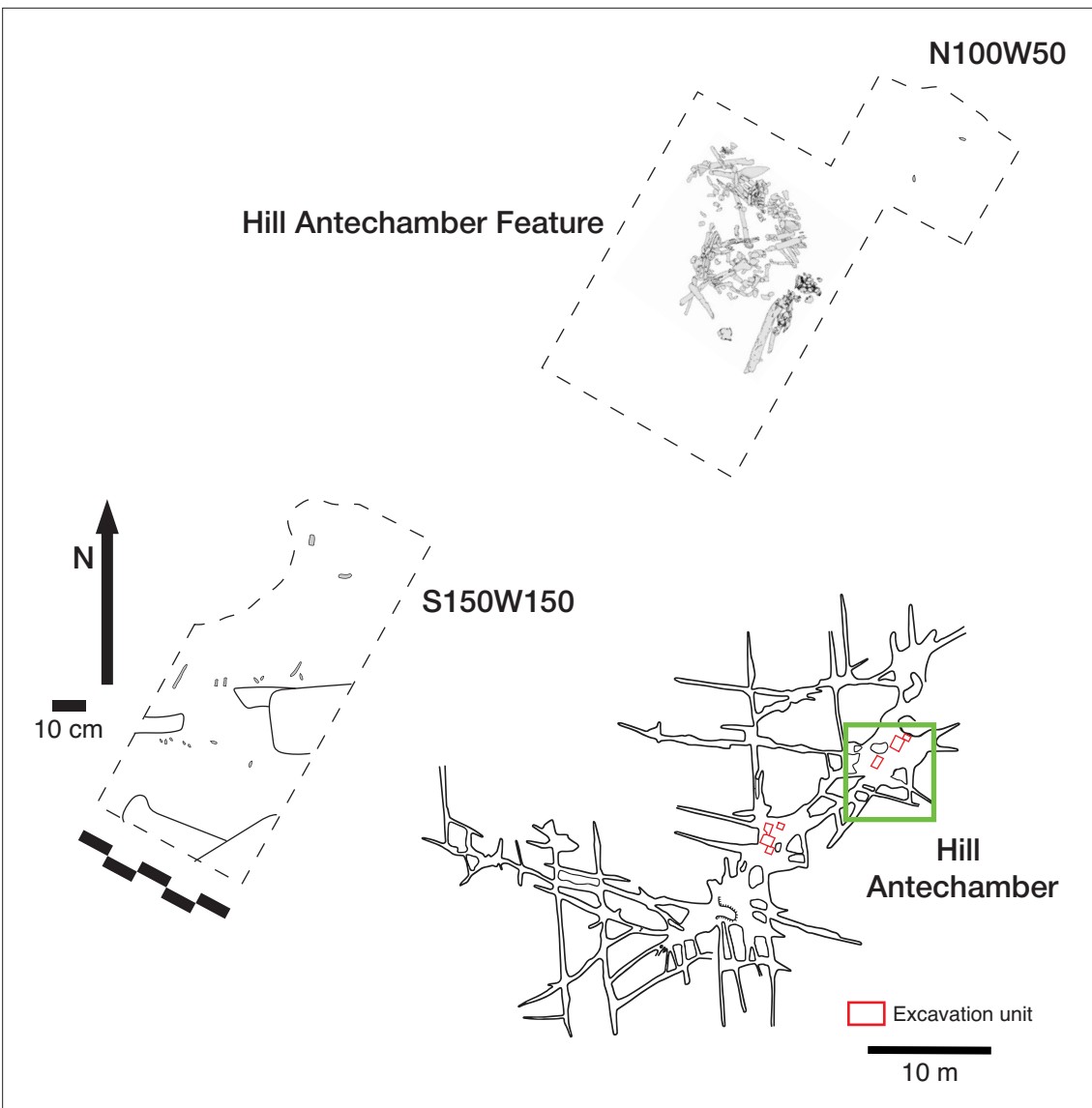

**Figure 4.** Excavation plan in the Hill Antechamber. The location of the three excavation units described in this study is shown, with the skeletal and dental material of the Hill Antechamber Feature in place. The other two excavation areas each produced little skeletal material, with only unidentifiable fragments in N100W50 and 10 teeth with few long bone fragments in S150W150, discussed in text.

## Hill Antechamber excavation

We initiated fieldwork in the Hill Antechamber with a surface search, which identified a few small fossil hominin fragments. Within the Antechamber, the team opened three excavation units (*Figure 4*). The unit at grid square N50W100 was the nearest full 50 cm square to the base of the Chute. It is this unit that produced the Hill Antechamber Feature and is discussed in detail below.

A second unit at grid unit N100W50 was a half unit of 50 cm by approximately 25 cm against the north wall of the Antechamber. The surface in this location slopes away from the wall with an angle of approximately 20 degrees, and from east to west along the wall with a slope of approximately 15 degrees. This unit is diagonally adjacent to the Hill Antechamber Feature. The unit was excavated to a depth below surface of approximately 20 cm. The content of this unit was composed of Unit 3 sediment with a coarse, rubbly texture and including orange LORM clasts as observed previously in other parts of the subsystem. The coloration of this sediment is a darker brown within 10 cm of the chamber wall, and lighter brown somewhat further away. This unit was largely devoid of skeletal material. Only unidentifiable bone fragments were found within this sediment, making up six specimens

(three of these specimen numbers include a few tiny fragments collected together) all collected within 10 cm of the surface. These are all small (<1 cm) fragments and cannot be attributed to element.

The third unit, S150W150, was lower on the slope near the opening to the parallel narrow passages that lead to the Dinaledi Chamber, and our investigation of this area extended to the next grid square to the south, S200W150. This is a complex area near a wall of the antechamber on its west side, with somewhat overhanging rock within 1 meter above the unit. The floor slope across this area is approximately 30 degrees, similar to that of neighboring areas. Prior to opening this excavation unit, the area had several flat rocks on the surface. Surface investigation revealed two hominin teeth and bone fragments. After this material was documented and removed, this excavation proceeded ultimately to a depth of approximately 20 cm. Beneath the surface were some large boulders or protrusions of the bedrock floor, with large (>20 cm) channels of sedimentary content. This was Unit 3 sediment of similar texture and content as the upslope units. This area contained a scatter of skeletal material and teeth of *H. naledi* throughout the excavation depth, numbering a total of 112 specimens. Most of these were small and unidentifiable. Sixteen bone fragments were identifiable to anatomical region or element, including fragments of tibia or fibula, an intermediate cuneiform, phalanx fragments, rib fragments, and vertebral fragments. The area included 10 isolated teeth, all attributable to a single maxillary dentition of *H. naledi* but not recovered in contact with each other. The area included one identifiable faunal element, a partial maxilla of a cercopithecoid monkey consistent with *Papio* (U.W. 101–1992).

On the whole, this excavation area appears to represent a reworked assemblage of bone fragments and teeth. None of the bone fragments were recovered in anatomical order or in articulation with each other. Most are small (<1 cm), with many crushed or disintegrating into the surrounding sediment. No conjoins have been identified among the identifiable fragments. The teeth recovered in this unit include anatomically contiguous elements with interproximal facets and occlusal attrition indicating that they derive from the maxillary dentition of a single adult individual. These teeth were found at a range of depths within the unit, from the surface to 10 cm below the surface, and speak to a process of reworking causing the displacement of elements that were originally conjoined across some horizontal and vertical extent. It is not evident whether the flat rocks on the surface of this area have any association with the skeletal remains. The subsurface material did not include rocks of similar size, suggesting that some size-sorting occurred during the emplacement of sediment, rock, and skeletal remains in this location. Finally, this is the only location within the subsystem where a faunal element occurs in a subsurface context where *H. naledi* remains are also present. The reworked nature of this area leaves the question open whether this faunal element is associated with the *H. naledi* material.

In addition to these three excavation areas, the team investigated a small pocket of sediment on the north wall of the chamber and also excavated the thin layer of sediment resting upon the flowstone shelf at the base of the subsystem entry passage. The sedimentary volume in each of these locations was small and no skeletal material was recovered.

## Hill Antechamber Feature
### General description

The Hill Antechamber Feature (*Figure 5*) was uncovered during excavations conducted in 2017 and 2018. The excavation area that produced this feature commenced in grid square N50W100 and extended into the neighboring square to the south, S50W100, and to the east at N50W50. This area is more than 15 m from the nearest excavated area in the Dinaledi Chamber. Prior to opening this excavation area, no hominin material or other material had been recovered from the surface at this location (*Berger et al., 2015*; *Elliott et al., 2021*).

At a depth of 20 cm below datum and 5 cm below the sloping surface, the team encountered a localized accumulation of bone fragments and skeletal elements across a roughly triangular space some 45 cm by 55 cm. From this exposed surface, it was evident that some dental remains were in relative occlusal position and ordering, some ribs were present in anatomical position and order, and some carpal remains were spatially localized or articulated (*Elliott et al., 2018*). These observations suggested that substantial parts of an articulated or semi-articulated skeleton might be present. The fragile state of the material made it preferable to study it in laboratory conditions, and the team prepared the entire feature for removal *en bloc* from the cave system. This process involved expanding

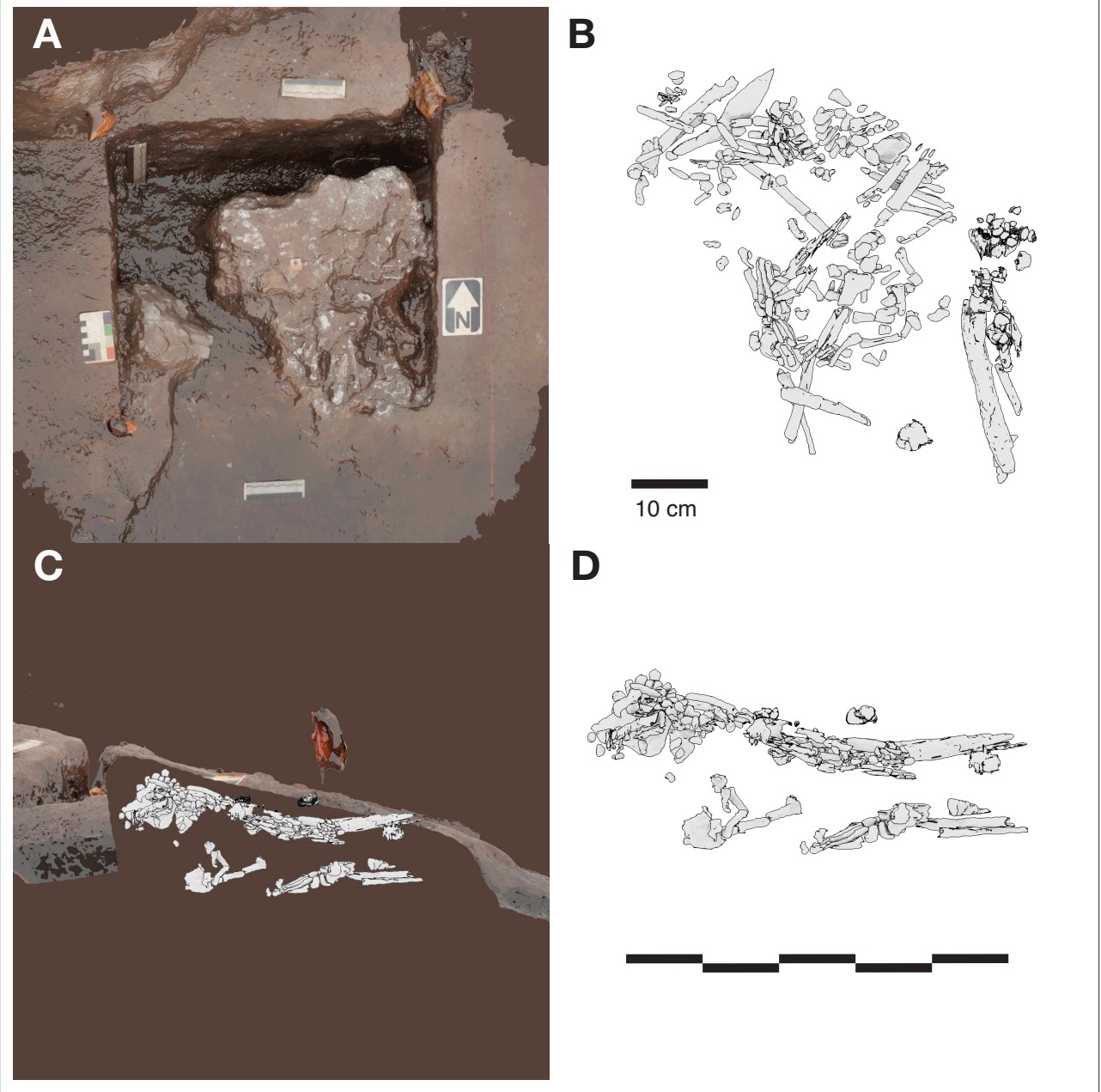

**Figure 5.** Hill Antechamber Feature. (**A**) Photogrammetric model of the feature during the course of excavation with surface partially exposed. North arrow in this image depicts grid north, which is offset by 20 degrees from magnetic north. (**B**) Plan view of skeletal material identified from CT segmentation. (**C**) Transect of photogrammetric model showing the floor surface of the profile flanking the feature excavation unit. A model of the segmented skeletal material within the feature is superimposed on the floor profile in the same orientation to illustrate the position of remains relative to the floor surface. (**D**) Skeletal material segmented within the feature as viewed from the west of the excavation. The material occupies volume across a substantial thickness of sediment.

the excavation to 75 by 75 cm to identify the full outline of the feature. The expanded excavation enabled the team to pedestal the feature to a resulting thickness of approximately 20 cm. To enable its removal securely through the space constraints of the cave system in the vicinity of the 'Chute', the team separated the feature into one larger and two smaller subsections, each of which was encased within a plaster jacket for removal. After removal from the cave system, the two smaller subsections were scanned on a micro-computed tomography (micro-CT) scanner (see Materials and methods). The largest subsection could not be accommodated within this micro-CT unit, and so we obtained

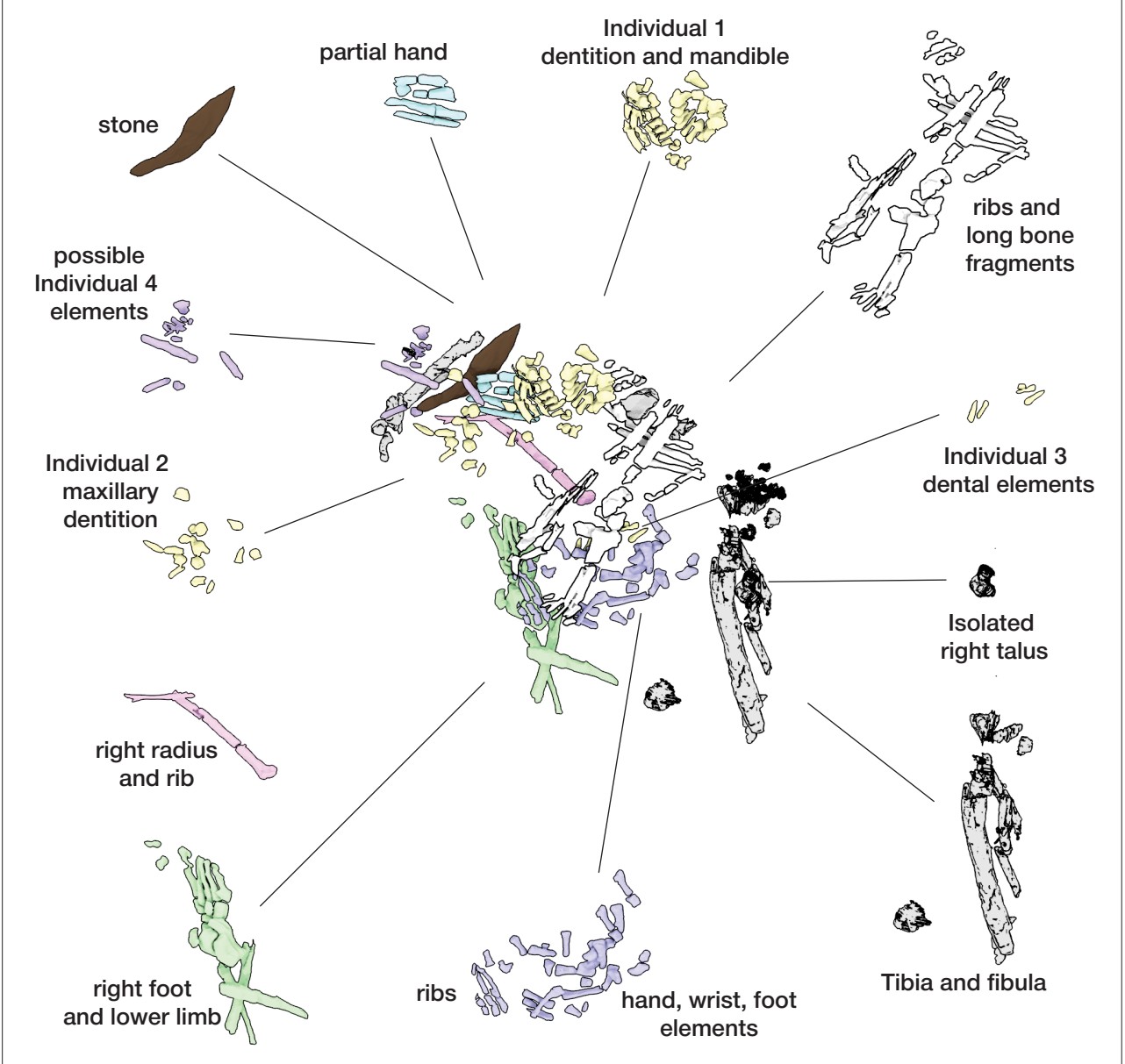

**Figure 6.** Hill Antechamber Feature associated elements. Groups associated by spatial position, articulation, or clustering of contiguous anatomical elements.

a clinical CT scan of this specimen with 0.5 mm voxel size (see Materials and methods). Synchrotron scanning of these specimens is being conducted in order to visualize the specimens in greater resolution as part of future work.

## Skeletal material

After segmentation of the CT scans of the feature, we identified a minimum of 90 skeletal elements and 51 dental elements throughout the volume of the excavated material (*Figure 6*). This is a minimum value based upon identifiable elements from CT data; additional skeletal elements and fragments are visible in the data but are not unambiguously attributable to element, and we have taken a conservative approach to identification. In part, this lack of identifiability of elements is due to the limited resolution of the scan data, but a larger factor is the fragmentation or partial dissolution of bone elements. Prior to jacketing, the dissolution of bone was visible across the feature surface, including a large area

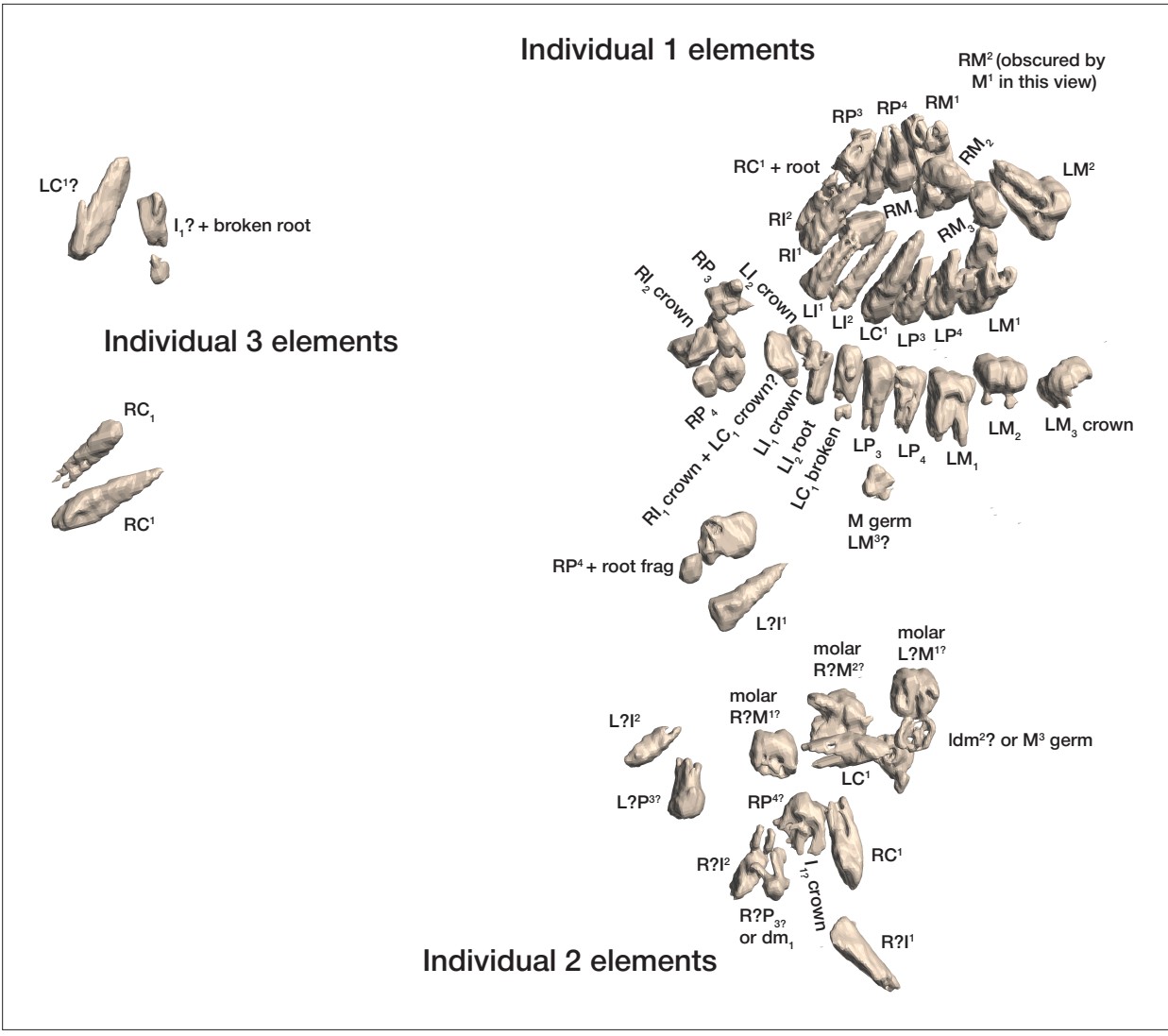

**Figure 7.** Dental elements identified within the Hill Antechamber Feature, viewed from below in their relative positions in the feature. Identification of elements from the CT data is uncertain in a number of cases, represented here by '?'. These provisional attributions are provided to evaluate the minimum number of individuals and likely individuation of elements.

of powdery residue at the highest point of the feature. Within the feature, smaller elements such as wrist and hand bones are sometimes complete, but larger elements are highly fragmented, often with only small parts remaining. The preservation of elements varies greatly by trabecular versus cortical bone representation and cortical bone thickness; for example, none of the axial elements with high trabecular bone content such as vertebral bodies or ilium are identifiable. In comparing these data with fully excavated samples, such as the samples from the Dinaledi Chamber, it is important to note that the two approaches yield different data resolution.

The dental elements include at least 30 teeth from a single individual, which we have designated as Individual 1. All maxillary teeth of this individual are spatially contiguous, in correct occlusal order, and are consistent with being positioned within what was once a complete maxilla that either has degraded or does not present a contrast in density with surrounding sediment (*Figure 7*). The mandibular teeth of Individual 1 are in two disjunct parts. The left mandibular dentition and right $I_1$, $I_2$, $P_3$, and $P_4$ are in correct order and in situ within a mandibular fragment; these teeth are approximately 1 cm from occlusion with the left maxillary dentition. The right mandibular molars are in near occlusion with the right maxillary molars, making clear that the mandible of this individual was in articulation with the skull when these teeth arrived at their current positions. The bone of the right mandibular corpus and

ramus is not evident in the CT data. The third molars of Individual 1 are partial crowns without roots consistent with a late juvenile developmental stage (*Bolter et al., 2018*).

At least fourteen of the remaining teeth represent a second late juvenile individual designated Individual 2. These teeth are localized within the feature, all within a 10 cm diameter, but are not in any anatomical order (*Figure 7*). The identifiable elements are all maxillary teeth. Based on the incomplete roots of the canines, premolars, and second molars, Individual 2 was a slightly younger juvenile than Individual 1, at or shortly after second molar eruption. Four additional teeth represent a third individual designated Individual 3 (*Figure 7*). These teeth include left and right permanent maxillary canines, a right permanent mandibular canine, and a probable mandibular incisor, all localized within an 8 cm area. On the basis of these data, it is not possible to exclude that these teeth represent the same individual identified by 10 maxillary teeth in the downslope S150W150 excavation unit. Testing this will require higher-resolution data capable of visualizing the attritional characteristics of the unexcavated teeth. Review of the CT data suggests that an additional individual of infant developmental stage may also be represented within the feature, but the data resolution is not sufficient to enable segmentation and identification of these possible infant elements.

The identifiable postcranial elements in the feature exhibit substantial anatomical ordering with many articulated parts and other parts showing relative anatomical positioning after loss of trabecular bone portions (*Figure 6*). The spatial distribution of the remains is consistent with the hypothesis that most of the postcranial elements are associated with the Individual 1 craniodental remains. Among the identified material, there is only one duplicated element: an isolated right talus in the easternmost highest level of the feature duplicates the talus of the articulated right foot in the lowest central area of the feature. Many of the elements within the feature are in articulation or sequential anatomical position, including a substantially articulated right foot, ankle, and adjacent lower limb bones, two series of partial ribs, a partial right hand, and ulna and radius. Other elements are in anatomical proximity but not articulated; for example, many identifiable elements of a left hand are within a 7 cm radius of each other, fragments of right tibia and fibula are adjacent to each other with some displacement, and metatarsals of a left foot are localized within a small radius. Elements with high trabecular volumes such as vertebral bodies, tarsals, and pelvic elements are less evident in the CT data; this may either reflect a lack of density contrast with surrounding sediment or the postdepositional degradation of these elements. The excavated skeletal sample from the Dinaledi Chamber underrepresents these same elements, which is interpreted as consistent with postdepositional processes (*Dirks et al., 2015*; *Dirks et al., 2016*).

Although no additional duplications have been identified, some postcranial elements within the feature appear developmentally inconsistent. A number of small unidentifiable diaphyses can be seen in the northwest corner of the feature, which do not appear consistent with adult elements based on size and relative translucency. We hypothesize that these may represent a very young immature individual as suggested for the dental elements above, but not clearly discernible on medical grade CT scans.

## Taphonomy and body position

The evidence above indicates that soft tissue was present at the time of deposition of at least one primary individual (Individual 1). For this reason, we describe what the evidence demonstrates about the postburial dynamics of this individual and other skeletal material. For this purpose, we follow archaeothanatological analysis of element relationships. The CT volume indicates that a combination of labile and persistent joints is preserved. These enable an interpretation of both initial body position (disposition) and subsequent postmortem change (necrodynamics). It is important to note that the position and number of bones can inform questions about the processes of decay and destruction, while the postmortem narrative of all the processes affecting the cadaver is required (after *Knüsel and Robb, 2016*) – this includes both taphonomic/thanatological data as well as sedimentary/geological data. We provide additional discussion in Appendix 5 ('Postmortem change and *Homo naledi*: Understanding decomposition in the Rising Star burial environment') for the framework for this analysis.

Skeletal material within the feature interpreted as belonging to Individual 1 retains persistent joints in anatomical association or articulation including the right talocrural and talocalcaneal joints. Preserved labial joints are represented by the temporomandibular joint (evidenced from dental occlusion), and both the right manual and right pedal skeletons (carpal, metacarpal, tarsal, metatarsal, and

phalangeal joints). Additionally, we consider the preserved position of the Individual 1 dentition to present both labile and persistent components - gomphoses of the anterior dentition are labile, whilst the posterior dentition is persistent. The relative position of the joint elements (*Figure 6*) and presence of labile joints suggests the following hypothesis: The body was deposited in a supine position (evidenced from costal position and orientation), with the head rotated facing to the left. The lower limbs displayed significant flexion at the knee and ankle in a cross-legged position, with the right foot conforming to the lowest level of the burial feature and the left foot placed above the right in a semi-squat. The right arm was tightly flexed, with the right hand placed close to the right auricular region of the head. The left arm was likely placed across the thorax and abdominal region. The labile elements of the right hand and foot are in anatomical association indicating that they were supported by sedimentary matrix before decomposition of the soft tissues. Displacement of the bony elements of the left hand and foot is consistent with decomposition above (hand) and below (foot) of the abdominal and pelvic structures. Secondary voids are created by soft tissue decomposition allowing movement of the elements following loss of connective tissue support. In this case, the bony elements of the left hand disassociated and fell into the lower abdomen and pelvic regions. We interpret the lack of anatomical association in the left foot as due to a higher position relative to the right foot. The presence of decomposition effluvia would have saturated the sediment around the left foot with leachates and putrefaction products, leading to localized instability during the active and advanced decomposition phases. This narrative is based upon archaeothanatological principles (*Duday, 2005*; *Duday, 2009*; *Knüsel et al., 2022*).

## Additional material

In addition to the skeletal remains, the feature contains a single stone object. This stone object is not isolated or separated from the skeletal material in the feature. The closest skeletal material to this stone is bones of a right wrist and partially articulated right hand, with the possible Individual 4 represented by elements recognized on the western side of the feature near this stone also (*Figure 6*). This object is emplaced within the feature at an angle of 25 degrees from horizontal in a southeast-northwest slope. This orientation is a different orientation from any of the skeletal material and is near a reverse angle from the slope of the chamber floor. This stone remains with the skeletal and dental material within its context in the feature. To examine it, we obtained a high-resolution scan of this object from the European Synchrotron Research Facility, and we describe it in the Appendix ('Hill Antechamber Artifact').

## Sedimentary context

The present sediment floor of the Hill Antechamber is a steep slope of between 20 and 30 degrees. The Hill Antechamber Feature was excavated near the top of this slope, approximately 50 cm from the northeast wall. In this area, the floor slope proceeds from the northeast toward the southwest, with both the floor surface and subsurface layers sloping away from the north wall and along that wall from east to west. As a result, the entry point into the chamber is not the highest point on the sediment floor (*Figure 3*). As in the Dinaledi Chamber, the floor of the Hill Antechamber is composed of sub-unit 3 a and sub-unit 3b unlithified mud clast breccia, including laminated orange-red mud (LORM) clasts. The available data on the profile of surrounding sediment includes profiles on three sides in addition to a profile recorded upon the exposed south face of the feature itself during excavation (*Figure 8*, *Appendix 6—figures 6 and 8*). The excavation profiles at the east and south edges of the feature show distinct alternation of LORM clast-rich and LORM clast-poor layers that are roughly parallel to the current chamber floor, at the same 25 degree slope.

Even at lower resolution, the CT data enable some examination of the stratigraphic and microstratigraphic situation of the feature (*Figures 8 and 9*, *Figure 10*). Layers of clast-rich Unit 3 sediment that are evident in the surrounding profiles do not continue within the feature. Instead of the sloping layers of the surrounding subsurface, the sediment in contact with the skeletal material exhibits two clast-rich zones. The bottom of the feature is defined by a roughly bowl-shaped concave layer of clasts and sediment-free voids. At the south end of the feature, this concave clast layer is sloping in the inverse direction compared to the chamber floor. The right foot and lower limb material is supported by this concave layer, sloping southwest-to-northeast approximately 15 degrees. Above the foot is additional sediment with a high density of clasts and small air pockets or voids, separating it from

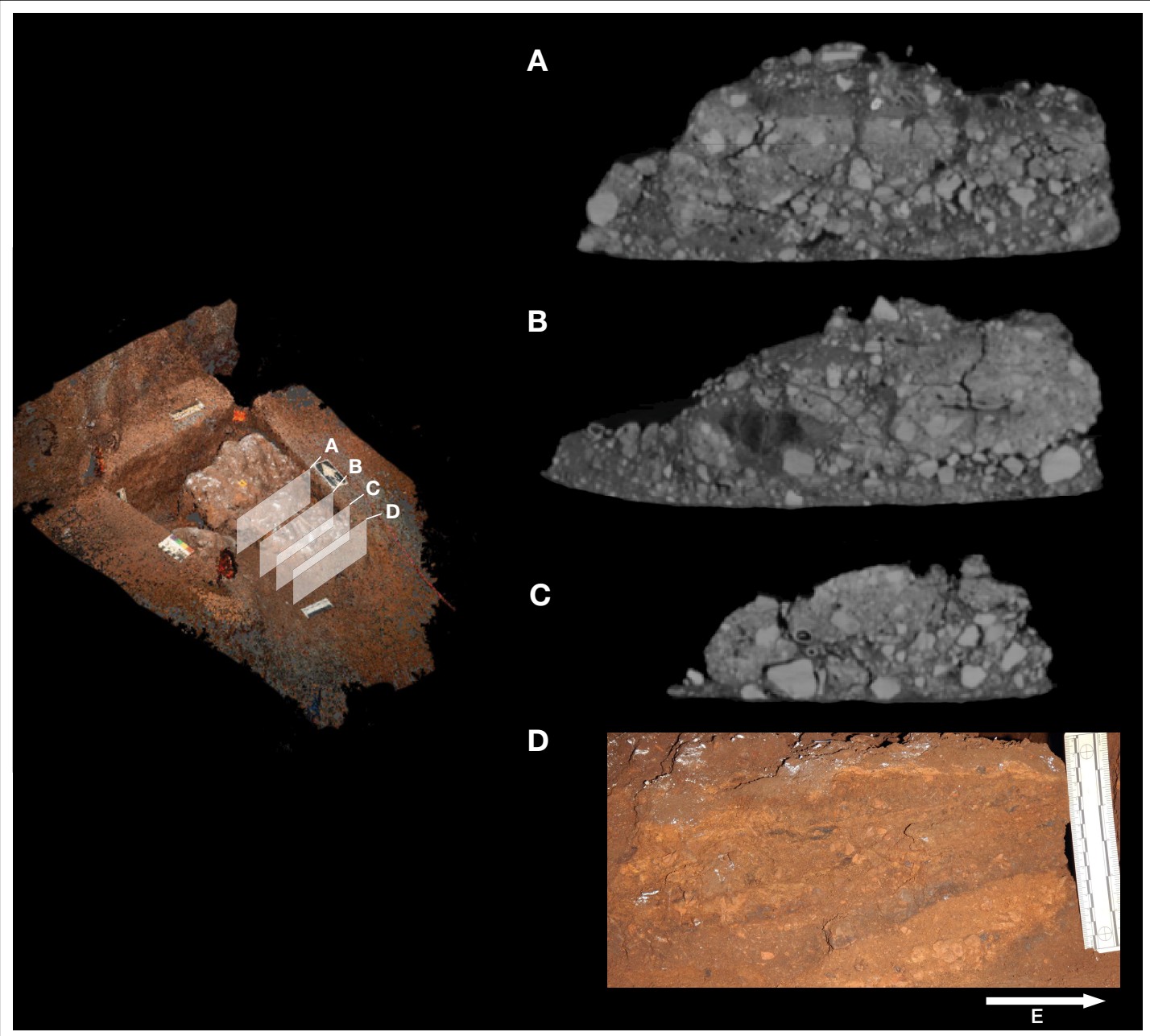

**Figure 8.** Stratigraphic situation of the Hill Antechamber Feature. At left, a photogrammetric visualization of this excavation area shows the locations of four cross-sections labeled A, B, C, and D, which are pictured at right. These are arranged from north to south across the designated feature, with the final one being the excavation profile exposed as sediments to the south were cleared during the pillaring of the feature. All sections are shown at the same scale. (**A**) Section 20 cm from south end of feature. Four digits of the articulated foot are visible at lower left of this section. A concave rubbly layer along the bottom of the excavated unit slopes upward at both sides of the section, while an area of large clasts occupies the center of the section above and to the right of the foot. (**B**) Section approximately 10 cm from south end. Tarsals of the articulated foot at center-left. Here, the rubbly layer is a concave region below the tarsals and bone cross-sections at far left. (**C**) Section approximately 5 cm from south end of feature, at same approximate scale. The bottom half of this section is dominated by the same rubbly clast-rich area. Cross-sections of tibia and fibula are visible at center-left of section. (**D**) Sediment profile at immediate south end of feature. Orange layers are rich in LORM content, with many visible clasts, and these alternate with darker-colored layers. The layering is roughly parallel with the east-west slope of the chamber floor in this profile, somewhat increasing in slope in layers below 10 cm. This pattern to the outside of any skeletal remains is not parallel to the situation where skeletal remains are present, despite being only a few centimeters from the section shown in panel C, and an additional 5 cm from the section shown in panel B.

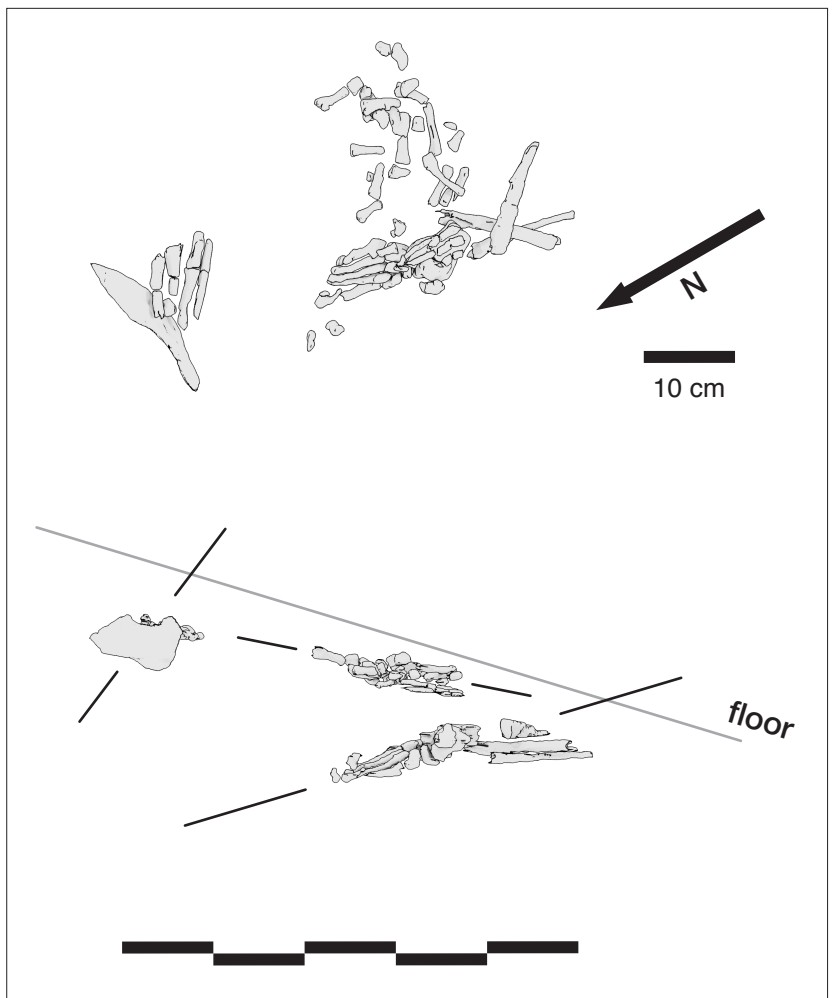

**Figure 9.** Vertical separation and angulation of articulated and semi-articulated material in the Hill Antechamber Feature. Top: Plan view of articulated hand (next to stone), articulated foot with lower limb elements, and disarticulated hand and foot elements. Bottom: View of same elements from the west. The angle of the floor in a north-south profile is indicated, which contrasts with the angulation of the articulated foot, the disarticulated hand and foot elements, and the orientation of the stone.

the upper limb and torso material. This skeletal material follows a slightly concave plane that slopes northeast-southwest, opposite the slope of the foot and leg, and different from the 30 degree slope of the chamber floor (*Figure 9*). The spatial characteristics of the skeletal remains and associated infill contrast with the immediately surrounding sediment (*Figure 8*).

Two smaller portions comprising the easternmost side of the feature were separated at the time of excavation, and microCT data were obtained of their contents and composition (*Elliott et al., 2018*). These specimens are designated as U.W. 101–2074 and U.W. 101–2075. These specimens enable a higher-resolution examination of the sediment context immediately surrounding skeletal material (*Figure 10*). The sediment within these parts of the feature is Unit 3 unlithified breccia with clasts of laminated orange-red mud (LORM) as described in the subsystem by previous work (*Dirks et al., 2015*; *Dirks et al., 2017*; *Wiersma et al., 2020*). These clasts are not layered in this part of the feature, and they occur at random orientations with angular outlines as described elsewhere. Some isolated skeletal elements are supported within this Unit 3 sediment (*Figure 10e*). In some portions, skeletal elements are immediately surrounded by sediment with lower radiodensity that contrasts with the undifferentiated Unit 3 sediment. For example, the U.W. 101–2074 jacketed specimen contains an area with trabecular bone fragments embedded within such low radiodensity sediment (*Figure 10a and c*). This sediment has a similar density

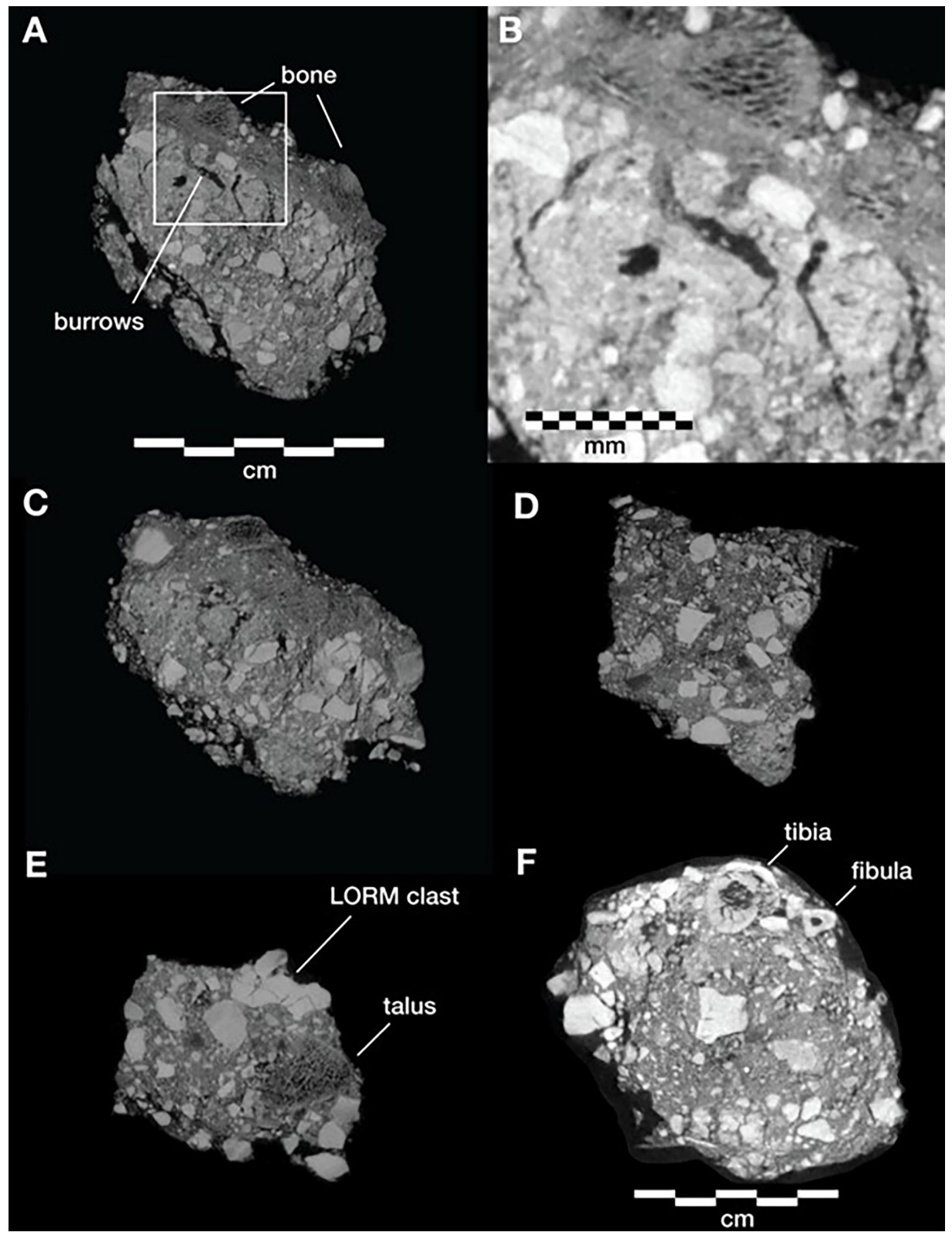

**Figure 10.** Microtomographic sections within the Hill Antechamber Feature. (**A**) Cross-section through the U.W. 101–2074 jacketed mass. The excavation surface is top; the right side of this section was in contact with the larger U.W. 101–2076 mass prior to excavation. In (**A**) small, partially fragmented bone elements are visible at the top of the section, embedded within a sediment mass with low radiodensity (darker on this image). Below these bones and low radiodensity sediment is higher radiodensity sediment characteristic of the Hill Antechamber Feature as a whole. The

*Figure 10 continued on next page*

*Figure 10 continued*

highest radiodensity (lightest in color) objects in the section are LORM clasts, with a distribution characteristic of the feature as a whole. Square shows the region illustrated in panel B. (**B**) Detail of section shown in panel A showing sediment interface below bones, with invertebrate burrows visible. (**C**) Section near that shown in panel A, with similar features but gradient of sediment radiodensity. The interface shown in these panels is interpreted as invertebrate activity and diffusion or percolation of material associated with decomposition of soft tissue or diagenesis of bone. (**D**) Section of U.W. 101–2074 with the same orientation as in panels A and C, but without bone evident. Texture and composition are typical of Unit 3 sediment throughout the subsystem. (**E**) Section of U.W. 101–2074 with an isolated right talus visible at right. No sediment contrast or halo is evident near this bone, which was likely isolated at the time of burial in this position. (**F**) Section of U.W. 101–2075, with orientation as in other panels but different scale. The top of this section includes tibia and fibula shafts. As in panel D, the Unit 3 sediment here underlying skeletal material shows no evident layering and chaotic orientation of LORM clasts.

to bone and may itself be the result of diagenesis of bone in the deposit; alternatively, it may represent the mixture of Unit 3 sediment with organic material associated with this area of skeletal remains. The sediment surrounding this mass of bone and low radiodensity sediment includes hypothesized invertebrate burrows. These are sinuous in form with diameters of around 2 mm. These are air-filled in some places, and in other places, they are filled with the lower radiodensity sediment. These burrows within the U.W. 101–2074 jacketed specimen are common in areas near to and underlying the mass of trabecular bone fragments and associated sediment; they are not visible in areas of Unit 3 sediment without such bone distributions (*Figure 10d and e*). Similar burrows appear to be present within the larger U.W. 101–2076 specimen, although the available scan resolution does not allow a close examination of their structure and content.

## Dinaledi Chamber excavation

The Dinaledi Chamber was the subject of previous research in the subsystem, including most of the collection of skeletal material from the floor surface and the limited excavation of the area known as the Puzzle Box area. In this study, we report on three new excavation areas. Two of these immediately flank the initial Puzzle Box excavation area on its southeast and northwest edges (*Figure 11*). The northern area of excavation led to the discovery of Dinaledi Feature 1, described below. The southeast unit at grid square S1050W500 was very different from either of the other two areas in its low abundance of subsurface fossil material. A third excavation area was opened during 2019 in a test of GPR methodology within the chamber. This area at S875W475 was excavated to a depth of 8 cm. Only three fossil specimens were recovered in the excavation of this area, two unidentifiable and one long bone fragment consistent with *H. naledi*. This area, which is largely devoid of fossil material, provides information about the stratigraphic profile in areas without dense subsurface bone.

The excavation of the northern unit at grid square S1050W100 continued in 5 cm levels until a depth of 20 cm was reached, bringing its base into line with the Puzzle Box excavation. This yielded 58 fossil specimens. Most are fragmentary, including cranial fragments and long bone fragments. A few complete pedal elements and four teeth or tooth germs consistent with a single juvenile *H. naledi* individual were also found. The skeletal material in this unit was localized near the boundary joining this excavation unit to the Puzzle Box excavation unit, across the complete range of depths. Elements within an anatomical region were in some cases clustered spatially, including three teeth from an immature individual. However, the evidence does not make clear whether all this material may represent a single individual. Considering the spatial proximity to the Puzzle Box excavation, these elements may comprise one or more parts of the more complex array of remains from that area.

## Dinaledi Chamber Feature 1
### General description

The Dinaledi Feature 1 (*Figure 12*) was uncovered in 2018 in an excavation unit within the Dinaledi Chamber just to the north of the Puzzle Box area. Within this excavation, beginning at a depth of 8 cm, a concentration of bones and accompanying soil disturbance forms a roughly oval-shaped distribution. On its south side, which is nearest to the Puzzle Box excavation, the roughly 50 cm space between this feature and the earlier excavation area included three fragments of different long bones. On the west and north sides, the oval shape of Feature 1 is delimited from surrounding sediment in which few small bone fragments occur. On the east side, the feature is at the edge of the excavated

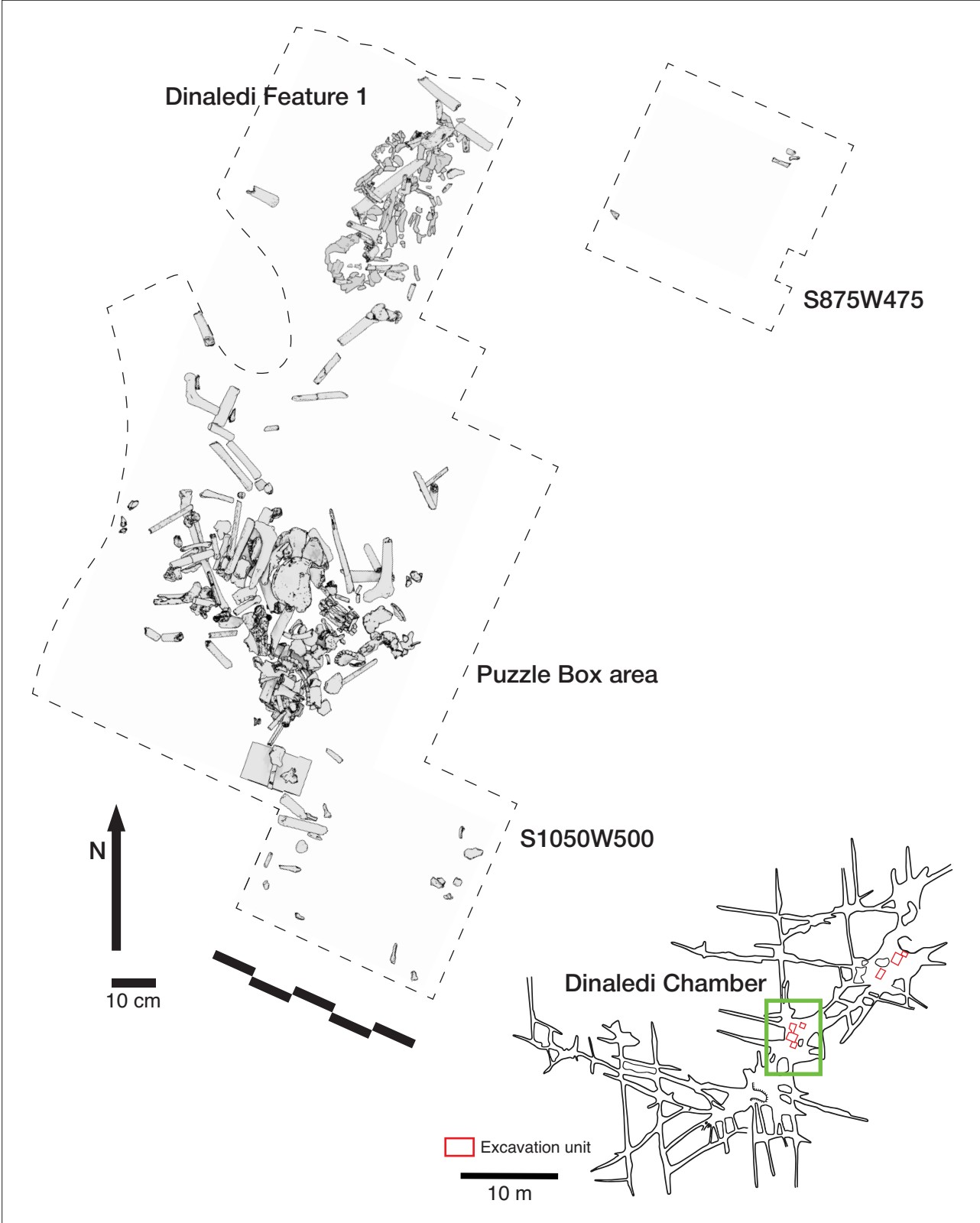

**Figure 11.** Excavation plan in the Dinaledi Chamber. The location of the three excavation areas described in this study is shown, together with a plan view of the skeletal material in the Puzzle Box area and the Dinaledi Feature 1 area.

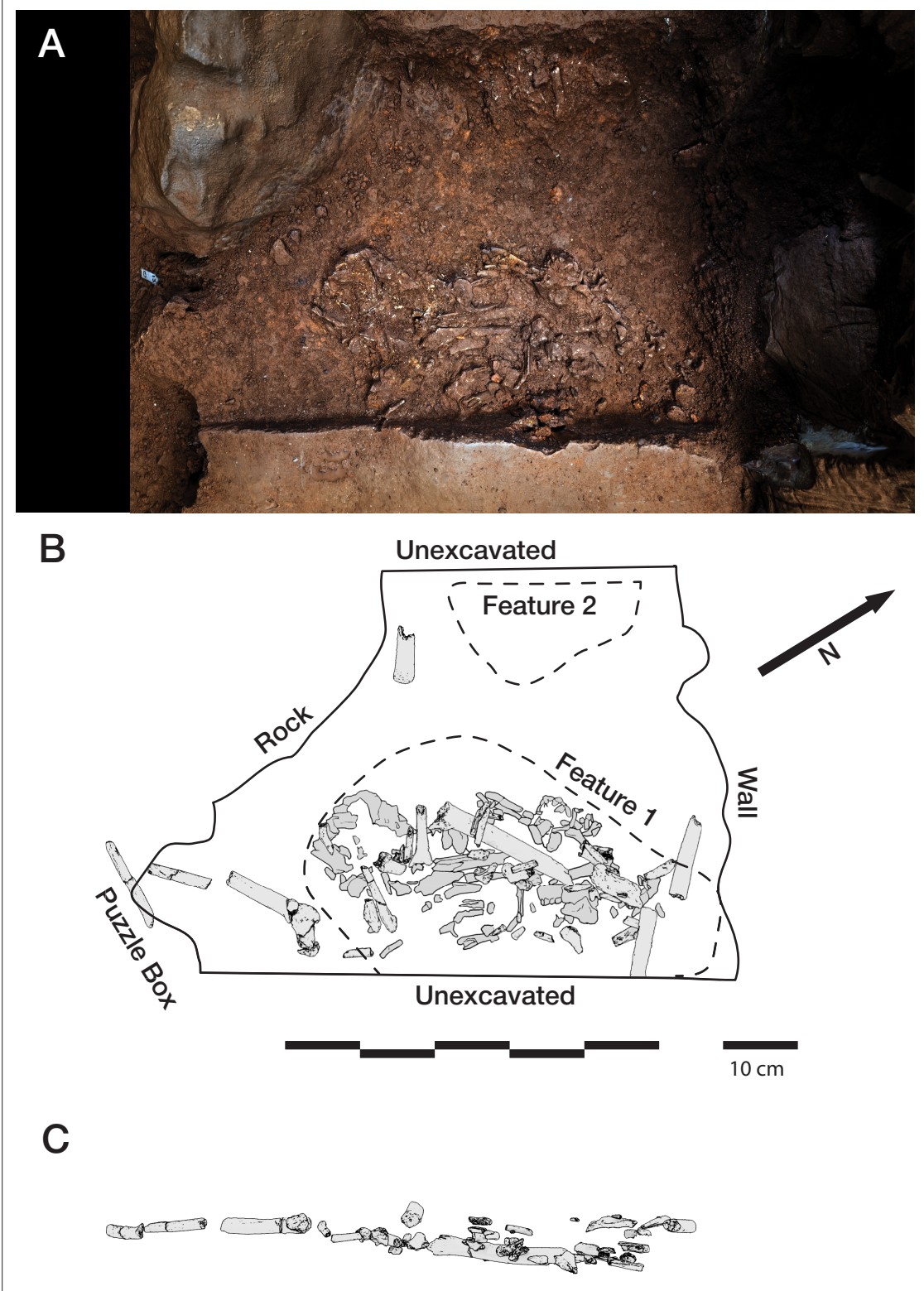

**Figure 12.** Dinaledi Feature 1. (**A**) Photogrammetric model of excavated unit including Dinaledi Feature 1 and part of Dinaledi Feature 2. These data derive from a time before the cutting of the profile on the south edge of the feature as shown in *Figure 15*. (**B**) Schematic showing excavated area, neighboring areas, and skeletal material. All skeletal remains are shown, including those identifiable within the unexcavated feature and those excavated from above the current level. While there is additional material visible within the unexcavated portion of the feature, the boundaries and remaining depth of such material are not yet known. (**C**) Positions of excavated elements as viewed from east excavation face. Elements remaining in the feature are not included in this view.

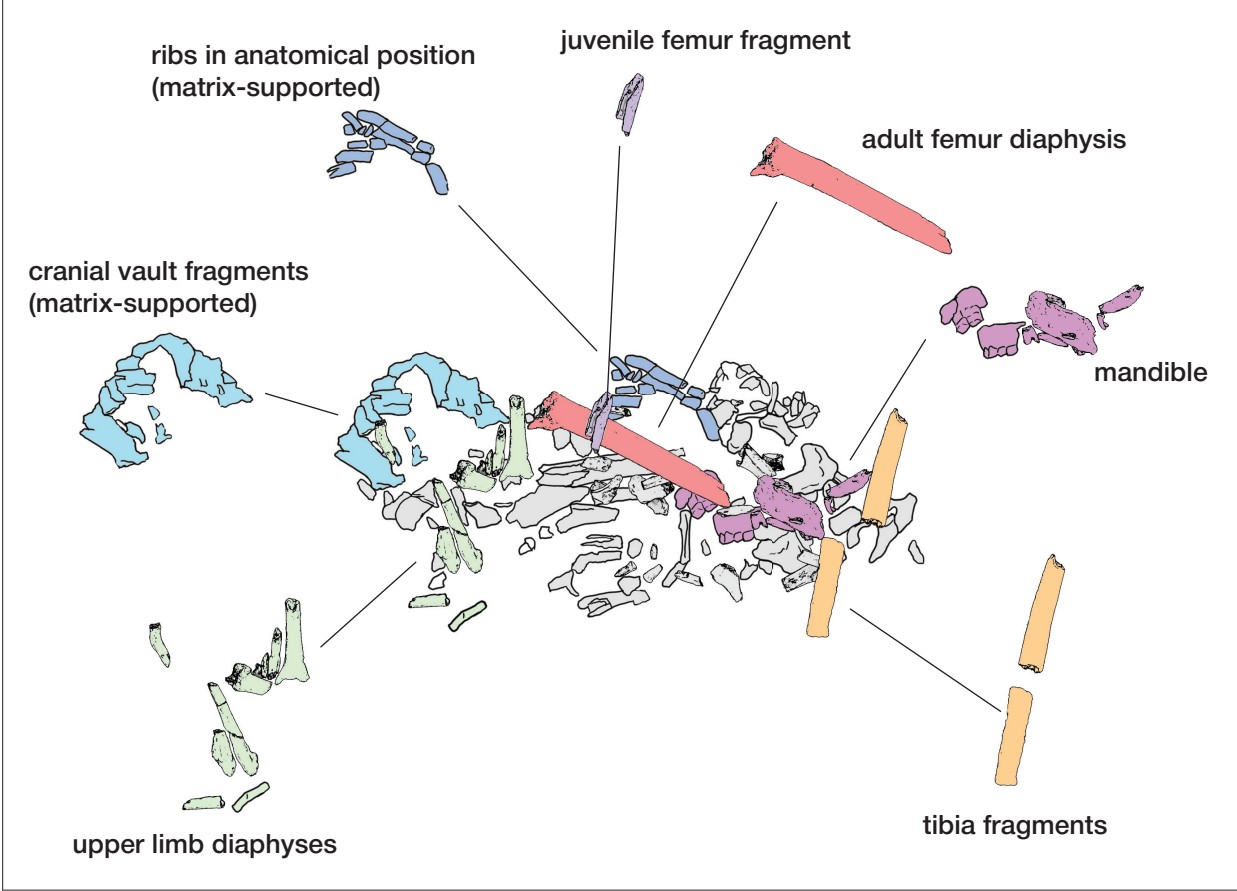

**Figure 13.** Skeletal element localization in the Dinaledi Feature 1.

area, with part of it unexcavated. The exposed area of the feature measures 50 cm in total length and 30 cm in breadth. This area of the chamber floor has a slope of ~11 degrees from magnetic north to south, and the feature itself has a horizontal orientation beneath this sloping surface.

Once the excavation work exposed the length of the feature, it became clear that articulated and matrix-supported elements are present. At this point of recognition, we ceased excavation to study the remains in place and leave witness sections for future hypothesis testing. Despite being in roughly planar view, some of the material left in place is readily identifiable from the exposed anatomy; however, substantial fossil material has not been excavated to an extent that would render individual elements or fragments identifiable. At the southernmost extent of the feature, we excavated a profile into underlying sediment that reveals a remaining depth of skeletal material of 3 cm in this location. Given that this profile is at the edge of the feature, the data do not show whether this depth is consistent across the remainder of the feature or what its total vertical extent may be at its deepest point.

## Skeletal material

Above and within the exposed circumference of the feature, excavation yielded 108 skeletal and dental fragments, of which 83 are identifiable to element. All identifiable elements are consistent with *H. naledi*. Some of the excavated elements were in direct contact with underlying fossil material, while others were above the concentration of bone that comprises the feature, separated by up to 5 cm of sediment. The excavated hominin remains (Appendix) include a fragmentary adult left hemi-mandible with associated dentition, partial adult right femur, fragments of both left and right humeri, left and right ulnae, left and right radii, vertebral and rib fragments, ischium fragment, and some cranial fragments (*Figure 13*). These excavated remains do not include any specimens in articulation, although some were consistent with anatomical positioning.

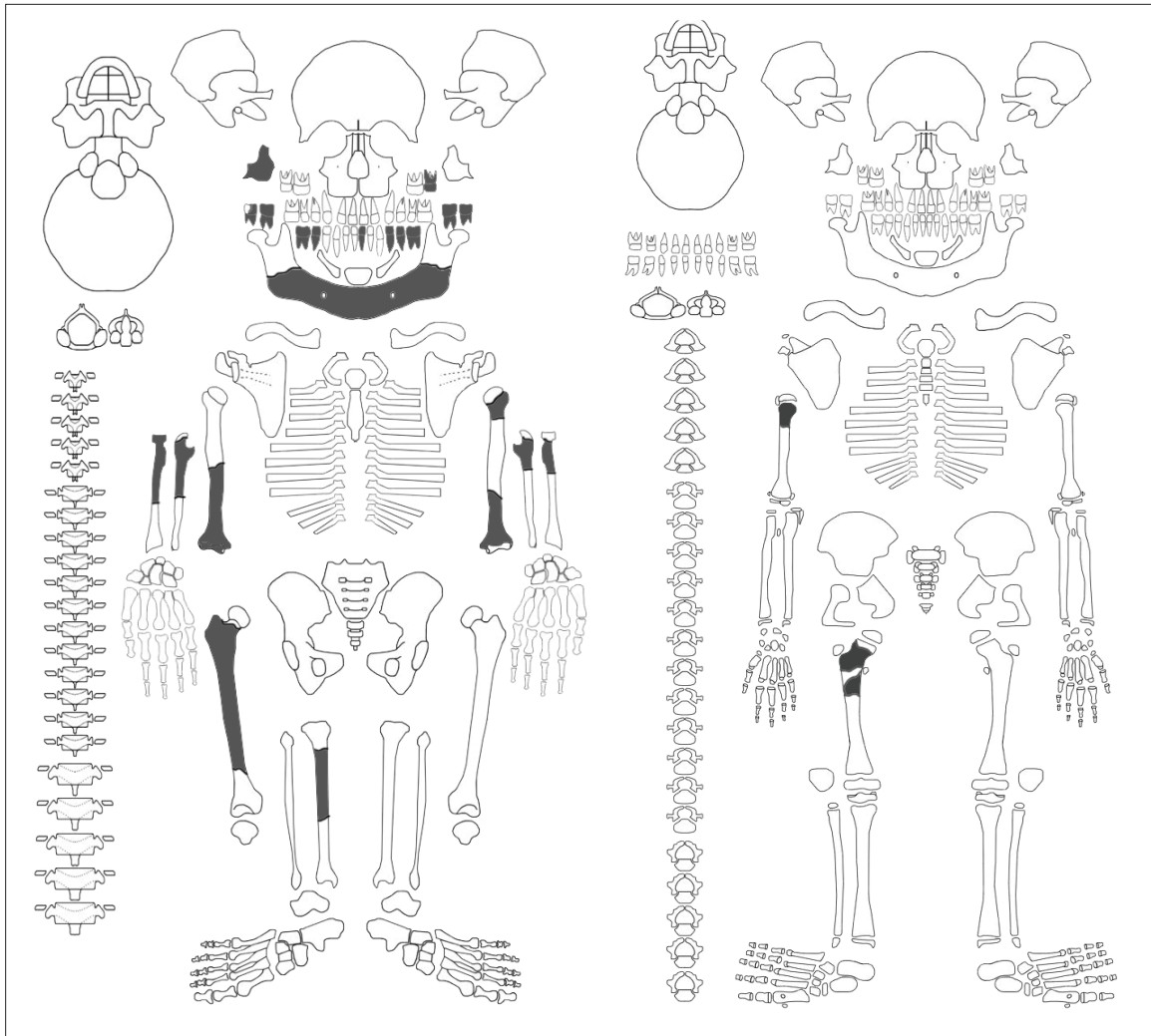

**Figure 14.** Skeletal part representation in Dinaledi Feature 1. Excavated fragments are shown in dark gray, in addition to the portions of mandible that are unambiguously identifiable within the site. Other material, both within the deposit and within the excavated collection, is identifiable to region but not necessarily to element. For example, a partial cranial vault is in place, many ribs are represented, some partial phalanges and fragments of metacarpals, and some vertebral fragments. These are not indicated in the diagram; none of them duplicate any identified element. Excavated elements are listed in the Appendix. At least two postcranial elements attributable to a juvenile individual are represented; these are indicated at right. Recording forms after *Roksandic, 2002*.

These recovered skeletal remains represent only a small fraction of the feature's contents. Most of the skeletal material is still in place in the Dinaledi Chamber. Although much of this in situ skeletal material is obscured by sediment, many elements within the unexcavated portion of the feature are identifiable (*Figure 13*), and some are in articular order, including a series of ribs that were supported by matrix. None of this visible material duplicates elements that were recovered during excavation of the feature. Long bone fragments that are visible are primarily lower limb elements that do not duplicate the previously excavated femur or upper limb material. Fragments of a right hemi-mandible are present that are consistent with the excavated left hemi-mandible. A large fraction of a cranial vault is also present.

The skeletal representation and spatial relationship of elements are consistent with the hypothesis that Feature 1 includes predominantly the remains of a single body (*Figure 14*). Three of the fragments that we excavated directly above Feature 1 represent two elements of at least one additional individual. An immature proximal right femur was excavated, touching the adult femur but not in the same orientation. A fragment of an immature proximal left humerus also was excavated above the feature.

### Condition of the skeletal remains

All fragments recovered from this excavation unit were subject to a minimal preparation protocol. This enabled identification of bone fragments and characterization of fracture patterning, but the protocol was not designed for full examination of small-scale surface modifications such as cutmarks or marks from invertebrates. The condition of all skeletal material recovered from this feature is fragile. The recovered portion of the mandibular corpus is friable, light in mineral content, and characterized by microfracturing across its entire extent. Trabecular-rich bone portions are poorly preserved, and no evidence has been found from this feature of bone portions with predominantly trabecular content such as humeral or femoral heads. Portions of phalangeal and metacarpal shafts and portions of upper

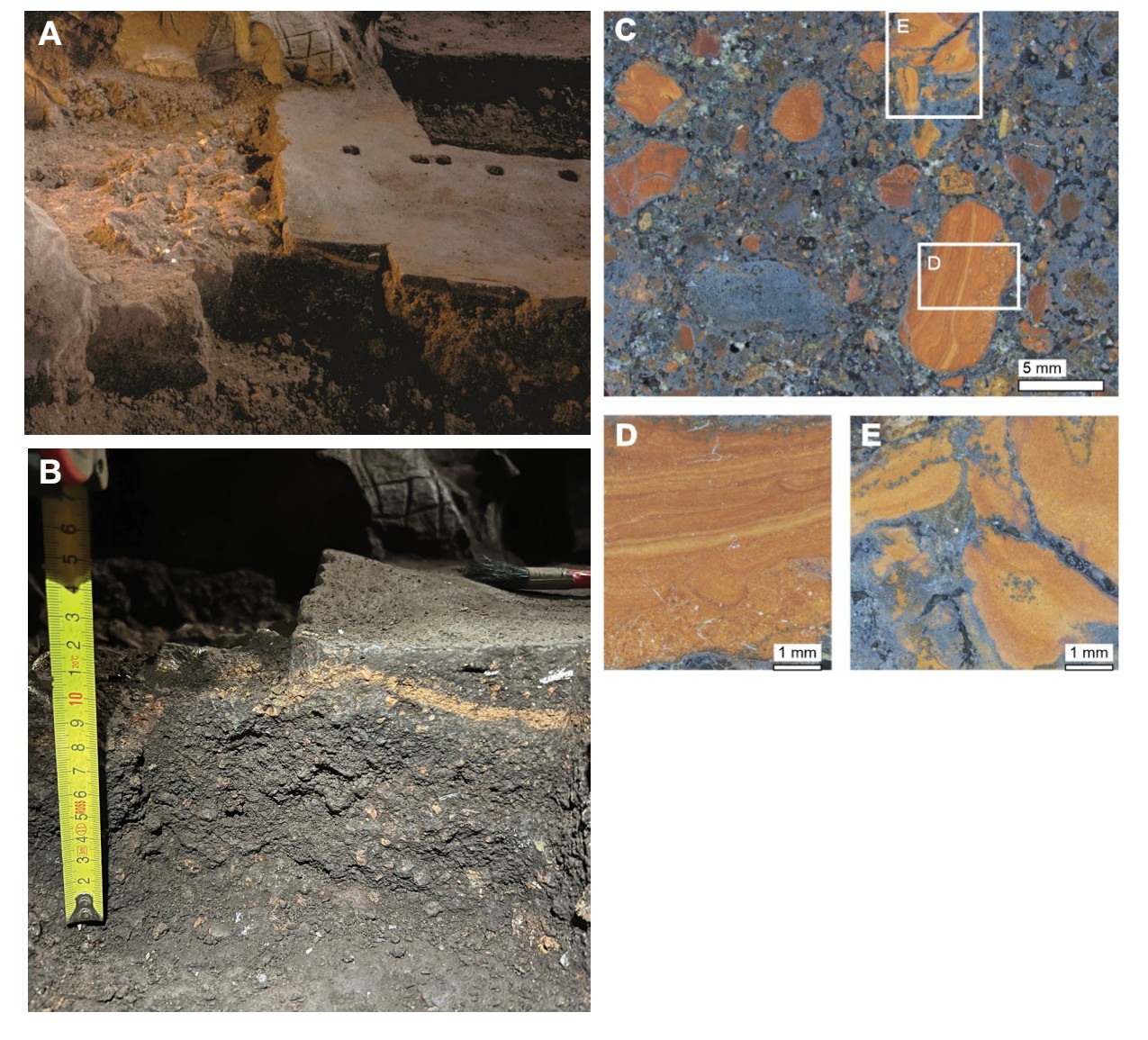

**Figure 15.** Sedimentology and stratigraphy of unlithified mud clast breccia and laminated orange-red mud clasts surrounding Feature 1. (**A**) North-facing overview of Feature 1 showing the relation of the sediments around the fossils and height of profile. (**B**) Profile view. Feature 1 occurs within unlithified mud clast breccia (UMCB) rich in orange-red clasts. A continuous laminated orange-red mud layer beneath the unexcavated floor surface dips near the feature, where it becomes fragmented and muddled. (**C**) Photomicrograph of sediment beneath the burial feature showing the in situ poorly sorted fabric of the unlithified mud clast breccia. (**D**) Close-up photomicrograph of a laminated orange-red mud clast. The clast contains up to 30% sand and has mm-scale laminations. (**E**) Close-up photomicrograph of laminated orange-red mud clasts coated and impregnated with secondary Mn- and Fe-oxyhydroxides in brown-gray silt and clay matrix of the unlithified mud clast breccia. Note that (**D**) and (**E**) have been intentionally rotated 90° right from their in situ position in (**C**) for easier viewing of microstratigraphy of LORM.

limb shafts are relatively well preserved, but proximal ulna and radial heads are not present. Cortical bone portions such as long bone shafts exhibit columnar fractures consistent with postdepositional fracturing (*Lyman, 1994*). No observed fractures are indicative of perimortem fracturing (*Christensen et al., 2022*).

This pattern of preservation of skeletal material is similar to that observed in the Puzzle Box area where material was excavated previously (*Berger et al., 2015*; *Dirks et al., 2015*). The low representation of bone portions with high trabecular content and low mineral density is consistent with postdepositional processes. Several observations rule out perimortem processes such as carnivore mastication or pre-depositional trampling, most notably the lack of any evidence of tooth marks, punctures, or trampling marks on any remaining bone surfaces (*Dirks et al., 2015*).

## Sedimentary context

The feature is within the unlithified mud clast breccia (UMCB) of sub-unit 3b, the fossil-bearing sub-unit in the Dinaledi Chamber (Appendix; *Dirks et al., 2017*; *Wiersma et al., 2020*; *Robbins et al., 2021*). UMCB is mostly chaotic and unstructured without any observable microstratigraphy and consists of orange-red angular mud clasts derived from the laminated orange-red mud (LORM) facies from Unit 1 deposits (*Figure 15*; *Wiersma et al., 2020*). Around the feature, we found a laterally continuous layer that was not noted during previous excavations in the Puzzle Box area where the orange-red angular mud clasts were reported (*Figure 15*; *Dirks et al., 2015*; *Dirks et al., 2017*). This layer occurs at a depth of between ~3 and 7 cm below the current surface. This layer is identical to LORM in that it has a rusty orange color with visible layering of alternating lighter and darker sub-mm bands (*Figure 15c and d*; *Dirks et al., 2017*). In some places, it has lenses of mostly silt-sized grains of mainly quartz or impregnation with Fe-Mn-oxihydroxide concretions (*Figure 15c–e*). We have designated this laterally continuous layer as the laminated orange silty-mudstone unit (LOSMU) to distinguish it from the remnant LORM, in other parts of the cave system, not yet transformed to UMCB or not having any association with UMCB. LOSMU is disrupted in the sediment profile at the southern extent of the feature as if it is a lens of LORM, but continuous in the southern wall profile as the layer continues to the east of the feature, shown in *Figure 15a and b*. At the intersection of this southern profile with the east profile of the Puzzle Box area shown in *Figure 15b*, the LOSMU layer continues further toward the south at the same depth. The abundance of laminated orange-red mud clasts with grain sizes >2 cm around Feature 1 is hypothesized to emanate from LOSMU. These clasts occur without layering or structure and at random orientations, as noted within the earlier Puzzle Box excavation (*Figure 15c*; *Dirks et al., 2015*). Some of these LOSMU clasts are in direct contact with skeletal material. In plan view, a curving line of sediment exhibiting textural and color contrast is evident just to the west of the edge of the skeletal material. This line of contrast is shown in *Figure 12a* and is demarcated in the schematic in *Figure 12b*. At its southern end, this contrast boundary is in contact with a cluster of cranial fragments; from there, it first extends away from the skeletal material, and at its northern extent, curves back east toward the exposed skeletal remains, merging with their edge (*Figure 13*). The area between this line and the edge of Feature 2 is devoid of skeletal material, except for one long bone fragment recovered at a depth of 5 cm.

We used particle size distribution (*Appendix 6—table 1*) characteristics to investigate if there are variations and any resolvable microstratigraphy in the UMCB sediments around Feature 1 using five sediment groups. These sediment groups are DF, sediments removed above the feature; SB, sediments from within the feature; SA, sediments to the east of the feature; SC, sediments between Features 1 and 2 at the same level of the features (to the west of Feature 1); SE, sediments from a vertical profile south of the Feature 1 (see Materials and methods). The distribution curves of the particles in the samples of each of these sediment groups are largely identical (*Figure 16a*; *Appendix 6—figure 17*). There are also no statistically significant variations in the median particle distributions between the sediment groups as indicated by a Kruskal-Wallis H test statistic (henceforth, H test statistic) of 0.66 and p-value of 0.96 (a p-value higher than 0.05 indicates that there is no significant statistical difference). Visually using violin plots, SB has a distinctive variation of grain size and sorting markedly different from SA and SC, but similar to DF and SE (*Figure 16b and c*). The mean grain sizes for DF, SA, SC, and SE range from 344 to 353 µm while SB has the lowest value of 319 µm. All groups

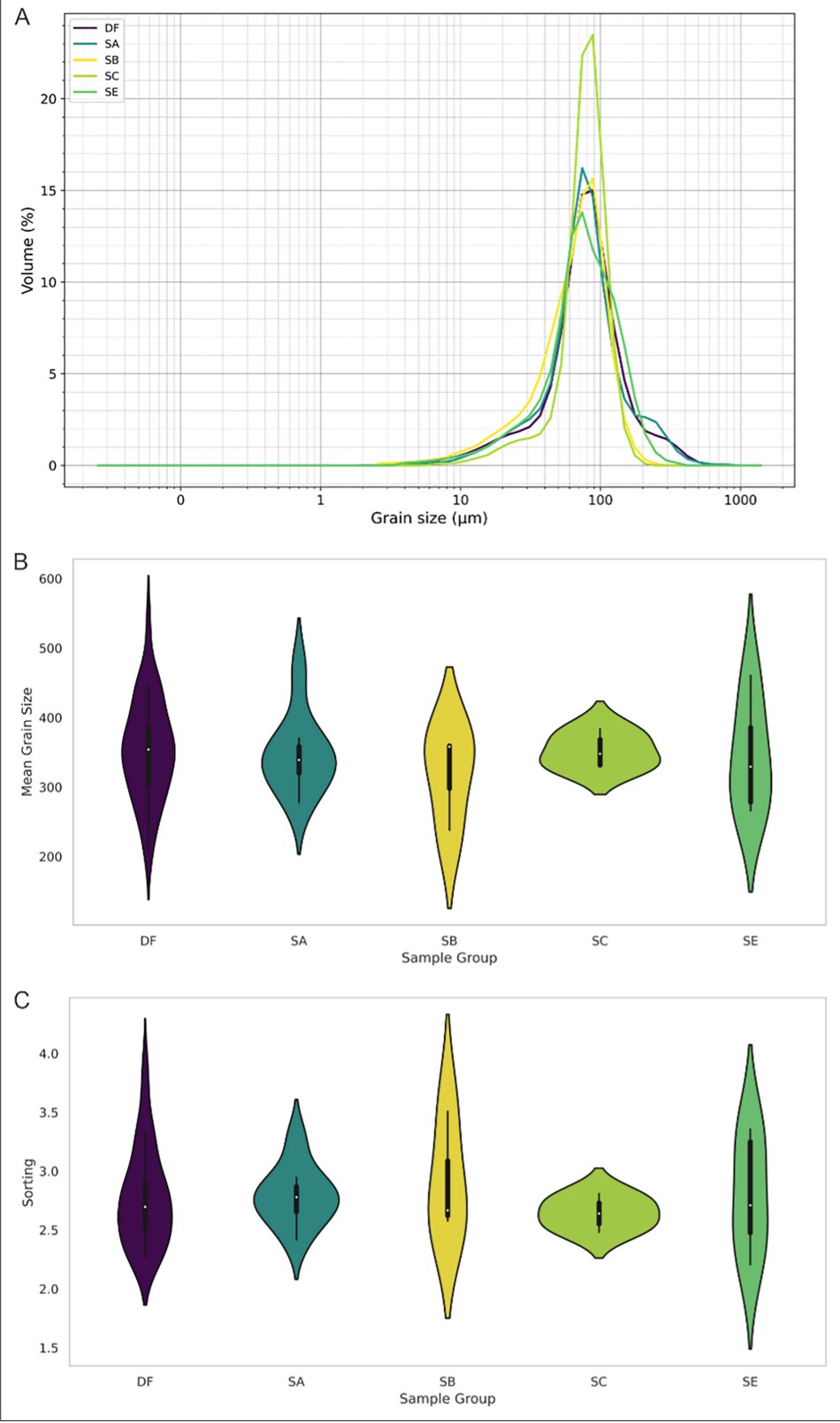

**Figure 16.** Comparative analysis of sediment particle size distributions and sorting characteristics. (**A**) Particle size distribution curves for sediment around Feature 1, showing volume percentage as a function of the mean grain size. (**B**) Violin plots representing the mean grain size in μm for each sample group. (**C**) Violin plots illustrating the sorting of sediments in each sample group. In violin plots, internal box plots show the interquartile range and white dots denote the median sorting value.

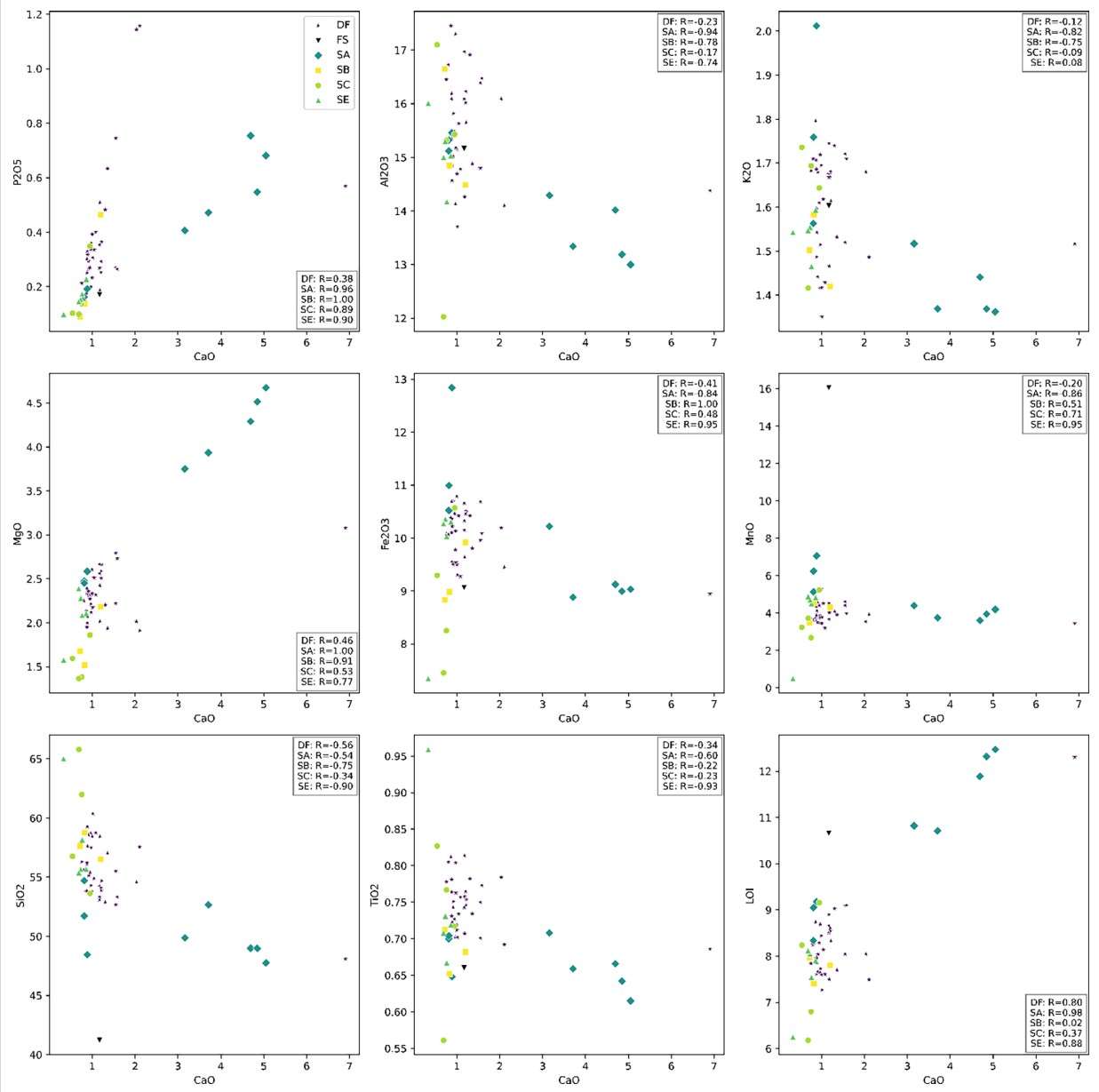

**Figure 17.** Harker variation plots showing the relationship between CaO content (wt.%) and other major oxides (wt.%) and loss on ignition (LOI, wt.%) for the five sediment groups analyzed within and around Feature 1. The Pearson correlation coefficient (**r**) for each sediment group is provided, indicating the strength and direction of the linear relationship between CaO content and the respective major oxide or LOI.

have mean sorting values that fall within a relatively narrow range of 2.64–2.92 (*Figure 16c*). SB has the highest median mean grain size in contrast to having the lowest mean grain size, indicating a skewed distribution towards larger grain sizes in the sediments within the feature. These larger grains presumably derived from LOSMU. DF has the widest range in mean grain sizes and the second highest median mean grain size. However, there are no statistical differences in the median values of the mean grain sizes (H test statistic of 0.67 and p-value of 0.96) and sorting (H test statistic of 1.04 and p-value of 0.90) across all sample groups. Overall, the grain size analysis results confirm that UMCB is structureless and without any resolvable microstratigraphy (*Dirks et al., 2015*; *Dirks et al., 2017*; *Wiersma et al., 2020*). This means that the cut profile of LOSMU is the only available physical indication that sediments around Feature 1 were disturbed and mixed, with SB (sediments within Feature 1) composed of mainly sediments from LOSMU.

## Geochemical context

We further investigated the five sediment groups (DF, SA, SB, SC, and SE) to see if there are geochemical variations between them and if those variations would be consistent with the discernible delimitation of Feature 1. A sample (FS2280) of UMCB from the southern end area without fossils in the Puzzle Box excavation was added for comparison (*Figure 11*). Given that the sediments are not calcified, and calcium is the major component of fossil bones, we examined the major oxides in relation to CaO. This examination revealed two main signatures in how major elements (*Appendix 6—table 2*) vary in the sediment groups when correlated with CaO. The first signature is characterized by positive correlations between CaO and $P_2O_5$, MgO and the loss on ignition (LOI) for all sediment groups, and with some sediment groups for $Fe_2O_3$ and MnO (*Figure 17*). There are strong positive correlations between CaO and $P_2O_5$ for SB ($r$=1), SA ($r$=0.96), SC ($r$=0.89), and SE ($r$=0.90), while for DF, it is a weak correlation ($r$=0.38). The positive correlation between CaO and MgO is strong for SA ($r$=1) and SB ($r$=0.91), and moderately strong for SE, SC, and DF ($r$=0.77, 0.53, and 0.46, respectively; *Figure 17*). SB, SC, and SE are the only sediment groups that exhibit positive correlations with $Fe_2O_3$ and MnO, with these correlations being predominantly strong (*Figure 17*). The strong positive correlations observed between CaO and $P_2O_5$, MgO, and LOI are interpreted as indicating that the presence of these elements is more significantly influenced by fossil bones rather than by the sediments themselves. The second signature is characterized by negative correlations between CaO and $Al_2O_3$, $K_2O$, $SiO_2$, $TiO_2$ for all sediment groups, and also with $Fe_2O_3$ and MnO in specific sediment groups (*Figure 17*). For $Al_2O_3$, $K_2O$, $SiO_2$, and $TiO_2$, DF has weak to moderately strong ($r$=–0.12 to –0.56) negative correlations while SA has moderately strong to strong negative correlations ($r$=–0.54 to –0.94). SB has strong negative correlations ($r$=–0.75 to –0.78) with $Al_2O_3$, $K_2O$, $SiO_2$ (*Figure 17*). The strong negative correlations between CaO and $Al_2O_3$, $K_2O$, and $SiO_2$ highlight that $Al_2O_3$, $K_2O$, and $SiO_2$ are heavily influenced by the clay-rich composition of the sediments, in contrast to CaO, which is significantly influenced by fossil bones.

The trace elements have diverse relationships with CaO across all the sediment groups. SB and SC exhibit similar concentrations of trace elements that often result in close clustering, except in the cases of Zn and Cu where SB is distinctly separated (*Figure 18*). DF and SE display variable trace element concentrations that scatter broadly around SB and SC (e.g. Sr and Pb), but cluster more closely to SB and SC in the case of Zn and Ba (*Figure 18*). SA has similar trace element concentrations to the other sediment groups but has CaO concentrations that are more variable such that SA plots separately and distinctly (*Figure 18*). FS2280 has CaO and trace element concentrations comparable to all five sediment groups, with the notable exception of Zn, Cu, Sr, and Ba where the levels are distinctly higher. The relationships between CaO and rare earth elements (REEs) mirror those observed with trace elements across all sediment groups (*Appendix 6—figure 18*). However, two significant observations are noteworthy: Firstly, SB has the highest concentrations of most REEs. Secondly, the REE concentrations in FS2280 are predominantly similar to those found in SB. A close relationship between CaO and Zn in SB is evident, especially given that SB exhibits higher Zn concentrations than any other sediment group and is the only trace element to show a strong positive correlation ($r$=0.91) with CaO. Examining the concentrations of trace elements relative to Zn reveals that SB plots distinctly apart, while other sediment groups tend to overlap or are mingled for most trace elements, and FS2280 is distinctly separated from sediments around Feature 1 (*Figure 19*). The observation that SB and FS2280 plot distinctly also holds true for REEs when they are examined relative to Zn (*Appendix 6—figure 19*).

We further examined the major and trace elements data using principal component analysis (PCA) to determine if there are statistical variations and patterns in the dataset. The sediment groups are separated into two clusters based on principal component 1 (PC1) of the major oxides (*Figure 20a*). Most SA samples and a single DF sample (DF8) are spread at higher positive values on PC1, whereas the other sediment groups, including three SA samples, are spread in the center of the PC1 axis (*Figure 20a*). Two SC samples (SC1 and 2) and SE3 appear to be spreading away from the larger central cluster towards the negative end of PC1. FS2280 plots between the central cluster and the cluster with higher PC1 positive values, leaning more distinctly towards the latter. In terms of principal component 2 (PC2), there is great variability between all sediment groups along the PC2 axis except the SB group (*Figure 20a*). There is overlap in the range of values along PC2, but these ranges and their distributions are not identical. This suggests that the variability between the sediment groups is

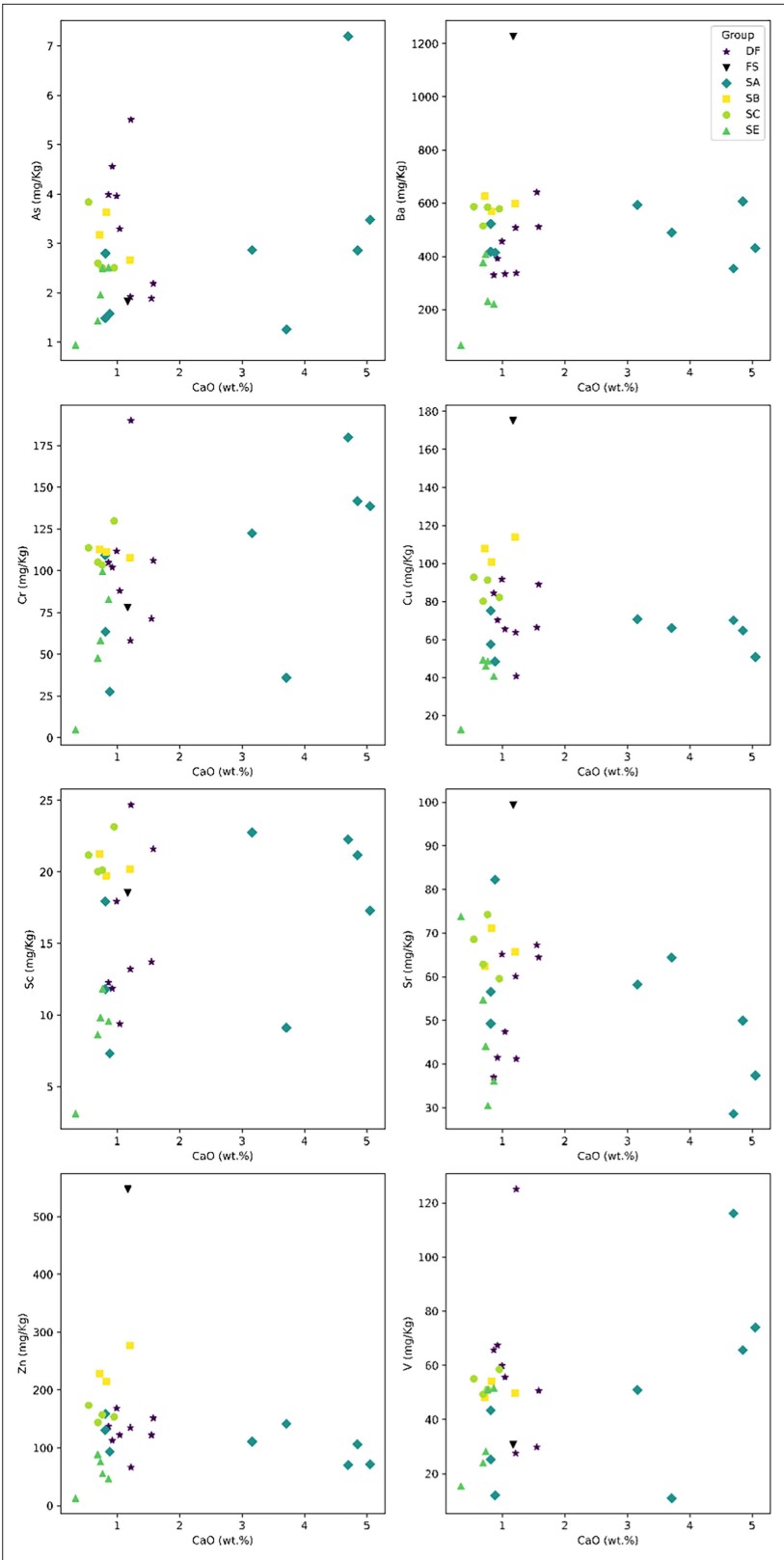

**Figure 18.** Harker variation plots illustrating the relationship between CaO and selected trace elements for the five sediment groups analyzed within and around Feature 1.

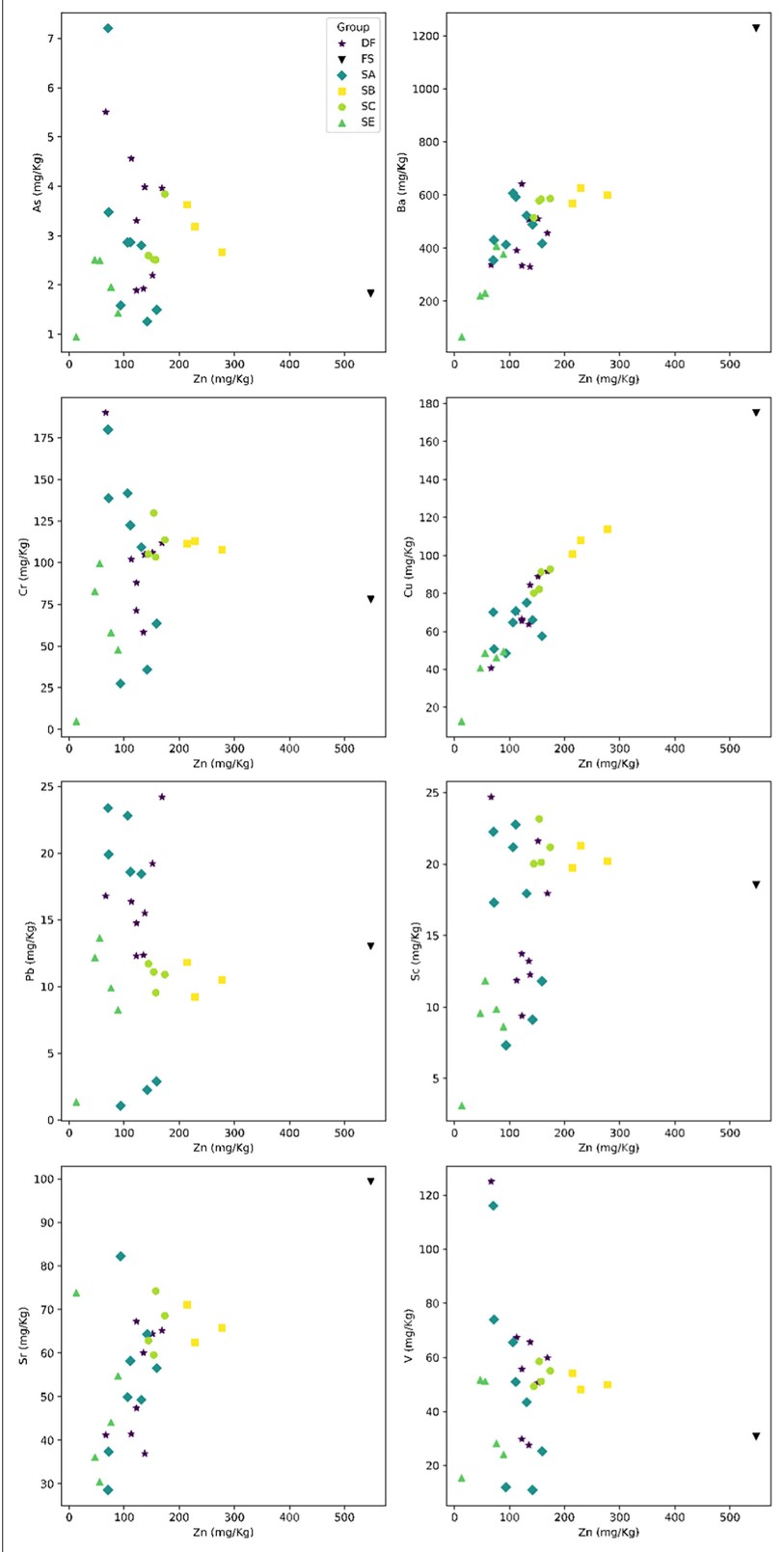

**Figure 19.** Harker variation plots illustrating the relationship between Zn and selected trace elements for the five sediment groups analyzed within and around Feature 1. SB plots distinctly apart from other sediment groups, which tend to overlap or are mingled for most trace elements.

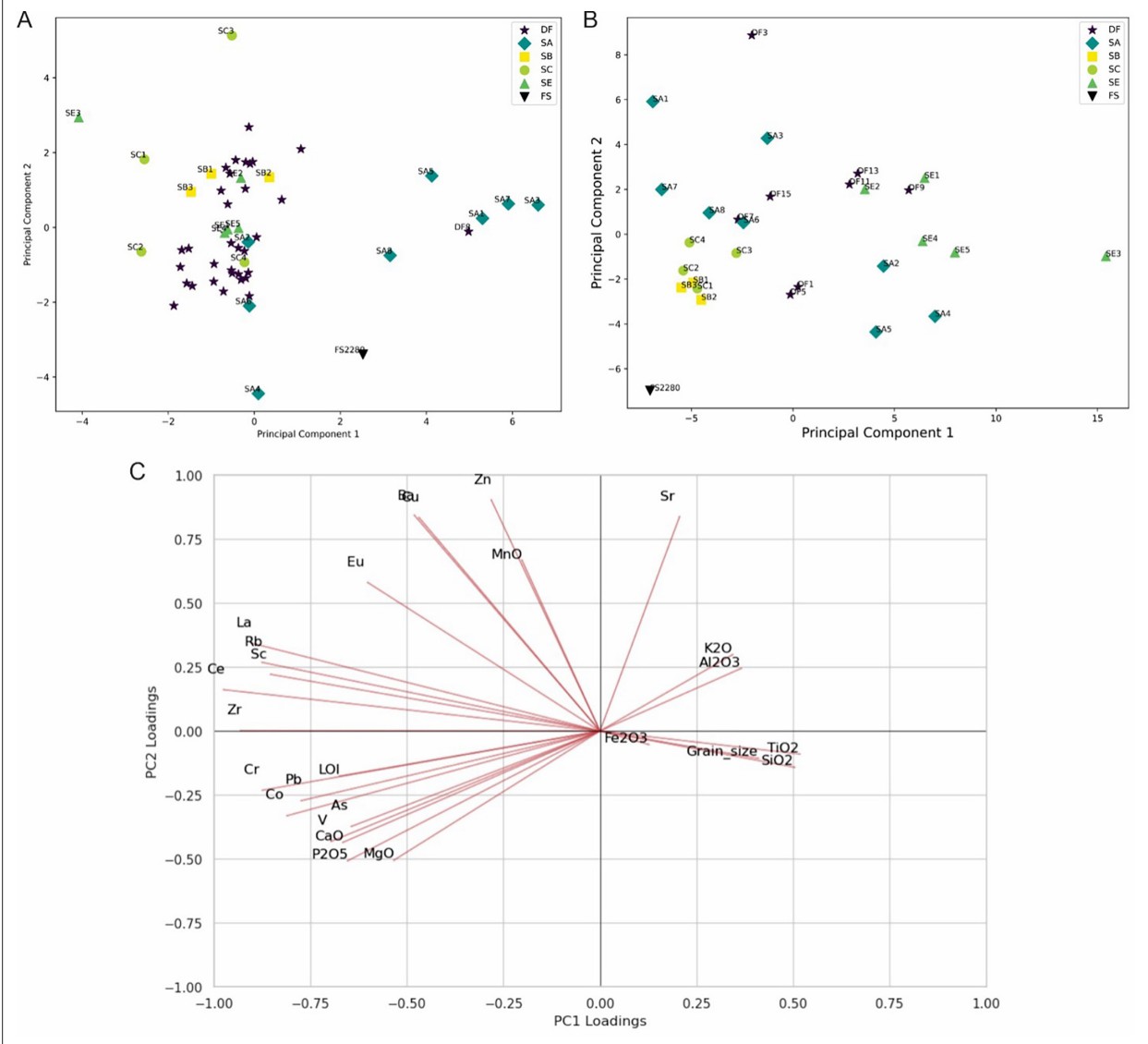

**Figure 20.** Principal component analysis (PCA) of the geochemistry of the five sediment groups analyzed within and around Feature 1. FS2280 is a sample of unlithified mud clast breccia (UMCB) from the Puzzle Box excavation area. (**A**) Scatter plot of PCA showing the distribution of major oxides and loss on ignition (LOI) over the first two principal components (PC1 and PC2). (**B**) Scatter plot of PCA showing the distribution of trace elements and rare earth elements (REEs) over PC1 and PC2. (**C**) Biplot of PCA loadings for major oxides, selected trace elements and REEs, LOI and mean grain size.

significant with respect to the characteristics or variables represented by PC2. DF shows the widest spread along PC2 compared to other sediment groups, but this spread can also be characterized as two distinct clusters (*Figure 20a*). SB samples share similar PC2 values with one of the DF clusters and samples SC1, SE2, and SA5. The FS2280 sample stands out from all sediment groups along PC2, positioning itself towards the negative end of PC2 and closer to an outlier, specifically SA4.

The PCA of trace elements, including REEs, reveals that along PC1, there are two distinct clusters spread on either side of the PC1 axis, with two DF samples plotting on the PC1 axis between these clusters (*Figure 20b*). On the positive side of PC1, all SE samples are spread out, together with three samples each from SA and SE, while the negative side of PC1 comprises all SB and SC samples, joined by FS2280 and the rest of the SA and SE samples (*Figure 20b*). Along PC2, all sediment groups exhibit significant variability, except for SB and SC, which cluster more closely together on the negative side of PC2. A few SA and DF samples, along with most SE samples, also plot on the negative side of PC2 (*Figure 20b*). On the positive side of PC2, there is a spread of the rest of the DF, SA, and SE

samples. FS2280 stands apart from all sediment groups, distinctly positioned at the negative extreme of PC2 (*Figure 20b*). The PCA loadings plot shows that PC1 is strongly influenced by $SiO_2$, $TiO_2$, $Al_2O_3$, $K_2O$, and grain size, which are variables that represent the sediments (*Figure 20c*). For the PC2, the PCA loadings plot shows two orthogonal influences of either trace elements or major oxides. Trace elements such as Zn, Ba, and Cu have positive loadings, whereas major oxides such as CaO, P2O5, and MgO have negative loadings (*Figure 20c*). We interpret the characteristics or variables represented by PC1 as indicative of the fossil bones' influence on the sediment composition, whereas those associated with PC2 are seen as representing processes occurring within the fossil bones themselves, which subsequently affect the composition of the sediments. The PCA distributions suggest that fossil bones impact the geochemistry of the surrounding sediments, with their influence diminishing as the distance from Feature 1 increases. It is meaningful for most SA samples to be separated along PC1 from the other sediment groups more proximal to Feature 1. DF samples that have the same PC2 values like SB may be from the sediment directly above Feature 1, making them more proximal to the fossils like sample SC1. In this manner, the variability seen for FS2280 and the other samples, except SA5 and SE2, becomes consistent with their locations relative to Feature 1.

## Evidence for postdepositional reworking

The skeletal material within this feature was vertically compressed as indicated by the pattern of fracturing and subsequent displacement of fragments. The evidence for vertical compression is visible in several areas of the feature. Examples include an ulna shaft that was fractured with downward deflection of fragments from the high point supported by an underlying radius shaft; rib fragments that are fractured with fragments deflected downward, and a proximal humerus that is crushed in the vertical dimension relative to its orientation in the feature.

At the same time, some material within the feature has been maintained in subvertical orientations by the support of sediment matrix, and overlying elements were supported by underlying elements to some degree as they fragmented. For example, the exposed and fragmented pieces of cranial vault at the south extreme of the feature do not retain the shape of the vault but appear to have been crushed downward. However, the vault fragments with the lowest position in the visible material remain situated with a curve approximating their anatomical position. This suggests that these elements were supported by the matrix during the time that the skeletal material above them compressed downward onto them. Above this vault, we recovered upper limb bones including partial radius and ulna, with fragmented shafts supported by the underlying vault bones. A similar pattern is found in the center of the feature, where we recovered a partial femur and distal humerus fragment, but where other material remains in place. Here, a partial ribcage is visible at the western edge of the skeletal material. Six visible ribs are in anatomical order, highly fragmented, with three of these elements retaining subvertical orientations where they must have been supported by matrix as overlying material compressed downward above the ribcage (*Figure 21*). The identifiable material in the center of the feature includes additional lower limb fragments and a hemi-mandible that crushed downward into the ribs.

These examples of fracturing and crushing are all consistent with downward pressure from overlying sediment and material. All fractures are consistent with postdepositional fracturing subsequent to the decomposition of soft tissue and partial loss of organic integrity of the bone. The exposed portions of the ribcage, in particular, suggest that an intact ribcage was enclosed within sediment and later crushed by pressure from above. The positions of other bone fragments adjacent to matrix-supported ribs within the area of the ribcage can only have been attained by downward movement as the underlying ribcage collapsed. Hence, the major process responsible for the fragmentation and vertical displacement of material in the feature was the collapse that accompanied decomposition and resulting downward gravitational movement of overlying elements and sediment.

Finally, some material within and immediately above the feature was recovered in positions that are not indicative of decompositional collapse of a body. An example is a shaft fragment of immature femur that was recovered in contact with and directly above the large fragment of adult femur shaft in the center of the feature. This element representing a second individual provides evidence of reworking of the deposit. This reworking may have preceded or accompanied the deposition of other material; as for example, if this element was part of the sediment load that came to overlie the remains. A second example of reworking is the left hemi-mandible recovered a short distance above

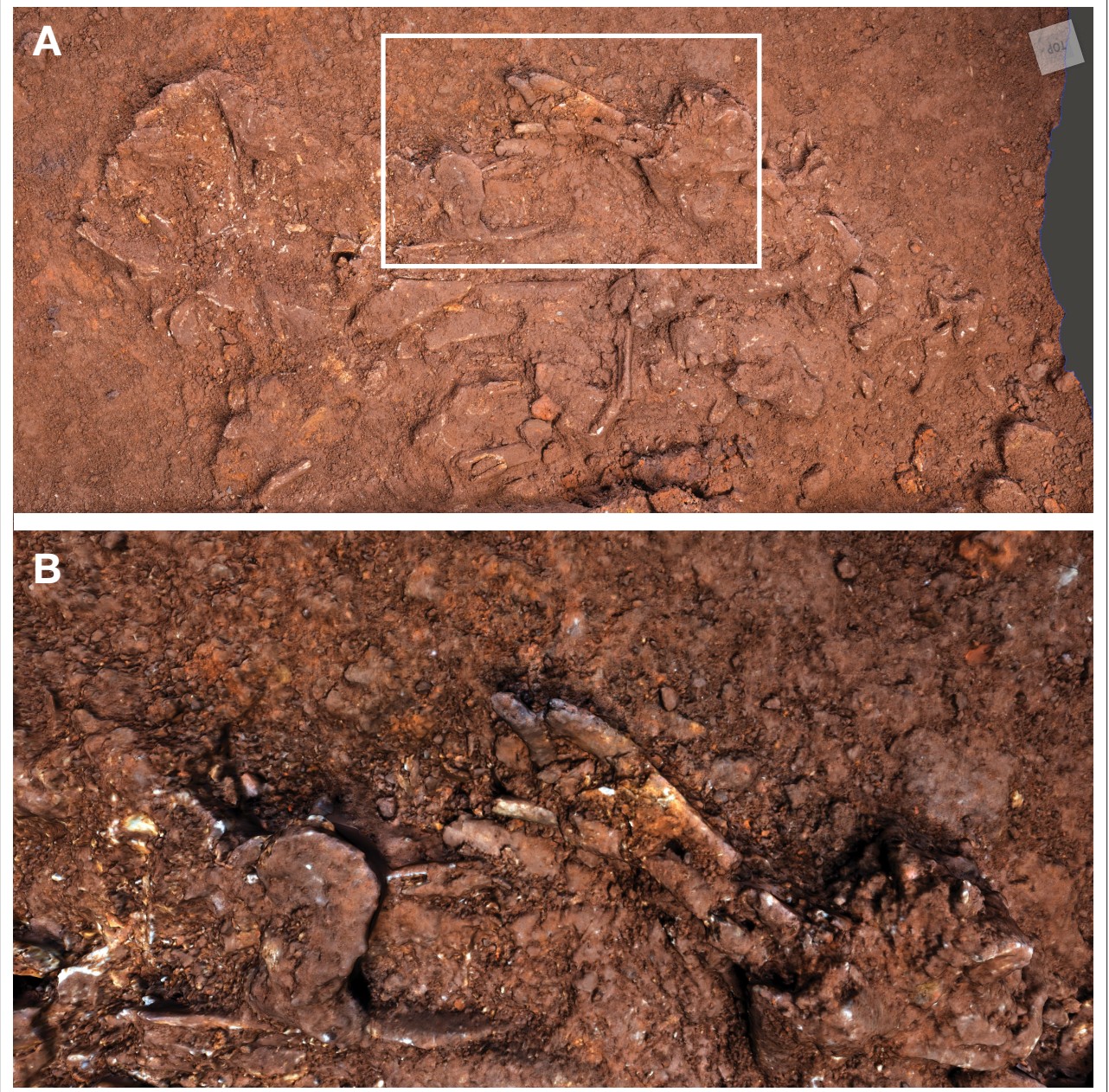

**Figure 21.** Matrix-supported elements within Dinaledi Feature 1. (**A**) Photogrammetric model of unexcavated elements. (**B**) Detail showing ribs including those matrix-supported in subvertical orientation.

the feature. This hemi-mandible is more than 10 cm in horizontal distance from the fragments of right mandibular corpus that are at a lower level within the feature. The fracturing and horizontal displacement of this hemi-mandible is not easily explained by gravitational collapse of underlying material. It suggests that selective postdepositional reworking of the deposit occurred.

## Reanalysis of the Puzzle Box Feature and associated context

Primary excavation of the Puzzle Box Feature (*Figure 22*) was carried out in 2013 and 2014, with some additional regularization of the excavation unit walls and excavation of a small sondage in the center of this area in 2015 and 2016 (*Berger et al., 2015*; *Dirks et al., 2015*; *Dirks et al., 2017*). Several previous studies have presented spatial data on the distribution of *H. naledi* remains within this excavation area (*Dirks et al., 2015*; *Kruger et al., 2016*; *Kruger et al., 2016*; *Bolter et al., 2020*).

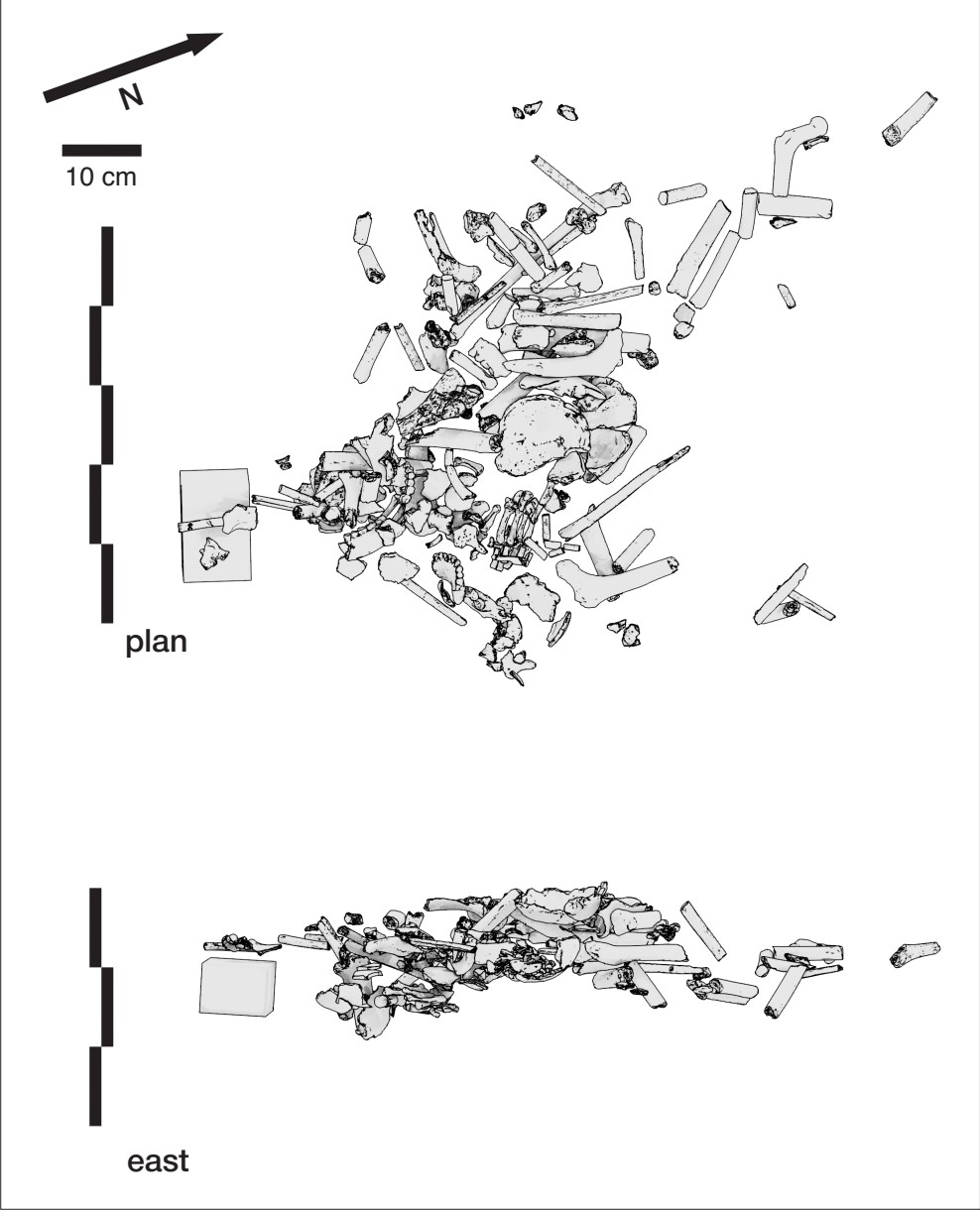

**Figure 22.** Skeletal material from the Puzzle Box area in position. Elements included in these views are greater than 2 cm in their maximum dimension; isolated teeth and smaller fragments are not included. The rectangular box at left represents the location of a partial infant skeleton, discussed in text. Top: plan view. Bottom: View from east side of excavation area.

Due to the limited area of excavation, it was not possible at the time of these studies to determine whether this excavation area had the same or different distribution of skeletal remains compared to other areas of the sediments in the chamber. The excavation results from 2018 and 2019 established that some areas adjacent to or near the Puzzle Box area are very different in subsurface presence of fossil material. The Dinaledi Chamber floor is not a homogeneous bone bed but instead is host to multiple concentrated features of skeletal remains (*Kruger, 2017*). The Puzzle Box excavation area is the largest and most complex area yet explored. The distribution of material within this area has new relevance in light of the new excavation results that reveal the subsurface composition of other areas. Here, we briefly summarize previous work and focus on new data and aspects that are relevant in comparison with other newly excavated areas.

## Skeletal material

The subsurface collection from this area includes skeletal remains of at least six individuals of *H. naledi*. No macrofaunal remains are present in this area. The anatomical characteristics of most elements have been described previously, with associations where these were supported by direct refitting or articulation. Work to associate skeletal remains with each other to reconstruct individuals has continued and remains ongoing (following *Huber, 2011*). Individuation has been based on multiple lines of evidence including the developmental stage of elements (*Bolter et al., 2018*; *Bolter et al., 2020*; *Delezene et al., 2023*), the ongoing program of direct refitting, and analysis of spatial proximity (*Kruger, 2017*; *Bolter et al., 2020*). This work provides substantial additional information about the process of deposition of the remains and subsequent processes of decomposition and reworking of material. Here, we focus on aspects of the preservation, individuation, and spatial distribution of remains that are relevant to processes of deposition and reworking.

High-resolution spatial recording methods enable the reconstruction and analysis of the spatial distribution of skeletal remains; orientations of individual fragments, and associations among anatomically contiguous parts (*Figure 23*). Skeletal elements within this excavation unit exhibit three-dimensional spatial clustering without any sign of layering, calculated using a 3D density-based clustering algorithm (*Hahsler et al., 2019*; *Ngoloyi et al., 2020*; *Figure 24*). Elements occur primarily at lower plunge angles but with a wider range of angles compared to elements mapped from across the Dinaledi Chamber's surface. Some elements occur at steep angles within the deposit (*Figure 24b*; *McPherron, 2018*). Elements are oriented across the full range of compass bearings with only a very small bias towards the N-S orientation across the sample (*Figure 24c*). These spatial characteristics constrain the possible processes of deposition and subsequent reworking to those that do not result in sorting or flow biases. Elements with steep plunge angles were supported by the matrix after arrival into these positions, possibly with some subsequent settling or compression of sediment over time.

Many of the skeletal elements were in anatomical articulation at the time of excavation. Some of the articulated structures have been described in previous work (*Dirks et al., 2015*; *Kivell et al., 2015*; *Harcourt-Smith et al., 2015*; *Kruger et al., 2016*; *Williams et al., 2017*). Previously described articulated material includes a near-complete hand in flexed articulation; a partial foot, an ankle and partial foot with tibia and fibula, two thoracic vertebrae with rib, and some sets of dental elements recovered in occlusal position and order but without remaining bone anchoring them. Articulated elements occur at all levels of the deposit from the surface to 25 cm depth.

The epiphyses of long bones are not well preserved in this area, and most long bone articular surfaces are absent, with few exceptions. The lack of evidence of corresponding articular surfaces impedes the recognition of elements that may be derived from the same individual. More rapid progress has been possible for immature individuals. *Bolter et al., 2020* associated elements of an immature individual based on evidence of developmental stage and spatial distribution of remains, which they designated as DH7 (*Figure 25*). This individual is represented by upper limb, lower limb, thoracic and mandibular elements as well as dentition. Elements attributed to this individual exhibit anatomical ordering within part of the Puzzle Box area. Fragments of the right ischium, right femur, and near-complete lower leg elements including the tibia, fibula, and articulated ankle are in anatomical order within a 40 cm space. The right tibia is broken with cleanly conjoining pieces recovered at different depths and displaced 20 cm from each other. Hand elements, a partial mandible, and many isolated teeth of this individual occur within 30–35 cm from the lower limb at a similar depth (*Bolter et al., 2020*). Some elements of this skeleton are in or near articulation, while others are fractured with closely spatially displaced fragments that refit cleanly.

Near the southern edge of the Puzzle Box area, a partial skeleton of an individual of infant developmental stage was recovered, which is under study. The skeleton includes a partial skull and partial mandible with near-complete dentition, most of the vertebral column, rib fragments, and other postcranial elements. Most of the elements of the vertebral column of this individual are represented, with vertebral bodies absent but most of the unfused laminae present. The partial cranium was excavated *en bloc* and prepared in the laboratory, while the vertebral elements were recovered by bulk collection of sediment collected from a single small area followed by sieving. The data show that this partial skeleton was localized within a diameter of 20 cm across a depth of between 12 and 20 cm, in contact with elements of the DH1 cranium.

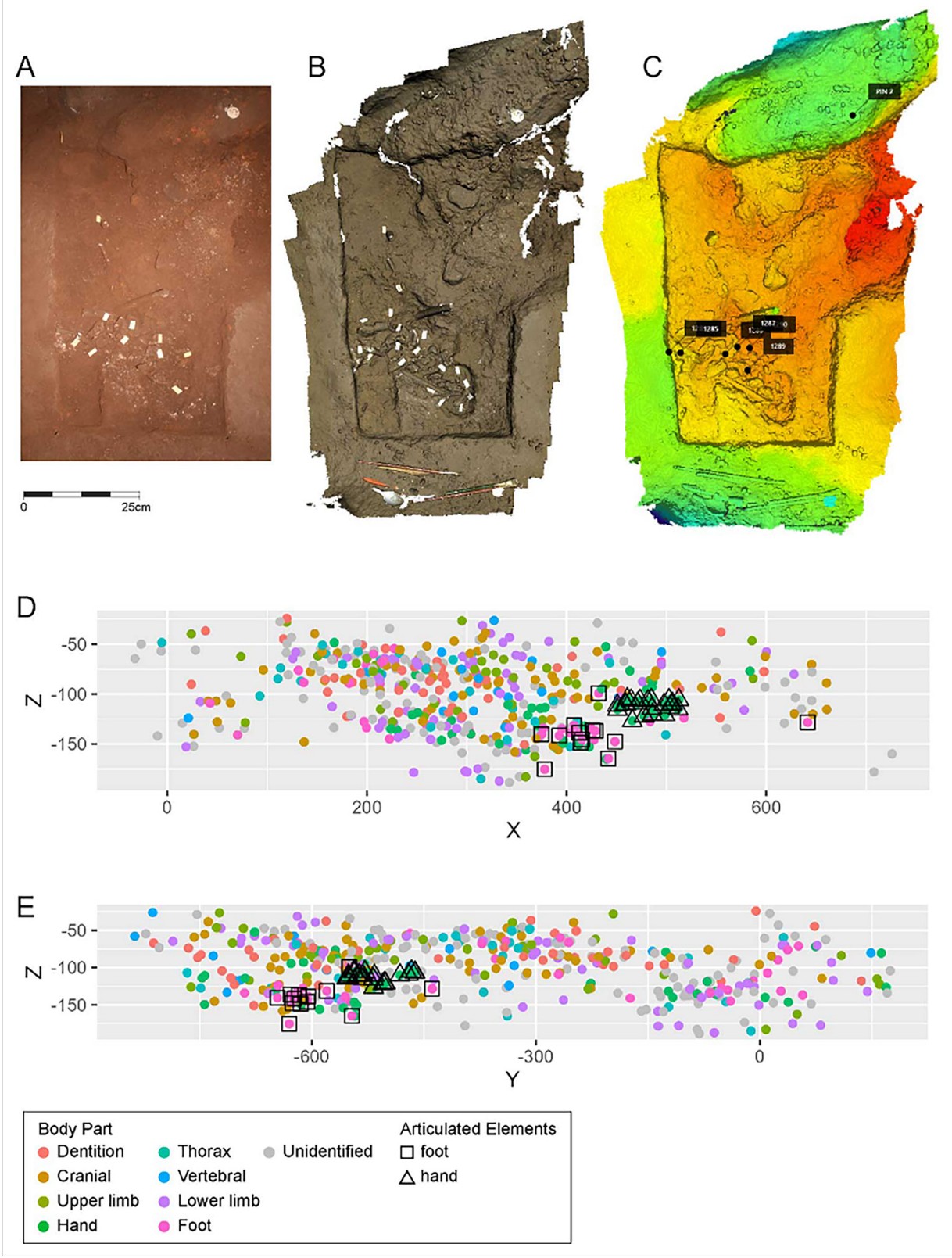

**Figure 23.** Spatial data on element positions within Puzzle Box area. (**A**) Photo taken during excavation. (**B**) The combination of white-light surface scans and high-resolution laser scans representing the same excavation stage as photo in A. (**C**) Elevation data for elements and surrounding context. (**D**) Three-dimensional coordinates of identified elements in plane with X and Z dimensions. (**E**) Three-dimensional coordinates of identified elements in plane with Y and Z coordinates. Scale in mm.

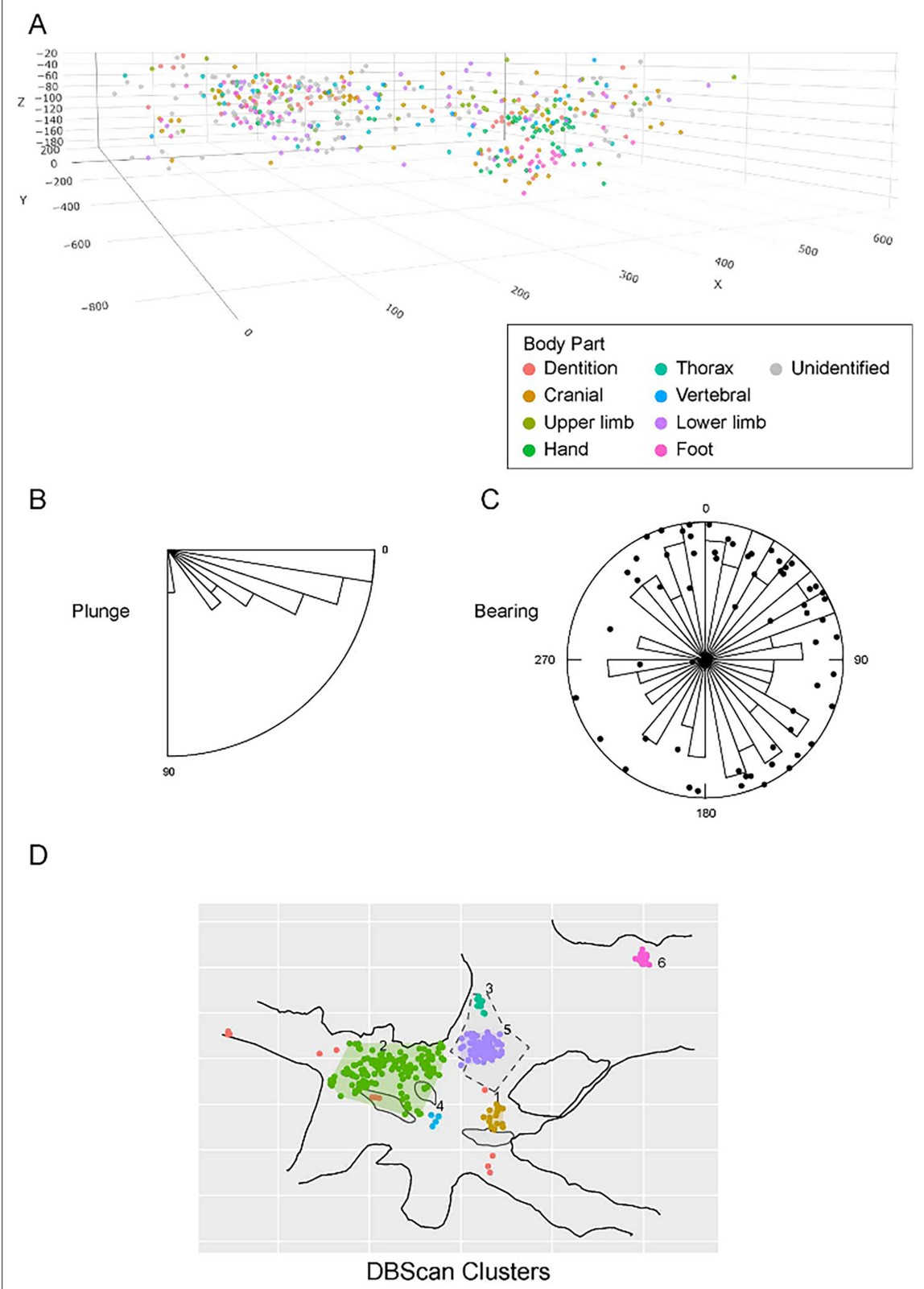

**Figure 24.** PLACEHOLDER Spatial orientation and clustering of elements from the Puzzle Box area. (**A**) Spatial locations of elements excavated from the Puzzle Box area in 2013 and 2014 in oblique 2.5D view. For larger specimens mapped with two end points (n=79), we calculate (**B**) their plunge angle and (**C**) planar orientation frequencies (*16*). (**D**) 3D density-based cluster analysis of fossil material collected on the surface of the Dinaledi Chamber together with excavated remains. A single high-density cluster comprises the majority of excavated fossils (green) with two smaller peripheral clusters (purple, gold) and outliers (red points).

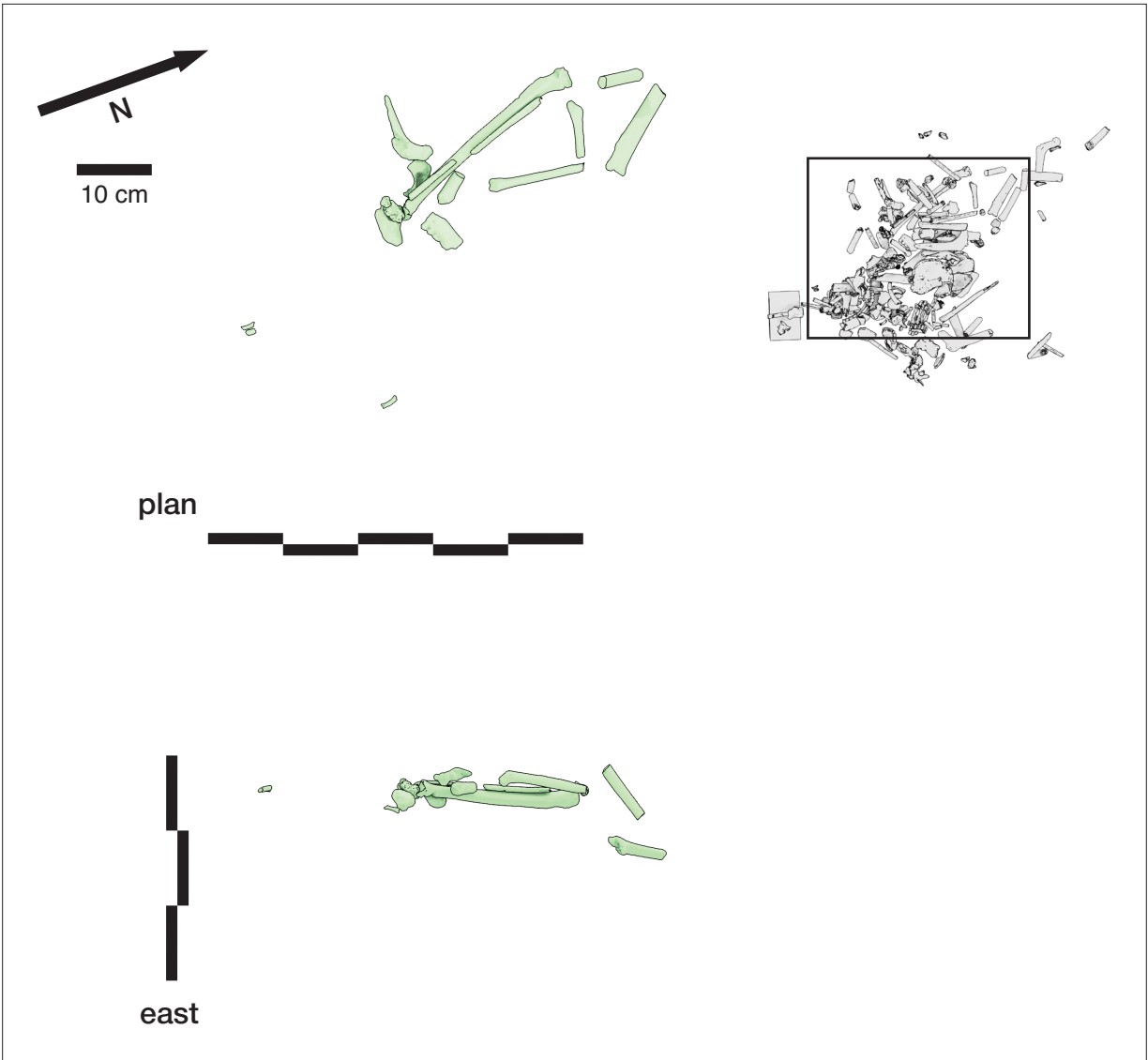

**Figure 25.** Spatial positions of elements attributed to the DH7 skeleton. Most recognized elements are localized within the western part of the Puzzle Box area, particularly ischium and lower limb elements, with a mandibular fragment and humerus fragments all within 10 cm. The more distant elements are hand bones.

In addition to this infant, four crania of older individuals also occur within the Puzzle Box area. These four crania have been designated as DH1, DH2, DH3, and DH4 (*Berger et al., 2015*; *Laird et al., 2017*). Each of these four crania was recovered as multiple fragments with displaced fractures that refit cleanly. The separated fragments of each skull that occur within the feature are spatially concentrated (*Figure 26*). Two of the skulls, DH2 and DH3, are located within a few centimeters of the floor surface. The DH3 skull has teeth and small refitting fragments that were recovered on the surface of the Puzzle Box area, as well as refitting fragments collected on the surface up to 1.5 m southwest of this area. The DH2 skull also has one conjoining fragment recovered on the surface 1.5 m southwest of the Puzzle Box. No association has yet been established between this skull and dental material. The preserved portions of the DH1 and DH4 crania were excavated at depths of 15–25 cm below surface, within the eastern part of this excavation area. Fragments of each of these two skulls are separated by both horizontal and vertical distances of several centimeters. The preservation of vault fragments is very good, with many of them conjoining along fractures that preserve matching edges, despite being displaced from each other in the deposit. However, basicranial portions are not well preserved,

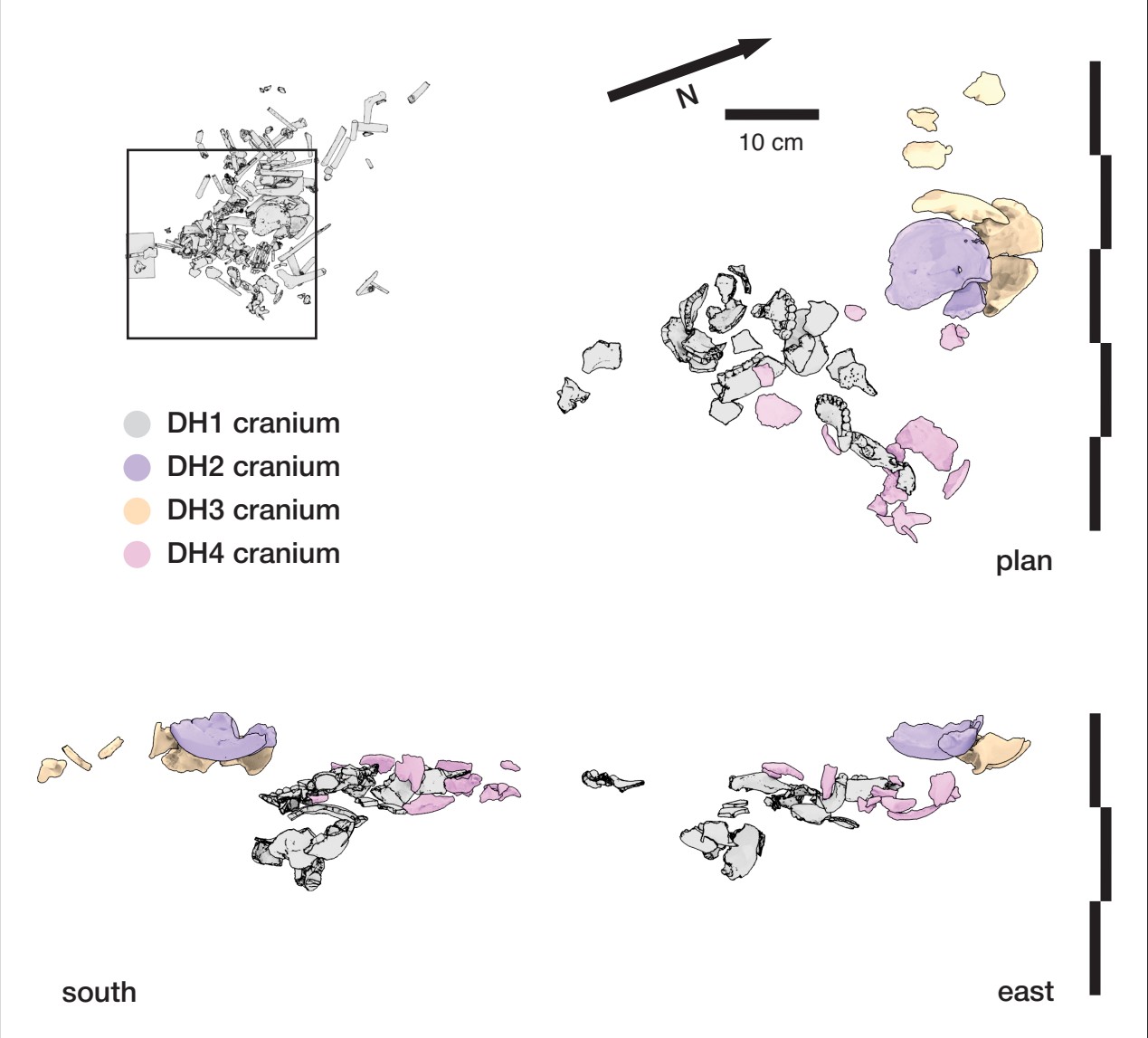

**Figure 26.** Spatial positions of elements attributed to the DH1, DH2, DH3, and DH4 crania within the Puzzle Box area. An outline of the DH2 vault outcropped upon the surface at the time of excavation, and the portions preserved from the feature were enclosed in sediment. This partial skull was underlain by the left half of the DH3 cranium, with the left hemimandible in contact with it and with other fragments located within 20 cm at a similar depth. Fragments of the DH4 cranium were recovered at greater depth, localized toward the east side of the area, with pieces of the left frontal approximately 15 cm from a concentration with most other fragments. The DH1 cranium is the most complete but was highly fragmented at the time of discovery, with many refitting parts displaced across 35 cm and a range of depths, including the occipital and temporal fragments in the deepest part of the deposit.

and only DH1 has any preservation of the maxilla. Most of the dentition of the DH1 cranium has been recovered, with associations based on the in-situ teeth of the mandible and left maxilla (*Delezene et al., 2023*); the DH4 cranium has not yet been associated with dental material.

It is not yet possible to associate postcranial and cranial elements from single individuals except for the two immature partial skeletons discussed above. It is probable that articulated postcranial structures come from individuals that are also represented by other postcranial, cranial, and dental elements that may have been displaced by reworking. This suggestion is supported by the refitting of fragments of crania and postcranial elements, which include examples that illustrate the reworking described below.

## Sedimentary context

As described for Dinaledi Feature 1, the Puzzle Box area includes *H. naledi* fossil material embedded within Unit 3 sediment. The Puzzle Box excavation area is the basis for the unit description by *Dirks et al., 2015*, and that description includes photos, images of thin sections, scanning electron microscope images of microsections, and electron microprobe point and ID geochemistry. The sediment in this area contains clasts of orange Unit 1 material, and as in other situations in Unit 3 throughout the subsystem, these clasts retained angular outlines, with laminations at random orientations, and are consistent with breakdown in dry cave conditions (*Dirks et al., 2015*; *Wiersma et al., 2020*). No layering or stratigraphic ordering of the sediments within this excavation area was noted. Clasts were noted in contact with *H. naledi* skeletal elements. Small voids were also noted within the top 25 cm of this deposit during its excavation (*Dirks et al., 2015*).

*Homo naledi* skeletal material has been found only within the upper 25 cm of the deposit in this location (*Figure 22*; *Dirks et al., 2015*). Sediment of sub-unit 3b that underlies the skeletal material is consistent in composition as observed in the sondage at least as deep as 80 cm below surface.

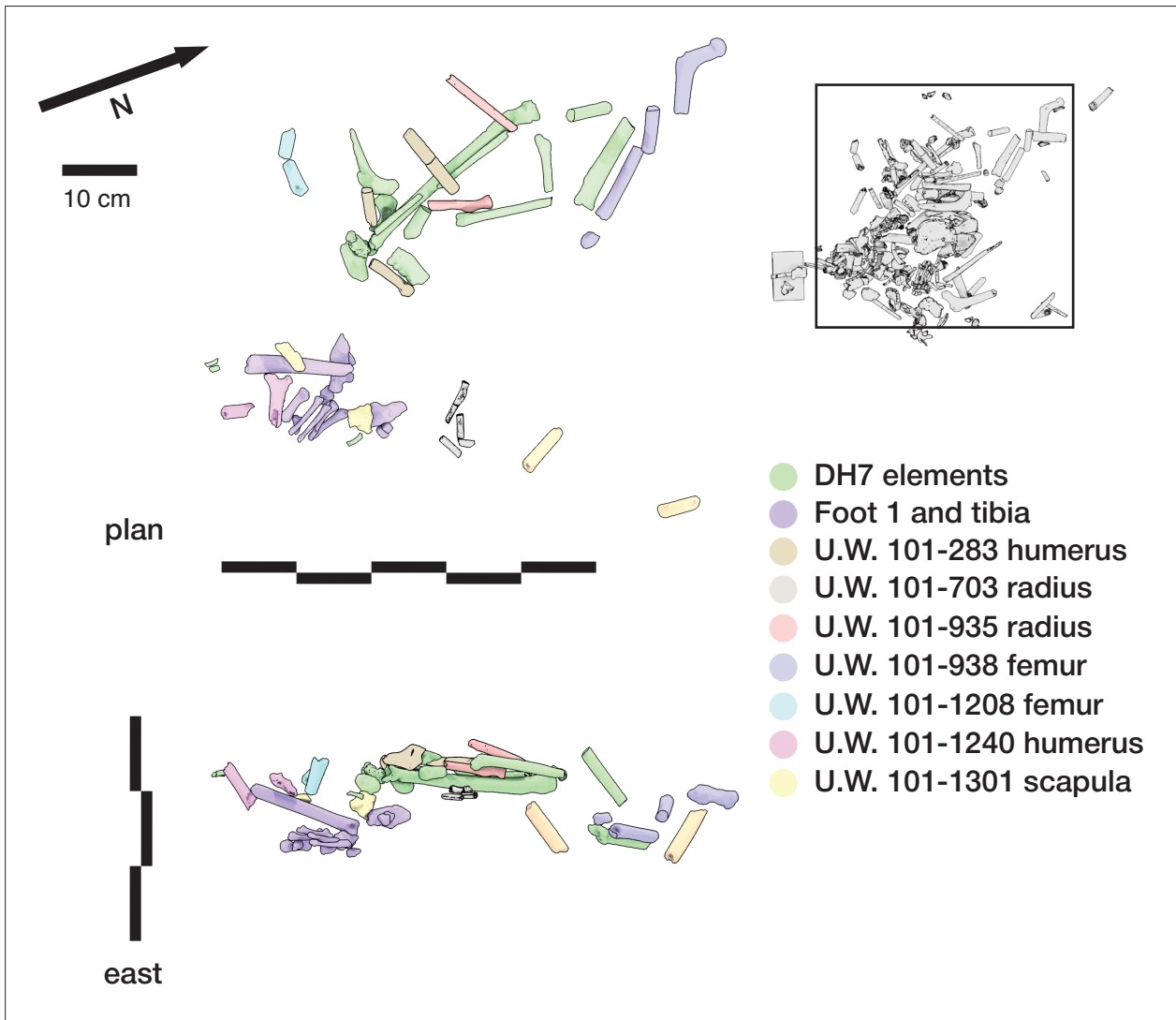

**Figure 27.** Postcranial elements for which conjoining fragments are spatially displaced from each other in the Puzzle Box area. The processes leading to displacement of these fragments did not result in significant loss of detail of fracture surfaces, enabling direct refitting to match them. The processes causing movement of these fragments after fracturing were capable of changing depth, compass bearing, and plunge angle of fragments, including those not supported by contact with other underlying fragments. However, those processes did not move any of these fragments more than 15 cm apart.

Sedimentological and geochronological evidence shows that this sediment was in position from at least 635±148 ka (*Dirks et al., 2017*), which precedes the period that *H. naledi* remains were deposited, encompasses the known range of ages of *H. naledi* material, and the period afterward.

## Evidence for postdepositional reworking

Multiple processes led to the fragmentation and spatial displacement of elements from the Puzzle Box area. The spatial situation of particular elements and anatomical structures illustrates these processes. Some parts of the Puzzle Box area underwent minimal or no reworking after the primary deposition of fleshed remains. For example, Dinaledi Hand 1 (H1) is a nearly complete right hand, found at a depth of 170 mm below the cave floor surface, with the palmar surface facing upwards (*Kivell et al., 2015*; *Kruger et al., 2016*; *Kruger, 2017*). H1 was found almost completely articulated, and the position of the reconstructed hand suggests that it was flexed (closed) or semi-closed during the process of skeletonization. Dinaledi Foot 1 (F1) is a semi-articulated adult right foot found in an inverted position (i.e. resting upon the dorsum) with anatomical disassociation having taken place within the broad pedal structure. This foot exhibits selective reworking, including fracture of the associated tibia and lateral displacement of the calcaneus. The articulated left lower limb attributed to the DH7 skeleton includes articulated tibia, fibula, talus, and navicular (*Bolter et al., 2020*). The right ischium and proximal femur attributed to DH7 were in near-articulation suggestive of the loss of connective tissue and in situ collapse of elements following decomposition.

Immediately in contact with such intact elements are highly fragmented remains. Direct refitting of fragments has begun to reveal the processes of reworking that the entire assemblage underwent after the primary deposition of bodies. The refitting program has identified many conjoining fragments, particularly long bone shaft fragments, that comprise in some cases nearly complete elements. To date, this program has taken a conservative approach matching only fragments along fracture edges that clearly refit without substantial erosion or loss of relief of the fracture surface. A number of these refitting fragments were recovered in spatially separated positions and at varied orientations (*Figure 27*). Some portions of the DH7 skeleton are included in this number, with the right humerus and right tibia each composed of conjoining fragments that were excavated in spatially displaced positions. Other postcranial elements have similar recovery situations. Many conjoining fragments are vertically separated, differing by as much as 10 cm in depth within the area. Fragments do not retain the same orientation; conjoining fragments differ in both compass bearing and in plunge angle from each other. Yet in all cases where conjoining fragments have been identified from the Puzzle Box area, these fragments were localized at the time of excavation within a 15 cm distance from each other. Two exceptions are postcranial elements for which some fragments were excavated from the Puzzle Box area but at least one fragment was recovered from the surface, in one case at a distance of approximately 1.5 m, and in the other at a distance of 4 m. Conjoining fragments that were excavated and not collected on the surface do not exhibit substantial displacement.

A similar observation holds for the cranial material discussed above. Each of the five crania within the Puzzle Box area is spatially localized despite their high degree of fragmentation. Many conjoining elements fit together without substantial erosion or loss of surface relief of fractures. The fragments that constitute each cranium were localized at the time of excavation within 30 cm diameters; the only exceptions are the surface-collected fragments of DH2 and DH3. The cranial and postcranial remains are highly localized spatially within the deposit, which is also reflected by the cluster analysis of spatial coordinate data.

Despite this low degree of spatial displacement of remains, there are several cases of striking juxtaposition of different elements that mismatch in anatomical placement, fragmentation, and plunge angle. For example, occipital and parietal fragments of the DH1 cranium are among the deepest skeletal remains within the Puzzle Box area. Several fragments were matrix-supported at angles above horizontal, and fragments of the occipital bone, mastoid region of the temporal bones, and adjacent portions of the parietal bones were recovered in a configuration that roughly matches a bowl in shape, although with displacement that indicates crushing into a smaller volume than the undistorted skull would occupy. In the center of this cluster of cranial bones was excavated the U.W. 101–1475 proximal femur fragment, with shaft near vertical (*Figure 28*). Other parts of the DH1 cranium, such as the remainder of the parietals and portions of the frontal bone, are displaced laterally above this bone cluster in both north and south directions; none directly overlie the vertically emplaced femur

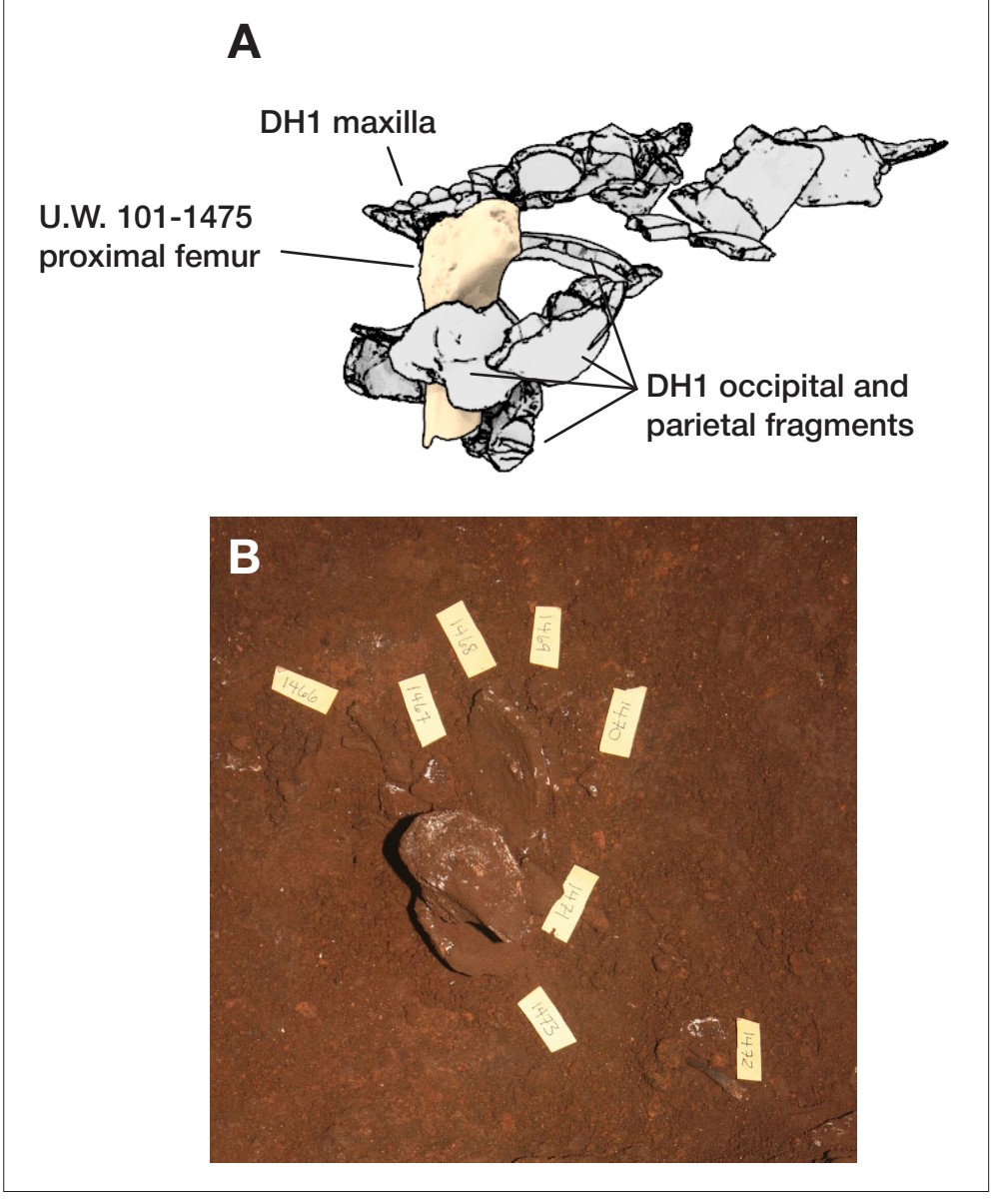

**Figure 28.** Context of DH1 fragments with U.W.101–1475 femur. The DH1 occipital and parietal bones are fragmented into pieces that vary from ~1 cm to >6 cm in diameter. (**A**) Diagram showing all DH1 fragments with U.W. 101–1475 from south direction. U.W. 101–1475 is a proximal femur emplaced vertically in the deposit, with its distal end in contact with multiple fragments of DH1 occipital and parietal bones. (**B**) Photograph at time of excavation of the deepest DH1 fragments showing contact and relative positioning of these fragments with U.W. 101–1475.

fragment. The vertical emplacement of the femur and orientation of these DH1 fragments were maintained by sediment support throughout the period after burial in this position. The collapse of overlying sediment into a void created by decomposition of the cranial contents could not by itself create this configuration, both due to the overlapping of fragments comprising the base of the bowl-shaped cluster and the horizontal displacement of other parts of the cranium.

The direct contact between elements with strongly different plunge angles is a notable aspect of the Puzzle Box area, which in part inspired the name. For example, the U.W. 101–1284 femur shaft sits above and near the area with DH1 cranial fragments, at an angle of approximately 50 degrees from horizontal. Touching it or in its immediate surroundings are many long bone shaft fragments, a partial

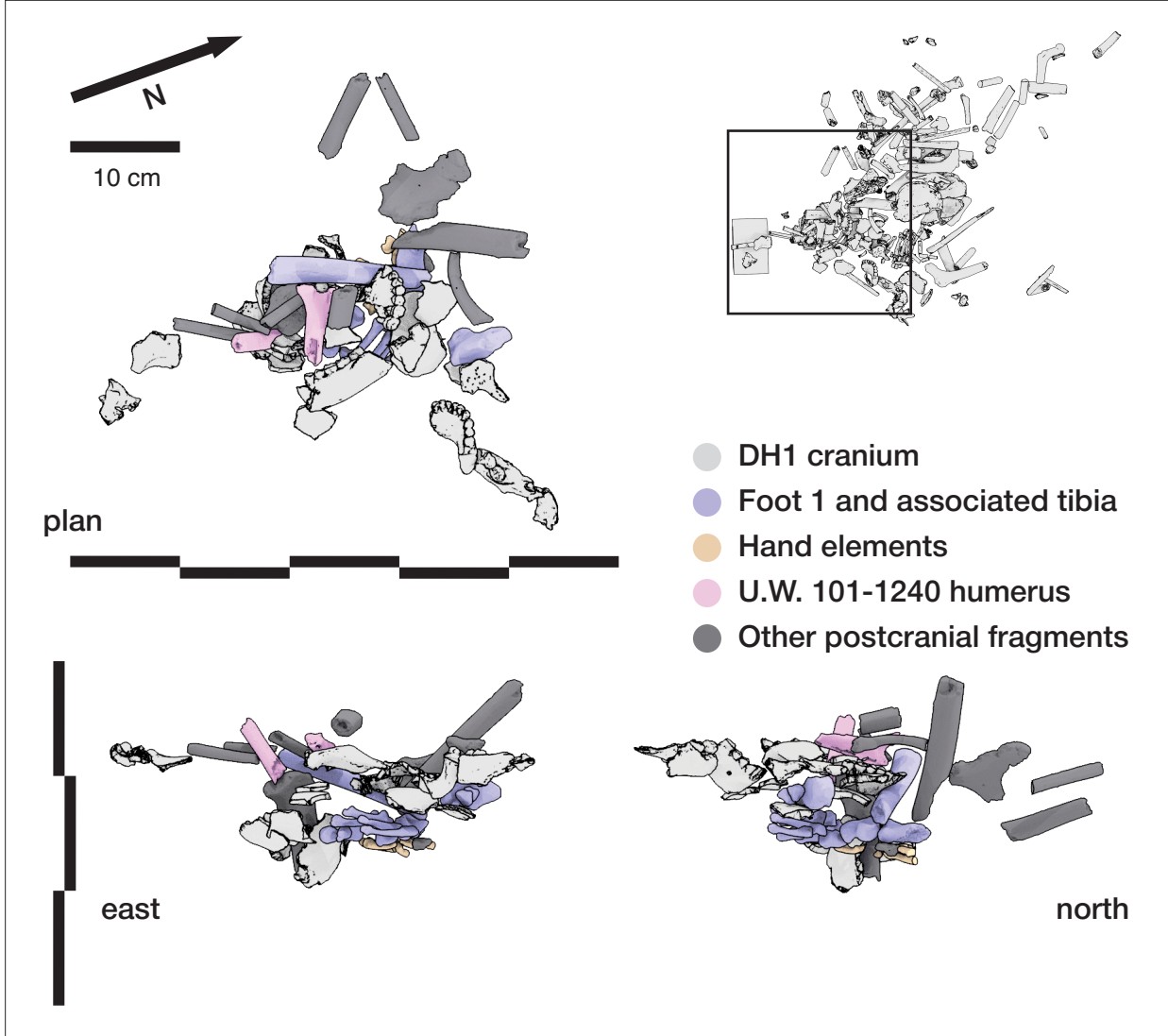

**Figure 29.** Postcranial shaft fragments in the vicinity of DH1 cranial fragments and articulated Foot 1. The range of plunge angles of these fragments could not have been maintained without sediment support at the time they attained these positions.

clavicle, and a partial ilium all at ranging in angles from horizontal to an angle of 70 degrees in the opposite direction, as well as the vertically emplaced U.W. 101–1475 femur fragment discussed above (*Figure 29*). The orientations of these fragments must have been supported by sediment at the time they arrived in this configuration.

Some skeletal structures were excavated in articular position, with little or no displacement of fragments from their situation in the body. These could not have been moved substantially or reworked after they were covered and supported by sediment in the positions where they were excavated. Yet in at least three cases, articulated structures occur within 2 cm of skeletal elements that are fragmented and displaced from conjoining fragments. These cases occur in different parts of the Puzzle Box area, at different depths. Adjacent to the DH1 cranial remains and immediately beneath the array of disordered long bone fragments discussed above is the articulated partial Foot 1 (*Figure 29*). A skull was broken into pieces, a femur shaft vertically emplaced within the broken fragments, and a jumble of other long bone fragments arrived immediately above these structures without disturbing some parts of the articulated foot located less than 5 cm away. A few elements of this otherwise articulated foot were displaced by 10 cm toward the northeast and 3–5 cm toward the surface, including the calcaneus, intermediate cuneiform, and lateral cuneiform. Two fragments of the distal tibia including the articular surface were recovered in close contact with the talus, suggesting that this foot arrived in

this location while still attached to the lower leg; the nearby tibia shaft fragment U.W. 101–1288 refits these fragments but is displaced at an extreme angle from them. All of these aspects of the distribution of skeletal material in this part of the Puzzle Box reflect localized and selective reworking after the deposition of the foot, and likely after the primary deposition of the DH1 cranium.

## Relationship of features and surroundings

An additional category of evidence is the spatial relationship of the various features and other evidence within the subsystem. Some hypotheses for the formation of the deposits presuppose the possibility of reworking or gravity-driven movement of bodies, bones, and sediment from one location within the subsystem to another. Our surveys and observations within the subsystem quantify the distances and pathways that constrain any possible movement of bodies or skeletal remains.

### Discreteness of features

The features described here, including Dinaledi Feature 1, the Puzzle Box area, and the Hill Antechamber Feature, are not part of a continuous bone bed. Neither are they localized areas that happen to have high subsurface bone density within a floor that has bone at some density throughout. The excavation work described here shows that some areas of the sediment floor are devoid of subsurface hominin bone elements beneath the surface at depths where other areas have large bone concentrations. Some of the areas lacking significant skeletal material are directly adjacent to areas with high concentration of *H. naledi* skeletal remains.

The two areas in the Dinaledi Chamber are within 1 m of each other and share most aspects of their sedimentary and geological context. However, the characteristics of these two bone assemblages are very different. Dinaledi Feature 1 is a bounded oval shape and exhibits very little evidence of commingling of material from multiple individuals. Only three fragments superficial to the feature clearly are bones of a second individual. In contrast, the Puzzle Box area presents a complex subsurface assemblage with clear spatial concentrations of skeletal material. Each area of concentration presents some articulated body parts, and skeletal material attributable to different individuals is localized in different concentrations. The Hill Antechamber Feature is at least 15 m in horizontal distance from these features and 4 m higher in vertical profile. In its complexity, this feature is intermediate between the two Dinaledi Chamber features. Remains of multiple individuals are present within the Hill Antechamber Feature, again in different spatial portions of the feature, but with a greater degree of anatomical contiguity of the hypothesized principal individual.

Within all the features, bone is characterized by postdepositional fracturing, some postdepositional displacement of fragments, and the introduction of sedimentary clasts within concentrations of skeletal remains. Adjacent areas with no subsurface bone are different from this. In the area of the Dinaledi Chamber adjacent both to the Puzzle Box and Feature 1, the horizontal LOSMU extends horizontally beneath the surface. In the Hill Antechamber, the subsurface adjacent to the Hill Antechamber Feature has layering of LORM-rich and LORM-poor sediment. While the excavations cover only a limited extent, the data are sufficient to establish that the features described here are discrete and bounded. They are not random spatial concentrations within sediments that have bone throughout their volume.

At the same time, other excavated units outside the features had bone fragments or teeth present at much lower density. The presence of *H. naledi* fragments outside of the highly concentrated features suggests that postdepositional processes dispersed small elements, fragments, and teeth from their primary context. By far, the largest quantity of material outside of features was represented upon the surface of the chamber floor. The subsystem had more than 300 elements or fragments of *H. naledi* bone and teeth on the floor surface at the time excavations began. The extensive surface presence of material has been noted in previous work, and much of this material was collected and described. Analysis of the surface distribution of bone has shown spatial concentrations of this material (***Kruger et al., 2016***; ***Kruger, 2017***). This patterning of skeletal material present on the floor surface does in part correlate with subsurface patterning, as reflected by the high concentration of surface elements immediately above the Puzzle Box area. Additionally, the occurrence of some conjoins between fragments excavated from the Puzzle Box area and fragments on the nearby surface shows that some skeletal material on the surface likely derives from postdepositional reworking of subsurface material and potentially deflation and compaction of sediments during body decay processes. On the other

hand, Dinaledi Feature 1 did not have any concentration of material on the floor surface immediately above it. Elements could not persist on the floor surface immediately above or near the Hill Antechamber Feature due to the high slope. The data do not provide evidence about whether fragments or elements originating from this location may have been reworked downslope, although this is possible. The incorporation of such reworked fragments into the floor sediment may explain the presence of fragments at low density in excavated units that have no high concentration of bone.

## Assessing hypotheses for the deposition of *H. naledi* remains and associated material

The results of these excavations and analyses provide a rich and complex picture of the surface and subsurface distribution of fossil material, the sedimentary context immediately surrounding this material, and the spatial relationship of these deposits within the surrounding subsystem. The new data add substantially to the information yielded by previous excavations and analyses within the system. In this section, we consider how these new data relate to the disposition of bodies within the Dinaledi Subsystem. We distinguish hypotheses for the subsequent disposition of bodies from hypotheses for the entry of *H. naledi* bodies into the subsystem, although these may be related to each other. *Dirks et al., 2015* and later work (*Randolph-Quinney et al., 2016*; *Hawks et al., 2017*; *Elliott et al., 2021*) addressed hypotheses for the entry of material into the Dinaledi Subsystem; these are discussed in the Appendix ('Hypotheses tested in previous work'). The data show that whole bodies of *H. naledi* were introduced into the subsystem prior to the decomposition of soft tissue, at least in some clear instances. Here, we examine hypotheses for how these bodies came to be buried in sediment after their introduction into the subsystem.

### Deposition upon a talus with passive burial

One of the three scenarios discussed by *Dirks et al., 2015* for the deposition of the *H. naledi* remains was the dropping of bodies into the entry Chute of the subsystem, with resulting accumulation of fleshed and semi-fleshed remains as a talus deposit. In this hypothesis, *H. naledi* bodies entered the subsystem into the Hill Antechamber and collected as a talus below the subsystem entry. The presence of articulated remains in the Dinaledi Chamber under this hypothesis would be attributable to slumping downslope movement of fleshed and semi-fleshed remains before the decomposition of soft tissue. Burial of the articulated and other material would have resulted from the continued inflow of sediment and bodies onto the talus, with subsequent incorporation of sediments and skeletal material into the structure of the talus. This hypothesis relates to the cartoon illustration of the Dinaledi Chamber in Figure 3 of *Dirks et al., 2015*, which portrays skeletal material throughout the floor deposits of the subsystem.

The results reported here, together with data previously collected from the subsystem, show that the continuous bone bed hypothesis is false. The subsurface representation of skeletal remains is heterogenous both within the sloping floor of the Hill Antechamber and within the sampled areas of the Dinaledi Chamber. While there are substantial *H. naledi* remains within the Hill Antechamber, these are localized in particular spatial areas. The sloping floor of this antechamber may comprise a talus, as suggested by the appearance of layering of clast-rich and clast-poor strata observed in the sediment profile, but this deposit was not composed in the main of bodies or parts of bodies. The presence of some *H. naledi* remains within a downslope excavation unit and on the antechamber surface is consistent with some degree of gravity-driven reworking downslope, but not to a sufficient degree to relocate body parts further into the subsystem than the base of this sediment slope. The narrow and choked passageways to the Dinaledi Chamber could not have been conduits of gravity-driven sediment slumping from the Hill Antechamber, nor could sediment have slumped uphill against gravity a further distance into the Dinaledi Chamber. The articulated remains within the Puzzle Box area, and the articulated remains within Dinaledi Feature 1, cannot have been transported passively by gravity from the entry Chute of the system. The material at U.W. 110 also cannot have been transported passively by gravity from the entry Chute, nor could this material have been transported from the Puzzle Box or adjacent areas by gravity.

While previous observations may have been sufficient to reject this hypothesis, it is valuable to summarize the total pattern of data, which is not compatible with the continuous accumulation of

bodies dumped onto a receiving talus. Previous work suggested that this hypothesis might be credible *Dirks et al., 2015*, and the well-known fossil deposit from Sima de los Huesos, Spain *Sala et al., 2024*; *Arsuaga et al., 2014* was noted as a possible parallel. However, despite some superficial similarities, including the large accumulation of many hominin individuals, the data show that the Dinaledi Subsystem situation is not like Sima de los Huesos. The data falsify the hypothesis that *H. naledi* bodies were dropped into the entry of the subsystem and transported passively by sedimentary movement from that location.

## Slow sedimentary burial of remains on cave floor surfaces or within natural depressions on cave floors

*Dirks et al., 2015* discuss two scenarios in which *H. naledi* bodies may have been emplaced within the subsystem at varied points instead of collecting passively immediately below the entry point to the system. In one of these scenarios, living *H. naledi* individuals entered the subsystem and died within it, the position of the bodies resulting from their deaths. In the other scenario, living *H. naledi* individuals transported dead bodies into the subsystem and placed them on the subsystem surface. These scenarios represent two extremes with possible intermediate hypotheses representing different patterns of social interactions with dead bodies. For example, *Nel et al., 2021* hypothesized that *H. naledi* adults may have used the subsystem as a 'sleeping site', sometimes dying natural deaths within this space, while the bodies of *H. naledi* infants and juveniles may have been carried into the subsystem by adults after their natural deaths outside the cave. All these scenarios entail the deposition of bodies on the floor of the subsystem at varied locations, followed by the encasement of the bodies in sediment by natural sedimentation processes. However, different patterns of sedimentation may result in different configurations of skeletal remains and associated sediments. We consider two boundary conditions for natural sedimentation: slow and gradual sedimentation versus rapid gravity-driven sedimentation (slumping). In this section, we test the hypothesis of slow and gradual sedimentation covering bodies on surfaces or within shallow depressions.

In the slow and gradual sedimentation scenario, bodies remained uncovered after their primary deposition until naturally derived sediments gradually covered them. The decomposition of a body that is sub-aerially exposed (i.e. not encapsulated by burial sediments) is driven by a combination of intrinsic and extrinsic factors, with temperature, microbial action, and insect predation being primary drivers of decomposition (*Forbes et al., 2017*; *Ody et al., 2017*). When a sub-aerially exposed body decomposes, gravity causes the bones to fall into positions that conform to the surface underlying the body. Actualistic experiments demonstrate that this also occurs when sub-aerially exposed bodies decompose while partially constrained by a pit wall (*Mickleburgh and Wescott, 2018*; *Mickleburgh et al., 2022*), a condition similar to sub-aerial decomposition in a natural depression. Uncovered decomposition may leave bones in general anatomical order, but elements lose articulations as they move to conform to the underlying surface, though they may reconnect (i.e. resume anatomical position) as part of this settling process (*ibid:* 165). These patterns are observable even after such remains are covered subsequently by sedimentation.

The nature of the sediments covering the remains provides additional ways to test the slow and gradual sedimentation scenario. Covering sediments in the Dinaledi subsystem do not include exogenous sediments introduced into the cave system but instead are endogenous in nature, derived either from in situ weathering or percolation through small fissures (*Dirks et al., 2015*; *Dirks et al., 2017*). In situations where a body is slowly encapsulated by sediment on an open surface, or bounded by a pit or depression, sediments are expected to present evidence of slow accumulation and cross-sectional morphology (stratification) in profile which reflects the pattern and tempo of encapsulation of the body. If sediments accumulated gradually upon the bones after decomposition, fine lamination of the sediments may be present, including within the sediments in direct contact with the remains. Sediment that encapsulates a concentration of skeletal remains that were buried by slow accretion must have arrived there by percolating around or infiltrating between elements and should therefore have texture representing the low energy required to transport and deposit such sedimentary material. In caves, clasts may be present from roof fall or spalling or overlying geology, but these too would be incorporated into the burial environment by gradually accumulating sediments. However, the occurrence of large clasts *within* the structures of the body, particularly large clasts that occur between

elements, is inconsistent with slow gradual sedimentation covering skeletal remains upon a surface or within a shallow depression. The slow deposition scenario also predicts that subsurface commingling of remains of different individuals may also have occurred on the surface prior to incorporation into the burial environment.

## Hill Antechamber Feature

The sedimentary situation of the Hill Antechamber floor is not consistent with an ancient depression filled by skeletal material and slow subsequent burial. The present-day floor surface of the Hill Antechamber has a slope of approximately 30 degrees from its highest point near the northeast wall down to the southwest. The subsurface deposits revealed by excavation in the area surrounding Feature 1 have alternating layers of LORM-rich and LORM-poor sediment (*Figure 8*). Where excavation exposed these sediments on the east and south walls of the unit containing Feature 1, these layers exhibit the same slope as the floor immediately above them. If there had been a natural depression at the time of emplacement of hominin remains in Feature 1, these layers would be expected to dip under the skeletal material. No such dip was encountered during the excavation of the unit. LORM-rich layers following the same angle were exposed immediately against the edge of the feature during excavation, on the east, south, and west sides (*Figure 8*, *Appendix 6—figures 6 and 8*). This layering shows that the ancient floor was sloped in the same orientation throughout the time of formation of these sediments.

The hominin skeletal material intrudes into and truncates the surrounding layered sediment (*Figure 8*). A series of east-west profiles within the feature are illustrated in *Figure 8*, a north-south profile is illustrated in Figure S7, and adjacent profiles outside the feature are illustrated in *Figure 8*, *Appendix 6—figures 6 and 8*. Within the feature, the skeletal remains occur at different angles from the horizontal and do not occupy a single settling layer that could correspond to skeletal material decomposing within, and coming to rest upon, the base of a depression or ancient surface (*Figure 9*). The total vertical extent of the remains is more than 18 cm, which is sufficient to inter a small-bodied individual and within the range of depths for clandestine forensic graves (*Hunter and Cox, 2005*; *Manhein, 1996*) and near-surface archaeological burials (*Crawford, 1999*; *Channing and Randolph-Quinney, 2006*). The articulated and anatomically associated right foot, ankle, and lower limb bones that are located in the base of the feature are separated by sediment fill from the skeletal material overlying them. These structures are generally considered labile (non-persistent) joints, and the retention of articulation suggests matrix support before decomposition. The immediately surrounding sediment fill does not exhibit sedimentary layering or morphology indicative of slow sedimentation, neither at large scale (*Figure 8*) nor small scale (*Figure 10*). The presence of large clasts shows that this sediment did not filter slowly into its current location, including where these clasts occur immediately below and above skeletal material. These observations indicate that the body and sediment were emplaced at the same time within this location; the body parts are supported by sedimentary matrix which was deposited as a single fill contemporaneous with the deposition of the body.

The single stone object is at an angle and position relative to skeletal material that could not have occurred on a surface without encasement with sediment. The remains of the cranium and mandible, at the highest point of the feature, are at a position on the slope of the remaining material that would be highly precarious on a sloping surface without encasement by surrounding matrix. In summary, the Hill Antechamber Feature cannot have been produced by slow sedimentation of skeletal remains that lay exposed upon a surface or within a depression.

## Dinaledi Feature 1

The sedimentary situation of Dinaledi Feature 1 is likewise not compatible with the deposition of remains upon a surface or within a shallow depression, followed by slow sediment covering. The sediment that is in immediate contact with the skeletal material in this feature is consistent with sub-unit 3b sediment from other parts of the subsystem, including clasts of up to 2 cm in diameter at random orientations and with angular boundary shapes. Clasts of this size cannot infiltrate or percolate gradually into gaps and spaces left around skeletal elements (*Figure 15*). The LORM composition of the

Unit 3 sediments in the Dinaledi Chamber is inferred to reflect the autobrecciation of Unit 1 sediments during the formation of Unit 3, and therefore the LORM presence within sediment predates the formation of Unit 3 and deposition of the *H. naledi* material. The LORM clasts within this feature must have been deposited upon or among the *H. naledi* elements by some process other than slow subaerial sedimentation.

There is no laminar patterning or layering of the sediment in contact with skeletal remains within Dinaledi Feature 1. The feature itself contrasts with the sediment immediately adjacent to it, which does not contain bone, but which does exhibit a distinct pattern of layering. The LOSMU layer that is visible in the sediment profile adjacent to the Dinaledi Feature 1 is interrupted by this feature (*Figure 15b*). The lower contour of the feature in this location, therefore, diverges from the appearance of a former floor surface. The interruption of an appearance of stable layering in this area of the chamber is not consistent with a stable surface that existed both before and after the deposition of the remains. In profile, this interruption is consistent with truncation into pre-existing layered sediments.

Several elements visible within Dinaledi Feature 1 are supported by surrounding sediment matrix in positions that could not occur upon a surface or within a shallow depression. These include a series of articulated ribs supported in a near-vertical anatomical orientation where the excavation has exposed them, as well as the matrix-supported portions of cranial vault (*Figure 21*).

## Puzzle Box area

Several aspects of the Puzzle Box area are incompatible with the hypothesis of slow sedimentary action covering remains on a surface or in a shallow depression. As in the case of the Hill Antechamber Feature and Dinaledi Feature 1, the sediment in direct contact with the hominin skeletal remains is composed of a brownish clay matrix containing clasts of LORM up to 2 cm in diameter. This composition is not consistent with slow sedimentation after deposition of the skeletal remains; the presence of such clasts within concentrations of skeletal remains is not consistent with percolation into and around subaerially exposed bones.

The skeletal remains within this area are in spatial configurations that are not consistent with deposition and decomposition on a flat surface or in a shallow depression. Articulated remains occur within this area at varied depths from the floor surface above them, ranging from 5 cm up to as much as 20 cm (*Kruger et al., 2016*; *Kruger, 2017*). Some articulated elements, including the near-complete Hand 1 and Foot 1, are at contrasting depths but close horizontal distance to each other. Any floor surface that ran beneath both of these articulated structures would have been highly sloped and would have provided no stable platform for decomposition and subsequent sedimentation (after *Kjorlien et al., 2009*; *Morton and Lord, 2006*; *Pokines and Baker, 2013*). If the articulated remains resulted from multiple depositional events upon surfaces that were accumulating over a long time, with slow sedimentation separating them, then the occurrence of repeated deposition in this one location, one on top of another, without gravity-induced displacement or down-slope scattering, would require some explanation.

Finally, the occurrence of sub-vertically oriented fragments would not be possible for exposed skeletal remains resulting from sub-aerial decomposition on a floor surface. These sub-vertically oriented elements must have been matrix encapsulated during the decomposition process in order to retain their final disposition (after *Mickleburgh et al., 2022*). The fact that these are juxtaposed near more horizontally oriented elements at varied levels cannot be a result of slow sedimentation. These elements were supported by the matrix in their excavated orientations, which means the encasement with the matrix was rapid and not slow.

## Summary

Across the features and other excavation areas, several key observations reject the hypothesis of deposition of bodies on a surface or within a natural depression followed by slow sedimentation. Body parts and fragments in all features occur at angles well out of horizontal in positions that could not be maintained without matrix support. In several cases, subvertical elements are in close contact with other elements at more horizontal orientations, arrangements that could not occur unsupported even within a depression. Large clasts are intermingled with skeletal remains in a way that could not reflect gradual sedimentation processes. Sediments adjacent to features contrast in layering and

composition from sediments within the features in a way inconsistent with the occurrence of natural depressions underlying remains. One observation that covers all areas with skeletal material is that the remains occur across a vertical depth of 20–30 cm but are covered by only a few centimeters of overlying sediment. This incongruity between a rapid sediment buildup during the period of fossil deposition, followed by extremely slow sediment accumulation after deposition, would be another inconsistency in the depositional history of the area.

## Rapid sediment slumping

An alternative hypothesis to slow and gradual sedimentation is a rapid gravity-driven slumping. Under this scenario, large clasts may become part of the deposit immediately surrounding and within skeletal remains because they are emplaced rapidly with the movement of a large mass of sediment. This scenario differs from the slow and gradual sedimentation scenario by the high kinetic energy associated with the sediment movement. High kinetic energies can be generated only in deposits where sediments of high potential energy exist, as for example in highly sloped deposits, deposits near input channels from other cave areas, and deposits with underlying collapses or sediment drains where rapid downward movement of sediment is possible. Water flow or mud flow may also generate high kinetic energy, but as noted previously, the evidence from micromorphology and sediment composition excludes water or mud flow as formation processes for the Dinaledi Subsystem deposits. Mud-flow depositional dynamics have been noted in other South African hominin-bearing localities, most notably at the site of Malapa, where hominin-bearing facies contained allochthonous material mixed with cave-derived sediment and coarse blocks deposited as a single debris flow (*Dirks et al., 2010*). Such a lithology is not present in the Dinaledi Chamber. Instead, the characteristics of the Unit 3 unlithified mud-clast breccia formed in dry cave conditions by autobrecciation of preceding Unit 1 sediments (*Wiersma et al., 2020*). The evidence against water or mud debris flow is further discussed in the Appendix.

Throughout this section, it should be kept in mind that the hypothesis of rapid sediment slumping is one to explain the emplacement of the remains and their rapid encasement by sediment. Selective slumping or gravitational movement of skeletal elements can also occur after the burial of a body. The breakdown and decomposition of soft tissue will leave voids in a buried body, particularly in the thorax and appendicular units with large cross-sectional diameters, such as the muscle compartments of the upper arm and the thigh. These subsequently collapse as soft tissues decompose and the bones lose structural integrity, and sediment may percolate around bones with overlying material gradually moving downward. Such a selective pattern of movement is considered in the following section. In this section, we consider the initial burial of a body or body parts. Burial by slumping requires high energy across a relatively short time, and as such, requires different conditions than the movement associated with postburial diagenesis.

The skeletal material from the S150W150 area in the Hill Antechamber may possibly represent an example of accumulation resulting from downslope movement. The context and material exhibit several aspects consistent with downslope movement with high energy. The skeletal remains are in isolated patches and mostly consist of small unidentifiable fragments. No articulation is evident in the assemblage. Yet this deposit is not without order. The majority of the maxillary teeth from a single individual are present, buried within a radius of 25 cm at varied depths. If these teeth were displaced from their original depositional location, they were constrained in direction or distance to a narrow endpoint. If slumping or erosion from upslope was the mechanism of burial in this location, that movement was selective and constrained. Alternatively, it is possible that these teeth were carried together near to the location where they were excavated while still rooted in the maxilla. If so, that bone was lost and the teeth subsequently scattered into the excavation area.

## Hill Antechamber Feature

This feature is in a position near the top of a steep talus slope and near an ingress point to the subsystem. This position is one where sediment with high potential energy exists in the subsystem. However, slumping or gravity-driven movement downward out of this elevated position is more possible than slumping upward into this location. Slumping into this location at the top of the slope would have to be a collapse along the top of the sloping floor from the southeast or northwest

directions. Several observations reject the hypothesis that the Hill Antechamber skeletal remains were rapidly buried by slumping sediment. The skeletal remains, including articulated elements, are interspersed within the feature with vertical separation of elements. This means that a slump or collapse could not have suddenly covered material that was already resting decomposed and disarticulated on the sloping floor; the skeletal remains including body parts that retained connective tissue intact must have already been incorporated within the sediment mass that slumped.

The sudden emplacement of a mass of sediment incorporating articulated body parts and skeletal remains into a depression on a slope would create a different pattern of evidence from the Hill Antechamber Feature in two ways. First, the stratigraphic evidence surrounding the feature shows layering that is interrupted by the sediment infill of the feature. This shows that some process disrupted the floor sediments by excavating or truncating them prior to the infill of sediment and hominin remains. Both the prior disruption of the talus deposit and the subsequent encasement of articulated skeletal remains require explanation; a single large slumping event cannot explain both. Second, the arrangement of the remains is not consistent with mass movement from another location, as seen at sites such as Malapa (after *Dirks et al., 2010*). This is clearly evidenced in terms of the anatomical association of preserved manual and pedal elements, which are directly articulated in the case of the Puzzle Box and Hill Antechamber Feature, while disarticulated but in local spatial association as seen in Malapa Hominin 2 (*contra Val et al., 2015*: 9; see Appendix 5.1: 'Spatial taphonomy of Puzzle Box – archaeothanatology of Hand 1 and Foot 1').

The Hill Antechamber remains present some evidence of postdepositional fragmentation and displacement that may be consistent with a small degree of sediment shifting. For example, the tibia and fibula associated with the articulated right foot, which are angled opposite the floor slope, show fragmentation and slight displacement of fragments away from each other by distances of up to a few centimeters and at angles up to 15 degrees. This may have resulted from a number of different causes, including pressure from overlying sediment, slight downslope shifting of sediment, movement of sediment within the plaster jacket during transport out of the cave, or reworking of the sediment by *H. naledi*. Other aspects of the spatial configuration show that their burial prevented further high-energy downslope movement. For example, the cranium which occurs near the highest point of the feature appears to have been flattened by overlying sediment with resulting loss of structure but without any evidence of downslope shifting, and with maxilla and hemimandible retaining their occlusal positions. Considering the location of the feature, it is remarkable that the skeletal remains remain in position without significant further slumping or downslope movement.

## Dinaledi Feature 1 and Puzzle Box area

These areas are considered together because of their spatial proximity; they share immediate surroundings and are within adjacent areas of the chamber floor. Both are emplaced in a floor that has a local slope of 11 degrees without notable increase in slope at the flanking chamber walls. A fissure passage exits the Dinaledi Chamber some 1.5 m north of Dinaledi Feature 1, and the floor of this passage continues the slope of the chamber floor. A closed fissure passage occurs approximately 1 m to the west of both features and is characterized by Unit 1 sediment within it. Potential sediment flow from this niche is currently prevented by a large rock or projection of underlying bedrock that protrudes from the chamber floor and that separates this fissure passage from both features. There are no evident sources of high potential energy sediment input into the location of these features.

In previous work describing the Puzzle Box excavation area, *Dirks et al., 2015* considered the possibility of slumping or movement. In that work, they did not distinguish the Dinaledi Chamber from the Hill Antechamber. They considered gravitational slumping down a talus (now recognized within the Hill Antechamber) as a possible source of sediment and skeletal material for the Puzzle Box area. They also considered floor drains as a possible conduit of sediment movement or slumping. The motivation for these hypotheses was the evidence for reworking within the deposit, which includes the presence of skeletal elements with subvertical orientations, large clasts of LORM, and presence of elements that show evidence of movement subsequent to postdepositional fracturing. There was also an inherent predilection at the time to explain movement and disturbance via solely 'natural' geological causes. *Dirks et al., 2015* recognized, though, that it was difficult to explain this evidence of reworking without a mechanism that could generate both horizontal and vertical movement of

sediment. In the Puzzle Box area, no mechanism for such reworking was evident from the surface or excavation area. The excavation at that time could not exclude the possibility that floor drains below the Dinaledi Chamber deposit might exist, providing a mechanism for gravity-driven localized reworking.

Subsequent work (*Dirks et al., 2017*) provided additional data by revealing the sediment structure immediately below the Puzzle Box Feature, while (*Wiersma et al., 2020*) outlined a mechanism for sediment formation that explained the presence of large Unit 1 clasts within the mud-clast breccia of Unit 3 sediments. The present study further helps to clarify the situation around the Puzzle Box Feature and thereby provide the opportunity to test the slumping hypothesis with better data.

The sediment column immediately below the Puzzle Box Feature consists of sub-unit 3 a sediment that extends unbroken to a depth of 80 cm below the present surface. This excavated area shows no evidence of slumping or downward sediment movement. The sediment from this area includes the tooth crown of *Papio* that provided a combined US-ESR age estimate of 635±148 ka. This evidence shows that the sediments immediately beneath the Puzzle Box Feature were in place prior to the arrival of the *H. naledi* skeletal material and that these sediments remained in place throughout their subsequent history. This provides no opportunity for slumping of the *H. naledi* skeletal remains downward through the sediment column subsequent to their deposition. If these remains were deposited on the floor surface of the cave, this evidence shows that the remains were not subsequently displaced downward by slumping or collapse. Nor is there any source for sub-unit 3 a sediment above or upslope from this area that might potentially have encased such remains from above or from the side. The LOSMU layer that occurs adjacent to both the Puzzle Box area and Dinaledi Feature 1 is disrupted at the edges of these features. As in the case of the Hill Antechamber Feature, this disruption occurred prior to the introduction of skeletal remains and cannot have been the result of slumping into these two locations. With very little floor slope and a disrupted horizontal layer beneath the surface, this area did not have the potential energy to spontaneously excavate holes or depressions into the surface, nor to emplace sediment and articulated bodies or body parts together in a large slumping episode.

Emplacement of these features by postdepositional slumping is also contraindicated by the skeletal remains themselves. The skeletal remains in the Puzzle Box area show clear evidence of postdepositional reworking, but that reworking could not have been the result of slumping. Three key observations are (1) presence of articulated body parts; (2) presence of large anatomical units with anatomical ordering; (3) elements in subvertical orientations surrounded by elements in other (including horizontal) orientation; and (4) lateral, vertical, and angular movement of fragments after postdepositional fracturing. Combined with each other, these aspects of the assemblage falsify the hypothesis that their configuration was the result of gravitational slumping. If fragments were moved after deposition by slumping downward, as a result of the collapse of underlying sediment, then articulation of elements throughout varied parts of the Puzzle Box area would have been disrupted. In particular, largely intact structures including Hand 1 and Foot 1 lie *beneath* reworked material; if the reworking was the result of subsurface slumping, these articulated structures would themselves have been reworked before or simultaneously with the overlying material. If subvertical fragments were shifted into this pitch by downward slumping or collapse, then the adjacent fragments would share a similar pitch and orientation, reflecting the direction of subsidence. The 'puzzle box' arrangement of elements in that feature could not reflect downward sediment slumping. Instead, it reflects selective reworking of skeletal remains after their deposition.

## Natural mummification

Earlier excavations in the Rising Star cave system uncovered many instances of skeletal elements found in articulation or semi-articulation with each other (*Dirks et al., 2015*; *Kivell et al., 2015*; *Harcourt-Smith et al., 2015*; *Kruger et al., 2016*; *Kruger et al., 2016*; *Hawks et al., 2017*). The present study shows further examples of articulated and matrix-supported skeletal elements in the Hill Antechamber Feature and in Dinaledi Feature 1. The retention of articulations in such material by itself demonstrates that the soft tissue connecting bones to each other must have still been present at the time the remains became encased within the surrounding sedimentary matrix. As noted above, this evidence is inconsistent with subaerial decomposition of bodies upon a cave floor or within a shallow depression.

Some authors and commentators have previously hypothesized that the presence of articulated *H. naledi* body parts might be consistent with the natural mummification of bodies or parts of bodies (e.g. *Val, 2016*; *Durand, 2017*; *Nel et al., 2021*; *Martinón-Torres et al., 2024*). Mummification is defined as the preservation of bodily soft tissue via natural processes and can include desiccation, or in the case of 'bog bodies' by immersion in a preservative environment such as a sphagnum peat bog (*Booth et al., 2015*; *Piombino-Mascali and Carr, 2020*). The former type of mummification occurs when the rapid desiccation of tissue delays or prevents decomposition (*Piombino-Mascali and Carr, 2020*). Soft tissue, therefore, survives in a desiccated state instead of undergoing total decompositional breakdown, which sometimes enables the persistence of bone articulations. Some degree of natural dry mummification sometimes occurs in varied settings in southern Africa, noted in both historic and contemporary human cases (*Finaughty and Morris, 2019*; *Esterhuysen et al., 2009*; *Karodia et al., 2016*; *Nel et al., 2021*). *Nel et al., 2021* documented baboon (*Papio ursinus*) remains from Misgrot Cave, South Africa, where remains were collected from the surface of a cave chamber near a natural cave entrance. These remains were in varied states of decomposition, including some desiccated and partially mummified bodies. These authors hypothesized that some adult baboon individuals died naturally in this cave setting while infant or juvenile individuals may have been carried there by adults or subject to infanticide within the cave. They further hypothesized that similar desiccated remains might retain articulations if they were in a situation where they could be quickly covered by sediment, different from the Misgrot depositional situation.

To avoid any possible confusion, it is important to note that the *H. naledi* remains as excavated are not mummies. No evidence of any remaining soft tissue has been noted during the excavation or collection of *H. naledi* remains. Total soft tissue decomposition ultimately did occur in every case of *H. naledi* remains observed in the Rising Star system. Articulations of these remains persisted after the decomposition of soft tissue, which means that the remains were encased within sediment prior to soft tissue decomposition. The sequence of encasement by sediment followed by soft tissue decomposition did occur, regardless of whether some of the remains were desiccated prior to encasement by sediment. Thus, natural mummification or desiccation does not provide any alternative pathway to the preservation of articulations within these remains. Encasement by sediment prior to soft tissue decomposition explains the presence of articulations.

What the hypothesis of desiccation might conceivably provide is a delay of some period of time between death and subsequent encasement by sediment. We can consider three possible scenarios. In one scenario, a natural mummy within a shallow depression on the cave floor might retain the relative positions of skeletal elements including articulations for some period of months or years, long enough to be gradually covered by sedimentation. In a second scenario, a partially mummified body might be dislocated together with sediment by gravity-driven slumping, resulting in the rapid encasement of the body in sediment at some time months or years after the death of the individual. In a third scenario, hominin individuals collect a partially mummified body from the landscape outside the cave at some time after its death, bring the mummified remains into a cave chamber, dig a hole, and bury the remains in sediment. In each case, the proximal agent of preservation of soft tissue and joint integrity is the process of natural mummification. It is evident that the occurrence of desiccation of the remains does not distinguish these possible scenarios from each other; rather, it is the process of encasement within sediment that may distinguish them, thus providing a way to circumvent issues of equifinality and under-determination (after *Perreault, 2019*).

As noted in previous sections, the sedimentological and geoarchaeological evidence rejects the first two scenarios for the encasement of remains within sediment. The evidence from Dinaledi Feature 1, the Puzzle Box Feature, the Hill Antechamber Feature is inconsistent with the gradual encasement of bodies by slow subaerial sedimentation. None of these features is consistent with rapid sediment slumping. These scenarios are excluded whether or not the remains were desiccated prior to encasement in sediment. Additionally, the conditions within the cave system are not conducive to natural mummification. In some outdoor settings, precocious mummification may occur more frequently and faster during summer conditions than winter (*Finaughty and Morris, 2019*). In cave settings, some degree of natural mummification has been noted in areas near surface entrances where remains are protected from scavengers including small carnivores. The environmental conditions where natural mummification is possible in caves are moderate to high temperature, low humidity, and natural ventilation (*Piombino-Mascali and Carr, 2020*). The Dinaledi Subsystem is far from any cave entrance and

is characterized by a high moisture content in the air. While cycles of higher and lower moisture within the cave system would have occurred in the past, at no time was rapid desiccation of a body likely to have occurred in this setting.

The sedimentological evidence does not exclude the third scenario, in which some *H. naledi* remains were naturally desiccated outside the cave system, then collected and buried by other individuals. The possibility of preburial desiccation of some of the *H. naledi* remains is an interesting hypothesis that we do not test further in this study, but such a scenario has been raised by *Schotsmans et al., 2022* in relation to forensic observations. In this case, a body naturally mummified for 28 days at the Australian Facility for Taphonomic Experimental Research (AFTER) was subsequently moved to a grave and buried. Schotmans and colleagues suggest a scenario (*ibid:* 513) where thousands of years later, when all soft tissue has disappeared, archaeologists would be tempted to classify such a buried mummy as a primary deposition based on the skeleton being fully articulated. Technically, from an archaeothanatological perspective, this would be classified as a secondary deposition, even though the act of burial carries significant importance as a mortuary act.

Several aspects of the surface condition and preservation of *H. naledi* remains may bear upon the hypothesis of preburial desiccation. While insect activity has been noted to damage mummified remains in many contexts, evidence of bone surface modifications by insects or other invertebrates is rare in some contexts of mummified remains (e.g. *Nel et al., 2021*). The *Homo naledi* material from the Dinaledi Chamber has a high frequency of surface modifications from invertebrate activity including bone surface markings attributable to beetles and markings attributable to snails (*Dirks et al., 2015*). Additionally, the sediment in direct contact with skeletal material in the Hill Antechamber Feature has evidence of invertebrate burrows that suggest an interaction with tissue or bone subsequent to burial. Disarticulation patterns also shed light on this issue, specifically patterns of 'paradoxical disarticulation' (*Duday and Guillon, 2006*) which refers to a reversal in the relationship between labile and persistent joints, where if labile connections are maintained and persistent joints disarticulated, the individual may have been mummified prior to deposition into the burial environment (*Maureille and Sellier, 1996*; *Sellier and Bendezu-Sarmiento, 2013*). The case of the Hill Antechamber Feature includes evidence of articulation of both labile and persistent joints, suggesting that paradoxical disarticulation does not apply in this case.

We conclude that the hypothesis of natural mummification does not provide an alternative depositional scenario to those considered above. *Homo naledi* remains were encased within sediment prior to soft tissue decomposition, and the sedimentological evidence addresses the pattern of encasement by sediment irrespective of whether remains may have been desiccated at the time of burial. Natural mummification of bodies after their entry into the Dinaledi subsystem is unlikely due to the conditions within this part of the cave system. The possibility that some *H. naledi* remains may have been desiccated prior to their introduction and burial within this part of the cave system is difficult to test. At a minimum, the transport of desiccated remains into the Dinaledi Subsystem would be unlikely without resulting in damage, disarticulation, or loss of parts, unless the remains were bound or enclosed in some kind of covering. The evidence from the *H. naledi* skeletal remains includes no specific indications of mummification or desiccation.

## Primary cultural burial

Previous work has not considered the hypothesis that *H. naledi* itself was an agent of reworking the sediments within the Dinaledi Subsystem. Burying bodies or body parts is one way that *H. naledi* may have reworked or modified sediments and skeletal remains. The decomposition of bodies after burial results in localized shifts in the position of skeletal elements and sediment brought about when soft tissue breaks down, creating voids where sediment and overlying skeletal material can collapse. This reworking by decompositional collapse would leave some remains in articulation while creating the circumstances where some fragments may be displaced into subvertical or oblique orientations supported by sediment and underlying skeletal material. Digging by *H. naledi* within sediments where remains had previously been interred would also result in localized reworking. Such digging would fragment skeletal material, cast some skeletal fragments onto the surface, and cause commingling of elements from different individuals as bone-containing sediment was used to cover bodies. Such a process is very common in historical cemetery contexts, particularly where the position of graves is

unmarked. In such cases, later mortuary acts (burials) cut through earlier ones, causing disturbance, disarticulation, and possibly dry bone breakage; some or all of the disturbed material may be placed back into the grave fill of the truncating burial or may be dispersed into the surrounding sediments (after *Channing and Randolph-Quinney, 2006*).

To enable a test of this hypothesis, we consider a minimum definition of cultural burial. We use the term *cultural burial* for clarity that we are discussing burial processes that result from cultural activities of hominins, as opposed to burial processes that may occur without hominin agency. Burial of hominin remains by other hominins is a cultural act, although the meaning and form of burial both differ greatly in different cultural contexts. We use a minimum definition that involves three components:

1. A hole or pit is dug by hominins into sediment.
2. A body or parts of bodies are placed into this feature by hominins.
3. The remains are covered (backfilled) by hominins.

The hypothesis of cultural burial is testable with reference to evidence from the spatial characteristics of skeletal material and their sedimentary context. If evidence proves that one of these three actions did not occur, then the hypothesis is falsified.

This minimal definition is different from the diagnostic criteria that have been employed in some other studies for the identification of burial of hominin remains. Some studies have relied upon the presence of 'grave goods', or cultural artifacts that are intentionally placed within the burial, even in the absence of compelling taphonomic or geoarchaeological evidence. Some have emphasized the importance of deliberate or ritualized positioning of the hominin remains. Others have noted the importance of 'special' or culturally significant surroundings as verification of cultural burial. We recognize that these elements may help to substantiate the cultural nature of burial evidence. But we observe that each of these criteria is specific to particular human cultures. In many contemporary human cultures where burial is practiced, these elements are absent. We have no reason to assume that *H. naledi* activity patterns would include elements that have such cultural specificity. We also recognize that some cases of Pleistocene burial are cases in which no hole was dug by hominins but instead remains were placed into a natural or pre-existing depression or space and then covered with sediments (*Pomeroy et al., 2020a*). Our definition could readily be extended to include such cases. We include a more detailed consideration of the minimal definition and these additional criteria in the Discussion. For the purposes of our analysis, we noted aspects of the evidence that might be consistent with these additional criteria without relying upon them to test the hypothesis of cultural burial.

## Hill Antechamber Feature

Sedimentary evidence within and around this feature suggests that a hole was dug through existing stratigraphy (*Figure 8*, *Figure 9*, *Appendix 6—figures 6–8, 11–13*) truncating the floor locally. The sedimentary line at the base of the feature is curved outside of the horizontal and departing from the slope of the surrounding floor and sedimentary profile (Figure S7). This configuration indicates that the skeletal remains and associated sediments are intrusive into the talus floor deposit rather than being part of a talus. The profiles within and outside the excavated unit evidence a cut which truncates sediment layers that were in place prior to the introduction of hominin remains.

The configuration of the skeletal remains in this feature is complex (*Figures 5 and 6*), with the body of Individual 1 being in a flexed position at the time of interment, with the right foot and right hand near or at their current spatial positions. The proximity and anatomical attitude of the left humerus, radius, and ulna, and left hand and wrist material is consistent with flexion of the left upper limb. Based on their current position, we conclude that the decomposition of muscle and integument created a void space into which the elements collapsed, with further spatial displacement following loss of connective tissues within the joints, suggesting the limb was resting at a somewhat higher level during the early postmortem period. Collectively, remains of the torso and upper limbs are compressed toward a plane that represents their collapsed state, consistent with archaeothanatological principles. The torso and upper limbs are separated from the lower limb and right foot by sediment that must have been in place prior to decompositional collapse of the upper body. The absence of such a matrix layer (i.e. if the burial was configured as an open pit) would have ensured that the torso and limb groups would have settled in a single layer at the base of the cut (*Mickleburgh and Wescott, 2018*), and

we would expect to see sedimentary laminations which reflect the shape of the pit in cross-section, with laminations deepest at the edges and margins of the pit and shallowest at the center overlying the decomposed body. Such a pattern, termed a 'secondary depression', is commonly observed in shallow graves (*Hunter, 2014*). Instead, the vertical separation between lower limb and upper body material is indicative of their support by sedimentary matrix. Decompositional slumping of the body and/or subsequent matrix compression has resulted in postdepositional fracturing visible on all identifiable rib bones. The upper limb elements exhibit multiple transverse and step fractures consistent with the effects of postdepositional matrix compression and other effects from the burial environment or our movement of the specimen. The maxillary dentition, with the mandible in or near articulation, is presently in a semi-inverted position, located above several rib shafts and near the right hand. This position is consistent with the displacement of the skull from a higher position, following decomposition of the nuchal musculature and cervical connective tissue, allowing the head to disarticulate from the axial skeleton, accompanied by rotation, inversion, and some horizontal displacement.

Several aspects of the hominin material and associated sediment show that the remains were covered prior to decomposition. The articulated right foot, articulated partial hand, articulated and associated maxillary and mandibular dentition, and partial ribcage were all encased in sediment prior to the loss of soft tissue. Some of the material, in particular the upper limb material, rests on top of and is supported by fragmented ribs and thorax material. This situation shows that the remains were subject to horizontal constraint as they were compressed downward. The articulated foot and lower limb elements are vertically separated from other material by sediment, which supports the material above. This situation is impossible for a body on a sloped surface or in a shallow depression. The remains appear to have been covered rapidly after deposition and prior to decomposition.

The single larger stone within the feature also demonstrates the rapid encasement by sediment. This object is at an angle of 25 degrees from horizontal, in a plane roughly orthogonal to the slope of the chamber floor. Beneath and lateral to it are skeletal material, and there is skeletal material above it. None of these elements are in direct contact with the object, other than a metacarpal that overlies it, and therefore none of them could have supported this object in its non-horizontal position without the presence of the sediment matrix that surrounds them. Whether this object was intentionally placed with the remains or was unintentionally interred as part of the sediment infill, its spatial position shows that the sediment infill was rapid and coincident with the emplacement of the skeletal remains that surrounds it. As the hand material is in articulation, this rapid sediment fill must have happened prior to soft tissue decomposition.

In their current situation, it is not possible to inspect the surfaces of the Hill Antechamber bones for other signs of taphonomic modification, and so we cannot rule out further involvement of hominins in the disarticulation of the remains. It is worth noting that the amplitude of movement away from the original anatomical position will depend upon the position of the remains when deposited, and the available space in which any movement can take place; such spaces can be created by the decomposition of surrounding soft tissues in cases of interment in sediment, and the amplitude of movement represents the extent to which a bone can move in any of the possible directions. Primary dispositions (especially burials) tend to place bones in a relatively stable position, and thus there will be little if any movement of individual bones or elements following the loss of soft tissue. When disposition occurs where bones are placed in an unstable position, they will move in accordance with gravity and the shape of the surrounding space once soft tissue is lost (*Roksandic, 2002*). As such, the spatial distribution and alignment of the Hill Antechamber remains are consistent with decompositional processes taking place after their deposition into a sediment-filled pit feature (*Mickleburgh and Wescott, 2018*).

Additional criteria that may be relevant to cultural burial include the possibility of manipulated body positioning and possible grave goods. Without complete excavation, we cannot be certain whether the remains that we attribute to Individual 1 were deposited in a fetal or seated position, either of which may be culturally meaningful positions in recent *Homo sapiens* burials. The stone in the feature is intriguing in both form and its position within or near the hand. While we are not in a position to identify if this stone was intentionally modified or selected by *H. naledi*, its shape is distinctive in comparison to other rocks collected from the Hill Antechamber, enhancing the possibility that it served some function. If it is a culturally placed object, it would provide additional evidence of cultural practice.

The four anterior teeth attributed to Individual 3 are positioned within the postcranial skeletal material attributed to Individual 1. No other indications of cranial or mandibular material that could be attributed to the adult Individual 3 are in evidence within the feature, although the resolution of the data does not reject the possibility that some postcranial material may represent this individual. The situation of these teeth is different from those of Individual 2, which is represented by most of the maxillary dentition. One hypothesis to account for the Individual 3 material is that these teeth were incorporated incidentally within the sediment used in the burial of Individual 1. However, the data do not rule out the hypothesis that these teeth may have been intentionally placed with the remains of Individual 1.

## Dinaledi Chamber Feature 1

The observed condition and spatial position of the remains is consistent with the emplacement of fleshed remains into a hole prior to decomposition. The spatial positioning of elements is not consistent with a supine position, positioning with extended limbs, or positioning on the side with limbs curled in one direction. Limb elements have been recovered from the highest part of the feature where they were fragmented and supported by underlying material from the head and torso which remains in anatomical positioning or articulation. The situation of elements recovered above the current excavated level reflects substantial postdepositional fracturing, loss of trabecular bone portions, and vertical compression. The breakdown and movement of skeletal material is largely consistent with the collapse and compression of skeletal elements and sediment into voids that were left by the decomposition of soft tissue. Such a pattern of collapse is characterized by the displacement of fragments and joint articulations, leaving fragments in positions where they are supported by underlying material and coarse collapsed sediment.

The sedimentary material within the feature is relatively homogenous in composition and texture, consisting of coarse, unconsolidated clasts of clay and some LORM clasts. We carried out new elemental analysis to evaluate the composition of the sediment within Feature 1 in comparison with surrounding sediment outside the feature, as well as sediments from other parts of the subsystem and broader cave system. This analysis shows that the sediment within this feature is similar to the makeup of sub-unit 3 a sediments lacking skeletal remains that surround and underlie it, with the exception of the presence of some material derived from bone. This analysis makes it clear that the feature was not filled over a long time by sediments with a different source or composition from the sub-unit 3 a sediment surrounding the feature in the same level, or underlying the feature. The composition is consistent with the excavated material from the hole having been used to cover the hominin remains.

The spatial positioning of some skeletal elements suggests that the remains were covered with sediment prior to the decomposition of soft tissue that connected them. As noted above, some elements remain in a vertical or sub-vertical orientation reflective of matrix support prior to decomposition. A portion of the ribcage is visible with some ribs oriented near vertical, others broken and collapsing along a craniocaudal axis, all apparently constrained by matrix support near the edge of the feature (*Figure 21*). Near the south end of the feature, cranial fragments are supported by matrix in sub-vertical orientations.

The presence of three fragments from a second individual is also consistent with the rapid burial of the remains. These fragments were in contact with skeletal material from the adult individual at the top of the feature, in a position that could not have been maintained upon bones that were not encased with sediment at the time of their emplacement. These fragments are easily explained if hominins excavated a hole and then filled it with the excavated sediment incorporating fragments from one or more other individuals that were either on the floor, nearby, or shallow subsurface.

Considering the possibility that these fragments were incorporated into the feature as backfill when covering the body, we must also acknowledge that the present configuration of skeletal remains at the top of the feature may in part reflect disturbance by *H. naledi* activity within the chamber subsequent to the burial, such as digging nearby burials or trampling. For example, the breakage and spatial displacement of the two halves of the mandible might be explained by such disturbance. Below, we consider this hypothesis in greater detail for the Puzzle Box feature. For Feature 1, the evidence does not distinguish this possibility from the postdepositional fragmentation, breakdown, and partial loss of bone described above.

## Puzzle Box Feature

No previous work has considered the hypothesis of hominin reworking or cultural burial as a possible explanation for the Puzzle Box Feature. We consider this hypothesis here for the first time. In doing so, we accept previous analyses of the hominin assemblage and context, which have provided evidence consistent with the hypothesis that whole bodies of hominin individuals were present in this area and that the hominin remains were subject to reworking subsequent to deposition (*Dirks et al., 2015*; *Dirks et al., 2017*). This reworking can be recognized as postdepositional due to the fracture patterning, pattern of fragment displacement, and selective effects within the assemblage. Many fragments were displaced and moved within sediment for distances up to 15 cm without erosion or loss of detail of refitting fractures. The reworking left some elements and fragments in subvertical orientations while being immediately in contact with horizontally oriented elements and fragments. The reworking also commingled bone elements from different individuals into close spatial contact with each other. Despite this evidence of reworking within the Puzzle Box Feature, many elements and body parts remain in complete articulation. In particular, there are areas where fragmented and reworked material is in direct contact with articulated body parts such as a complete hand and near-complete foot (*Figure 29*).

The selective reworking of this area is compatible with digging activity by hominins. A hominin digging into a deposit of sediment and articulated skeletal remains would exert localized disruption within the deposit. If digging was sustained to create a hole within the deposit, the disruption would be greatest near the surface, with a narrower area of disruption at the deepest part of the hole. Skeletal material at the deepest levels might remain unaffected; articulated elements that were not contacted by the digging implement would remain in place and undisturbed. Large structures like crania or long bones would be least likely to remain intact simply due to the area they occupy. If the hominin filled the hole with the excavated sediment, it would result in commingling of elements, displacement of elements at varied levels and angles, limited to the circumference of the hole. The data are compatible with this hypothesis.

Establishing the full sequence of reworking within this complex area is beyond the current evidence. A reasonable hypothesis is that the least disturbed of the skeletal remains may come from the last individual to be buried in this area. The partial infant skeleton that is strongly localized within the southeast part of the Puzzle Box area may represent the last episode of reworking. The placement of this skeleton would explain the fragmentation and displacement of fragments of the DH1 skeleton, the entry and vertical emplacement of the U.W. 101–1475 femur fragment, and the varied plunge angles in postcranial remains immediately above this area.

## Secondary cultural burial

Another form of cultural burial is the deposition of remains made when the skeletal elements are partially or completely disarticulated after decomposition of soft tissue (*Boulestin, 2012*; *Boulestin and Duday, 2006*). Skeletal remains in such a secondary burial may be in spatial arrangements that would be impossible or unlikely for the decomposition of a fleshed body covered in sediment. Secondary burials can sometimes be recognized by a different pattern of skeletal part representation than primary burials, often involving a winnowing of small elements such as hand and foot bones that are missed when collecting remains from their primary depositional situation (*Knüsel et al., 2022*). However, minimal criteria for burial do not easily distinguish this hypothesis from a primary burial in the presence of significant postdepositional reworking of a primary inhumation; the presence of disarticulations and intra-grave displacement does not necessarily mean the deposition is not a primary one (*Duday and Guillon, 2006*), simply that the original disposition of remains has been disturbed and distorted by biotic or abiotic taphonomic factors (*Bristow et al., 2011*). Digging a hole and covering remains reflects a cultural process whether these acts apply to primary or secondary cultural burials. Both activities represent mortuary acts with agency in those individuals who are manipulating the dead (*Leclerc, 1990*; *Knüsel, 2014*).

Evidence from all features considered here indicates postdepositional fracturing and reworking of skeletal material after burial. None of the evidence suggests perimortem fracturing prior to burial. Commingling of elements from different individuals, separation of fragments that conjoin at clean fractures, and spatial interruption of bodies might all result from secondary burial; however, these

may also be explained by postdepositional reworking of primary burials. Postdepositional diagenesis and resulting loss of low-density bone elements also can explain much separation of preserved elements of single individuals. In an isolated burial, the spatial disruption of a body and juxtaposition of parts from multiple individuals might be considered as evidence for secondary burial. In a setting with multiple bodies buried close to each other, with some disrupting previous burials, reworking may provide a better explanation for these observations.

The anatomical ordering of the skeletal material within the Hill Antechamber Feature is not suggestive of a secondary burial. Hand and foot elements are not underrepresented in this feature, and parts of a single skeleton appear to be anatomically ordered. The material at higher levels within the feature does manifest substantial compression and postdepositional fracturing, but none of this is suggestive of the emplacement of bones that had previously been deposited elsewhere. Little can be said about the situation of remains attributable to other individuals within the feature, such as the maxillary teeth of Individual 2. While it is possible that some of this material may represent a palimpsest of earlier burials prior to the emplacement of Individual 1, the evidence does not rule out the possibility that some of this material may have been brought into this location secondarily.

The situation of the skeletal remains in Dinaledi Feature 1 is indicative of the presence of connective tissue at the time remains were deposited in this position. The evidence includes anatomical positioning and subvertical orientation of ribs and articulation of vertebrae. At the same time, most of the excavated elements display a high degree of postdepositional fragmentation and are not in articulation. The mandible that is present in this feature is fragmented with part of it displaced toward the floor surface, suggestive of some reworking subsequent to deposition in this position. Still, the remains generally exhibit spatial positioning compatible with decompositional collapse of a body. Other aspects of the remains that might inform about primary versus secondary burial are not observable with the present evidence. For example, hand and foot remains are not strongly represented in the excavated material, but the data do not exclude that such elements may occur within the unexcavated portion of the feature. This evidence does not exclude the possibility of secondary positioning of a body during the process of decomposition, but does seem to exclude the hypothesis that remains were defleshed prior to burial in this location.

The Puzzle Box area was clearly the burial location of some whole bodies. The spatially contiguous infant skeletal remains with tiny elements such as vertebral laminae were not carried in disarticulated state into the site. The immature DH7 skeleton likewise does not exhibit consistency with secondary burial of disarticulated remains. Within this feature, as in the other features, the remains do not exhibit winnowing of small elements; in fact, hand and foot remains are among the most highly represented anatomical areas. While it is not possible to exclude that some bones may have been emplaced in this location after decomposition elsewhere, the evidence does not point to secondary burial as an explanation for the majority of the deposit.

One location within the subsystem does present evidence suggestive of the possible curation and movement of skeletal remains. The U.W. 110 locality deep within the fissure network has teeth and cranial fragments from a young child (*Brophy et al., 2021*). A skull or partial skull was evidently placed in this location where it subsequently fragmented. Without complete excavation of this difficult-to-access locality, it is not possible to establish whether additional remains are present. It is therefore not possible to rule out that this material is the result of secondary burial or funerary caching.

## Discussion

The results of this study provide additional evidence that *Homo naledi* was itself involved in the deposition of bodies in the Rising Star cave system (*Dirks et al., 2015*; *Randolph-Quinney et al., 2016*). Previous work on the context of the hominin remains and surrounding sediments largely considered large-scale questions including how sediment entered the subsystem, how the different sediment units formed, and the extent of movement or slumping of sediments within the subsystem (*Dirks et al., 2015*; *Dirks et al., 2017*; *Wiersma et al., 2020*; *Robbins et al., 2021*). This work developed a chronology for the subsystem and provided some data that constrained hypotheses about the entry of *H. naledi* remains. The current work provides evidence at a smaller scale, focused on particular excavation areas through the examination of *H. naledi* skeletal material and directly associated sediments. These two different scales of examination are relevant to different kinds of processes, and in a situation with the complexity of the Dinaledi Subsystem, it can be challenging to evaluate how the different

scales of information compare with each other. The evidence from three excavation areas exhibits commonalities that inform as to the deposition and burial history of *H. naledi* remains in these areas.

This evidence falsifies the hypothesis that the overall pattern of evidence can be explained by the introduction of bodies accumulated passively within the Dinaledi Subsystem with subsequent transport of bodies or partial bodies through the subsystem by gravity, as had previously been suggested (*Dirks et al., 2015*). The evidence described in this study includes cases that are not consistent with the decomposition of *H. naledi* bodies on the floor surface of the subsystem without rapid burial. In at least three locations in the Dinaledi Subsystem, bodies were encased in sediment shortly after their arrival in the cave system and prior to soft tissue decomposition. In these cases, the evidence rejects the hypothesis that bodies were covered in sediment slowly and gradually after death. Instead, these bodies were covered rapidly and selectively by sediment. The evidence from these cases also rejects the involvement of water, mud flow, or sediment slumping as mechanisms for the burial of skeletal material.

Here for the first time, we have considered the hypothesis that *Homo naledi* was directly involved in the burial process of bodies. This hypothesis explains many aspects of the data from these skeletal remains and sediments that were previously left unexplained. The cultural burial hypothesis can explain the evidence of selective reworking that we originally observed in our excavations of the Puzzle Box area. The hypothesis explains the preservation of articulations in each of the features that we describe here, as well as the matrix-supported nature of some articulated and preserved body parts. The hypothesis explains several otherwise problematic aspects of the sediments associated with the remains, including the different patterns of layering within and at the edge of the Hill Antechamber Feature, the truncation of a laminated orange silty-mudstone unit (LOSMU) layer and other sediment contrasts at the edges of Dinaledi Feature 1, the presence of undifferentiated and unlayered fill within all features with skeletal material, and the intermingling of remains with LOSMU and LORM clasts. Selective reworking of the sediments by hominins after burial provides a credible explanation for the pattern of fracturing and movement of remains within the Puzzle Box area, with subvertical orientation of elements directly adjacent to horizontal elements and articulated remains commingled with fragmented elements.

Beyond these delimited areas, the Dinaledi Subsystem evidence is more extensive and includes other processes besides hominin activity. For example, the downslope excavation area in the Hill Antechamber included a skeletal and dental assemblage that may be consistent with downslope erosional accumulation or accumulation by sediment slumping or trickling from higher elevation in this chamber. The evidence from this area contrasts strongly with the other subsurface areas with more substantial accumulations of *H. naledi* remains. The formation of flowstone units that date to periods later than the *H. naledi* fossil remains speaks to conditions in the subsystem after *H. naledi* activity. Some *H. naledi* fossils that were located on the floor surface at the time of discovery of the subsystem, along with the few faunal elements collected on the surface, likewise may reflect processes that followed long after the introduction of *H. naledi* into the cave (*Dirks et al., 2015*; *Dirks et al., 2017*).

The Dinaledi evidence is different in some respects from many other early examples of cultural burial. Of course, all sites are unique in certain ways, and there is no reason to assume that any particular ancient cultural practice must be identical to others at different times and places. The Rising Star case involves a hominin species with many anatomical differences from recent humans. It is therefore natural that different researchers may approach the data with different expectations and assumptions, placing higher value on some observations and greater skepticism on others. In the following sections, we examine the theoretical issues relevant to the interpretation of cultural burial in the past and consider how these apply to the current case.

## Hypothesis testing and parsimony

Researchers have diverse approaches toward hypothesis testing of mortuary activities by humans and other hominins. A few have advocated for the strong position that an archaeologist should never consider burial as a hypothesis until all possible alternative explanations have been rejected. For example, *Gargett et al., 1989* wrote: "It is not enough to say that humans *could have* produced a given deposit; it must be shown that nature could not" (*Gargett et al., 1989*: 161, emphasis in original). Recent researchers sometimes have suggested that all 'natural' (i.e. noncultural) explanations

must be taken as a null hypothesis, and cultural explanations should not be considered until all other hypotheses are rejected (e.g. *Martinón-Torres et al., 2024*).

One obvious drawback of this 'null hypothesis' approach is that it confuses two separate issues: first, whether there were biological limits that would prevent ancient individuals from carrying out mortuary behavior; and second, whether the physical evidence from a particular excavation was consistent with mortuary behavior. This confusion has led many researchers to demand a higher standard for Neandertal burial evidence than for recent human burials. (*Gargett et al., 1989*: 157) made this double standard explicit: acknowledging that in most cases the claim of purposeful burial was 'probably well-founded' but arguing that researchers should nonetheless treat instances from the Middle Paleolithic with greater skepticism. A second drawback of this 'null hypothesis' approach is that the credibility of alternative hypotheses is always subjective. A clever person may be able to imagine a series of noncultural events to account for any pattern of skeletal remains and context. As a series of natural accidents becomes longer and more contrived, the probability of the entire scenario is necessarily lower. Where should we draw the line between such convoluted 'natural' series of events and the much simpler hypothesis of hominin agency? The answer to that question is subjective; different researchers draw the line in different places.

A recent workshop on Neandertal burial evidence, including a large international group of experts, explicitly considered what should be the null hypothesis for cases of possible burial by Neandertals and other hominins (*Pomeroy et al., 2020b*). These authors emphasized that mortuary behavior occurs in other living species, and they did not recommend any null hypothesis for examination of Neandertal burial evidence. This represents an evidence-based understanding of nonhuman cultural behavior and that behavior of extinct hominins. As evidence of this advance in thinking, recent research studies covering burial evidence from Paleolithic contexts have not taken the approach that 'natural' deposition is a null hypothesis (e.g., *Maloney et al., 2022*; *Vandermeersch and Bar-Yosef, 2019*; *Kacki et al., 2020*; *Balzeau et al., 2020*; *Sparacello et al., 2021*; *Pomeroy et al., 2020a*; *Fewlass et al., 2023*).

An outlier among these recent studies has been previous work on *H. naledi*. Up to now, the *H. naledi* research has followed an approach in which cultural processes were considered only after first rejecting non-cultural processes. *Dirks et al., 2015*, including many of the present authors, did not discuss the intentional interment of remains even as a possible hypothesis and did not consider that *H. naledi* may have been an agent of reworking within the Dinaledi Chamber deposit. Instead, that study considered four 'natural' scenarios for the presence of *H. naledi* remains in the Dinaledi Chamber, tested each of them, and found that none could explain the evidence (*Dirks et al., 2015*; discussed further in Appendix). Upon rejecting these scenarios, the study presented a hypothesis of deliberate deposition without burial as an alternative that the data did not reject. Notably, even after accepting *H. naledi* as an accumulating agent, the study still considered only non-hominin sedimentation and transport processes within the Dinaledi Subsystem. Cultural burial was never considered.

That approach to the evidence may have seemed conservative, but the current study suggests that it was a mistake to assume that *H. naledi* was not itself involved in reworking the sediments. The work we report here illustrates that neither gravity and resulting sediment slumping, nor downslope movement of bodies on a talus, nor slow gradual sedimentation, nor any other 'natural' process previously hypothesized can account for the position and context of the *H. naledi* features in the Dinaledi subsystem. If we had considered the cultural burial hypothesis in our initial work, much confusion and some unnecessary work might have been avoided.

The present study follows a more balanced approach in which hominin agency at various stages of the depositional history is considered as one alternative among many possible processes that may affect hominin remains. This approach involves the critical examination of all relevant hypotheses and does not privilege any one favored hypothesis. This is a parsimony approach, recognizing that the most parsimonious hypothesis is the one that explains the entirety of the data with a unified explanation and a minimum of extraneous assumptions. One additional consideration in the parsimony approach is the consilience of multiple sets of observations. This study has examined three features in detail, combining geoarchaeological and skeletal evidence for each of them. Each of them presents different aspects of evidence, but together they must be compatible with a single system of explanation. While we have attempted to be exhaustive in considering realistic scenarios,

it is possible that future work will uncover additional alternative scenarios that we have not considered here.

## Possible variation in mortuary behavior

The finding that cultural burial was part of the overall pattern of evidence from the Rising Star cave system prompts us to evaluate the broader array of *H. naledi* remains for other evidence of mortuary activities. *Homo naledi* remains occur more widely in the cave system than the areas considered in this study, including some areas where no excavation has occurred (*Elliott et al., 2021*; *Brophy et al., 2021*). Evidence from these situations ranges from the LES1 partial skeleton from the Lesedi Chamber to the partial cranium in the U.W. 110 locality, as well as isolated bones in other locations. It will require further investigation to understand whether these varied situations may represent a single pattern of behavior or multiple isolated events.

The Hill Antechamber Feature, Dinaledi Feature 1, and Puzzle Box area are very similar in some respects. In particular, the depth of skeletal remains, pattern of disruption of surrounding sediment, degree of anatomical articulation of skeletal material, lack of intrusion of nonhominin faunal material, and preservation of elements is similar in each situation. However, the three settings are not identical. The number of individuals present is different across the three excavation settings, and there is no evidence of stereotypical body position or orientation. The evidence from the Puzzle Box feature suggests that *H. naledi* in this case reworked burials in later acts. Whether this may suggest a lack of attention to the locations of previous burials or a deliberate repeated interment is not evident.

The lack of stereotyping across these settings has parallels in other sites where multiple cultural burials of *H. sapiens* occur, including Qafzeh Cave and Skhūl Cave (*Tillier, 2022*). We must also consider the possibility that *H. naledi*, as a complex cultural hominin, may not have approached every mortuary event in any single stereotypical way. Future work may elaborate on this possibility. The presence of engravings (*Berger et al., 2023*) may also represent cultural activities within the space that *H. naledi* cultural burials took place.

## Additional criteria for cultural burial

The literature on burial evidence from Paleolithic sites provides diverse and sometimes contradictory criteria for identifying and studying burials. Recent authors who have reviewed the subject have reflected upon the contentious nature of definitions and criteria (*Courtaud and Duday, 2008*; *Pettitt, 2013*; *Zilhão, 2015*; *Stiner, 2017*; *Pomeroy et al., 2020b*; *Maureille, 2022*). In this study, we tested the burial hypothesis with three criteria: evidence for a hominin-dug hole, evidence of a body or body parts in the hole, and evidence that the body or body parts were rapidly covered by sediment. These criteria generally follow those suggested by *Pettitt, 2013* and either these or similar criteria have been part of diagnostic criteria for burial both in archaeological and forensic contexts. Going somewhat further, *Gargett et al., 1989*; *Gargett, 1999* asserted that the only unambiguous evidence of intentional burial is the formation of a distinct stratum containing the remains, demonstrably distinct from underlying and surrounding stratigraphic context. The formation of such a stratum is one possible result from the excavation of a hole followed by infill with body and sediments. However, *Pettitt, 2002* disagreed with this criterion, pointing out that a grave may be backfilled by the excavated sediments taken from the pit, and not distinct from overlying sediments.

The various contexts within the Dinaledi subsystem present several kinds of evidence for the presence of a distinct stratum containing the remains. The Puzzle Box area, Dinaledi Feature 1, and Hill Antechamber Feature each have a concentration of skeletal remains, including elements in anatomical articulation, within a volume that is differentiated from adjacent areas of the subsurface sediment. Within each area, the sediment is disordered and lacks structures associated with deposition over time, such as horizontal layering or particle sorting. Adjacent to each area are undisturbed floor sediments that exhibit visible layering roughly parallel to the floor surface.

While the three criteria considered in this study are basic to many instances of cultural burial, nevertheless, there are many cases accepted by archaeologists as evidence for cultural burial by Neanderthals or modern humans that lack one or more of the criteria. Few burials from Middle Paleolithic contexts present stratigraphic or other geoarchaeological evidence of a burial pit. As noted by *Pomeroy et al., 2020b*, a hominin that scoops a hole with hands or simple tools is unlikely to dig very deeply or to cut vertical edges. When a hominin backfills the hole using the excavated sediment, the

burial fill may present no geochemical contrast from surrounding sediment. In some cases, the presence of partially articulated skeletal remains has been assumed as *prima facie* evidence that a pit was present and the body covered prior to decomposition. In a survey of 32 Neanderthal individuals interpreted as burials, **Riel-Salvatore and Clark, 2001** noted 10 that are not accompanied by evidence or deduction of pits having been present, and out of 13 Middle Paleolithic-associated modern human burials, they noted 4 without evidence of pits.

Some cases that archaeologists have recognized as cultural burial by Neanderthals or early modern humans have been supported by criteria other than the ones used in this study. Such criteria include evidence for grave goods, evidence for differential preservation of hominin compared to faunal remains, evidence for special positioning of the body, and evidence that the place where remains were buried may have been special in some way. In this section, we discuss the ways in which such criteria may apply to the Dinaledi and Hill Antechamber evidence.

### Grave goods

Sometimes ancient hominins who buried bodies included special objects that they placed with the body at the time of burial, known as grave goods. When researchers recognize such objects in association with skeletal remains, they may interpret them as evidence of intention or symbolic purpose of a burial. Such evidence in some cases has led researchers to accept that cultural burial was practiced even where clear stratigraphic evidence of a pit is absent and where hominin remains may be highly degraded or disarticulated. That is, while a skeleton may be insufficient for some researchers to substantiate a cultural burial, grave goods in association with any skeletal remains may in some cases be sufficient. But grave goods are rarely found in cases of cultural burial older than around 40,000 years ago. In particular, among approximately 40 examples of cultural burial attributed to Neanderthals, only a handful have possible evidence of grave goods, and each of these is contentious (**Jaubert et al., 2022**). Whether an object was intended as an offering or accompaniment to a buried body is subject to interpretation. In some cultural settings, grave goods were stereotyped, distinctive, or abundant and therefore easily recognized, as for example in the Gravettian tradition of Europe (**d'Errico and Backwell, 2016**). Such evidence is lacking in many other cultural settings. Thus, grave goods may in some cases be sufficient for recognizing cultural burial, but they are not necessary evidence. **Pettitt, 2002** distinguished simple inhumation without grave goods from elaborated primary burial with grave goods or ritualized placement of the body.

The evidence from the Dinaledi Subsystem does not include unambiguous grave goods. Further analyses of the stone in the Hill Antechamber Feature are planned after synchrotron imaging reconstruction that will guide excavation. This stone is interesting in its placement in direct contact with the skeletal remains and its orientation, which shows that it was emplaced with sediments and supported subsequently. Apart from the evidence showing synchronous placement with fleshed remains, the current data do not test the possibility of intent.

### Contrasting preservation of faunal remains

One criterion commonly applied in studies of Paleolithic burial is the differential treatment and preservation of hominin remains compared with faunal remains. In several sites, archaeologists have noted that hominin remains are articulated and well-preserved while faunal remains are fragmented, weathered, or ravaged by carnivores or scavengers (**Rendu et al., 2014**). Such a contrast has particularly been noted in some cases where infant hominin skeletons were found. By contrast, some sites with articulated and well-preserved hominin skeletons also preserve articulated remains of other animals. Just as differential preservation has been used to support the hypothesis of cultural burial, non-differential preservation of faunal and hominin remains has been cited as a reason to be skeptical of cultural burial (**Dibble et al., 2015**).

This comparison cannot be made for the Dinaledi Chamber and Hill Antechamber material. Faunal material in these contexts is rare, and none is in stratigraphic context similar to the hominin remains (**Dirks et al., 2015**). The Lesedi Chamber has produced some faunal material, although none can be placed in the same stratigraphic context as the *H. naledi* remains in this chamber (**Hawks et al., 2017**). The LES1 skeleton stands out from the faunal remains in its high representation and surface preservation (**Hawks et al., 2017**). This difference may reflect a depositional history for the skeleton in which

it was protected by burial, while faunal material was not. However, evaluating this hypothesis requires the recovery of faunal material that can clearly be shown to be associated with the same sediments, which is beyond the information from the current sample.

The paucity of faunal material in the Dinaledi Subsystem itself may provide evidence of selective access to this part of the cave system by hominins (*Dirks et al., 2015*; *Randolph-Quinney et al., 2016*). Exploration and excavations of the Dinaledi Subsystem have continued to observe that non-hominin macrofauna is very rare, and this work has produced no evidence of non-hominin remains in stratigraphic association with *H. naledi*. The present sample of non-hominin macrofauna *Tyto alba* (*Kruger and Badenhorst, 2018*) and *Papio* (*Dirks et al., 2017*; *Elliott et al., 2021*), both represented by few elements. Only one element comes from a subsurface context with *H. naledi* present, and this area of the Hill Antechamber (S150W150) may be subject to downslope accumulation of reworked sediment as noted above. The lack of non-hominin macrofauna suggests that this part of the cave system was not an occupation area for *H. naledi* or for other species (*Dirks et al., 2015*). It contrasts with fossil deposits in many other caves that are more accessible from surface entrances.

## Conclusion

Mortuary evidence from the Dinaledi and Hill Antechamber exceeds the age of the earliest evidence of cultural burials by *H. sapiens* in Africa (*Martinón-Torres et al., 2021*) by as much 160,000 thousand years. Both are substantially more recent than the mortuary behavior by early Neandertals found at Sima de los Huesos, Spain, at ~430,000 years (*Sala et al., 2024*; *Arsuaga et al., 2014*). Neanderthal mortuary behavior is well documented at a number of sites from the later Middle and early Late Pleistocene (*Jaubert et al., 2022*). While evidence of primary cultural burial is not currently known earlier than the Rising Star evidence, a number of authors have suggested that evidence of other mortuary behavior goes back as far as *Australopithecus* (*Pettitt, 2013*). Altered behavior in the presence of dead conspecifics is found widely among various nonhuman primates and other mammals (*Pomeroy et al., 2020b*).

What does such behavior mean for *H. naledi*? Some authors have argued that mortuary behavior is unlikely for *H. naledi*, due to its small brain size (*Val, 2016*). The evidence compels us to conclude that this cultural behavior was not a simple function of brain size. We cannot at this time exclude that *H. naledi* may have been part of the ancestral makeup of humans, but its overall morphology suggests a more distant phylogenetic placement. Our common ancestor with *H. naledi* may have lived in the Early Pleistocene, well before the divergence of Neanderthals from African lineages that gave rise to modern humans (*Dembo et al., 2016*; *Argue et al., 2017*; *Caparros and Prat, 2021*; *Thackeray, 2015*). This raises the possibility that burial or other mortuary behavior may have arisen much earlier than present evidence for them, or that such behaviors evolved convergently in minds different from our own. Understanding such behaviors will require comparative study of all hominin lineages in which they occur.

Some archaeologists have suggested that the recognition of cultural burial by Neanderthals has deep implications for the evolution of modern human behavior, or 'behavioral modernity' (*Dibble et al., 2015*). The reasoning behind this suggestion is the idea that burial is one aspect of modern human culture, and the occurrence of burial in Neanderthals must mean that the common ancestors of modern people and Neanderthals themselves manifested modern human culture also. While we accept that the observation of cultural burial in Neanderthals and *H. naledi* is valuable evidence about their cultural practices, it is oversimplistic to assess burial or other cultural practices as 'traits' inherited from the common ancestor of these groups. Cultural behavior results from a complex interplay of social learning, demographic history, and brain development. The conditions for cultural burial may have arisen in Neanderthal populations multiple times, as they did in African ancestors of modern people, in early modern humans of the Levant, in *H. naledi*, and possibly in other hominins. There is no need to assume an

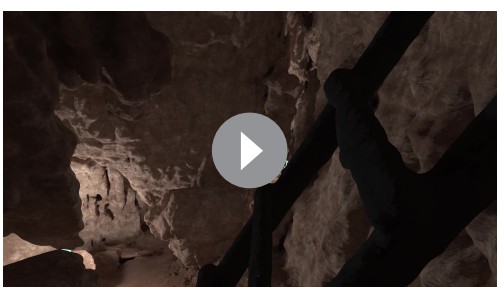

**Video 1.** Augmented virtual reality of the hill antechamber and the dinaledi chamber.
https://elifesciences.org/articles/89106/figures#video1

unbroken line of cultural practices connecting these groups. One conclusion based on the overall picture of cultural burial among these groups is that hominins became capable of responding to the deaths of individuals that they knew in varied culture-specific ways. Ethnographic and historical evidence shows that modern people have elaborated upon such cultural practices with an enormous variety of symbolic, artistic, and emotive expressions. What we can learn from this ethnohistorical evidence about the possible lives of earlier hominins is not yet clear. In the case of *H. naledi*, cultural burial is one piece of evidence about the cultural breadth of social relationships, joining other kinds of evidence that unite many members of the genus *Homo* (*Fuentes et al., 2023*).

## Materials and methods
### Fossil material and its location
The features and fossils described here were situated within naturally formed chambers of the Rising Star cave system (*Figure 1*). Within the broader system, the Dinaledi Subsystem represents a contiguous series of chambers and fissure passages including the Hill Antechamber and Dinaledi Chamber (*Figure 2*; *Elliott et al., 2021*). We also discuss material recovered from the Lesedi Chamber (Figure S6; *Hawks et al., 2017*) and a fourth locality in a remote passage of the Dinaledi Subsystem, designated as U.W. 110 (Figure S5; *Brophy et al., 2021*) in the University of the Witwatersrand's fossil locality catalogue system (*Zipfel and Berger, 2009*). An augmented virtual reality flythrough of the cave is provided as *Video 1* that allows a detailed examination of the Hill antechamber and Dinaledi Subsystem.

Images and notes for this work have either been published or are held by the University Curator of Collections at the University of the Witwatersrand, Johannesburg, South Africa. All sediments recovered from these excavations have been catalogued and preserved and are held onsite at the Rising Star site and are available for study. The fossil remains described here, and from all previous excavations are available for study by application to the Fossil Access Committee of the University of the Witwatersrand through the Curator of Collections. 3D shape files of more identifiable material may be downloaded at https://Morphosource.org and https://human-fossil-record.org/ under a Creative Commons License by attribution.

### Field methods
#### Hill Antechamber excavation
The Hill Antechamber (U.W. 107) is the portion of the Dinaledi Subsystem nearest to the present-day entrance access into the subsystem (*Elliott et al., 2021*). In September 2017, we established a datum for the subsystem near the northeast extreme of the Hill Antechamber and established a grid based on a local north extended throughout the subsystem (*Appendix 6—figure 2*). This local north diverges from magnetic north by 35 degrees NW; maps here are presented relative to magnetic north. The grid system was projected into the spaces with laser level and laser rangefinding equipment. The spatial location of each fossil fragment was recorded from measurements prior to recovery, with scaled photographs taken in situ. Photography for photogrammetry was carried out approximately daily and sometimes at shorter intervals. Metashape 1.8.1 Standard (Agisoft) was used to generate 3D models of the excavation from these series of photographs. The resulting 3D models were correlated with 3D point cloud data generated by a Faro laser instrument.

Excavation of the Hill Antechamber units followed methodology previously described for work in the Rising Star cave system (*Berger et al., 2015*; *Hawks et al., 2017*). No metal instruments were used for excavation. On the highly sloped units in the Hill Antechamber, we proceeded with a terraced approach to bring the excavation surface down in horizontal steps of 5 cm. During the excavation of the unit at N50W100, we encountered a substantial area of fragmented and powdery bone material beginning at a depth of 5 cm below surface, near the north end of the unit (*Appendix 6—figure 16*). Leaving this material in place, we continued to excavate within the designated unit to reveal its extent. As we exposed this material further toward the south, we found that it included some more complete fossil elements and extended across more than half the horizontal area of the 50 cm x 50 cm unit. At this stage, we decided to delineate the edge of this feature and excavate more deeply around it to form a pedestal that could be removed *en bloc* from

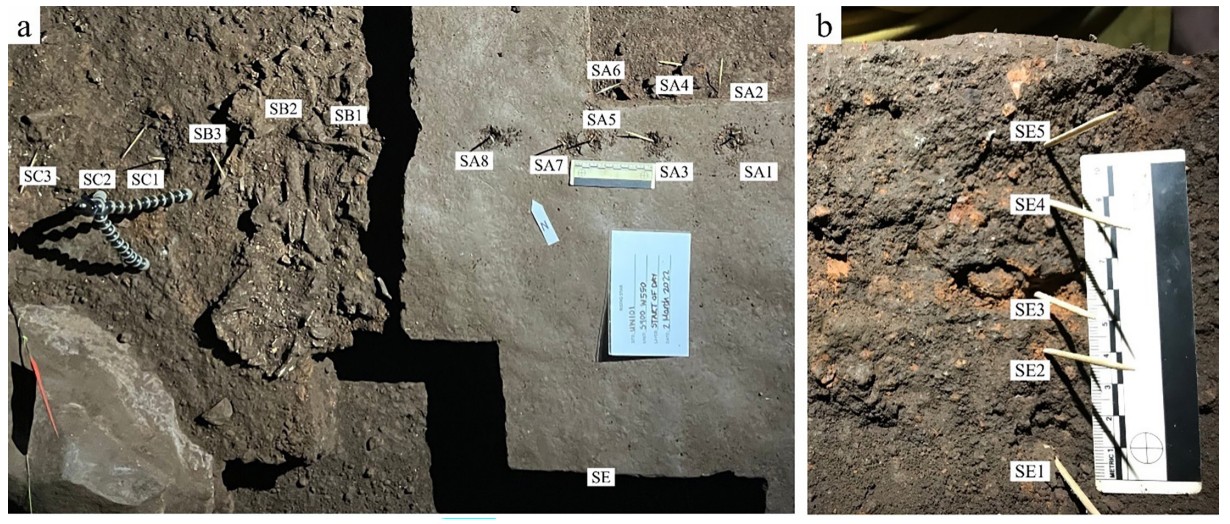

**Figure 30.** In situ sampling localities around Feature 1 on the Dinaledi Chamber floor. (**a**) Top view of the Dinaledi floor showing the exposed Feature 1 and the group A (SA) samples from areas outside any feature, group B (SB) samples from inside Feature 1, and group C (SC) samples between Features 1 and 2. (**b**) Vertical wall where profile group E (SE) samples were collected.

the cave system for further work in the laboratory (***Appendix 6—figure 17***). The material extended to the east and south edges of the N50W100 square, and so we opened half of the neighboring grid squares at N50W50 and S50W100, and the northwest quadrant of S50W50 to fully expose the boundaries of the feature. By the end of excavation in March 2018, an area of 75 cm x 75 cm was excavated down to a horizontal level 40.5 cm below datum, leaving the bone-containing feature as a pedestal. Only a small number of fossil elements and fragments occurred within the surrounding excavated area.

All material and people entering and exiting the Dinaledi Subsystem must pass through the ~18 cm constraint of the subsystem entry passage, which places a severe size limit on any excavated blocks of material. The overall size of the Hill Antechamber Feature was incompatible with removal from the subsystem in a single block. We therefore planned for a segmentation of the feature into a minimal number of pieces. The contour of the surface of the feature presented an upper surface dense with bone elements and bone material through most of the western two-thirds, while the eastern third of the feature lacked any bone at this level and presented deeper, less uniform bone material (***Appendix 6—figure 18***). This suggested a division of the feature to separate the eastern portion would be most appropriate. We applied a plaster jacket to the east wall of the feature and allowed it to solidify. We then excavated into the feature gradually to separate this portion from the overall mass, using toothpicks to confine its lower and western boundaries. Once separated, this block (U.W. 101–2074) measured 9 by 23 by 9 cm. We followed a similar approach to remove a second somewhat larger block that partially underlay the first. The resulting block, U.W. 101–2075, measured 21 by 26 by 11.5 cm. Finally, we jacketed the top and sides of the remaining mass of the feature (***Appendix 6—figure 19*** jacketed block). We used two thin metal sheets to separate the feature from the underlying sediment, then inverted it, applied a rigid plastic sheet to the bottom surface, and covered this bottom with plaster. The resulting mass, designated as U.W. 101–2076*, measured 40 cm by 49 cm by ~18 cm at its maximum dimensions. We wrapped all plaster-jacketed masses in bubble wrap and additional padding, placing them in waterproof cave haul bags. The team brought them out of the cave system without incident (***Appendix 6—figure 20***).

## Dinaledi Chamber excavation

The Dinaledi Chamber is the location of our excavations from 2013 to 2016. During the course of these excavations, our team collected skeletal material from across the present-day surface of the chamber

and opened an excavation unit that finally reached 80 cm by 80 cm to a depth of 20 cm, and a small sondage to a total depth of 80 cm at its center (*Figure 4*). This excavation yielded abundant skeletal remains of *H. naledi*. In November 2018, our team initiated renewed investigation of the Dinaledi Chamber. We opened two new 50 cm by 50 cm excavation units within the Dinaledi Chamber, flanking the previous excavation area on the northwest corner (S950W550) and southeast corner (S1050W500). These areas were selected to evaluate several questions: (1) whether other portions of the floor of the Dinaledi Chamber have similar subsurface concentrations of *H. naledi* skeletal material as found in the previous excavation unit; (2) whether subsurface skeletal material bears any relation to remains that were recovered from the present-day surface; and (3) to further understand the stratigraphic situation in the chamber. Excavation and recording methods applied in this chamber were the same as those used in the Hill Antechamber.

## Geochemistry and particle-size distribution of sediment samples

### Materials

Samples for studying the sediments around Feature 1 were obtained from (1) the sediments collected during the excavations that uncovered Feature 1 and 2, and (2) the sediments still in situ within and around Feature 1 (*Figure 30*). A total of 34 samples denoted DF group samples were subsampled from bags of sediments stored after excavation above the burial Features 1 and 2. Each bag of sediment corresponds to a particular area and depth level of the gridded excavation pit (*Figure 11*). These stored sediments were sieved for recovery of fossils after excavation and thus were already well mixed at the time of subsampling for geochemical studies of the sediments. For the in situ samples, a total of 22 samples were collected from within and around Feature 1 on the Dinaledi floor (*Figure 30*). These samples were categorized into four groups: SB, 3 samples of sediment from within Feature 1; SA, 8 samples of sediment to the east of Feature 1 on a floor that has not yet been excavated and from inside an excavated area; SC, 4 samples of sediment between Features 1 and 2 at the same level of the features (to the west of Feature 1); and SE, 5 samples of sediment from a vertical profile south of Feature 1 (*Figure 30*). A single sample (FS2280) of UMCB from the southern end area without fossils in the Puzzle Box excavation was added for comparison with the sediments in and around Feature 1.

### Method for particle-size distribution

The grain sizes of the sediments were assessed by carrying out particle-size distribution (PSD) analysis using the Microtrac S3500 laser-diffraction particle size analyzer at the Department of Metallurgy at UJ. The samples were pretreated by agitation in water using ultrasound bath treatment, and complete de-agglomeration may not have been achieved. The obtained results were reduced using GRADI-STAT, the grain size and statistics package (*Blott and Pye, 2001*), and selecting the Folk and Ward Method to obtain the mean grain size, sorting, skewness, and kurtosis.

## Method for geochemistry

The quantification of the major oxide bulk chemistry and the trace elements (including rare earth elements, REEs) chemistry was done using X-ray fluorescence (XRF) and inductively coupled plasma mass spectrometry (ICPMS), respectively, on finely milled (<75 µm) sediments. Both were carried out at the Spectrum Analytical Facility of the University of Johannesburg (UJ). XRF analysis was carried out as follows: Approximately 1 g of each sample was dried at 105 °C for a duration exceeding 24 hr. The dried samples were ignited in glazed porcelain crucibles in air for 30 min, with the temperature reaching 930 °C. Thereafter, borate fusion discs were prepared in Pt-Au crucibles and casting dishes from mixtures of ca. 0.7 g of the ignited sample material, 0.1 g $LiNO_3$ and 6 g '50/50 flux': 49.75 mass % lithium tetraborate ($Li_2B_4O_7$), 49.75 mass % lithium metaborate ($LiBO_2$), 0.5 mass % LiBr (lithium bromide; *Watanabe, 2015*). These mixtures were fused at 1050 °C in an electric fusion machine (Claisse TheOx). The discs had a diameter of 30 mm and were about 4 mm thick and were analyzed with a Malvern Panalytical MagiX PRO wavelength dispersive X-ray fluorescence spectrometer (WDXRFS). Instrument parameters and conditions are provided in *Supplementary file 1*. The analyses were calibrated with mixtures of pure chemicals (*Bennett and Oliver, 1992*) and with the certified reference materials AN-G, BE-N, FeR-2 and 3, IF-G, JSy-1, Nod-A-1 and SARM 1, 2, 5, 6, 11,

12, 16, 17, 32, 39, 40, 44, and 47 – more than 50 fusion discs in total (*Abbey et al., 1983*; *Govindaraju, 1995*). The following interferences *ij* of *j* on *i* were corrected in the SuperQ software of the MagiX PRO: AlBr, MgAl, MgBr, BaTi, TiBa, VBa, VTi, CrBa, CrV, and MnCr. Matrix corrections used a fundamental parameters model in SuperQ. LLDs during the calibration varied between about 40 and 250 mg/kg at 95% confidence level depending on the element. Limits of determinations in unknowns were taken as 0.05 mass %, and lower values were replaced with a dash. Measurements of unknowns included samples to correct for instrumental drift since the last calibration. Results of XRF measurements on fusion discs of ignited material were recalculated to refer to the dried samples (using the LOI that had been measured gravimetrically).

The concentrations of trace elements were determined using inductively coupled plasma mass spectrometry (ICP-MS). For this analysis, ca. 0.1 g of finely milled sample was completely dissolved with a mixture consisting of 4 mL hydrofluoric acid (HF) and 2 mL nitric acid ($HNO_3$) on a hot plate with heating up to 140 °C at atmospheric pressure for 48 hr. Subsequently, the $HF/HNO_3$ mixture was evaporated to near dryness, and the samples were treated concentrated with 1 mL $HNO_3$ three times, each time evaporating the mixture to near dryness. To dissolve any precipitates formed during the digestion process, the samples were taken up in 10 mL of 20% $HNO_3$ and placed on a hot plate overnight at 120 °C. The cooled samples were diluted to 50 mL using Milli-Q $H_2O$ to yield a solution in ca. 4% $HNO_3$. Measurements were carried out using a Perkin-Elmer NexION 300D instrument and instrument parameters are provided in *Appendix 6—table 4*. Duplicate analyses of samples were not performed; however, the quality assurance of the analytical process was maintained through the utilization of the standard reference material NIST SRM 2709 a, with recovery rates reported between 95% and 97%. Detection limits for the analysis were established at three times the standard deviation of measurements obtained from reagent blanks.

## Fossil preparation and analysis

Skeletal and dental material included in this study were recovered over the course of several expeditions ranging in date from 2013–2018. The recovery and laboratory preparation of fossil material from work in 2013–2017 has been presented together with published descriptions and analyses of those fossil remains (*Berger et al., 2015*; *Dirks et al., 2015*; *Hawks et al., 2017*; *Brophy et al., 2021*). Briefly, the protocol applied to skeletal material, dental material, and bulk sediments collected from the Rising Star cave system is designed to conserve the fossil material in curation as well as recover all spatial evidence from the excavation context. Fossil specimens are documented prior to collection using both traditional measurement methods, photography, and photogrammetry. After collection and cataloguing, fossil fragments are taken to a laboratory where they are allowed to remain in ambient temperature and humidity conditions with adequate ventilation until excess moisture from their burial context can gradually evaporate from the material. After drying in this way, adhering sediment is brushed from the surface, and where necessary, a light application of acetone is used as a solvent to assist in removing adhering sediment. All specimens are identified to element and conjoins between refitting fragments are noted. Most fossil elements enter curation at this stage without further preparation. In some cases, it is desirable for curation purposes to affix conjoining fragments to each other, and in these cases, a 50% solution of B-72 Paralloid in acetone is used as an adhesive to affix fragments to each other. In a few cases, it is necessary for curation and study to apply a consolidant to facilitate handling and prevent damage to the fossil material. In these cases, a 10% solution of B-72 Paralloid in acetone is applied to the surface of the fossils to act as a consolidant. Bulk sediment samples from the excavations are allowed to gradually dry in laboratory conditions with ambient temperature and humidity and after drying are sieved with a 1 mm mesh to recover small fossil fragments and elements. All bulk sediment samples are catalogued and stored after sieving.

## Hill Antechamber

Skeletal and dental material described in this study was collected from excavations that took place in 2017 and 2018. The fragments excavated from units in the Hill Antechamber were prepared using the methods described above. The surfaces of all fossil fragments identifiable to element were scanned using an Einscan-SP (Shining 3D) structured light scanner.

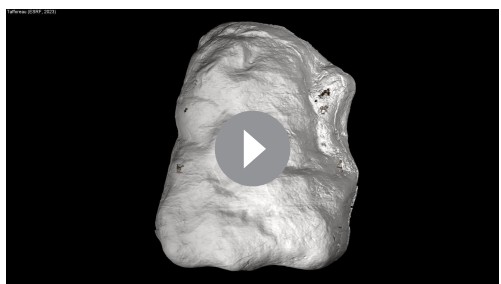

**Video 2.** HAA1 visualization movie.
https://elifesciences.org/articles/89106/figures#video2

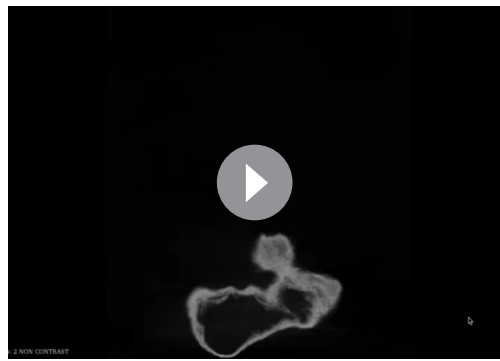

**Video 3.** Hill Antechamber Block movie.
https://elifesciences.org/articles/89106/figures#video3

Upon exposure of skeletal material from the excavation unit near the top of the sloping floor of the Hill Antechamber, it was immediately evident that the fossil material was highly fragile and had already experienced significant deterioration. We therefore extracted the entire feature in three parts by exposing the sediments surrounding its edges and then encasing all contents in three plaster jackets, one large and two smaller encasements. This was done so that the resulting sections would be small enough to be brought through the narrow confines of the Chute during ascent from the Hill Antechamber. The material was designated for CT scanning within the plaster jackets as described below. At present, the material remains in these plaster jackets in order to protect them and allow the application of technology to examine them in situ.

### Dinaledi Chamber

The fossil material recovered during excavation in 2018 was processed using an adapted preparation protocol intended to minimize the loss of possible biochemical or geochemical evidence that may persist on surfaces of the remains. All fragments were allowed to gradually dry after excavation and documentation as described above. After this, adhering sediment was removed with a dry brush, collected, and stored together with each fragment. The fragments were identified without further cleaning or consolidation. Each fragment was scanned using an Einscan-SP (Shining 3D) structured light scanner.

### CT scanning and segmentation methods

In the lab, we prioritized the plaster-jacketed masses for imaging. The smaller jacketed masses, U.W. 101–2074 and U.W. 101–2075, were suitable for imaging with the Nikon Metrology XTH 225/320 microtomography (microCT) scanner housed at the Evolutionary Studies Institute of the University of the Witwatersrand. These were scanned at a voxel size of 72 μm. This instrument was not suitable for scanning the larger jacketed mass, U.W. 101–2076, and so we obtained a clinical CT from a Siemens Somatom Definition AS 40 (Erlangen, Germany) with larger gantry size in the Charlotte Maxeke Johannesburg Academic Hospital (CMJAH), with a 0.5 mm voxel size (*Video 2*). We segmented these models with a combination of manual and density threshold segmentation strategies. All identifiable elements after segmentation were incorporated into a three-dimensional model of the feature.

A possible stone artifact was recognized on the clinical CT (*Appendix 6—figure 21*; Appendix 3; *Video 3*). It was then scanned as a focused target at the European Synchrotron Radiation Facility (ESRF) in Grenoble, France. This scan was performed using propagation phase-contrast on the beam-line BM18, using an isotropic voxel size of 16.22 μm and a propagation distance of 6 m. The average detected energy was 156 keV, obtained by filtering the white beam of the BM18 tripole wiggler with 2.62 mm of molybdenum. We used an indirect X-ray detector based on a LuAG:Ce scintillator of 2 mm thickness, coupled to a PCO edge 4.2 CLHS camera through a Hasselblad 120 mm macro objective.

## Acknowledgements

Permits to conduct research in the Rising Star Cave system are provided by the South African Heritage Resource Agency (LRB). Permission to work in the Rising Star cave is given by the LRB Foundation for Research and Exploration. We would like to thank the University of the Witwatersrand's Primate Fossil Access Committee for allowing access to the original material for study and the University Curator and Assistant Curator of Collections for assisting with logistics in studying the material. We would like to thank the Department of Diagnostic Radiology at the Charlotte Maxeke Academic Hospital for allowing access to their clinical CT scanner. We acknowledge the European Synchrotron Research Facility for providing access to facilities including the BM18 beamline. We would like to thank Matthew Caruana for discussions related to the stone object associated with the Hill Antechamber burial. We would like to thank the Spectrum Analytical Facility and the Department of Metallurgy at the University of Johannesburg for the analytical work on the sediments. Christian Reinke is specifically thanked for carrying out the XRF measurements (TM). The authors would like to acknowledge the funders of the various expeditions that recovered the fossil material and information, including 3d scanning and the production of AVR material described in this paper including the National Geographic Society (LRB, CJ, KB), the Lyda Hill Foundation (LRB) and the National Research Foundation of South Africa (LRB). Laboratory work, field work and travel was funded by the National Geographic Society (LRB, ME, AK, CJ, KB), the Lyda Hill Foundation (LRB, ME, AK) the Centre for Excellence in PalaeoSciences at the University of the Witwatersrand (now GENUS) (AK), the Fulbright Scholar Program (JH), John Templeton Foundation (LRB, JH) and National Research Foundation of South Africa (TM), University of Johannesburg Research Council (TM), and the Spectrum Analytical Facility at University of Johannesburg (TM).

## Additional information

### Competing interests

Angharad Brewer Gillham: Employee of Frontiers Media Limited. The other authors declare that no competing interests exist.

### Funding

| Funder | Grant reference number | Author |
| --- | --- | --- |
| The National Geographic Society | Explorer at Large | Lee R Berger |
| The Lyda Hill Foundation | | Lee R Berger<br>Ashley Krüger<br>Marina Elliott |
| The National Geographic Society | | Ashley Krüger<br>Marina Elliott<br>Kenneth Broad<br>Corey Jaskolski |
| National Research Foundation | | Lee R Berger<br>Tebogo Vincent Makhubela |
| Genus | | Ashley Krüger |
| Fulbright Association | | John Hawks |
| University of Johannesburg | | Tebogo Vincent Makhubela |
| Spectrum Analytical Facility, University of Johannesburg | | Tebogo Vincent Makhubela |

The funders had no role in study design, data collection and interpretation, or the decision to submit the work for publication.

## Author contributions
Lee R Berger, Conceptualization, Formal analysis, Supervision, Funding acquisition, Investigation, Methodology, Writing – original draft, Project administration, Writing – review and editing; Tebogo Vincent Makhubela, Formal analysis, Investigation, Methodology, Writing – original draft, Writing – review and editing, Funding acquisition; Keneiloe Molopyane, Marina Elliott, Becca Peixotto, Paul Tafforeau, Zubair Jinnah, Paul HGM Dirks, Investigation, Writing – original draft; Ashley Krüger, Investigation, Visualization, Methodology, Writing – original draft; Patrick Randolph-Quinney, Investigation, Visualization, Methodology, Writing – original draft, Writing – review and editing; Agustín Fuentes, Investigation, Writing – original draft, Writing – review and editing; Vincent Beyrand, Kathleen Dollman, Angharad Brewer Gillham, Juliet Brophy, Gideon Chinamatira, Elen Feuerriegel, Alia Gurtov, Nompumelelo Hlophe, Lindsay Hunter, Rick Hunter, Kudakwashe Jakata, Hannah Morris, Ellie Pryor, Maropeng Mpete, Mathabela Tsikoane, Steven Tucker, Kerryn Warren, Investigation; Kenneth Broad, Corey Jaskolski, Investigation, Visualization; Eric M Roberts, Jacqueline S Smilg, Colin D Wren, Investigation, Visualization, Writing – original draft; Dirk Van Rooyen, Investigation, Visualization, Methodology; Marc Kissel, Penny Spikins, Validation, Methodology, Writing – original draft; John Hawks, Conceptualization, Resources, Investigation, Visualization, Methodology, Writing – original draft, Project administration, Writing – review and editing

## Author ORCIDs
Lee R Berger ⓘ https://orcid.org/0000-0002-0367-7629
Tebogo Vincent Makhubela ⓘ https://orcid.org/0000-0002-6797-8359
Patrick Randolph-Quinney ⓘ https://orcid.org/0000-0003-0694-5868
Agustín Fuentes ⓘ https://orcid.org/0000-0003-0955-8214
Paul Tafforeau ⓘ https://orcid.org/0000-0002-5962-1683
Kathleen Dollman ⓘ https://orcid.org/0000-0002-5468-4896
Paul HGM Dirks ⓘ https://orcid.org/0000-0002-1582-1405
Marc Kissel ⓘ https://orcid.org/0000-0002-4004-1996
John Hawks ⓘ https://orcid.org/0000-0003-3187-3755

Reviewer #1 (Public review): https://doi.org/10.7554/eLife.89106.3.sa1
Reviewer #2 (Public review): https://doi.org/10.7554/eLife.89106.3.sa2
Author response https://doi.org/10.7554/eLife.89106.3.sa3

# Additional files

## Supplementary files
Supplementary file 1. Trace element and rare earth elements (REEs) results.

## Data availability
The original fossil materials and high-resolution scans are curated at the University of the Witwatersrand. All materials and scans are available for study through application to the Primate Fossil Access Committee of the University of the Witwatersrand. 3D shape files and images of HAA1 are available via Creative Commons License by Attribution on https://morphosource.org/. Videos and Augmented Virtual Reality data are published under Creative Commons License by Attribution on Figshare, with links provided in manuscript.

The following datasets were generated:

| Author(s) | Year | Dataset title | Dataset URL | Database and Identifier |
|---|---|---|---|---|
| Hawks J, Berger LR | 2025 | Hill Antechamber Dinaledi AVR movie.mp4 - Supporting material for *Homo naledi* burial | https://doi.org/10.6084/m9.figshare.29328911.v1 | figshare, 10.6084/m9.figshare.29328911.v1 |
| Hawks J, Berger LR | 2025 | *Homo-naledi*_tool-shaped_stone_in_burial_HD_tagged.mp4 - Supporting material for *Homo naledi* burial | https://doi.org/10.6084/m9.figshare.29329025.v1 | figshare, 10.6084/m9.figshare.29329025.v1 |

| Author(s) | Year | Dataset title | Dataset URL | Database and Identifier |
|---|---|---|---|---|
| Hawks J, Berger LR | 2025 | Hill Antechamber block visualization.mov - Supporting material for *Homo naledi* burial | https://doi.org/10.6084/m9.figshare.29328881.v1 | figshare, 10.6084/m9.figshare.29328881.v1 |

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

# Appendix 1

## Description of the sedimentary deposits of the Dinaledi Subsystem

The sedimentary deposits within the Dinaledi Chamber and Hill Antechamber have been categorized within three units defined on lithological but not necessarily chronological bases (*Dirks et al., 2015*; *Dirks et al., 2017*, *Wiersma et al., 2020*). An initial description was provided by *Dirks et al., 2015* and this description was revised by *Dirks et al., 2017*; *Wiersma et al., 2020* presented additional data and a hypothesis for the processes of autobrecciation of sediment deposits within the subsystem. This section reviews these previous descriptions so that readers of the current study have the relevant sediment descriptions available.

**Unit 1** consists of unlithified, horizontally laminated orange-red mudstone, abbreviated as LORM. Deposits of this unit formed when water laden with mud filtered into the cave chamber and deposited material that was suspended within it. The content of Unit 1 facies varies in grain size and composition, with some sandy lenses and inclusion of microfaunal remains. Outcrops of Unit 1 are localized within fissure passages extending from the chamber and on chert ledges on chamber walls. Unit 1 includes the oldest sedimentary deposits in the chamber as inferred from the presence of Unit 1 clasts within Unit 2 and Unit 3 sediments. But outcrops of Unit 1 on chert ledges and in fissure passages were deposited later and are still being deposited today, so that this unit is time-transgressive relative to the other sedimentary units in the subsystem (*Dirks et al., 2017*; *Wiersma et al., 2020*; see *Appendix 6—figure 1*).

Unit 2 and Unit 3 both include clasts of Unit 1 laminated orange-red mudstone (LORM) embedded within a brown clay matrix. These clasts are angular or subangular, they vary in size but are typically less than 3 cm in diameter, and they do not retain any orientation from their original deposition, with the originally-horizontal laminations now at random orientations.

**Unit 2** includes dolomite and chert fragments and is found only beneath a flowstone sheet near the entrance of the Hill Antechamber *Dirks et al., 2017*; the oldest part of this flowstone sheet (Flowstone 1 a) has reversed magnetic polarity and is older than 780 ka, which is a minimum age for these Unit 2 sediments.

**Unit 3** outcrops across the remainder of the floor of both the Hill Antechamber and Dinaledi Chamber. We have distinguished two sub-units of Unit 3: the lower **sub-unit 3** a which lacks *H. naledi* fossil material, and upper **sub-unit 3b,** which contains *H. naledi* fossils. The earliest known incidence of *H. naledi* in sub-unit 3b sediments is constrained at less than 335 ka by direct dating of *H. naledi* teeth (*Dirks et al., 2017*). A direct-dated baboon tooth 35–40 cm below *H. naledi* material suggests a maximum age for this part of sub-unit 3 a of 723±181 ka (*Dirks et al., 2017*). These dates demonstrate that the formation of Unit 2 and Unit 3 sediments was neither rapid nor simultaneous with the arrival and emplacement of *H. naledi* remains in sub-unit 3b sediments.

We have previously reported on the composition and micromorphology of sub-unit 3b sediments from two areas of the Dinaledi Chamber. This work included micromorphological and bulk sediment samples from a surface surrounding *H. naledi* remains and from within the 2013 excavation unit directly in contact with subsurface *H. naledi* remains (*Dirks et al., 2015*). We summarize these results briefly here. These samples exhibit the same grain size and shape distributions, predominantly composed of clay fragments with angular shapes, in random orientations without evident bias or preference to horizontal. The micromorphology and composition evident in these samples reflect the formation history of sub-unit 3b deposits that are in direct contact with *H. naledi* remains. These samples contrast strongly with sediments of the nearby Dragon's Back Chamber, which have a high input of detrital quartz and stony clasts from outside the cave system. Unlike the Dragon's Back Chamber sediments, sub-unit 3b sediments in the Dinaledi Chamber have little quartz content and are dominated by clay minerals and mica, with a texture composed of angular clay fragments and laminated orange-red mud (LORM) clasts (*Dirks et al., 2015*; *Wiersma et al., 2020*). These LORM clasts are remnants of Unit 1 sediments that were reworked during the formation of sub-unit 3 a sediments. The laminations within LORM clasts in sub-unit 3b are oriented randomly, and this orientation reflects the origin of sub-unit 3b from weathering and reworking of Unit 1 and sub-unit 3 a sediments. The micromorphology of sub-unit 3b samples demonstrates that the formation of this sub-unit occurred under low energy conditions with minimal lateral transport of material and without sufficient water saturation to dissolve LORM clasts or to remove sharp angles from them (*Appendix 6—figure 2*).

## Mud

Previous work has employed the geological terms 'mud', 'mudstone', and 'mud-clasts' in the description of sediments within the Dinaledi Subsystem (*Dirks et al., 2015*; *Dirks et al., 2017*; *Wiersma et al., 2020*). These terms have been misunderstood by some commentators, who have assumed that the term 'mud' must imply that water actively flowed through the Dinaledi Chamber at the time of deposition of *H. naledi* remains. This misconception about the nature of the Dinaledi Chamber deposits can be avoided by differentiating the different units and their formation history.

The laminations in Unit 1 sediments formed when water carried fine sediment into the subsystem, some of which percolated through cracks and deposited material on chert ledges and some in fissure passages. These laminations resulted from successive introductions of water-saturated sediments onto surfaces; at the time of their formation, the laminations were horizontal or parallel to underlying surfaces such as chert ledges (*Wiersma et al., 2020*). Unit 1 sediments remain in these places today. No *H. naledi* or other macrofaunal fossil material has been found in Unit 1 sediments. All *H. naledi* material occurs within Unit 3 sediments, interpreted as having formed by autobrecciation through the weathering and reworking of Unit 1 (*Dirks et al., 2015*; *Wiersma et al., 2020*). Where we excavated Unit 3 deposits and where they are exposed on the surface, they consist of an unconsolidated, loosely-packed clay matrix containing clasts and frequent air voids. We have divided Unit 3 into two sub-units: sub-unit 3b contains *H. naledi* remains while sub-unit 3 a lacks them. Both contain clasts derived from Unit 1, and these clasts are angular in outline and retain their laminations, but the laminations in different clasts are not aligned with each other or with a horizontal plane. In micromorphological thin sections, the LORM clasts appear at random orientations (*Dirks et al., 2015*). The micromorphology of sub-unit 3b sediments is inconsistent with flowing water, high-energy transport, or saturation sufficient to round the soft angular clasts or dissolve them. Sub-unit 3b sediments contain moisture: the air humidity within the cave averages 97–99%, there are drip points at various places with some speleothem deposition, and the *H. naledi* skeletal material contains moisture and is fragile. The use of the term 'mud' does not denote a formation history with flowing water, and the sedimentology of sub-unit 3b sediments rules out flowing water as a cause for their formation (Appendix figure S3).

## Reworking and autobrecciation

Reworking and autobrecciationPrevious work (*Dirks et al., 2015*; *Wiersma et al., 2020*) discusses the reworking and autobrecciation of sediments in the Dinaledi subsystem. The mud-clast breccia of Unit 2 and sub-unit 3 a in the Dinaledi Subsystem formed as a result of weathering, erosional breakdown, and reworking of Unit 1 sediments, resulting in unlithified breccia units containing clasts derived from Unit 1 sediments. This breakdown was a result of several processes. The composition of sediments includes clay minerals that are susceptible to swelling and contraction with greater or lesser moisture content. Fluctuations in moisture in the cave system such as those accompanying seasonal or longer cycles in the environment external to the cave resulted in the relative desiccation of parts of Unit 1 sediments and the formation of fine cracks (*Wiersma et al., 2020*). Such fine cracks in some areas led to infiltration of the Unit 1 sediments by calcite and aragonite, which is visible in some parts of the cave system today. Microbial processes and oxidation of iron and manganese also were part of this diagenesis of Unit 1 sediments. The area of the Hill Antechamber exhibits a substantial (~30 degree) floor slope today, and cycles of humidity with expansion and contraction might lead to some horizontal displacement of sediments on such a slope (*Dirks et al., 2015*; *Wiersma et al., 2020*).

In previous work, *Dirks et al., 2015* suggested that such cycles of humidity may have continued after the deposition of *H. naledi* remains and may have contributed to the fragmentation of *H. naledi* skeletal material, explaining in part the post-depositional fracturing of the material. In that work, it was noted that some skeletal elements were stained by mineral precipitates forming 'tide marks' that are recognizable on anatomically contiguous fragments after refitting. Some of these elements were further fragmented after the tide marks formed, and some of those fragments shifted in position. In addition, some skeletal elements were uncovered in positions that could not have occurred on a horizontal surface or within a shallow depression, including near-vertical orientations of long bone shafts (*Dirks et al., 2015*).

The evidence of post-depositional fragmentation and non-horizontal orientations appeared to contrast with other elements and skeletal regions that were deposited as whole limbs retaining

articulations of elements. These aspects of the deposit are not consistent with the deposition of bone on a near-horizontal floor surface followed by slow sedimentation. As we noted, these spatial properties of the sediment and bone must have resulted from some kind of highly selective reworking of the sub-unit 3b (*Dirks et al., 2015*; *Dirks et al., 2016*). Some *H. naledi* bones show evidence of invertebrate activity, and previous work has suggested the possibility of some bioturbation by these agents in the sediments (*Wiersma et al., 2020*).

The excavations that we report here have given us some additional information about the nature of reworking in Unit 3 sediments. Our excavation units in the Hill Antechamber include areas with a higher density of Unit 1-derived clasts, enough to form rough layers parallel to the sloping floor surface. These layers are interrupted by the intrusion of skeletal material. In the Dinaledi Chamber, our excavation encountered one horizontal layer rich in clasts that is interrupted by a feature containing skeletal remains.

In our previous considerations, we neglected two possible causes of reworking in our previous considerations of the Dinaledi Subsystem that could result in the patterns we observed: reworking of sediment due to collapse of spaces left by decomposition of soft tissue and reworking by *H. naledi*. In the current study, we consider these processes for the first time.

## Appendix 2

### Entry of *Homo naledi* into the cave system

2.1: Entrances to the cave system

The contiguous areas of the Rising Star cave system that are accessible today connect to the surface outside the cave via several entrances along the east and northeast-facing hillside above the system. The Dinaledi Subsystem is approximately 30 m beneath the present surface, and the nearest opening to the outside is via the Dragon's Back Chamber and Postbox Chamber, a horizontal distance of more than 100 m and vertical distance of 30 m (*Dirks et al., 2015*). Team members who work in the cave today usually follow a somewhat longer and easier route into the Dragon's Back Chamber via the Superman Chamber. Both these routes into the system involve vertical descents, sharp turns, and squeezes through rock passages of less than 30 cm.

The southwestern-most extent of the Dragon's Back Chamber is tall, narrow, and largely filled by a high, jagged fin of rock, approximately 10 m in length and 7 m in height, called the Dragon's Back. This structure is a dolomite septum that separates narrow spaces on either side, which become too narrow at the floor level to accommodate passage and which terminate before the chamber's southwestern end. The chert banding of the Dragon's Back itself is displaced approximately 60 cm downward relative to the bedrock of the surrounding chamber walls, indicating a shift in the position of this structure sometime in the past (*Dirks et al., 2015*; *Robbins et al., 2021*). Today it is possible to climb the Dragon's Back structure and then descend 12 m through a fissure network at its southwestern extent, which passes into the Dinaledi subsystem. This fissure network opens from above at several places along the top of the Dragon's Back. In 2013, Steven Tucker and Rick Hunter descended the most distal of these passages for the first time and recognized skeletal remains in the Dinaledi subsystem, leading to the exploration of this space and recovery of *H. naledi* (*Berger et al., 2015*; *Dirks et al., 2015*). This passage from Dragon's Back to the Hill Antechamber became known as the Chute, and it cannot be traversed by humans who cannot fit their bodies through a squeeze of less than 18 cm. There are other pathways through this fissure network, but exploration of them did not identify any passage that could accommodate any larger person (*Elliott et al., 2021*). One of these pathways is used today for cables that support our work in the Dinaledi subsystem.

The Lesedi Chamber is in a different part of the cave system. There is no path directly from the Lesedi Chamber to the Dinaledi subsystem; the closest route is more than 130 m. The Lesedi Chamber is approximately 30 m below surface and 86 m from the closest current outside surface entrance. This chamber can be entered from three different directions, and all of these involve vertical descents, tight turns, and squeezes of 30 cm or less.

The complexity of accessing these spaces today has brought a great deal of attention to the question of whether *Homo naledi* entered the cave system and these chambers by the same route as today's scientists. Cave systems change over time, and the Rising Star system is no exception. The time involved for these changes since the occurrence of *H. naledi* is more limited than for some nearby cave systems with remains of Pliocene and Early Pleistocene species such as *Australopithecus africanus* or *Paranthropus robustus*. Present evidence shows that some *H. naledi* remains entered the Dinaledi Subsystem between 335,000 and 241,000 years ago (*Dirks et al., 2017*; *Robbins et al., 2021*). No geochronological estimate yet exists for the Lesedi Chamber material, and the Dinaledi Subsystem results do not exclude the possibility that some individuals may represent a broader time range.

Most cave systems in the region were affected by lime mining activity during the early twentieth century. In the Rising Star cave system, the southern entrance of the cave and adjacent underground areas were affected the most strongly; neither of these areas is closer than 250 m through the cave system from the Dragon's Back Chamber. The eastern entrance to the system passes through the Skylight Chamber, and miners here blasted a small tunnel and constructed a ramp to gain easier access into this chamber. There is no visible trace of mining activity in the Dragon's Back Chamber or Postbox Chamber, or in the areas adjacent to the Lesedi Chamber.

Our investigations of the cave system, flowstone geochronology, and sedimentation processes have helped build a partial understanding of how the system changed in the period between 600,000 years ago and recent times (*Dirks et al., 2015*; *Dirks et al., 2017*; *Wiersma et al., 2020*; *Elliott et al., 2021*; *Robbins et al., 2021*). Detrital quartz and other material derived from surface contexts external to the cave system is abundant within lithified breccia deposits in the Dragon's

Back Chamber and unlithified sediment deposits that occur in the Postbox Chamber and Superman Chamber. The current nearest entrances to the system, such as the Skylight Chamber, are not situated in a way that sediment input from them could fill large areas of the Postbox, Superman, or Dragon's Back chambers from these sources. A survey of the upper areas of the Postbox Chamber suggests a possible scenario for the entrance of this external material into the system, described in *Robbins et al., 2021*. Much of the externally derived sediment is in positions where gravity could carry material from the highest parts of the Postbox Chamber, which are at the northeastern-most extent of the chamber. This area is below the point where two large chert horizons outcrop on the hillside above the Skylight Chamber. If an entrance once existed at this location, it must now be completely filled with sediment and is not obvious either from the surface or from within the Postbox Chamber. Any entrance at this location would have opened to a vertical descent of approximately 10 m into the upper part of the Postbox Chamber. From such an entrance, the distance from the surface to the Dinaledi Subsystem would be reduced by up to 20 m, making it around 80 m in total.

No stones or large grains of externally derived sediment input occur in the Dinaledi Subsystem sediments (*Dirks et al., 2015*; *Wiersma et al., 2020*). This indicates that no passive gravity-driven route for sediment input from the surface into this subsystem existed throughout the formation of Units 1, 2, or 3, inferred to be the last 700,000 years or more (*Dirks et al., 2017*). A 5-m sill occurs within the Dragon's Back Chamber, separating the main area of the chamber from the southwestern section in which the Dragon's Back ridge is situated (*Robbins et al., 2021*). This sill prevented direct sediment flow from the main area of the Dragon's Back Chamber up to the floor near the Dragon's Back itself, and thereby closed off sediment flow from there toward the Dinaledi Subsystem (*Robbins et al., 2021*). Sediment input into the Dinaledi Subsystem has occurred by slow introduction of fine sediment with dripwater at various points in the subsystem, as well as slow weathering of dolomite. The sediment floor level is highest at the northeasternmost extent of the Hill Antechamber, and this may suggest some input of material from the fissure network that separates this subsystem from the Dragon's Back Chamber. However, neither the sediment thickness nor the underlying bedrock floor elevation is known within the Hill Antechamber. The 30-degree floor slope is higher than the 15-degree southwest-trending dip of the dolomite bedrock, but may not indicate a very great accumulation of sediment in this area.

The accessibility of the Dinaledi Subsystem may have been altered over time by changes to the Dragon's Back structure itself. *Robbins et al., 2021* developed a chronology of the deposition of massive orange sand (MOS) and laminated orange sand (LOS) breccias within the Dragon's Back Chamber, establishing that the MOS breccia was formed between 290 ka and 225 ka. The MOS breccia contains large dolomite blocks, and these authors hypothesize that the formation of this breccia may correspond to a period of changes to the roof and structure of this part of the system. They further suggest that it is a reasonable hypothesis that the Dragon's Back structure itself may have taken its current form at that time. The 60 cm downward displacement and 10 degree rotation of this large block would have changed the accessibility of this space within the Dragon's Back chamber. *Robbins et al., 2021* suggest that passage beneath this block may have been possible prior to its displacement.

However, this displaced block alone does not form the separation between the Dragon's Back Chamber and the Hill Antechamber. These areas were connected prior to the block displacement as they are today: only through various small passages of a fissure network in the dolomite bedrock. The displaced block does not define the vertical descent that today provides the most common access into the Dinaledi Subsystem via this fissure. Passage beneath the block prior to its displacement may have provided greater access into this fissure network, but the extent to which this may have affected ingress into the Dinaledi Subsystem is not evident. Work to characterize the constraints of the present very challenging passageways continues.

In summary, *H. naledi* localities all are located a minimum of 86 m from the nearest identifiable current entrance into the cave system, and one hypothesized past entrance may reduce this minimum to approximately 80 m. All localities are accessible only through narrow passages that include vertical descents of at least 30 m, sharp turns, and squeezes of less than 25 cm and remain the case for any hypothesized past entrance. While many commentators have emphasized the physical difficulty of these passages for humans today, most aspects of the anatomy of *H. naledi* appear to be well-suited for movement within these spaces. With adult body mass of less than 50 kg, stature less than 160 cm,

thin and long lower limbs, shoulders with a cranially oriented glenoid fossa, strong and long thumb, and a powerful grip, the locomotor and body size adaptations of *H. naledi* are matched to climbing and moving into passages with a small diameter.

## 2.2: Hypotheses tested in previous work

In previous work, we have reported some ways that the hominin fossil material in the Dinaledi Chamber, Lesedi Chamber, and U.W. 110 locality departs from the situation in other fossil-bearing sites in the Cradle of Humankind, and we have addressed several hypotheses related to the deposition of these remains (*Dirks et al., 2015*; *Dirks et al., 2016*; *Hawks et al., 2017*; *Elliott et al., 2021*; *Brophy et al., 2021*). Remains of *H. naledi* are situated within chambers and spaces at substantial distances from each other, and at substantial distances from current entrances into the system. These localities have a limited representation of non-hominin faunal material and a high abundance of *H. naledi* with at least 16 individuals represented in the Dinaledi Subsystem prior to the current study, and at least 3 individuals in the Lesedi Chamber. Our initial analysis of both the Dinaledi Chamber and Lesedi Chamber situations documented some intact articulations in the skeletal remains from these localities (*Berger et al., 2015*; *Dirks et al., 2015*; *Hawks et al., 2017*). Subsequent work showed spatial clustering of elements within the Dinaledi Chamber subsurface excavation (*Kruger et al., 2016*) and correlated one of these clusters with elements from a single subadult partial skeleton (*Bolter et al., 2020*). The retention of anatomical articulations and clustering of material from particular individuals demonstrated that some skeletal material was emplaced in sediment within the Dinaledi Chamber and within the Lesedi Chamber prior to the decomposition of soft tissue (*Dirks et al., 2015*; *Hawks et al., 2017*). Excavation uncovered some skeletal elements of *H. naledi* in subvertical orientations while in close contact with other elements in near-horizontal orientations, a pattern inconsistent with the deposition of bones on a horizontal surface or in a shallow depression. We interpreted this pattern as suggesting multiple depositional events (*Dirks et al., 2015*).

In our past work, we have tested several hypotheses for the presence of hominin remains in various parts of the Rising Star cave system. Here we accept the results of this previous work and review the evidence that rejects several specific hypotheses.

### Carnivore accumulation

At a few other cave sites in South Africa, large carnivores interacted with some hominin remains, as evidenced by tooth scoring, punctures, and other carnivore damage on some hominin fossil bones (*Brain, 1985*; *Brain, 1993*). Some researchers have proposed that the remains of *H. naledi* from the Dinaledi Chamber may have been accumulated by large carnivores, pointing either to the pattern of fragmentation of the remains or the skeletal part representation (*Val, 2016*; *Egeland et al., 2018*).

The hypothesis of carnivore accumulation is testable with reference to the remains and their context. We have previously shown that no element from either the Dinaledi Subsystem or Lesedi Chamber has any tooth scoring or punctures, and none have breakage consistent with carnivore activity on fresh bone (*Dirks et al., 2016*; *Hawks et al., 2017*; *Brophy et al., 2021*). All fractures evident in the Dinaledi Chamber and Lesedi Chamber fossil material are consistent with breakage after deposition, burial, and subsequent loss of organic content (*Dirks et al., 2015*; *Hawks et al., 2017*). Additionally, non-hominin macrofaunal elements are rare and do not include other potential large prey animals (*Dirks et al., 2015*; *Hawks et al., 2017*). Excavations and surface examination of both areas have failed to yield any evidence of carnivore presence within them. There are no coprolites, no evidence of burrowing or other sediment disruption by large carnivores, and no large carnivore remains. In contrast to the Dinaledi Subsystem, the Lesedi Chamber assemblage does present evidence of small carnivores, including mongoose and small canids (*Hawks et al., 2017*), but such species have not been implicated in predation of hominins and cannot have transported or accumulated hominin bodies within a deep cave. These observations falsify the hypothesis that carnivores played any role in the presence of hominins in these parts of the cave system.

### Passive gravity-driven accumulation

*Dirks et al., 2015* discussed the hypothesis that hominin remains within the Dinaledi Chamber may have been introduced at or near our current access point, dropping into the Hill Antechamber and spreading from there under the influence of gravity alone. This scenario would resemble that proposed for the entrance of hominin remains into the Sima de los Huesos, Spain (reviewed by *Aranburu et al., 2017*). The evidence from previous work (*Dirks et al., 2015*) was sufficient to

raise doubts about this hypothesis for several reasons. Remains are found in the Dinaledi Chamber, more than 20 m from the nearest access point into the subsystem, and separated from the Hill Antechamber by a long (>7 m) and narrow (<0.5 m) fissure passage. This fissure passage includes the lowest point in the subsystem, adjacent to drains that could channel sediment downward (*Dirks et al., 2015*). Articulated bodies and parts of bodies within the Dinaledi Chamber could not have traversed this route under the force of gravity alone. Our subsequent work in the subsystem produced further evidence refuting this hypothesis. The presence of hominin remains in the fissure network distal to the Dinaledi Chamber, including the U.W. 110 locality, cannot have arrived in their present locations passively by gravity alone (*Elliott et al., 2021*; *Brophy et al., 2021*). This hypothesis for the Dinaledi Subsystem is further refuted by the results of the current study. The additional excavation units within the Dinaledi Chamber and Hill Antechamber show clearly that hominin remains are concentrated in specific features and do not form a talus or continuous layer in either the Dinaledi Chamber or Hill Antechamber. In addition, the hominin material within the Lesedi Chamber is within a different depositional space far from the Dinaledi subsystem, with different access constraints that are not consistent with passive, gravity-driven introduction of hominin remains.

## Occupation

OccupationThe long-term use of the Dinaledi Subsystem by *H. naledi* for occupation appears to be inconsistent with the fossil and sedimentary situation. No occupation debris or other evidence of occupation has been identified within this subsystem or within the Lesedi Chamber.

## Water transport and burial by fluvial action

Water transport and burial by fluvial actionAs discussed above, the sediments of sub-unit 3b within the Dinaledi Subsystem did not form in flowing water or with saturation sufficient to round the shape of LORM clasts and angular clay fragments which comprise this sub-unit (*Dirks et al., 2015*; *Wiersma et al., 2020*). These micromorphological observations are derived from sediment found in direct contact with surface and subsurface fossil remains of *H. naledi* (*Dirks et al., 2015*). Additionally, water transport of either bodies, large parts of bodies, or bones is refuted by physical constraints of the system and evidence from the surfaces of bone elements themselves (*Dirks et al., 2015*; *Dirks et al., 2016*). The Dinaledi Subsystem is separated from the adjacent Dragon's Back Chamber by a~5 m sill of dolomite that has been in place throughout the time of formation of Unit 3 (*Robbins et al., 2021*). This has prevented the movement of sediment directly into the Dinaledi Subsystem from Dragon's Back or other areas nearer to surface entrances and explains the sedimentological contrasts between the two chambers (*Dirks et al., 2015*). This leaves no possibility for bodies or bones to have been transported into Dinaledi by water flow. The transport of bodies by water from the Hill Antechamber into the Dinaledi Chamber is likewise not possible; bodies would have to flow down one of two narrow passages, past floor drains that would channel any water further down into the cave, and after passing through these passages would then have had to continue with minimal grade or even uphill for at least 15 m to their resting place in the Dinaledi Chamber. The transport of bodies or bones by water into the distal fissure network is likewise not credible (*Brophy et al., 2021*). Water transport of bodies into the Lesedi Chamber has not been suggested and cannot explain the distribution of skeletal material in the chamber.

## Mass mortality or death trap

Mass mortality or death trapOur initial exploration of the Dinaledi Chamber produced evidence of a minimum number of 15 individuals representing ages from infant to old adult (*Berger et al., 2015*; *Bolter et al., 2018*). The distribution of ontogenetic ages of this assemblage could be consistent with either attritional or catastrophic mortality distributions and did not distinguish them (*Bolter et al., 2018*). These data by themselves appeared to leave open the possibility that some or all of the assemblage might represent individuals that entered the chamber alive and then died there (*Dirks et al., 2015*; *Dirks et al., 2016*). Later workers suggested that baboon sleeping sites might provide an analogy for the accumulation of skeletal remains in the Dinaledi Chamber, with bodies gradually accumulating after natural deaths of individuals (*Nel et al., 2021*). However, we previously noted several aspects of the Dinaledi hominin assemblage that are inconsistent with an unburied mass death accumulation, including the presence of skeletal elements in subvertical orientations in close spatial proximity or contact with articulated parts of bodies (*Dirks et al., 2015*; *Dirks et al., 2016*). The presence of multiple hominin individuals in the Lesedi Chamber and within widely separated

localities in the Dinaledi Subsystem also implicates regular use of extensive parts of the cave system rather than a single mass death event (*Hawks et al., 2017*; *Brophy et al., 2021*). The localities with *H. naledi* remains contrast with known baboon sleeping sites in several ways, including their distance from cave entrances, lack of evidence of external input of sediment or organic material, and the extent of bias toward hominin (or primate) remains at the expense of other taxa.

# Appendix 3

## Description of the Hill Antechamber artifact 1 associated with the Hill Antechamber Feature

Within the Hill Antechamber Feature is a rock of 138.5 mm length which is located within 20 mm of a group of bones interpreted as anatomically associated metacarpals and phalanges of a right hand. Due to the proximity of this rock to the hand remains and its general resemblance to a large flake tool, we prioritized obtaining higher-resolution scan data of the stone to evaluate its morphology. We designated it Hill Antechamber Artifact 1 (HAA1), recognizing that at a minimum, its spatial association is of interest. At this writing, HAA1 still remains within the larger plaster jacket containing most of the Hill Antechamber Feature. This section presents a description of the object. Orientation is described based on an arbitrarily defined position of features illustrated in *Appendix 6—figure 21*. All measurements and descriptions are taken from 3D images produced from the synchrotron scans or from examination of high-resolution 3D prints of HAA1. Both scans and 3D shape files of HAA1 are available for download on https://Morphosource.org. A high-resolution movie of HAA1 is available as *Video 2*.

The shape of HAA1 is distinctive in comparison to other rocks on the surface or encountered during excavation of the Hill Antechamber or Dinaledi Chamber. HAA1 is 138.5 mm in total length. Its greatest width perpendicular to length is 49 mm just left of the middle of center. Its greatest diameter in the remaining dimension is 26.3 mm. HAA1 is roughly crescent-shaped coming to a sharp point laterally left and a more rounded but anteroposteriorly sharp edge right laterally. It presents a large flake scar of ca. 80 mm in length on the anterior-inferior surface from approximately 10 mm left laterally of its midline that travels to within 12 mm of the right lateral end. The lunate area of this flake scar occupies approximately three-quarters of the body of the artifact for most of its extent. A 6–7 mm wide ledge is observed in the superior one-third of the artifact that is created from the removal of rock flake ca. 84.5 mm in length that leaves a prominent hump occupying the middle 65 mm of the artifact. We cannot determine the type of stone unambiguously from scan data, but we hypothesize based on its surface characteristics that it is likely composed of dolomite, and there is no evidence that the rock is exogenous to the cave system.

Within the area of the lunate flake scar on the anterior surface, and opposite this area on the posterior surface, striations are visible that appear to be use wear or erosional marks (*Figure 12*). Some of these lines are as much as 1.5 mm wide, but most are sub-millimeter in width and are 20–25 mm in length. They travel predominantly in a lower-left to upper-right direction across this surface (*Appendix 6—figure 22*).

The posterior surface of HAA1 is dominated by a prominent ridge that runs from the left lateral point superolaterally at an approximate 15 degree angle, reaching the superior edge about 50 mm from the right lateral edge. The peak of the ridge forms the greatest width of the artifact, and this occurs approximately 55 mm from the left lateral end. Supero-inferiorly oriented erosional or wear lines are found on the right lateral half of the artifact roughly mirroring those on the anterior surface. There is a depressed area that may represent a worn flake removal in this same region that is not as large nor prominent as that found on the anterior surface.

The inferior edge of the artifact along the area of the anterior lunate flake scar is very sharp and under high-resolution imaging by synchrotron (6 μm), irregular serration is visible across the whole of the surface (*Appendix 6—figure 22*). No other serration is obvious along the edges of other sharp areas.

## Appendix 4

### Hominin skeletal material and element representation in the Dinaledi Feature 1 and 2 area

In this section we present a preliminary identification of skeletal elements within the burial features, with an assessment of skeletal part representation. We note the presence of articulated or spatially contiguous elements where this has been observed.

In both the Dinaledi and Hill Antechamber, we have left much of the evidence described here in situ or en bloc without unnecessary destructive excavation or preparation. The skeletal material that has been excavated from Dinaledi Features 1 and 2 has not been subjected to solvents or consolidants at this time (Materials and Methods). The Hill Antechamber feature rests within three plaster-jacketed blocks at the present time. These decisions limit the number of observations that we can gather on the skeletal remains from these features. Here it is worth a brief comment to clarify these protocols and discuss their relevance to the study of the burials.

Archaeological controlled excavation is a fundamentally destructive process (*Lucas, 2001*), one in which the spatial relationships between buried objects (including hominin fossils) is lost once the matrix surrounding such remains is disturbed and the buried objects lifted. Spatial relationships are necessary to determine temporal relationships in archaeological settings; they are also fundamental to understanding the processes of alteration that assemblages may have undergone after burial. Article 5 of the International Charter for Archaeological Heritage Management states that archaeological investigations can be carried out using a wide array of methods from non-destructive remote sensing, through sampling, to total excavation. It is widely recognized that total excavation is a recourse that should be adopted after consideration of other less destructive means of study (*Beaudet and Elie, 1991*).

3D modeling or non-invasive imaging is the preferred method for recording and understanding fragile, friable buried contexts where the precise spatial relationships between clastic components is considered more important rather than the necessity to view the surface morphology of such remainsi. Such a spatially-based approach has been used and advocated in recording of mainstream archaeology (*Roosevelt et al., 2015*), cremains (*Pankowská et al., 2017*), bone taphonomy (*Silver, 2016*; *Randolph-Quinney et al., 2018*), 4D documentation of pit burials (*Mickleburgh and Wescott, 2018*), virtual autopsy (*Bolliger and Thali, 2009*), recording fragile funerary goods and human remains (*Jansen et al., 2006*), through to whole burial chambers (*Krausse et al., 2017*), where information derived from the spatial relationships within the burial environment outweighs that from direct analysis of the bones or artefacts themselves (*Dell'Unto and Landeschi, 2022*).

In the Rising Star cave system we have employed study protocols that minimize the extent of excavation areas and maximize the collection of spatial data by multiple modalities (*Elliott et al., 2021*; *Dirks et al., 2015*; *Dirks et al., 2017*; *Kruger et al., 2016*). The total excavation surface inthe Dinaledi Chamber to date is 1.55 m$^2$ and in the Hill Antechamber is 0.75 m$^2$. *Homo naledi* skeletal and dental material has come from these excavated areas and from collection of skeletal remains from the floor surfaces of these chambers. That material now numbers more than 2500 pieces, many of them small fragments but including a large number of complete and well preserved elements. Nonetheless much of this skeletal material is highly fragile with loss of elements with thin cortical bone or substantially trabecular bone content. Our excavations in 2013 and 2014 in the Dinaledi Chamber identified buried skeletal material with articulated limb, cranial, and vertebral elements (*Dirks et al., 2015*; *Berger et al., 2015*) including portions that could be attributed based on spatial and developmental data to a partial juvenile skeleton (*Bolter et al., 2020*). This extensive evidence of skeletal morphology and spatial positioning of material has informed our more recent work in the system in a variety of ways (*Elliott et al., 2021*).

In many paleoanthropological contexts, the recovery of morphological or developmental observations on hominin skeletal material is a very high research priority. Additionally, some settings pose taphonomic questions that can be addressed only through collection of data from bone surfaces. To recover such data, hominin skeletal material has often been subjected to intensive preparation, cleaning, and consolidation. Much *Homo naledi* skeletal material from our earlier work has been treated with conventional methods of skeletal conservation and analysis, which has included microscopic examination of bone surfaces (*Dirks et al., 2015*). Having this published evidence in hand gives us the possibility of taking a less destructive approach to studying the burial

features that we identified in subsequent work. For these reasons, once we identified the possibility that the Dinaledi Feature 1 and Hill Antechamber Feature were burial contexts, we planned for non-destructive methods to record and preserve the spatial relationships between any remains entombed within these features. In the Hill Antechamber case, this involved lifting the feature en bloc for study, scanning, and possible preparation in the laboratory. In the case of Dinaledi Feature 1, the prior recovery of a similar burial feature from the Hill Antechamber argued for leaving this feature in situ within the cave system and recording data on its spatial and sedimentary context noninvasively.

## 4.1: Identification and assessment of skeletal remains from Dinaledi Feature 1

Here we list brief anatomical identification of all catalogued skeletal elements and fragments excavated within and above the area of Feature 1. These specimens all come from grid units S950W550 and S900W550. Some other material excavated from unit S900W550 is associated with Feature 2 and these are listed separately below.

These brief identifications do not focus on comparative morphology beyond the identification to element where possible. Many of the remains are small fragments that are not identifiable to element and these are included here to present a complete record. In the few cases where diagnostic morphology of *H. naledi* is present, this is indicated. Where evidence for developmental stage is present, this is indicated.

We have not fully cleaned sediment from this skeletal material or applied fixatives or other chemicals, in anticipation of possible analysis of biochemicals or trace evidence. This precludes a full examination of possible surface modifications or other surface taphonomy.

Nearly all elements identified in this list are compatible with belonging to a single adult individual. All identifiable adult elements are size-compatible; they do not appear to represent a mixture of individuals of different sizes. None of the adult elements duplicate each other; nor is there any duplication with identifiable elements that remain in situ within Feature 1 in the Dinaledi Chamber.

There are three notable exceptions that are immature elements and inconsistent with the single adult represented by the rest of this material. Two of these are fragments of juvenile femur, including the right proximal femur fragment included in U.W. 101-2250 and a fragment of femur shaft U.W. 101-2260. These two elements were excavated overlying and in physical contact with the bone concentration of Feature 1, both within 5 cm of each other near the center of the feature. The proximal right immature femur duplicates the proximal right adult femur fragment, also included within U.W. 101-2250. These two elements were excavated in immediate contact with each other with the immature femur fragment overlying the adult fragment. The other inconsistent element is an immature proximal left humerus fragment, U.W. 101-2243. This element duplicates another proximal left humerus within Feature 1, U.W. 101-2231. These two elements contrast in developmental status, with U.W. 101-2243 having evidence of an unfused proximal epiphysis and U.W. 101-2231 compatible with adult status. The two elements are also in different situations relative to the feature: U.W. 101-2231 is near other compatible humerus and upper limb material at the south end of the feature, while U.W. 101-2243 was out of anatomical placement overlying the uppermost part of the north side of the feature. We interpret these three fragments of immature elements as bone fragments that were on the surface or within pre-existing deposit that comprised the sediment fill of Feature 1.

## Specimen list

U.W. 101–2114

Thin plate-like bone fragment, broken on all edges, less than15 mm.

U.W. 101–2115

Two cranial bone fragments. One is possibly zygomatic or root of zygomatic arch on temporal, length 23.8. The other is possibly zygomatic at orbit, or frontal above orbit, bone surface is concave and slightly wavy. Preserved size 16.5x16.5.

U.W. 101–2116

Phalanx fragment with adhering clump of sediment. Fragment size is 18.9x9.0

U.W. 101–2117

Right proximal radius fragment. The head is present but broken around the visible edge, and one side is still obscured by sediment. The bone is very comparable in size with U.W. 101–2246 and they may be antimeres. 38.8 mm length

U.W. 101–2118

Two bone fragments less than 15 mm

U.W. 101–2119

Bone fragment less than 15 mm

U.W. 101–2120

Two cranial bone fragments less than 20 mm

U.W. 101–2121

Right proximal radius fragment, representing nearly the entire circumference of the bone. The head is not present. The distal break on this fragment is consistent with the break at one end of U.W. 101–2240 and may be the same bone. Preserved length 49.6. Shaft diameter 10.0x9.4.

U.W. 101–2136

Cranial or mandibular fragment, possibly zygomatic arch. Length 20.4

U.W. 101–2137

Three large clumps of sediment with spongy bone fragments embedded within them. Possibly distal femur or pelvic.

U.W. 101–2138

Sediment clump with embedded bone fragments. ca. 15 mm

U.W. 101–2139

Sediment clump with embedded fragments of cortical bone. ca. 15 mm

U.W. 101–2140

Tooth root. 11.5 mm

U.W. 101–2141

Tooth root. 13.2 mm.

U.W. 101–2142

Cortical bone fragment, 16.0 mm.

U.W. 101–2143

Three bone fragments less than 15 mm.

U.W. 101–2144

Mandibular incisor. Probably left I1, possibly I2. Occlusal wear is extensive and remaining crown height above cervix on distal face is only 2.2 mm. The wear is at an angle of approximately 60 degrees compared to the labial face. The wear has exposed dentin with MD 2.0 mm, LL 1.3 mm. LL breadth 5.9, MD 4.6. Root is nearly complete with a small section of the distal tip missing due to fracture.

U.W. 101–2145

Bone or dentin fragments, less than 5 mm, with some sediment.

U.W. 101–2146

Bone fragments in sediment clump, less than 10 mm.

U.W. 101–2147

Bone fragment with large cancellous structure in sediment clump. Less than 20 mm.

U.W 101–2148

Two bone fragments with sediment adhering. Smaller fragment 13.0, larger 21.3.

U.W. 101–2168

Sediment with bone fragments less than 5 mm.

The tibia shaft portion, 121 mm in length, represents midshaft. Consistent with adult. Diameter at point where posterior shaft flattens is 21.4x17.2. Other small shaft fragments, all less than 20 mm.

U.W. 101–2171

Long bone shaft fragment, 34.3x10.4

U.W. 101–2172

Fourteen long bone shaft fragments. Most represent a part of the bone circumference and are between 30 and 40 mm in length, 10–15 in width. Additional fragments in the bag.

U.W. 101–2173

Two bone fragments less than 15 mm.

U.W. 101–2174

Mandibular fragments. Two tooth crowns are present with their roots and three additional root fragments. The two crowns are both left mandibular molars. One is markedly more worn, with large dentin pools at protoconid and endoconid, and metaconid is entirely fragmented away. The rest of

the enamel rim has large fragments missing. BL 11.1 MD 12.1. This tooth has a distal interproximal facet; I interpret it as an M2. The other tooth is worn approximately flat with dentin exposed on mesial four cusps. BL 10.8 MD 12.1. This tooth is more triangular in shape distally and does not have an interproximal facet distally; I interpret it as an M3. One of the broken root fragments is a molar root, the others appear likely to be premolar roots. The morphology and dimensions of these teeth are compatible with *H. naledi*.

U.W. 101–2175

Lower left fourth premolar. Root is largely intact, broken at tip. Single root. Enamel is worn flat with large dentin pools for both buccal and lingual cusps. BL 10.7, MD 9.2. The morphology and size of this tooth are compatible with *H. naledi*.

U.W. 101–2176

Lower left third premolar. Two roots, both intact. BL 10.6, MD 9.4. Crown is worn flat, dentin exposed at protoconid and metaconid. This tooth possesses diagnostic anatomy for *H. naledi*.

U.W. 101–2177

Ulna shaft portion, 48 mm.

U.W. 101–2178

Two fragments. One is fibula shaft, 37.5 mm. The other is a flat fragment less than 20 mm.

U.W. 101–2179.

Bone fragment, possibly tarsal, maybe medial cuneiform fragment. Two additional pieces are bone fragments less than 10 mm.

U.W. 101–2180

Distal ulna shaft fragment, 44.6 mm.

U.W. 101–2181

Cranial fragment consistent with right zygomatic bone including orbital border. 34.8x24.7 mm.

U.W. 101–2182

Fibula or metacarpal shaft portion, 27 mm

U.W. 101–2183

Radius or humerus shaft portion 43.6 mm. Additional fragments of shaft.

U.W. 101–2184

Base of right mandibular corpus. 52.5 mm.

U.W. 101–2185

Enamel fragment, 5 mm.

U.W. 101–2186

Enamel fragment, corresponding to rim of molar, 8.9 mm

U.W. 101–2187

Tooth root fragment, 10.1 mm

U.W. 101–2188

Cranial vault fragment. One side is obscured by sediment. 21.4x15.4 mm, 5.4 thick.

U.W. 101–2189

Mesial part of the mandibular molar, with mesial root complete. Very little enamel on this crown. This looks like a mirror image of the U.W. 101–2174 M3 mesial root. Several additional cranial or mandibular fragments, mostly less than 20 mm.

U.W. 101–2190

Fragment of humerus head, with small slice of surface 22.0x8.3 mm.

U.W. 101–2191

Partial phalanx, probably manual intermediate phalanx. The proximal articular surface is eroded but present, shaft 16.5. Embedded in sediment chunk.

U.W. 101–2199

Phalanx shaft portion, or possibly fibula shaft portion. 23.5 mm.

U.W. 101–2200

Fibula shaft or metatarsal shaft portion, triangular cross-section. 22.6 mm

U.W. 101–2201

Three bone fragments, with some articular surfaces.

U.W. 101–2202

Bone fragment, shaped like possibly zygomatic or mandibular fragment, seems too thick to be vertebral or rib. 22.7 mm length.

U.W. 101–2203

Shaft fragment consistent with metacarpal. Length 28.5, diameter 5.9x5.2.

U.W. 101–2204

Bone fragment. 25.8 long, 14.8 broad, 6.8 thick.

U.W. 101–2205

Long bone shaft fragment, based on size and curvature consistent with humerus, representing approximately 40% of the circumference. Length 43.5, width 13.3.

U.W. 101–2206

Bone fragment, one side exposed trabeculae, the other rough cortical surface. Possibly distal femur or ilium. 24.5x17.9

U.W. 101–2207

Metatarsal or manual phalanx shaft fragment. 23.6 long, 7.9x5.3 near base.

U.W. 101–2208

Long bone shaft fragment, based on size and curvature, probably humerus, representing 30% of the circumference. 25.4x13.9.

U.W. 101–2209

Long bone shaft fragment, based on size and curvature, probably humerus, representing 30% of the circumference. 29.1x11.4.

U.W. 101–2210

Bone fragments, probably cranial, less than 30 mm.

U.W. 101–2211

Rib fragment. 34.3 long, diameter 8.6x6.1

U.W. 101–2221

Bone fragments.

U.W. 101–2222

Cranial vault fragment. 22.4x14.8.

U.W. 101–2223

Long bone fragment. 38.7x13.0

U.W. 101–2224

Left distal humerus fragment. The lateral supracondylar crest is well preserved and evident. Length 54.4.

U.W. 101–2225

Long bone shaft fragment with rounded circumference compatible with humerus or radius, including 50% of shaft circumference. 29.9 long, 12.0 wide.

U.W. 101–2226

Flat long bone shaft fragment. Consistent with distal humerus or proximal tibia. 38.5x15.8.

U.W. 101–2227

Long bone shaft fragment consistent with humerus. 29.1x15.1

U.W. 101–2228

Shaft fragment consistent with small long bone, metacarpal or metatarsal. 21.3x9.5

U.W. 101–2229

Long bone shaft fragment consistent with radius, ulna, or fibula. 27.4x10.2

U.W. 101–2230

Bone fragments in sediment.

U.W. 101–2231

Proximal left humerus shaft. Length 53. Diameter at surgical neck 18.2x15.8. A chip of enamel is adhering within the sediment that fills the proximal end of this fragment. The enamel chip is approximately 3 mm x 5 mm and could be molar or incisor. A second fragment refits the distal end of the first. Length 24.8x13.7.

U.W. 101–2232

Bone fragment 19x9 mm.

U.W. 101–2233

Bone fragment 18.7x12.8.

U.W. 101–2234

Bone fragment 20.6x8.4.

U.W. 101–2235

Tooth root, slightly bilobate toward crown. 15.1 mm.

U.W. 101–2236

Bone fragment with concave surface exposed on one side, other side trabecular fragments with sediment obscuring detail. 21x17.

U.W. 101–2237

Maxillary left molar. BL 12.3, MD 11.6. All roots are present and intact, all three curve distally, and the buccal two roots curve into each other strongly distally. Based on roots and crown morphology, this resembles an M3 more than either other molar. The crown is bigger than any of the M1s, and those teeth do not have the posteriorly directed lingual root that this one has. The occlusal surface has slight wear, no dentin exposure, and all cusps are salient.

U.W. 101–2238

Mandibular ramus fragment of right coronoid process. 35.9x14.8.

U.W. 101–2239

Shaft fragment of femur or tibia, relatively flat, consistent in size and thickness with U.W. 101–2259 femur. Length 48.3, width 19.4.

U.W. 101–2240

Shaft portion of radius or ulna. 43.1 long, diameter obscured by sediment. This fragment is consistent with the break at the distal end of the U.W. 101–2121 radius fragment and may be the same bone. A small, thin, and slightly rounded bone fragment was adhering to one end, not in anatomical position. This is thinner than the shaft fragment and probably belongs to some other bone.

U.W. 101–2241

Bone fragment, thin with adhering sediment on one side. 19.9x13.6

U.W. 101–2242

Proximal right ulna, lacking olecranon process. Length 50.2, diameter below coronoid process 12.9x12.4. Two additional shaft portions refit the larger fragment.

U.W. 101–2243

Proximal left humerus shaft. The proximal end of this includes a small portion of metaphyseal surface across the lateral 25% of this end of the bone. There is additionally a small (10 mm) piece of epiphysis that was adhering in position to part of this lateral edge of the proximal end. This is now detached, and its surface does correspond to the opposing surface of the diaphysis. Length 46.7. Shaft diameters at surgical neck 18.5x15.5.

U.W. 101–2244

Proximal end of a rib with the articular part of the head missing but the tubercle present. The neck is round in cross-section, and the break just lateral to the tubercle has a rib-like cross section. Length 25.9, diameter 6.3.

U.W. 101–2245

Bone fragment consistent with vertebral lamina. 13.3x9.3 x 5.6 thick at thickest point.

U.W. 101–2246

Proximal left radius, lacking head. Radial tuberosity and approximately 25% of shaft are present. The neck is broken and does not preserve a metaphyseal surface. It is size-consistent with adult material. That makes it comparable to the humerus elements nearby, including U.W. 101–2243. Length 55.7, diameter of neck 10.1x9.2.

U.W. 101–2247

Phalanx or rib fragment. 20.6x8.2 x 4.9.

U.W. 101–2248

Rib fragments less than 20 mm.

U.W. 101–2249

Right distal humerus, lacking trochlea and capitulum. Length 88. Diameter above epicondylar ridges 14.1x15.7.

U.W. 101–2250

Long bone remains that represent at least two different elements. One of these is a right proximal femur, including base of neck, lesser trochanter, broken below greater trochanter. Subtrochanteric diameters AP 20.9, ML 28.5. Length of fragment 68.4. A second large fragment is a portion of long bone shaft, 62.5 long, 16.0 in diameter. The cortical bone is thin relative to the shaft diameter, suggesting that it is more consistent with juvenile femur or tibia rather than adult humerus. In size, this shaft fragment may be compatible with the immature proximal femur fragment U.W. 101–2260 but there is no refit between these pieces. This bone is highly dark-stained with iridescent sheen and edge. Smaller additional fragment of the same 27.3x14.0. Additional bag contains bone fragments in sediment.

U.W. 101–2258

Rib fragment 23.1 long.

U.W. 101–2259

Femur shaft. Refits right proximal femur in U.W. 101–2250. Fracture is stained, not fresh. Length 152.9. Diameter same as 101–2250.

U.W. 101–2260

Immature proximal right femur. Includes neck, head is missing. It is possible that the metaphyseal surface for the head is present here, but sediment and possible erosion mask whether this is the case. The metaphyseal surface for the greater trochanter is partially present. None of the head is present, and the broken portion of the neck does not retain any of its metaphyseal surface. The lesser trochanter is projecting, the shaft is broken away irregularly, and the lateral border of the shaft is missing. Fragment 41 mm, neck 19.6x15.1.

U.W. 101–2261

Fragment with concave surface on one side, convex obscured by sediment on the other. Fragment 19.7x15.2 x 6.6 thick.

U.W. 101–2262

Long bone shaft fragment, ulna. 39x13.4.

U.W. 101–2263

First metacarpal head and 40% of shaft. Includes diagnostic morphology of *H. naledi*. 28.5 mm.

U.W. 101–2264

Shaft fragment of phalanx or metacarpal. 25 mm x 8.1 mm.

U.W. 101–2265

Bone fragment, possibly carpal fragment less than 15 mm.

U.W. 101–2266

Bone fragment, less than 20 mm.

U.W. 101–2267

Bone fragments within clump of sediment, possibly rib fragments, 32.7 mm.

U.W. 101–2268

Bone fragment, possibly rib fragment, 25.4x9.4.

U.W. 101–2269

Bone fragment with morphology and possible metaphyseal surfaces, possibly immature long bone fragment? 18.4 mm.

U.W. 101–2270

Bone fragments

U.W. 101–2271

Bone fragment less than 20 mm.

U.W. 101–2272

Long bone fragment 25.2x16.6 x 6.9 thick.

U.W. 101–2273

Shaft fragment of long bone, 28.5x17.4. Other associated fragments.

U.W. 101–2274

Manual phalanx. Head is eroded, base is missing. Probably proximal phalanx based on length. 28.0 mm long, diameters at midshaft 8.8x4.9.

U.W. 101–2275

Ischium fragment including lunate surface of acetabulum, broken superior to ischial tuberosity. 25.0x13.0 mm of lunate surface present. Fragment length 21.1 from acetabular border to inferior edge. Subacetabular sulcus 6.9 mm.

U.W. 101–2276

Mandibular corpus fragment, including the angulation between the base of the corpus and either the external or internal surfaces. Based on curvature, this seems likely to be external. This fragment is possibly consistent with U.W. 101–2184, which is clearly right mandibular corpus including the base and the swelling at the base of the ramus. Fragment dimensions 36.5x13.5.

U.W. 101–2277

Shaft portion of ulna or fibula. Length 37.3, diameter 9.9x9.4.

U.W. 101–2278

Left proximal ulna with shaft fragments. The olecranon process is present but eroded, and the coronoid process appears to be broken or missing. The coronoid process is certainly missing because of a break. This is consistent with adult ulna due to the extent of the olecranon process that remains without evidence of metaphysis, but its most proximal extent is abraded. Measurements below coronoid 11.9x11.5. Length 47.6.

## 4.2. Identification and assessment of skeletal remains from Dinaledi Feature 2

Very little skeletal material has been excavated from or above Dinaledi Feature 2. The feature remains largely undisturbed, with only a small semi-circular concentration of bone visible and an unknown portion remaining within the unexcavated S950W600 grid square. The identifiable elements are consistent with a single adult individual, but very little evidence has been recovered to date.

U.W. 101–2134

Bone fragment, thin cortical flake, 13.9 mm

U.W. 101–2135

Bone fragment consistent with cranium or mandible, 12.7 mm

U.W. 101–2166

Bone fragment consistent with cranium or mandible, 34.9x20.9 mm.

U.W. 101–2167

Bone fragment in sediment, possibly phalanx shaft. Less than 20 mm.

U.W. 101–2198

Flat thin fragment consistent with zygomatic arch or vertebral lamina. 15.2x7.4.

U.W. 101–2220

Femur or tibia shaft portion. Surface morphology and cross-section obscured by sediment. Length 80.7, diameter 24.9

# Appendix 5

## Postmortem change and *Homo naledi*: Understanding decomposition in the Rising Star burial environment

Decomposition can be described as the process by which the body physically breaks down and decays, ultimately resulting in skeletonization. However, it is important to firstly understand the nature of body as it decomposes, as although the distal outcomes (skeletonization and post-skeletonization phases) may be different in differing depositional scenarios, the initial proximal stages (the early postmortem period) are entirely driven by the linked processes of putrefaction and decomposition. Decompositional changes to a body that occur immediately after death are more rapid than those occurring later in the process, making it difficult to establish an exact time interval, but the general pattern of decomposition is well understood (*Bristow et al., 2011*; *Gill-King, 1997*; *Wilson et al., 2007*). The factors that affect the rate at which postmortem changes occur are classified into two groups *Prieto et al., 2004*; firstly, those dependent on the cadaver and intrinsic physiological factors, and secondly, those dependent on the postmortem environment of the cadaver and extrinsic factors such as temperature, humidity, insect activity, and scavenging with the codicil that decomposition can be retarded by several processes, including physical and chemical barriers and climatic factors (*Bristow et al., 2011*). Much of our understanding of human decomposition data is derived from forensic taphonomic research either based on real-world forensic case evidence and interpretation, or (more crucially) actualistic experimentation undertaken at human taphonomic facilities - such forensic approaches override classical inductive and intuitive palaeo and archaeo-taphonomic approaches which come with significant evidential shortcomings (*Bristow et al., 2011*; *Schotsmans et al., 2022*). Whilst the general process of decomposition is broadly similar across most mammalian taxa, specific differences in functional anatomy, body composition, physiology, biochemistry, microbiome, and skeletal ultrastructure (including histological pattern, mineralized and organic makeup, and biomechanical competency) between the human pattern (and by inference hominin and non-human primate) and that of other animals limit the use of inference from classical vertebrate taphonomic studies of animal decomposition and skeletonization (*Bristow et al., 2011*).

Decomposition processes are influenced primarily by temperature, but secondarily affected by many different variables, including humidity, insect activity, sunlight, rainfall, scavenging, and the burial (sedimentary) environment, among others (*Bristow et al., 2011*; *Henderson, 1987*). Temperature influences microbial degradation as the primary driver of decomposition (*Forbes et al., 2017*; *Ody et al., 2017*) with warmer temperatures increasing the rate of microbially-driven decomposition. In bodies exposed on the external ground surface (sub-aerially), insect activity is the key driver of soft tissue decomposition in conjunction with microbial action. Very cold or freezing temperatures inhibit and retard microbial action leading to tissue preservation or the drastic slowing of the decomposition process. Low temperatures and/or enclosed spaces (including deep cave systems such as Rising Star) also influence insect action and access - holometabolous insects (those with larval and pupal stages) are either inactive or have limited activity (*Anderson and Cervenka, 2002*; *Bachmann and Simmons, 2010*; *Benecke, 2005*; *Komar, 1998*; *Lutz et al., 2021*; *Michaud and Moreau, 2009*; *Mona et al., 2019*; *Myskowiak et al., 1999*).

It is important to understand the physical changes that the body (cadaver) undergoes between death and skeletonization, as these changes influence both the volume and integrity of body systems and parts (*Boulestin and Duday, 2005*; *Duday, 2009*; *Schotsmans et al., 2022*). The process of decomposition proceeds through six recognized stages (after *Wilson et al., 2007*): (1) Fresh - begins at death and includes rigor mortis, postmortem hypostasis, and cooling which continues until bloating of the cadaver is visible; (2) Primary bloat stage – bacterially mediated accumulation of gases within the body causes distension and bloating, with an increase in surface area and volume of the body. Whilst there is no disarticulation of body parts, the hair and epidermis are loose, and the soil-skin interface is visually and biochemically altered by microbial-sediment interaction. The body emits a strong odor caused by the release of volatile organic compounds (primarily putrescine and cadaverine); (3) Secondary bloat stage – the body is still bloated, and there is now rank-order disarticulation of joints and limb segments. Purging of decomposition fluids occurs from natural cavities such as the oral and nasal cavities, anus, or from damage to the integrity of the integument. The soil-skin interface is black. A mobile leachate plume may extend into the surrounding soil/sediment around and some distance from the body; (4) Active decay stage - deflation of the cadaver, disarticulation of joints, limbs, and head. Muscle and integument may still be present. The cadaver

is very wet with a strong odor; (5) Advanced decay stage - collapse of components of the rib cage and pectoral girdle into the thoracic cavity, with most of the flesh liquefied or gone; skin, bone, fat, and cartilage may remain and the cadaver and underlying sediments are very wet with leachates and decomposition fluids; and, (6) Skeletonization – muscles, skin, fat, and cartilage disappear, although some persistent ligaments may remain. Bone will present as 'dry' with a shift in biomechanical competency through the loss of organic (primarily collagen) content, and chemi-absorbed and physi-absorbed water, leaving the mineral bioapatite as the primary component of remaining skeletal tissue. Bone will fracture in a classic 'postmortem' and opposed to 'perimortem' fashion (*Lyman, 1994*; *Christensen et al., 2022*).

In the early postmortem period, enzymatic digestion (autolysis) is the primary driving factor. This initiates widespread cell degradation via anaerobic microorganisms such as bacteria, fungi, and protozoa (*Vass, 2001*; *Vass, 2011*; *Vass et al., 2002*) within the gastrointestinal tract and the respiratory system (*Carter et al., 2007*; *Carter et al., 2010Carter and Tibbett, 2008a*; *Carter et al., 2008b*), leading to saturation of body tissues (hemolysis) (*Fiedler and Graw, 2003*) with decomposition products. These products result in color changes seen on the dermis with concomitant bloating of the body, signaling the onset of the putrefactive stage of decomposition (*Bristow et al., 2011*). A continued internal build-up of decompositional gases eventually leads to a build-up of internal pressure, leading to purging through body orifices, such as the mouth, nose, and anus, and possible tissue rupturing (*Carter et al., 2007*). The active decay process is most often associated with a decrease in body mass (*Adlam and Simmons, 2007*). Putrefaction processes function under anaerobic conditions (*Adlam and Simmons, 2007*; *Cross and Simmons, 2010*; *Simmons et al., 2010a*; *Simmons et al., 2010b*). Further decay by bacteria and fungi is purely anaerobic, and it is this which leads to full skeletonization of the cadaver (*Fiedler and Graw, 2003*). The decay process will generally be accelerated in bodies in sub-aerial contexts, particularly with open access to insect faunas. Different depositional and burial environments will lead to a difference in observed or expected results since neither environment is static (*Wilson et al., 2007*).

In burial environments, there is a restriction of access to the remains from scavengers and insects who pose a major role in decomposition (*Adlam and Simmons, 2007*; *Cross and Simmons, 2010*; *Simmons et al., 2010a*; *Simmons et al., 2010b*), as well as restricting the effects of sub-aerial processes such as weathering, which can accelerate the decomposition and degradation of skeletal tissue itself (*Behrensmeyer, 1978*, *Hill, 1976*; *Manhein, 1996*; *Tappen, 1969*). Most of the scientific literature regarding the direct observations of decompositional processes of human remains refers to those that have been deposited on the surface of the ground (refs), or more recently in shallow or open pits (e.g. *Mickleburgh and Wescott, 2018*; *Mickleburgh et al., 2022*), where temperature and insect activity play significant roles in the decomposition process (e.g. *Megyesi et al., 2005*). In burial environments where the body is encapsulated, such as inside structures or buried in sediment matrix, there is restriction of access to the remains from scavengers and insects who would otherwise play a major role in the pattern and tempo of decomposition (e.g. *Cross and Simmons, 2010*; *Simmons et al., 2010a*; *Simmons et al., 2010b*). In sub-surface decomposition, soil temperature (*Carter and Tibbett, 2006*; *Carter et al., 2008b*), moisture content, soil texture and type (*Fiedler and Graw, 2003*; *Tibbett et al., 2004*), soil pH (*Haslam and Tibbett, 2009*; *Prangnell and McGowan, 2009*) and bacterial/microbial community structure (*Carter and Tibbett, 2006* and *Carter and Tibbett, 2008a*; *Hopkins, 2008*; *Hopkins et al., 2000*; *Sagara et al., 2008*; *Zhang et al., 2021*) all play an important role in the postmortem fate of the buried cadaver.

During and following the process of skeletonization, the body undergoes a process of disarticulation and scattering of elements, due to the decomposition of muscles, connective tissues (particularly ligamentous structures, but also tendons), and joint capsules. The process of disarticulation specifically relates to the destruction, decomposition, or removal of (primarily) soft tissues which hold bony elements or joint surfaces of a skeleton together. Depending on the organism under consideration (and anatomical region), these soft tissues may comprise tendons, ligaments, muscles or skin, or components of joint capsules such as synovial membrane; the breakdown of these anatomical components allows individual bones, or complete elements and limbs, to disarticulate from the body. Because soft tissue anatomy varies with the functional anatomy of each joint, the process of disarticulation is highly complex (*Roksandic, 2002*). Subsequent dispersal is the increase or the decrease of the distance between bones (*Duday and Guillon,*

*2006*). The movement of bones out of normal anatomical association (termed necrodynamics; *Mickleburgh and Wescott, 2018*) occurs through the combined factors of connective tissue and joint decomposition, the effects of gravity on the bones, physical disturbance, and sediment type, grain size, and stability. Movement can be affected by both physical processes (including swelling and contraction of sediment under moisture cycles; after *Pokines et al., 2018*), biotic agents (animal and plant bioturbation; *Armour-Chelu and Andrews, 1994*; *Gabet et al., 2003*; *Pokines and Baker, 2013*) or human agents (*Hunter and Cox, 2005*). Scattering can be extensive and spatially widespread in open sub-aerial contexts due to the transport effects of gravity and water (*Haglund, 1993*), or scavengers (*Berryman, 2002*; *Carson et al., 2000*; *Haglund et al., 1989*; *Haynes, 1982*) or minimal, where skeletal elements and body units (head, thorax, appendicular skeleton, etc.) are supported and encapsulated by matrix such as soil, volcanic ash, or anthropogenic materials such as concrete (after *Hunter and Cox, 2005*; *Pokines and Baker, 2013*). Displacement of body elements requires the presence of open space for the bones to move within and into. Such spaces are termed primary where the empty space is the burial chamber itself (which can include mortuary structures, coffins, pits, or surfaces), and secondary spaces generated by the decomposition of soft tissues (*Duday, 2009*).

However, the order and pattern of rank disarticulation in the human (hominin) body differs somewhat from that observed in quadrupedal animals (after *Lyman, 1994*), but has been difficult to quantify precisely. Initial observations based on an understanding of musculoskeletal anatomy and biomechanical joint function led researchers *Duday, 2005* and *Duday, 2009*; *Knüsel, 2014*; *Knüsel and Robb, 2016* to define two primary types of articulations termed persistent (durable) and labile (non-durable). Persistent joints are considered to be those which are mechanically stable and strong, and which play a role in important biomechanical functions such as weight bearing and locomotion (such as the atlanto-occipital, humeroulnar, sacro-iliac, and tibio-tarsal joints, and the structures of the thoracic and lumbar vertebrae). Labile joints are considered to be those prone to rapid decomposition due to a lack of stabilizing and regulating bony articulations or soft tissue support (such as those between cervical vertebrae, carpal-metacarpal-phalangeal joints, the costosternal and costovertebral joints, scapulothoracic joint, and tarsal-metatarsal-phalangeal joints). Subsequent studies *Duday, 2009*; *Knüsel, 2014*; *Knüsel and Robb, 2016* have indicated that some joints were misclassified (i.e. femoroacetabular) and that the relationship between biomechanical competency in life and the persistence of such joint structures after death is not straightforward or simple. Current consensus suggests that persistent joints are represented by the atlanto-occipital, humeroulnar, thoracic and lumbar inter-vertebral, lumbosacral, sacroiliac, tibiofemoral, talocrural, and talocalcaneal joints. Labial joints are represented by the hyoid and its anchoring attachments, temporomandibular, cervical vertebral, scapulothoracic, glenohumeral, costosternal, costovertebral, acetabulofemoral, femoro-patella, carpal, metacarpal, tarsal, metatarsal, and phalangeal joints (though see *Schotsmans et al., 2022*: 512 for a summary of ongoing disagreements in classification).

In the practice of archaeothanatology (*Duday, 2005* and *Duday, 2009*; *Knüsel, 2014*; *Knüsel and Robb, 2016*), it is the differentiation between articulations of labile and persistent joints that is used to distinguish primary and secondary burials. Archaeothanatology uses the relative sequence of joint disarticulation to separate natural (biotic and abiotic) processes from those relating to the placement and treatment of the body (such as rapid primary burial). However, both archaeothanatological and forensic actualistic studies have indicated that patterns of joint disarticulation and final bone position covary with depending on original body disposition/placement as well as secondary environmental effects (*Haglund, 1993*; *Rodriguez and Bass, 1985*; *Gerdau-Radonic, 2012*). The overall pattern suggests that disarticulation proceeds in a generally craniocaudal direction and from the extremities to the core (from the distal appendages to the thorax). Onto this must be mapped the specific patterns of association, disarticulation, rotation, and displacement of each joint, if an understanding of the postmortem narrative is to be achieved. However, based on actualistic experimental forensic taphonomy, *Schotsmans et al., 2022* suggest that labile and persistent joints disarticulate over different time intervals, irrespective of their classification, and that joint disarticulation is complex and influenced by many covariables, with the pattern of disarticulation being highly influenced by slight differences in body position.

## 5.1: Spatial taphonomy of Puzzle Box – archaeothanatology of Hand and Foot 1

In situ sequential scanning of the fossil deposits in the Puzzle Box was undertaken during the initial excavations of the Dinaledi Chamber (2013 and 2014). This was accomplished with structured light scanning to assist in spatial recording and visualization of the position and disposition of individual bones. The technical methods and workflow employed are detailed by *Kruger et al., 2016*. In particular, the recording methodology provides significant spatial taphonomic information about the decomposition of body parts within the Puzzle Box burial environment. Most informative are Dinaledi Hand 1 and Foot 1. Dinaledi Hand 1 (H1) is a nearly complete right hand, found semi-articulated with the palmar surface facing upwards with the bones in close spatial association. Dinaledi Foot 1 (F1) is a semi-articulated adult right foot, found resting on its dorsal (plantar) surface.

Scanning of the fossils within their sediment matrix was enacted with a handheld Artec Eva 3D surface scanner. A series of sequential scans were taken of H1 and F1 prior to excavation, and then at different stages during the excavation process as sediment was removed, prior to recovery of the fossils. Following excavation, lifting, cleaning, and conservation, each bone was surface scanned using a NextEngine 3D Laser Scanner, with data exported as ply format files. The individual bone scans were then imported into Artec Studio 10 Professional along with the sequential surface scans undertaken during excavation. Translation and rotation of each 3D model of the individual hand bones was guided and aligned by surface models of the excavation. By layering each sequential scan of the in-situ hand and foot, a virtual three-dimensional representation of the excavation area surrounding H1 and F1 was produced, representing a cube of sediment within which the precise anatomical resting and relative position of each bone element can be seen. The encapsulating sediment was then extracted from the scan volume, leaving the precise spatial arrangement of each element as they would have been in the ground, prior to excavation and recovery. This 'reverse engineering' of the fossil deposits allows for the investigation of spatial taphonomic patterns within the surviving assemblage, as well as clear visualization of small-scale anatomical articulations and disjunctions between bones. This produced an accurate 3D reconstruction of the placement of H1 and F1 as they lay in situ, by virtually placing the bones of H1 and F1 back into the excavation pit. This allows for a precise and accurate visualization of the relative position of each elemental part, which imparts significant spatial information that can assist in taphonomic analyses and interpretation (*Appendix 6—figure 20*).

The extracted (sediment-removed) anatomical volumes indicate that both Hand 1 and Foot 1 retain their labile joint structures. In particular, H1 was almost completely articulated, and the reconstructed hand indicates that it was flexed (closed) or semi-closed during the process of skeletonization. The overall configuration of F1 suggests that it was resting on the dorsal surface. To retain such fragile, labile anatomical structures observed in H1 and F1 is most parsimoniously explained by rapid encapsulation within the burial environment, with the position of individual structures supported by sedimentary matrix as the soft tissues of the hand and foot decomposed – this matrix support would ensure little or no movement of the individual elements of the hand and foot during decomposition, supporting the interpretation of rapid burial.

# Appendix 6

## Figures and tables

**Appendix 6—table 1.** Particle-size distribution (PSD) of sediments based on the Folk and Ward Method.

| Sample name | Folk and Ward Method (µm) | | | |
|---|---|---|---|---|
| | Mean grain size | Sorting | Skewness | Kurtosis |
| DF1 | 372.21 | 2.60 | –0.07 | 0.78 |
| DF2 | 279.24 | 3.22 | –0.07 | 0.85 |
| DF3 | 442.02 | 2.28 | –0.06 | 0.75 |
| DF4 | 373.84 | 2.90 | –0.23 | 0.91 |
| DF5 | 356.86 | 2.61 | –0.03 | 0.76 |
| DF6 | 444.55 | 2.28 | –0.08 | 0.75 |
| DF7 | 375.33 | 2.54 | –0.05 | 0.77 |
| DF8 | 433.52 | 2.29 | –0.05 | 0.74 |
| DF9 | 379.40 | 2.49 | –0.03 | 0.74 |
| DF10 | 443.43 | 2.27 | –0.06 | 0.74 |
| DF11 | 351.16 | 2.81 | –0.12 | 0.83 |
| DF12 | 336.17 | 2.68 | –0.02 | 0.74 |
| DF13 | 369.75 | 2.51 | –0.02 | 0.73 |
| DF14 | 416.46 | 2.34 | –0.04 | 0.73 |
| DF15 | 334.22 | 2.79 | –0.05 | 0.80 |
| DF16 | 306.38 | 2.89 | –0.02 | 0.77 |
| DF17 | 362.50 | 2.55 | –0.02 | 0.74 |
| DF18 | 369.84 | 2.52 | –0.02 | 0.73 |
| DF19 | 533.93 | 2.31 | –0.28 | 1.01 |
| DF20 | 385.75 | 2.46 | –0.03 | 0.74 |
| DF21 | 261.64 | 3.29 | –0.05 | 0.82 |
| DF22 | 305.33 | 2.87 | –0.01 | 0.75 |
| DF23 | 270.15 | 3.34 | –0.08 | 0.86 |
| DF24 | 302.06 | 2.90 | –0.02 | 0.76 |
| DF25 | 340.90 | 2.82 | –0.08 | 0.84 |
| DF26 | 331.13 | 2.97 | –0.14 | 0.84 |
| DF27 | 208.47 | 3.89 | –0.07 | 0.84 |
| DF28 | 288.48 | 3.21 | –0.09 | 0.85 |
| DF29 | 317.93 | 2.87 | –0.05 | 0.80 |
| DF30 | 442.62 | 2.28 | –0.07 | 0.75 |
| DF31 | 268.14 | 3.32 | –0.07 | 0.85 |
| DF32 | 222.71 | 3.69 | –0.06 | 0.83 |
| DF33 | 337.60 | 2.71 | –0.03 | 0.76 |
| DF34 | 442.46 | 2.63 | –0.28 | 0.94 |

*Appendix 6—table 1 Continued on next page*

*Appendix 6—table 1 Continued*

| Sample name | Folk and Ward Method (µm) | | | |
|---|---|---|---|---|
| | Mean grain size | Sorting | Skewness | Kurtosis |
| SA1 | 323.81 | 2.84 | –0.05 | 0.80 |
| SA2 | 468.58 | 2.42 | –0.21 | 0.92 |
| SA3 | 277.86 | 3.27 | –0.09 | 0.85 |
| SA4 | 370.36 | 2.64 | –0.08 | 0.81 |
| SA5 | 345.26 | 2.66 | –0.03 | 0.76 |
| SA6 | 332.60 | 2.72 | –0.02 | 0.75 |
| SA7 | 308.77 | 2.84 | –0.01 | 0.75 |
| SA8 | 354.38 | 2.95 | –0.16 | 0.91 |
| SB1 | 237.81 | 3.51 | –0.05 | 0.82 |
| SB2 | 360.13 | 2.58 | –0.03 | 0.75 |
| SB3 | 358.12 | 2.67 | –0.06 | 0.80 |
| SC1 | 362.99 | 2.58 | –0.03 | 0.75 |
| SC2 | 328.85 | 2.81 | –0.05 | 0.80 |
| SC3 | 383.76 | 2.48 | –0.04 | 0.75 |
| SC4 | 332.60 | 2.70 | –0.02 | 0.75 |
| SE1 | 329.15 | 2.71 | –0.01 | 0.74 |
| SE2 | 265.82 | 3.36 | –0.09 | 0.84 |
| SE3 | 460.70 | 2.21 | –0.06 | 0.73 |
| SE4 | 385.72 | 2.48 | –0.04 | 0.75 |
| SE5 | 278.21 | 3.25 | –0.08 | 0.84 |

**Appendix 6—table 2.** Bulk major oxide chemistry and loss on ignition (LOI) obtained from x-ray fluorescence (XRF) in weight percentage (wt.%).

| Sample name | Sample locality | Al2O3 | CaO | Fe2O3 | K2O | MgO | MnO | P2O5 | SiO2 | TiO2 | LOI | SUM |
|---|---|---|---|---|---|---|---|---|---|---|---|---|
| DF1 | | 16.39 | 1.55 | 10.69 | 1.72 | 2.79 | 4.41 | 0.27 | 52.67 | 0.75 | 9.1 | 100.34 |
| DF2 | | 16.97 | 1.18 | 10.15 | 1.75 | 2.42 | 3.66 | 0.19 | 54.23 | 0.814 | 8.51 | 99.87 |
| DF3 | | 15.65 | 1.22 | 10.49 | 1.61 | 2.66 | 4.25 | 0.37 | 54.69 | 0.754 | 8.33 | 100.03 |
| DF4 | | 16.09 | 1.17 | 10.33 | 1.68 | 2.67 | 4.32 | 0.27 | 53.11 | 0.758 | 8.66 | 99.05 |
| DF5 | | 16.23 | 1.21 | 10.51 | 1.68 | 2.51 | 4.30 | 0.29 | 53.89 | 0.746 | 8.55 | 99.91 |
| DF6 | | 16.02 | 1.21 | 10.46 | 1.67 | 2.59 | 4.29 | 0.25 | 54.10 | 0.765 | 8.61 | 99.96 |
| DF7 | | 16.47 | 1.58 | 10.09 | 1.71 | 2.73 | 3.98 | 0.27 | 53.34 | 0.773 | 9.1 | 100.04 |
| DF8 | | 14.38 | 6.90 | 8.95 | 1.52 | 3.08 | 3.45 | 0.57 | 48.09 | 0.686 | 12.31 | 99.93 |
| DF9 | | 15.63 | 1.04 | 10.42 | 1.62 | 2.51 | 4.51 | 0.34 | 54.90 | 0.734 | 8.14 | 99.85 |
| DF10 | | 16.09 | 1.17 | 10.66 | 1.67 | 2.56 | 4.53 | 0.35 | 53.33 | 0.743 | 8.9 | 100.00 |
| DF11 | | 17.45 | 0.86 | 10.38 | 1.80 | 2.13 | 3.70 | 0.18 | 53.75 | 0.812 | 8.74 | 99.80 |
| DF12 | | 16.91 | 1.30 | 10.42 | 1.74 | 2.20 | 4.09 | 0.48 | 52.89 | 0.782 | 9.03 | 99.84 |
| DF13 | | 15.82 | 0.92 | 10.70 | 1.68 | 2.31 | 4.52 | 0.27 | 55.15 | 0.751 | 8.04 | 100.16 |
| DF14 | | 16.45 | 0.76 | 10.08 | 1.68 | 2.08 | 3.70 | 0.21 | 56.30 | 0.778 | 7.84 | 99.88 |
| DF15 | DF group: Sediments from above Features 1 and 2 collected during excavation and opening of features. | 15.15 | 1.00 | 10.79 | 1.70 | 2.61 | 5.28 | 0.23 | 53.78 | 0.712 | 8.44 | 99.69 |
| DF16 | | 16.09 | 0.88 | 10.69 | 1.71 | 2.39 | 4.60 | 0.23 | 53.82 | 0.764 | 8.33 | 99.49 |
| DF17 | | 15.35 | 0.88 | 10.36 | 1.60 | 2.33 | 4.35 | 0.25 | 56.10 | 0.731 | 7.93 | 99.89 |
| DF18 | | 15.48 | 0.95 | 10.46 | 1.61 | 2.27 | 4.41 | 0.27 | 55.39 | 0.727 | 8.29 | 99.86 |
| DF19 | | 16.73 | 0.80 | 10.07 | 1.71 | 2.25 | 3.62 | 0.16 | 55.60 | 0.805 | 8.25 | 100.00 |
| DF20 | | 17.30 | 0.97 | 10.13 | 1.72 | 2.22 | 3.49 | 0.20 | 54.30 | 0.804 | 8.69 | 99.82 |
| DF21 | | 14.57 | 0.89 | 9.55 | 1.49 | 2.00 | 3.59 | 0.33 | 59.28 | 0.72 | 7.62 | 100.03 |
| DF22 | | 14.85 | 0.89 | 10.22 | 1.54 | 2.07 | 4.13 | 0.32 | 57.61 | 0.723 | 7.86 | 100.21 |
| DF23 | | 14.13 | 0.96 | 9.78 | 1.42 | 2.32 | 3.86 | 0.34 | 58.71 | 0.702 | 7.65 | 99.86 |
| DF24 | | 14.69 | 1.00 | 9.50 | 1.42 | 2.33 | 3.45 | 0.39 | 58.44 | 0.719 | 7.6 | 99.53 |
| DF25 | | 15.02 | 0.91 | 9.51 | 1.44 | 2.36 | 3.50 | 0.29 | 58.57 | 0.743 | 7.67 | 100.01 |
| DF26 | | 14.78 | 1.08 | 9.28 | 1.43 | 2.32 | 3.20 | 0.40 | 58.76 | 0.757 | 7.61 | 99.62 |
| DF27 | | 15.18 | 0.98 | 9.54 | 1.52 | 2.12 | 3.74 | 0.36 | 57.51 | 0.763 | 7.73 | 99.43 |
| DF28 | | 13.71 | 1.01 | 9.31 | 1.35 | 2.18 | 3.78 | 0.31 | 60.38 | 0.702 | 7.27 | 99.99 |
| DF29 | | 14.80 | 1.55 | 9.96 | 1.52 | 2.22 | 4.60 | 0.75 | 55.51 | 0.701 | 8.05 | 99.66 |
| DF30 | | 14.26 | 1.18 | 9.64 | 1.47 | 2.02 | 4.02 | 0.51 | 58.44 | 0.707 | 7.5 | 99.74 |
| DF31 | | 16.19 | 0.88 | 10.10 | 1.69 | 1.95 | 3.88 | 0.30 | 56.21 | 0.781 | 7.96 | 99.94 |
| DF32 | | 14.88 | 1.36 | 9.80 | 1.53 | 1.94 | 3.90 | 0.63 | 57.03 | 0.734 | 7.7 | 99.51 |
| DF33 | | 14.10 | 2.11 | 9.45 | 1.49 | 1.91 | 3.92 | 1.16 | 57.55 | 0.692 | 7.49 | 99.87 |
| DF34 | | 16.09 | 2.03 | 10.19 | 1.68 | 2.01 | 3.54 | 1.14 | 54.58 | 0.784 | 8.05 | 100.11 |
| Mean | | 15.59 | 1.30 | 10.08 | 1.60 | 2.33 | 4.02 | 0.38 | 55.53 | 0.75 | 8.28 | |
| STD | | 0.98 | 1.04 | 0.48 | 0.12 | 0.28 | 0.45 | 0.23 | 2.50 | 0.04 | 0.87 | |

*Appendix 6—table 2 Continued on next page*

*Appendix 6—table 2 Continued*

| Sample name | Sample locality | Al2O3 | CaO | Fe2O3 | K2O | MgO | MnO | P2O5 | SiO2 | TiO2 | LOI | SUM |
|---|---|---|---|---|---|---|---|---|---|---|---|---|
| SA1 | | 14.02 | 4.70 | 9.12 | 1.44 | 4.29 | 3.60 | 0.75 | 48.99 | 0.666 | 11.89 | 99.47 |
| SA2 | | 15.12 | 0.81 | 10.52 | 1.56 | 2.48 | 5.14 | 0.16 | 54.71 | 0.7 | 8.34 | 99.54 |
| SA3 | SA group: Sediment from sterile areas east of Feature 1 | 13.00 | 5.05 | 9.03 | 1.36 | 4.67 | 4.19 | 0.68 | 47.76 | 0.615 | 12.47 | 98.84 |
| SA4 | | 15.46 | 0.88 | 12.84 | 2.01 | 2.59 | 7.06 | 0.19 | 48.44 | 0.648 | 9.18 | 99.30 |
| SA5 | | 13.34 | 3.71 | 8.88 | 1.37 | 3.94 | 3.75 | 0.47 | 52.66 | 0.659 | 10.71 | 99.48 |
| SA6 | | 15.32 | 0.81 | 10.99 | 1.76 | 2.45 | 6.24 | 0.14 | 51.72 | 0.704 | 9.05 | 99.19 |
| SA7 | | 13.19 | 4.85 | 8.99 | 1.37 | 4.52 | 3.94 | 0.55 | 48.99 | 0.642 | 12.32 | 99.36 |
| SA8 | | 14.29 | 3.16 | 10.22 | 1.52 | 3.75 | 4.40 | 0.41 | 49.87 | 0.708 | 10.82 | 99.14 |
| Mean | | 14.22 | 3.00 | 10.07 | 1.55 | 3.58 | 4.79 | 0.42 | 50.39 | 0.67 | 10.60 | |
| STD | | 0.99 | 1.89 | 1.38 | 0.23 | 0.94 | 1.26 | 0.24 | 2.41 | 0.03 | 1.59 | |
| | | | | | | | | | | | | |
| SB1 | SB group: Sediment from within Feature 1 | 14.85 | 0.82 | 8.99 | 1.58 | 1.52 | 4.53 | 0.14 | 58.77 | 0.652 | 7.41 | 99.25 |
| SB2 | | 14.49 | 1.20 | 9.92 | 1.42 | 2.18 | 4.31 | 0.46 | 56.52 | 0.682 | 7.8 | 98.99 |
| SB3 | | 16.65 | 0.72 | 8.83 | 1.50 | 1.68 | 3.48 | 0.09 | 57.65 | 0.713 | 7.97 | 99.28 |
| Mean | | 15.33 | 0.91 | 9.24 | 1.50 | 1.79 | 4.11 | 0.23 | 57.65 | 0.68 | 7.73 | |
| STD | | 1.16 | 0.25 | 0.59 | 0.08 | 0.35 | 0.55 | 0.20 | 1.13 | 0.03 | 0.29 | |
| | | | | | | | | | | | | |
| SC1 | SC group: Sediment between Features 1 and 2 | 15.32 | 0.76 | 8.25 | 1.69 | 1.38 | 2.68 | 0.14 | 61.98 | 0.767 | 6.8 | 99.77 |
| SC2 | | 17.10 | 0.54 | 9.30 | 1.74 | 1.59 | 3.24 | 0.10 | 56.77 | 0.827 | 8.24 | 99.45 |
| SC3 | | 12.03 | 0.69 | 7.45 | 1.42 | 1.37 | 3.72 | 0.10 | 65.79 | 0.561 | 6.18 | 99.30 |
| SC4 | | 15.43 | 0.95 | 10.57 | 1.64 | 1.86 | 5.24 | 0.35 | 53.63 | 0.717 | 9.16 | 99.55 |
| Mean | | 14.97 | 0.73 | 8.89 | 1.62 | 1.55 | 3.72 | 0.17 | 59.54 | 0.72 | 7.60 | |
| STD | | 2.12 | 0.17 | 1.35 | 0.14 | 0.23 | 1.10 | 0.12 | 5.40 | 0.11 | 1.35 | |
| | | | | | | | | | | | | |
| SE1 | SE group: Sediment from vertical profile south of Feature 1 | 15.03 | 0.86 | 10.31 | 1.59 | 2.11 | 4.83 | 0.23 | 55.68 | 0.719 | 7.91 | 99.27 |
| SE2 | | 14.17 | 0.764 | 10.03 | 1.47 | 2.08 | 4.498 | 0.17 | 58.16 | 0.667 | 7.54 | 99.55 |
| SE3 | | 16.01 | 0.332 | 7.346 | 1.54 | 1.58 | 0.498 | 0.1 | 65.01 | 0.959 | 6.25 | 99.62 |
| SE4 | | 15.3 | 0.731 | 10.36 | 1.55 | 2.28 | 4.697 | 0.15 | 55.66 | 0.731 | 8.05 | 99.52 |
| SE5 | | 15 | 0.688 | 10.27 | 1.55 | 2.39 | 4.882 | 0.15 | 55.38 | 0.708 | 8.12 | 99.13 |
| Mean | | 15.10 | 0.68 | 9.66 | 1.54 | 2.09 | 3.88 | 0.16 | 57.98 | 0.76 | 7.57 | |
| STD | | 0.66 | 0.20 | 1.30 | 0.05 | 0.31 | 1.90 | 0.05 | 4.09 | 0.12 | 0.77 | |
| | | | | | | | | | | | | |
| FS2280 | Puzzle Box excavation | 15.16 | 1.17 | 9.065 | 1.6 | 2.17 | 16.05 | 0.17 | 41.21 | 0.66 | 10.66 | 97.92 |

**Appendix 6—table 3.** Trace element and rare earth elements (REEs) results.

Separate Excel spreadsheet. Appendix 4: Instrumental parameters for WDXRFS (MagiX PRO). Measurements were carried out under vacuum with a rhodium X-ray tube without tube filter, a sample spinner, 25 mm collimator mask, and a flow counter. Elements were analyzed in the order of decreasing X-ray energy; i.e., from Ni to Na.

| Element and X-ray line | Crystal | Collimator | Tube kV | Offset Bg1 (°2θ) | Offset Bg2 (°2θ) | PHD1 LL | PHD1 UL | Comment |
|---|---|---|---|---|---|---|---|---|
| Ni Kα | LiF 220 | 150 μm | 60 | −1.23 | 1.9 | 23 | 59 | |
| Fe Kα | LiF 220 | 150 μm | 60 | −3.25 | 5 | 18 | 60 | |
| Mn Kα | LiF 220 | 150 μm | 60 | −3.7 | 4.91 | 16 | 61 | |
| Cr Kα | LiF 220 | 150 μm | 50 | −2.4 | 2.95 | 14 | 60 | |
| V Kα | LiF 220 | 150 μm | 50 | −2.2 | 2.75 | 12 | 61 | |
| Ti Kα | LiF 200 | 150 μm | 40 | | | 36 | 62 | Used bg meas. of Ba. |
| Ba Lα | LiF 200 | 150 μm | 40 | −4.95 | 2.14 | 36 | 62 | |
| Ca Kα | LiF 200 | 150 μm | 30 | −4.7 | 3.55 | 36 | 62 | |
| K Kα | LiF 200 | 150 μm | 30 | −5.3 | 5.5 | 36 | 63 | |
| S Kα | Ge 111 | 550 μm | 30 | 5.7 | | 35 | 64 | Bg factor 1.16. |
| P Kα | Ge 111 | 550 μm | 30 | −10.24 | 6.5 | 35 | 64 | |
| Si Kα | PE 002 | 550 μm | 30 | −4.2 | 5.69 | 33 | 67 | |
| Br Lβ$_1$ | PE 002 | 550 μm | 30 | | | 33 | 67 | Used bg of Al. |
| Al Kα | PE 002 | 550 μm | 30 | −11.28 | | 32 | 67 | Bg factor 1.0. |
| Mg Kα | PX1 | 150 μm | 30 | -2 | | 34 | 66 | Used bg1 meas. of Na. |
| Na Kα | PX1 | 150 μm | 30 | −2.384 | 2 | 32 | 68 | |

Notes: LL and UL are the lower and upper level, respectively, of the pulse height analyser window. PX1 is a synthetic multilayer with a nominal 2d spacing of 5 nm.

**Appendix 6—table 4.** Instrument parameters for ICPMS (NexION, Perkin-Elmer).

| Sample introduction system | Cross-flow with Scott double-pass spray chamber |
|---|---|
| Nebuliser gas flow | Ca. 0.8–0.9 L/min |
| Sample uptake rate | Ca. 1 mL/min |
| Skimmer cones | Nickel |
| RF power | 1200 W |
| Data acquisition | Peak hopping mode, 20 sweeps per reading, 1 reading per replicate, 3 replicates. Dwell time 40ms. |

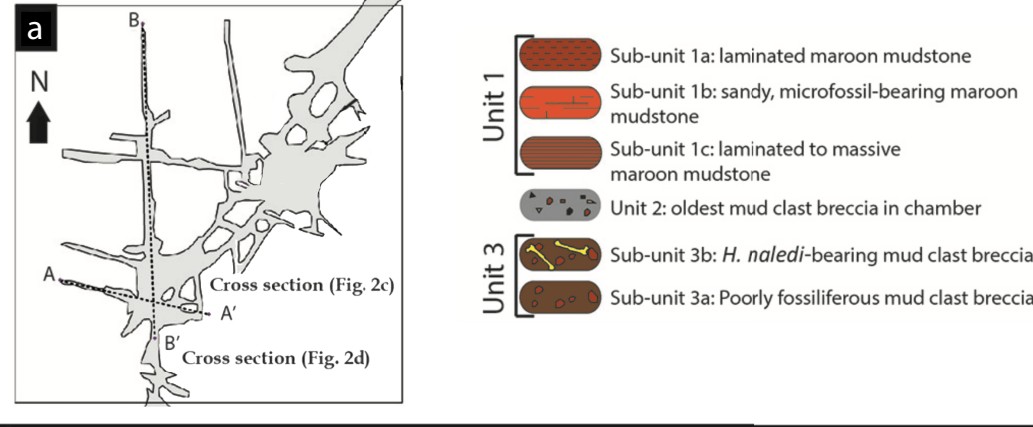

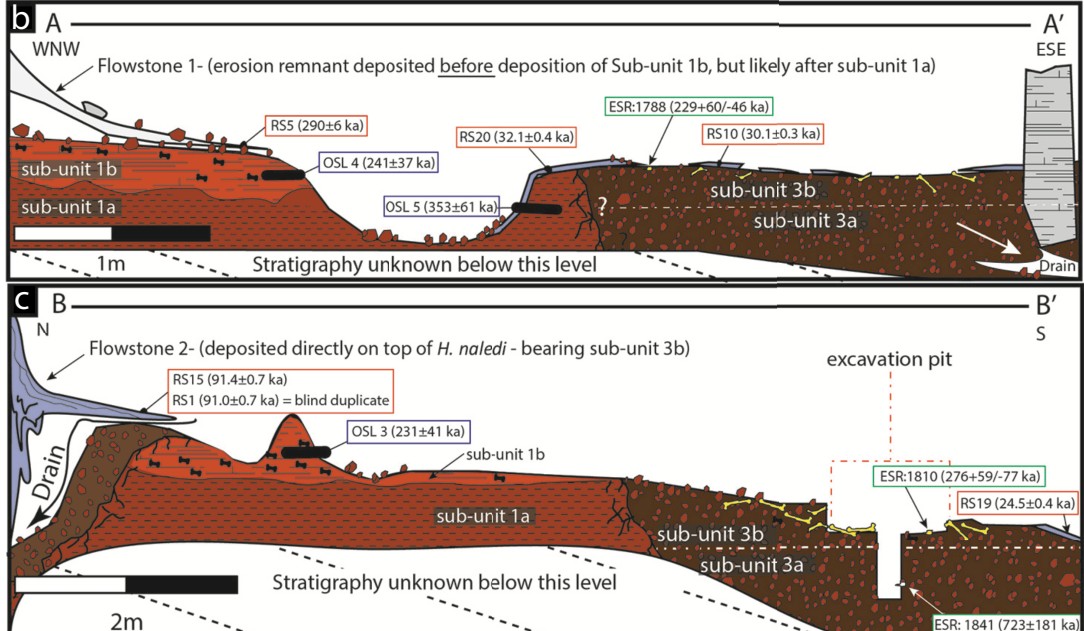

**Appendix 6—figure 1.** Geological face map and cross-sections through the sediments at different locations in the Dinaledi Chamber, illustrating the relationships between the flowstone groups and sedimentary units. Figure modified from *Dirks et al., 2017*; *Figure 2* to remove hypothesized subsurface floor drain, discussed in text.

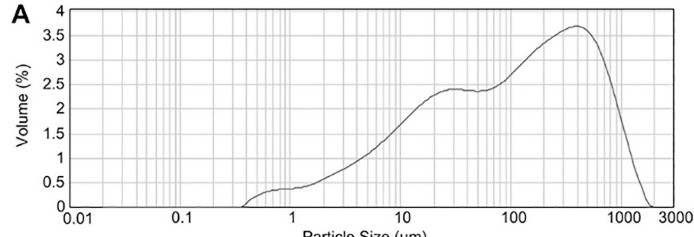

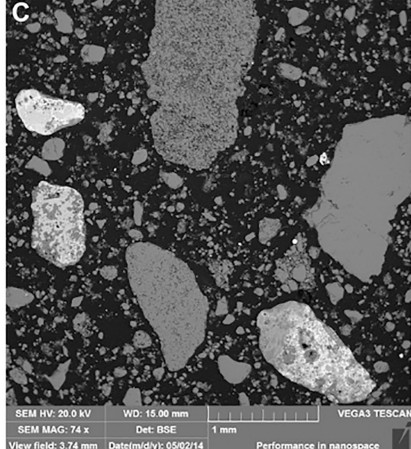

| | DB1 | UW 101-SO31 | UW 101-SO34 | UW 101-SO39 |
|---|---|---|---|---|
| SiO₂ | 84.68 | 51.33 | 51.16 | 51.79 |
| TiO₂ | 0.37 | 0.7 | 0.62 | 0.7 |
| Al₂O₃ | 4.33 | 15.85 | 13.96 | 16.2 |
| MnO | 0.88 | 3.92 | 4.26 | 4.32 |
| Fe₂O₃ | 3.73 | 10.43 | 10.7 | 10.95 |
| MgO | 0.46 | 3.21 | 4.01 | 3.1 |
| CaO | 0.62 | 2.13 | 2.4 | 1.06 |
| K₂O | 0.48 | 1.52 | 1.35 | 1.61 |
| Na₂O | 0.06 | 0.1 | 0.09 | 0.08 |
| P₂O₅ | 0.16 | 0.97 | 0.48 | 0.29 |
| BaO | - | 0.06 | 0.06 | 0.06 |
| SO₃ | - | 0.09 | 0.1 | - |
| LOI | 3.27 | 9.02 | 9.84 | 9.04 |
| Sum | 99.04 | 99.33 | 99.03 | 99.2 |

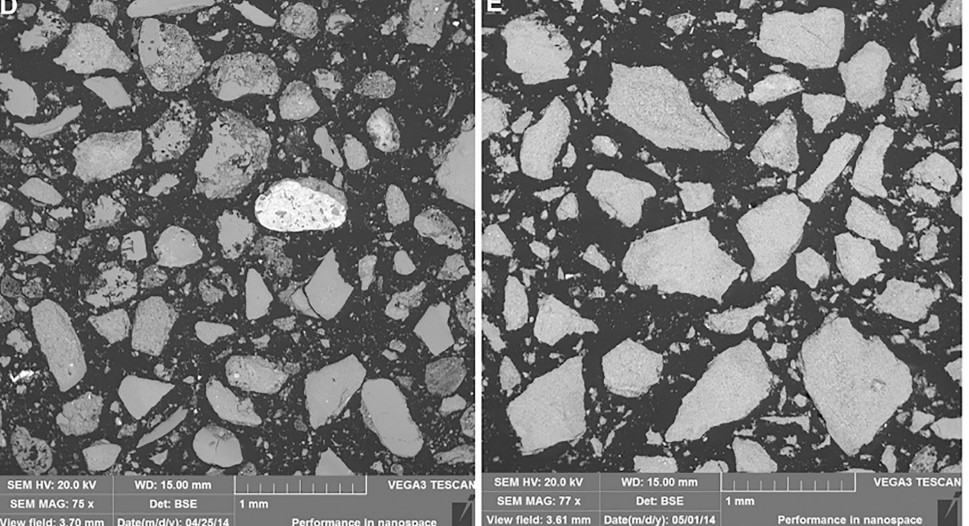

**Appendix 6—figure 2.** Data and characteristics of cave floor sediments (Facies 2) from the Dinaledi and Dragon's Back Chambers. Figure taken from ***Dirks et al., 2015***; ***Figure 5*** - https://doi.org/10.7554/eLife.09561.007. Original figure legend: (**A**) Grain size distribution of sample UW101-SO-39 (***Figure 2C***). The bulk of the sample material falls within a size fraction corresponding to silt and fine-grained sand. Some coarser mudstone fragments did not disintegrate when immersed in water, likely due to considerable Mn- and Fe-oxide micro-concretionary development in the orange mudstone. Because some mudstone fragments are well lithified, the particle size distribution is skewed towards the coarser grain-size values. (**B**) Results of XRF analyses of bulk samples of three floor sediments from the Dinaledi Chamber (UW101-SO31, −34 and −39) and one from the Dragon's Back chamber (DB-1). The sample from the Dragon's Back Chamber has a radically different composition from those of the Dinaledi Chamber, with the high SiO2 content reflecting its dominance of quartz. The Dinaledi samples have much higher Al2O3 and K2O contents than DB-1, indicating a higher content of clay minerals and mica, and higher CaO, MgO, MnO, and total Fe oxide contents which reflect alterations and inclusions. The higher P2O5 content of the Dinaledi samples is probably located in comminuted bone fragments which are seen macroscopically. The volatiles content (LOI) of the Dinaledi samples is also higher than in DB-1, in accord with a higher total clay mineral and mica content. (**C**–**E**) Backscattered electron (BSE) wide-field images of grain mounts from floor sediments. Brighter shades indicate the presence of heavier elements, mainly Mn and Fe in altered grains. (**C**) DB-1, Dragon's

*Appendix 6—figure 2 continued on next page*

*Appendix 6—figure 2 continued*
Back Chamber, large fragments are quartz and chert, partly altered. (**D**) UW101-SO34. (**E**) UW101-SO39. In these samples, the large fragments are almost exclusively clay; note their angular shape, which shows these to be locally derived.

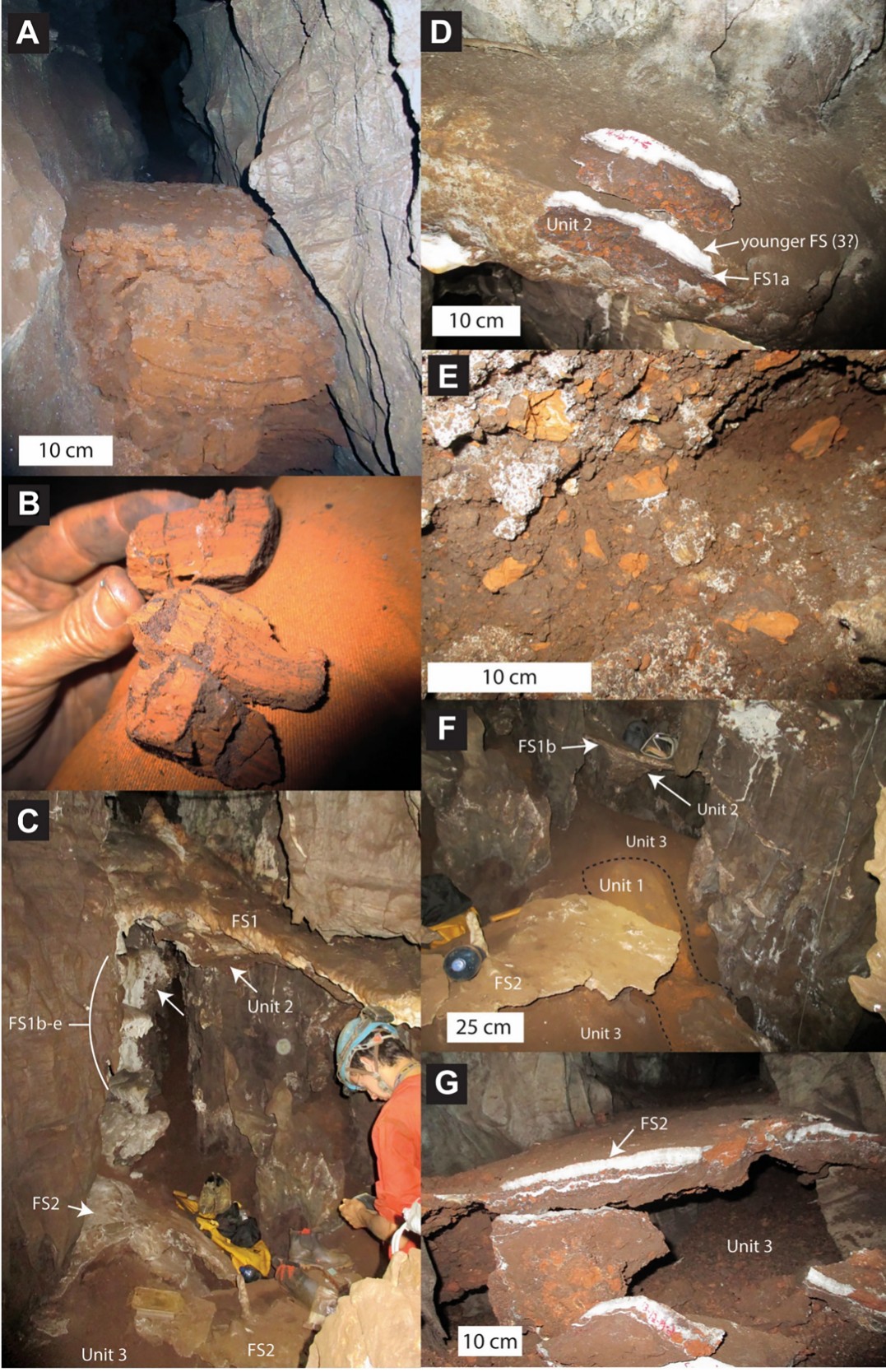

**Appendix 6—figure 3.** Stratigraphic units and flowstones observed in the Dinaledi Chamber, showing Unit 1 sediment lamination and laminated orange-red mud clasts. Figure taken from *Dirks et al., 2015*; *Figure 4 - Appendix 6—figure 3 continued on next page*

*Appendix 6—figure 3 continued*

https://doi.org/10.7554/eLife.09561.006.

Original figure legend: (**A**) Erosional remnant of horizontally laminated Unit 1 strata (Facies 1). (**B**) Close-up view of Unit 1 (Facies 1 a) showing fine laminations and small invertebrate burrows (note fine sand infilling in burrows). (**C**) Overview photo of the Dinaledi Chamber, directly to the east of the entrance point into the chamber. Photo shows distribution of Flowstones 1–3 and stratigraphic Units 2 and 3. (**D**) Close-up view of Flowstone 1 encasing sediment of Unit 2. Note that several generations of flowstone (Flowstones 1 a–e) are coating Unit 2. The thin, clear lower layer is Flowstone 1 a, and the overlying white flowstone is either Flowstone 2 or 3. (**E**) Close-up view of Unit 2, consisting of generally poorly-cemented Facies 2 sediment. (**F**) View of the chamber floor near the entry point. On the cave floor, a large erosional remnant of Unit 1 (orange laminated mudstone of Facies 1 a) is surrounded by mud-clast breccia of Unit 3 (main hominin bearing unit). Note that Flowstone 2 has been undercut by post-depositional erosion of Unit 3, which, in this location, has resulted in a lowering of the floor by as much as 25 cm. (**G**) Flowstone 2 overlying Unit 3 in one of the chamber's side passages. In this location, Unit 3 has also been partly eroded after deposition from underneath the flowstone drape, leaving a hanging remnant, with some indurated sediment of Unit 3 attached to its base. Note the continued deposition of sediment above Flowstone 2.

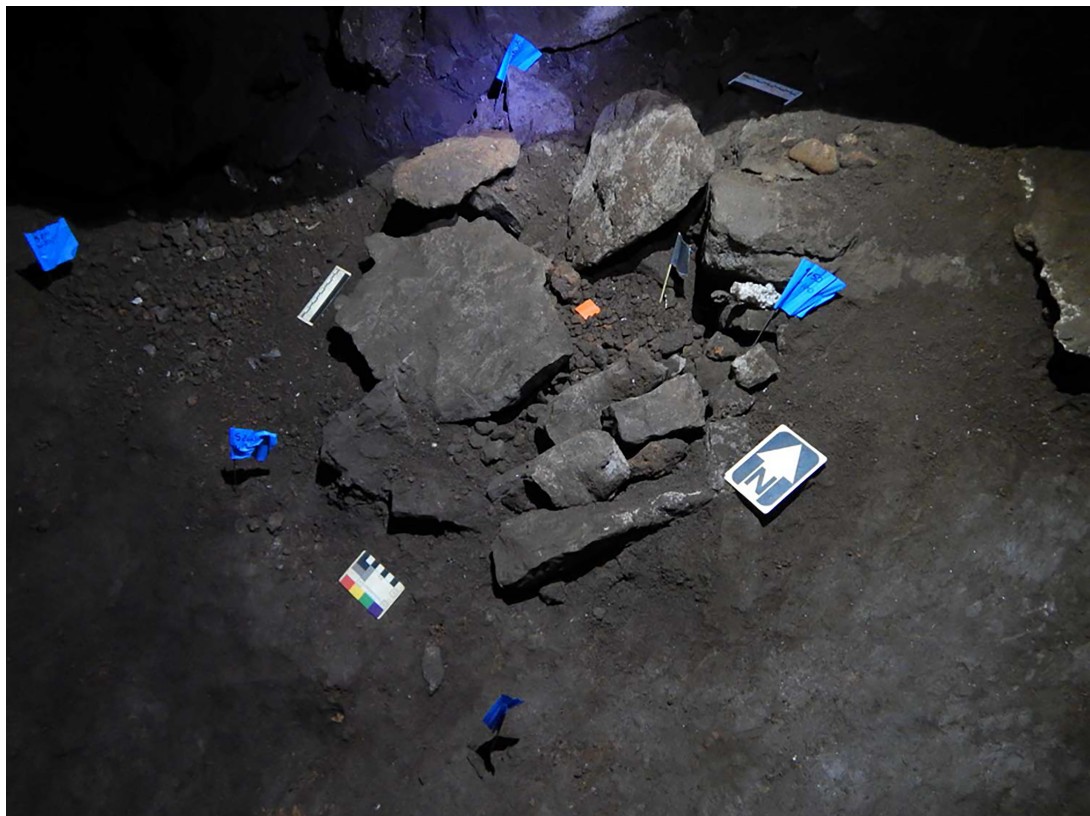

**Appendix 6—figure 4.** Hill Antechamber excavation unit S150W150 prior to opening excavation. This area had a collection of non-overlapping flat stones on the surface.

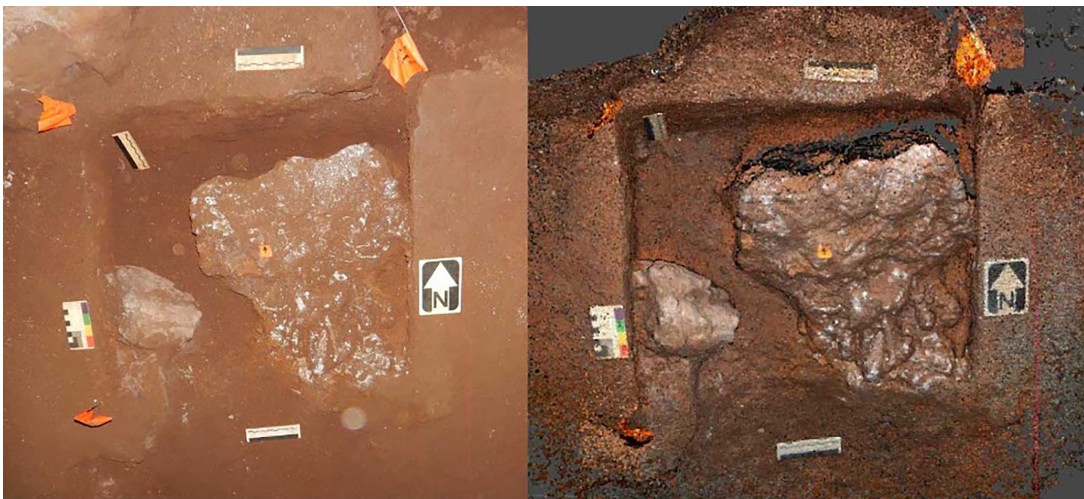

**Appendix 6—figure 5.** Top surface of the Hill Antechamber Feature after full exposure. Photo (left) and 3D model based on photogrammetry (right). The bone material at the northmost extent of the feature is powdery and highly fragmented, with more complete skeletal elements visible toward the south.

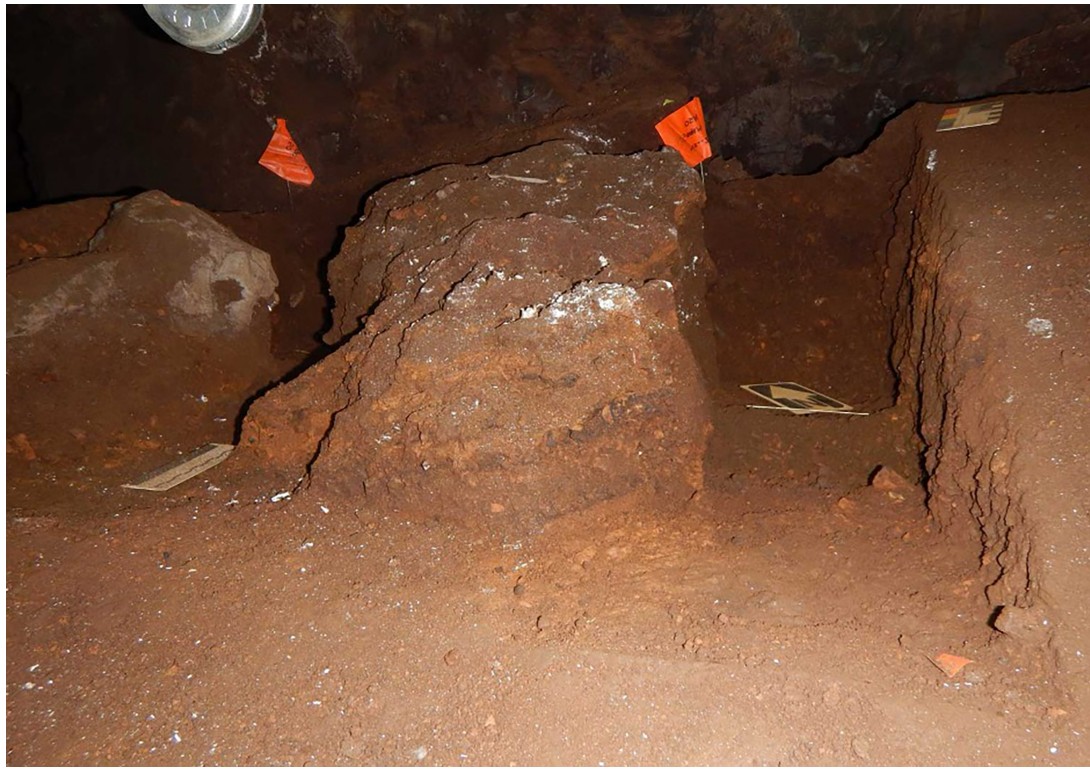

**Appendix 6—figure 6.** Hill Antechamber Feature after pedestaling and separation from surrounding sediment.

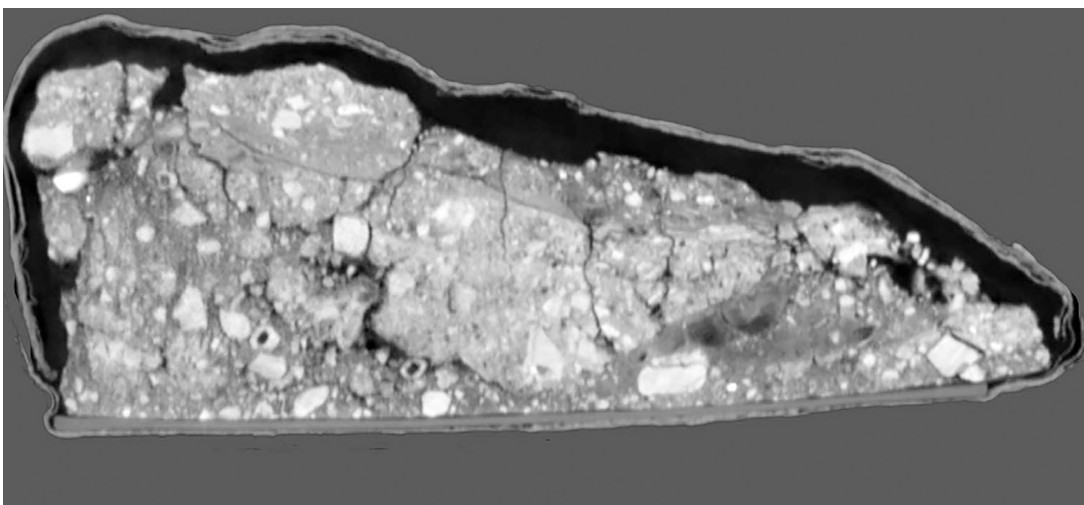

**Appendix 6—figure 7.** Sagittal (north-south) section of Hill Antechamber Feature. North is at the left of the frame. This section is at approximately 55% of the east-west breadth of the feature. The articulated foot is visible in longitudinal section at right of frame, with cross-sections of other bones and teeth further to the left of frame. The layer that constitutes the top of the feature is packed with bone material including articulated, semi-articulated, and loose material, flattened into less than 5 cm thickness.

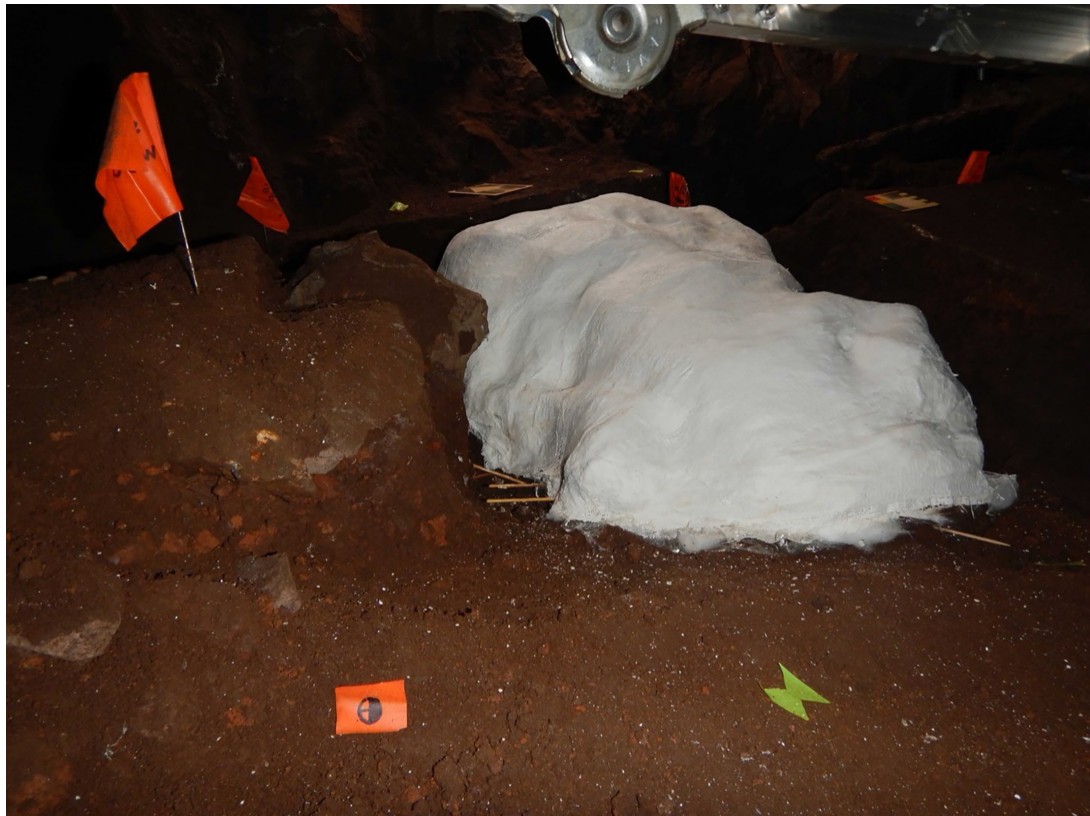

**Appendix 6—figure 8.** Hill Antechamber Feature after jacketing of largest block (U.W.101–2076) in six layers of plaster bandages, prior to separation from sediment at its base.

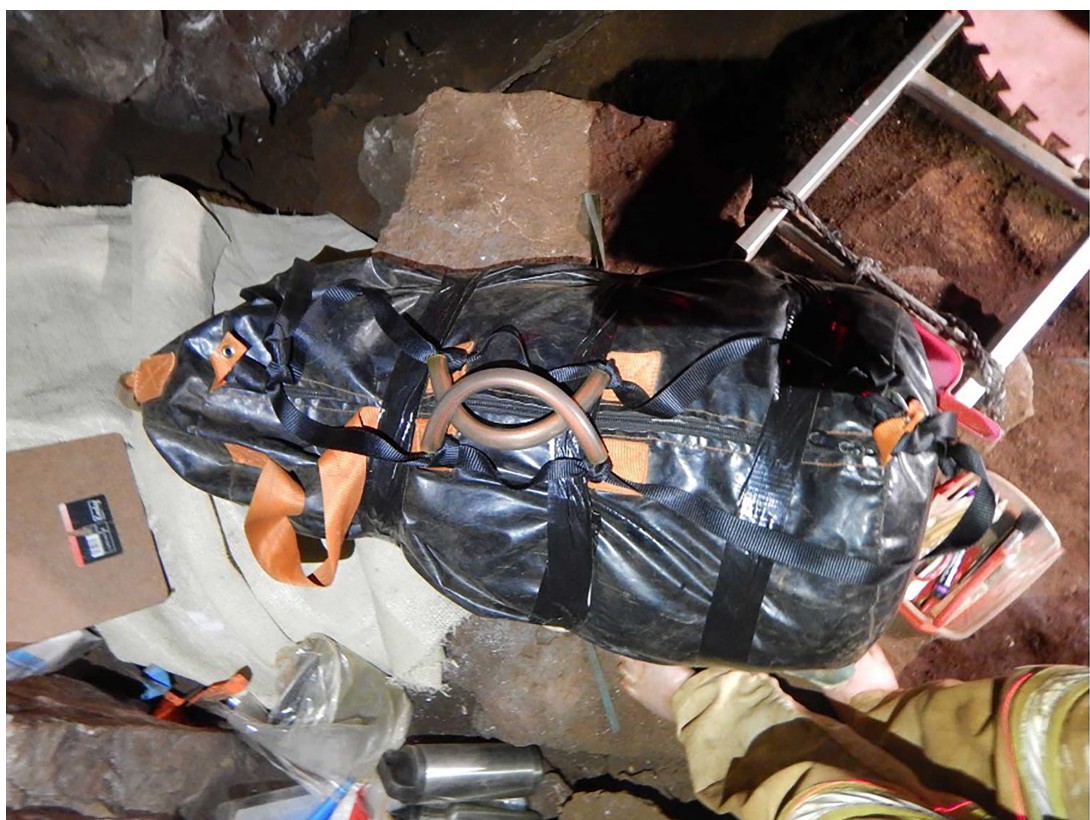

**Appendix 6—figure 9.** Fossil mass within plaster jacket after packing into waterproof caving bag for exit from the cave system.

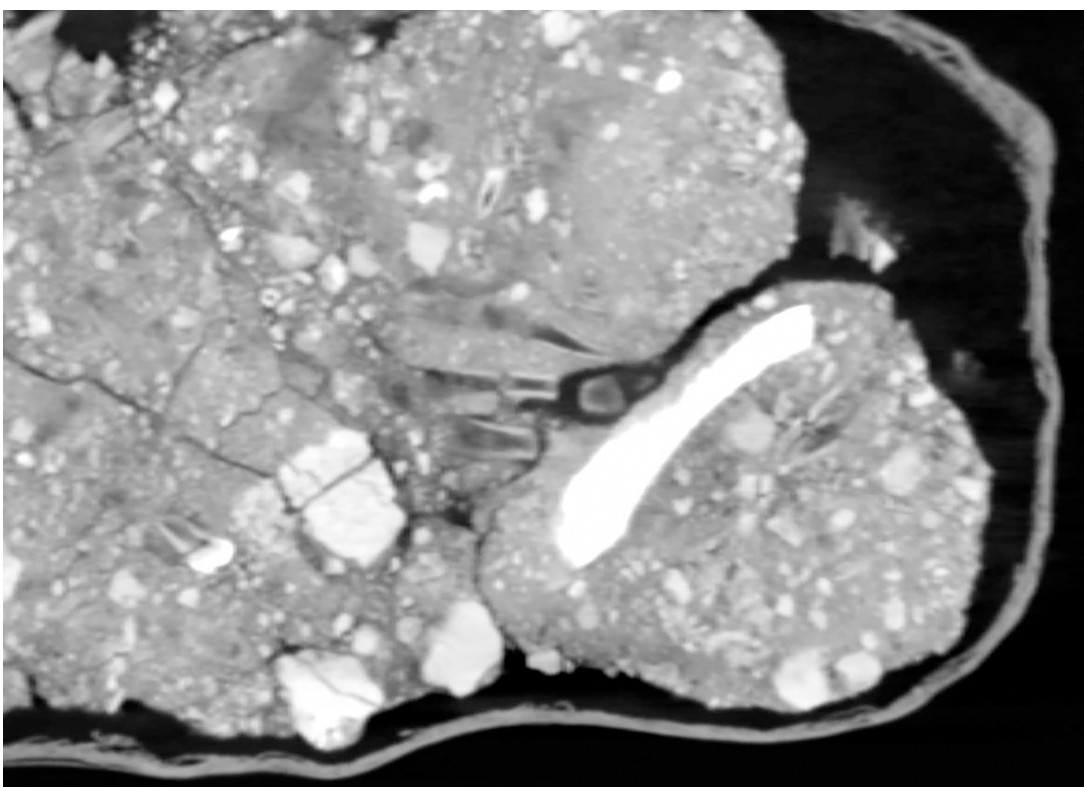

**Appendix 6—figure 10.** Detail of medical-resolution CT image of Hill Antechamber feature. This is a horizontal section with north at the bottom of the frame and west at the right of the frame. In this image, the bright object at lower left is a cross section of HAA1. At its left, cross sections of four rays of the articulated hand are visible; there is also a bone visible in the gap or space adjacent to the artifact that is a fragment of intermediate phalanx.

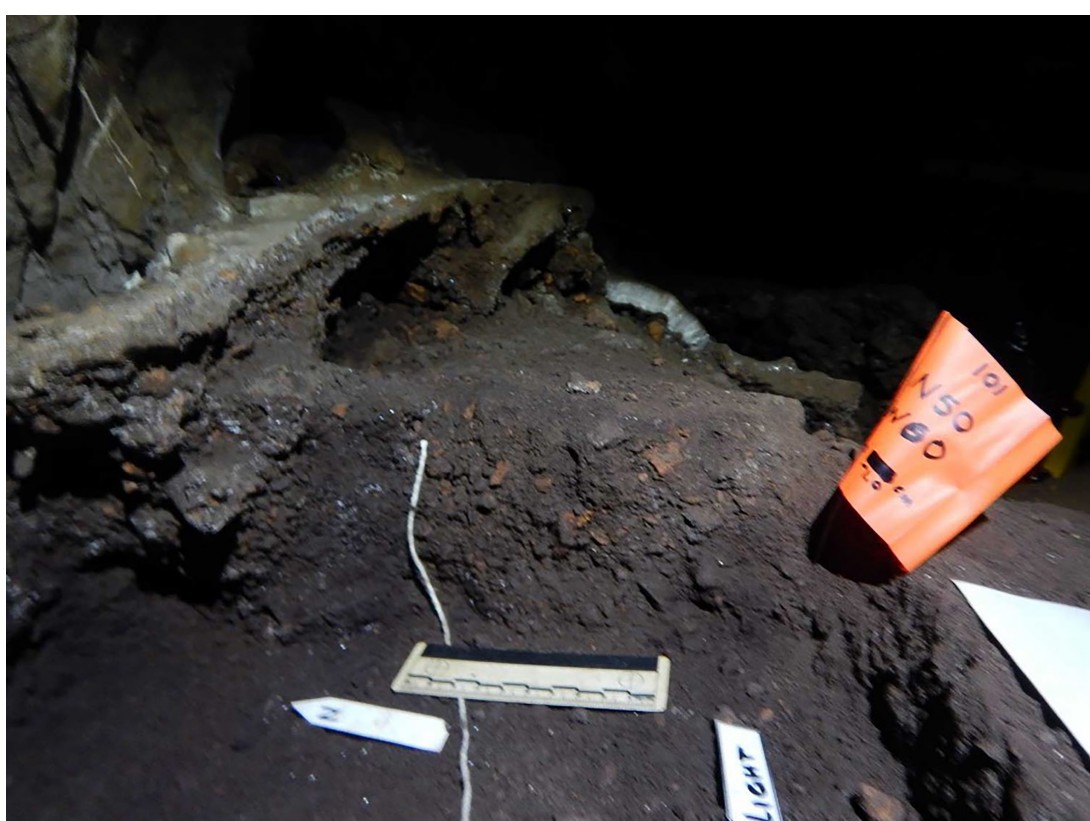

**Appendix 6—figure 11.** Sediment profile showing east wall of N100W50 excavation unit in Hill Antechamber. The sediment is a dark brown unlithified breccia containing laminated orange-red mud (LORM) clasts. In this unit, the clasts make up a small fraction of the sedimentary deposit with little evidence of layering or stratigraphic differentiation.

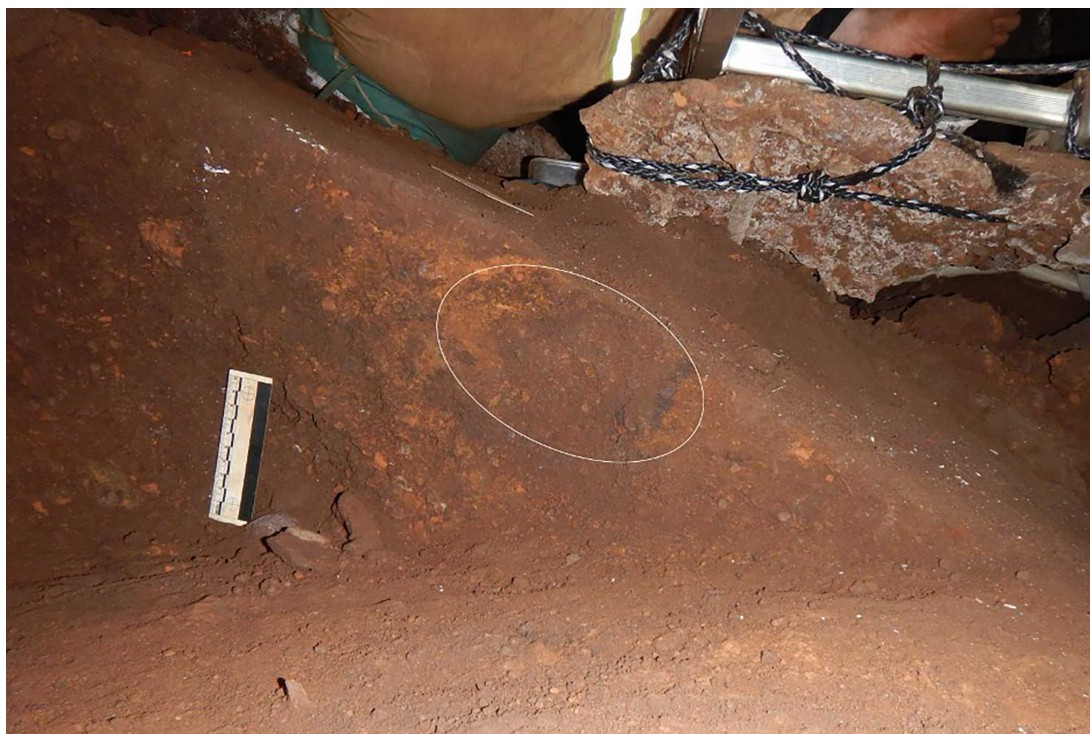

**Appendix 6—figure 12.** Hill Antechamber excavation east wall profile. North is at left of frame. The ellipse is drawn around a 15 cm by 10 cm by 5 cm collapse of the wall that accompanied removal of the plaster-jacketed block, resulting in some distortion to the profile in this localized area. The darker patch at the right side of the ellipse is a shadow from the collapse edge, not a dark-colored inclusion in the sediment. Layering of the unlithified mud clast breccia is visible, with some layers having a higher content of LORM clasts and laminae, with color variation less evident here than in the section shown in *Figures 11 and 13* The layering is approximately parallel to the slope of the chamber floor. LORM content and clasts are less toward the north edge of the excavation, at left of frame.

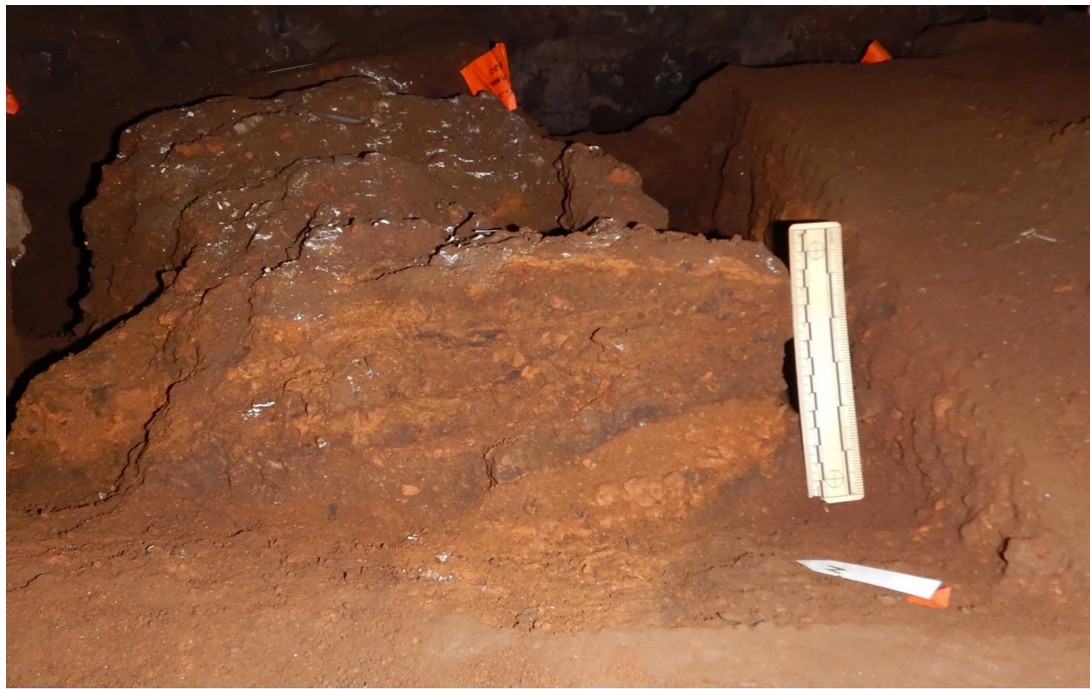

**Appendix 6—figure 13.** Stratigraphic profile of sediments directly adjacent to the south boundary of the Hill Antechamber Feature. At the left of the image is the west side of the unit. This profile represents the sedimentary structure of the S50W100 and S50W50 units before the excavation reduced the feature to its rounded south edge. Horizontal layering of unlithified mud clast breccia (UMCB) with denser orange-red LORM-clast bearing laminae is evident for the top 10 cm of the profile. This layering becomes subhorizontal with horizontal depth with an east-west trending slope. The lowest layer visible trends into the horizontal floor of the excavation unit. This layering is not paralleled by the skeletal material, fill, or LORM clasts visible within the feature itself.

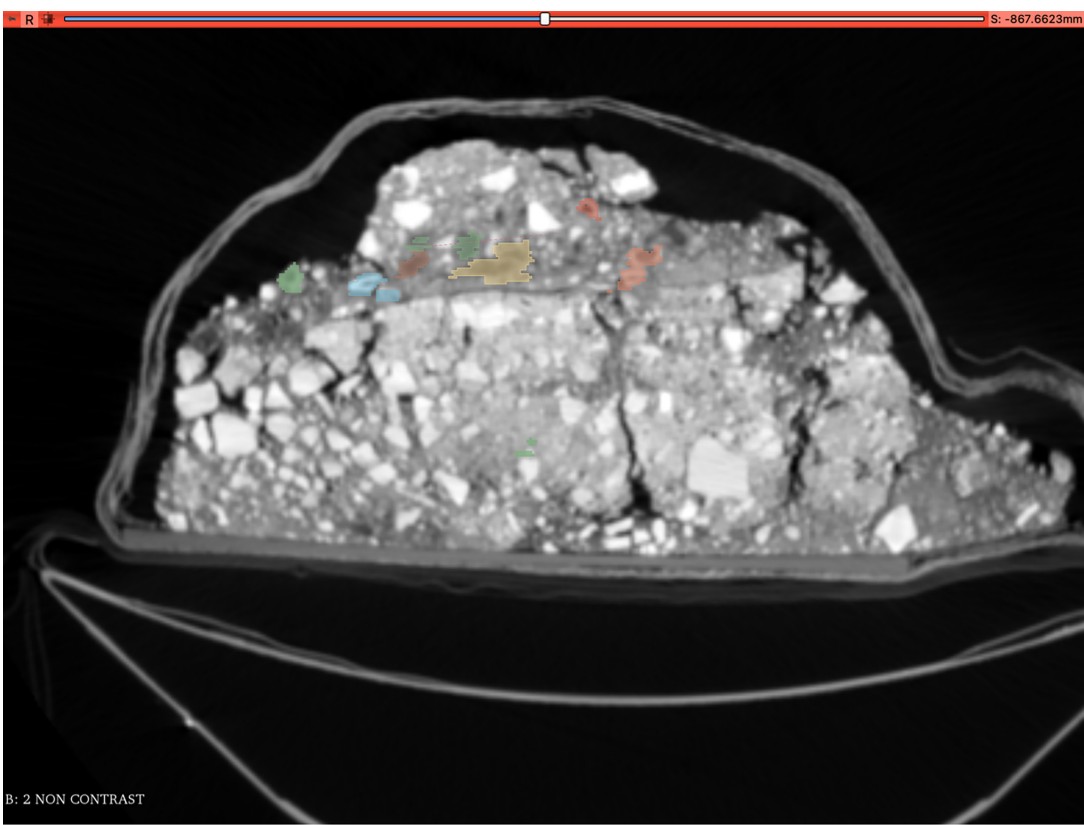

**Appendix 6—figure 14.** CT section of Hill Antechamber Feature. This east-west transverse section is at approximately 50% of north-south length of the feature. At the bottom of the section, many small LORM clasts are visible, with two notable voids taking the form of vertical cracks. The disordered array of LORM clasts continues to the right of the image with frequent voids (west side of feature).

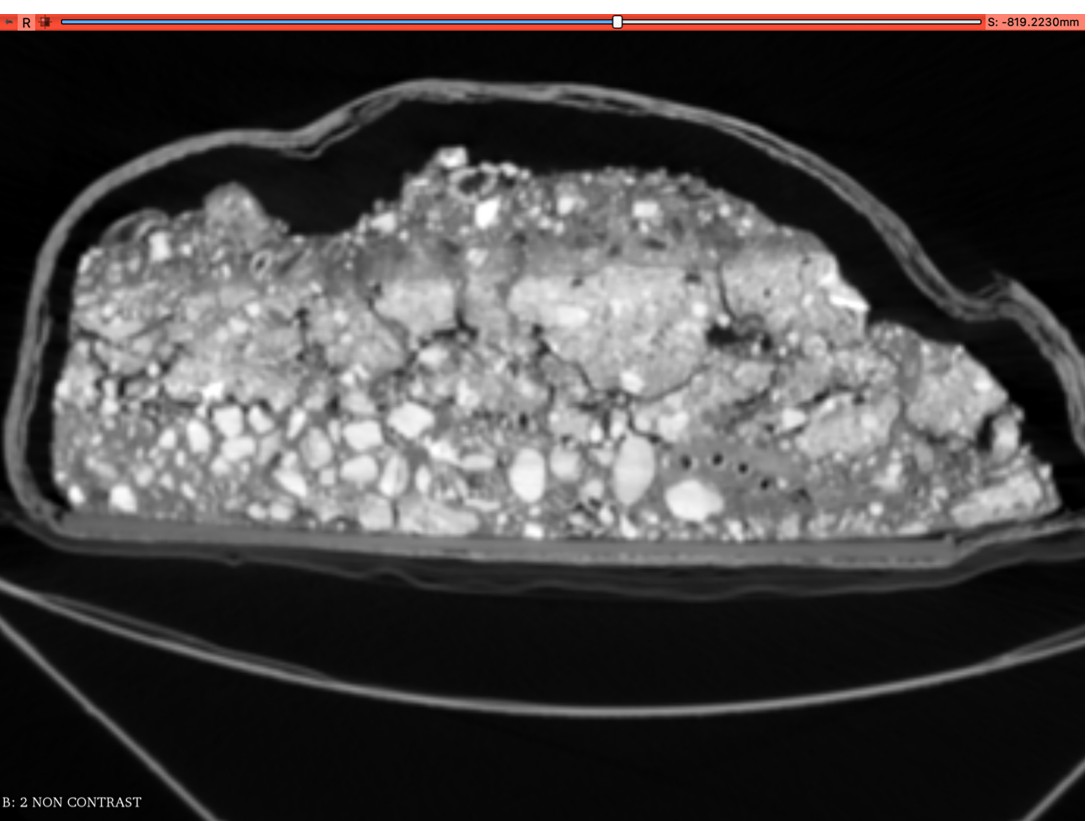

**Appendix 6—figure 15.** CT section of Hill Antechamber feature. This is a transverse section on the east-west plane at approximately 65% of the north-south length. The five rays of the articulated foot are visible at lower left of the section. This section cuts across the metatarsals. The bones of the foot are immediately surrounded by a halo of sediment that approximates the shape of the foot's soft tissue. This lower-density sediment separates the bones of the foot from surrounding, more radio-opaque LORM clasts and sediment. Above the foot, some small voids in the sediment are visible; small voids are also visible toward the left of this section directly above a disordered arrangement of LORM clasts.

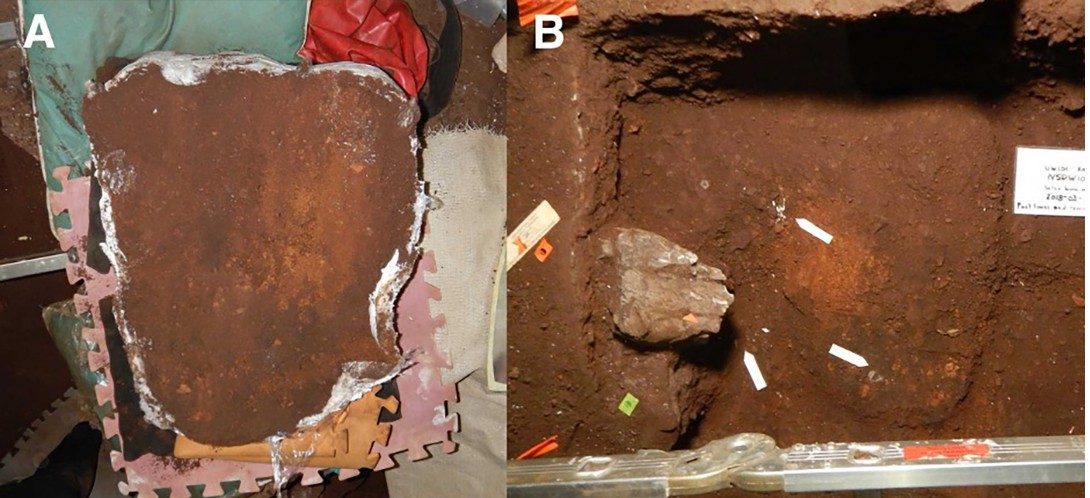

**Appendix 6—figure 16.** Bright orange LORM patch immediately beneath Hill Antechamber feature. (**A**) Bottom of Hill Antechamber feature after jacketed extraction and inversion. The bright orange patch is visible centrally slightly toward the right (west) side of the inverted feature (**B**) Excavation unit immediately after jacketed extraction of feature and cleaning of surface. The corresponding orange patch is visible at the center of the unit.

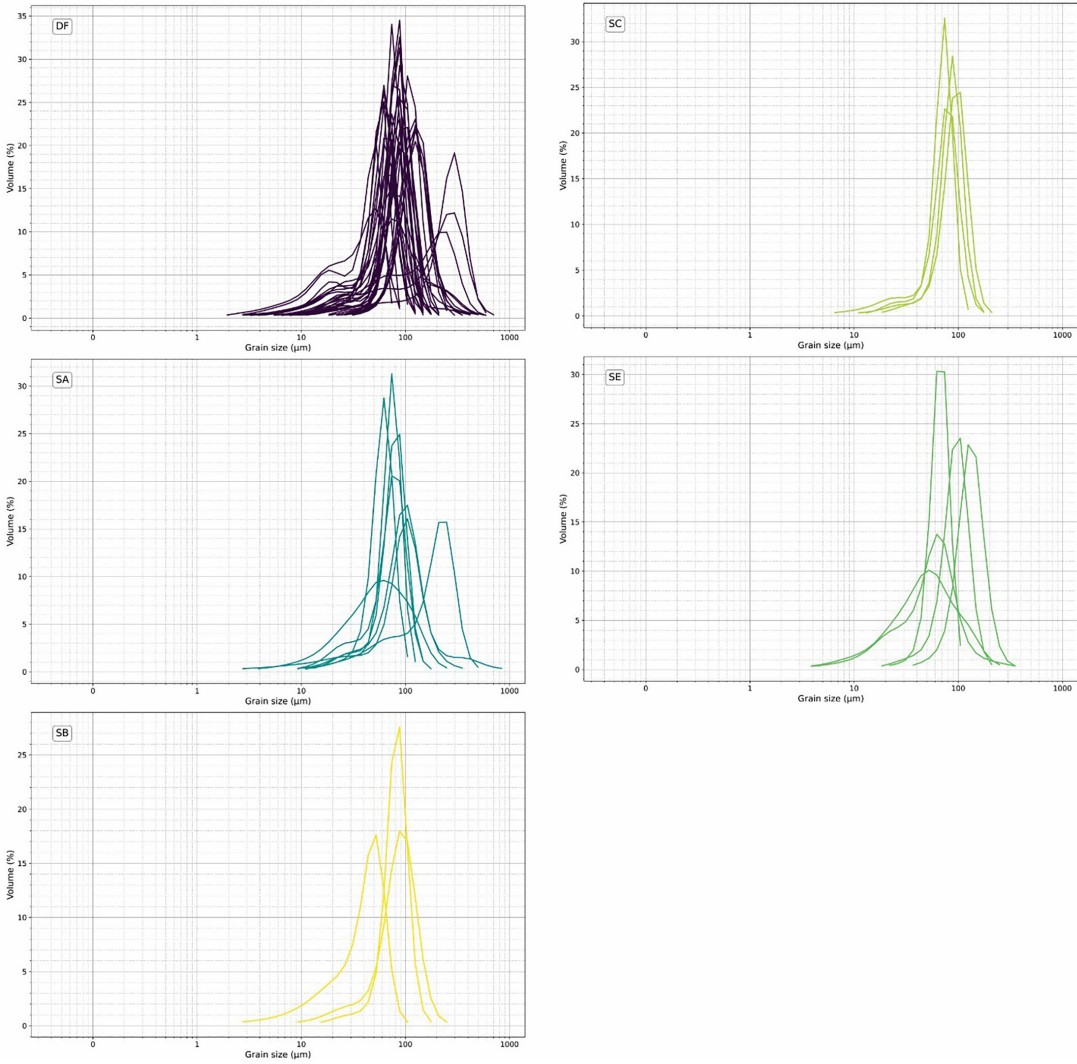

**Appendix 6—figure 17.** Grain size distribution curves of the five sediment groups analyzed within and around Feature 1.

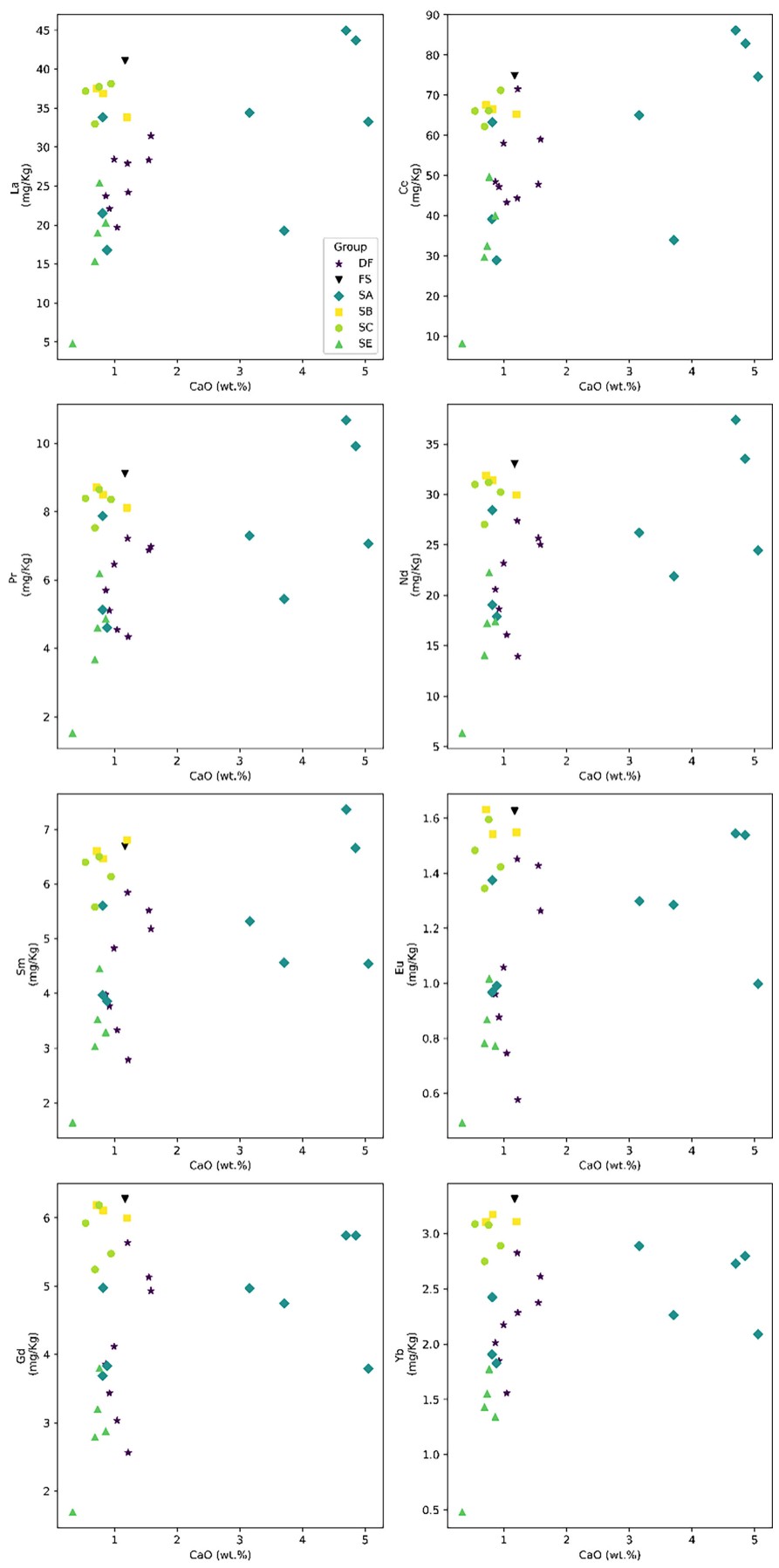

**Appendix 6—figure 18.** Harker variation plots illustrating the relationship between CaO and selected rare earth elements (REEs) for the five sediment groups analyzed within and around Feature 1.

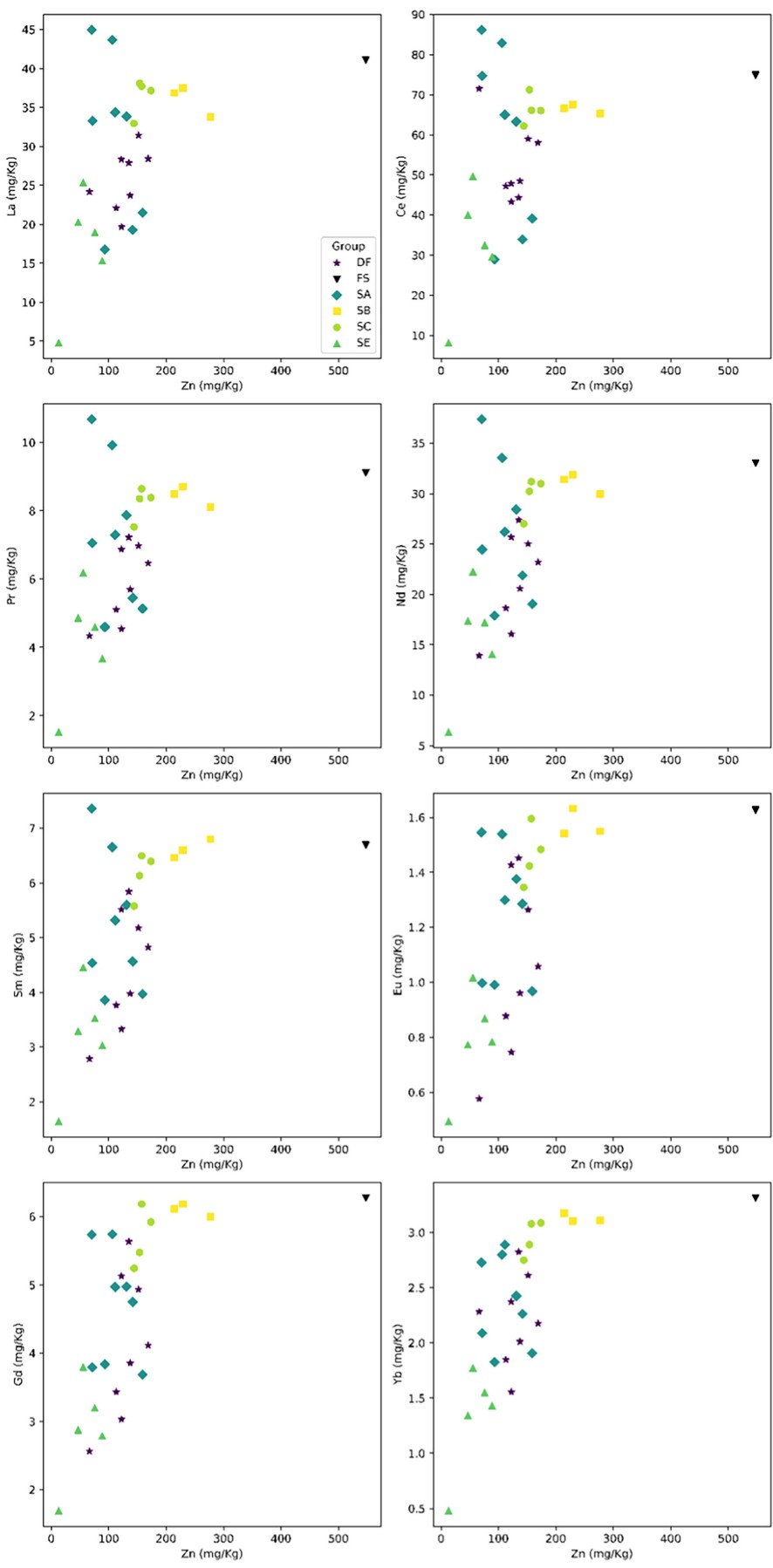

**Appendix 6—figure 19.** Harker variation plots illustrating the relationship between Zn and selected rare earth elements (REEs) for the five sediment groups analyzed within and around Feature 1. SB plots distinctly apart from other sediment groups, which tend to overlap or are mingled for most trace elements.

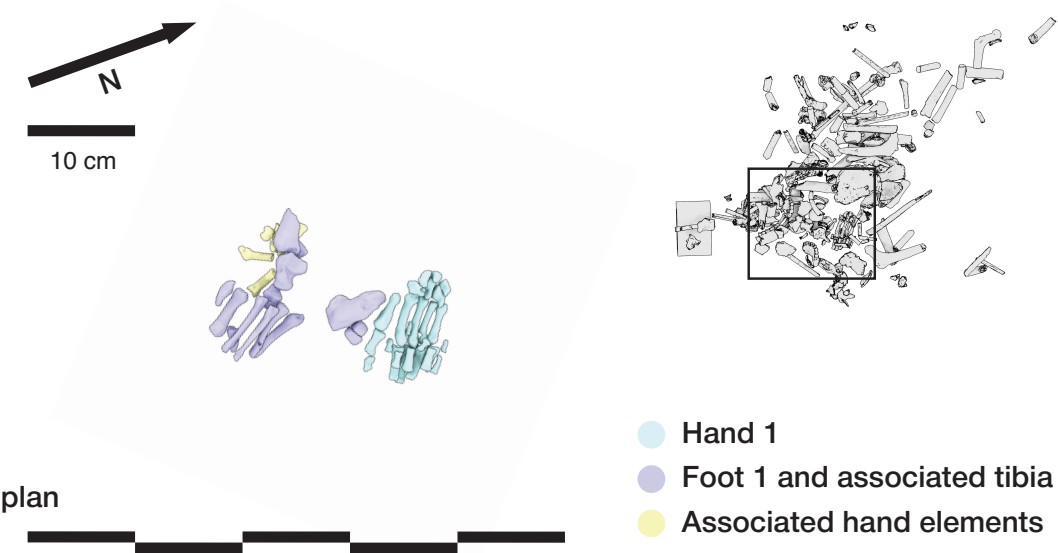

Hand 1
Foot 1 and associated tibia
Associated hand elements

**Appendix 6—figure 20.** Reconstruction of the burial position of Dinaledi Hand 1 and Foot 1. Hand 1 (blue) is nearly complete and shows the flexed (curled) nature of the fingers upon recovery. Foot 1 (purple) is less well preserved, but demonstrates the retention of the structures of the mid and hind foot. F1 is underlain by disarticulated manual elements not associated with Hand 1.

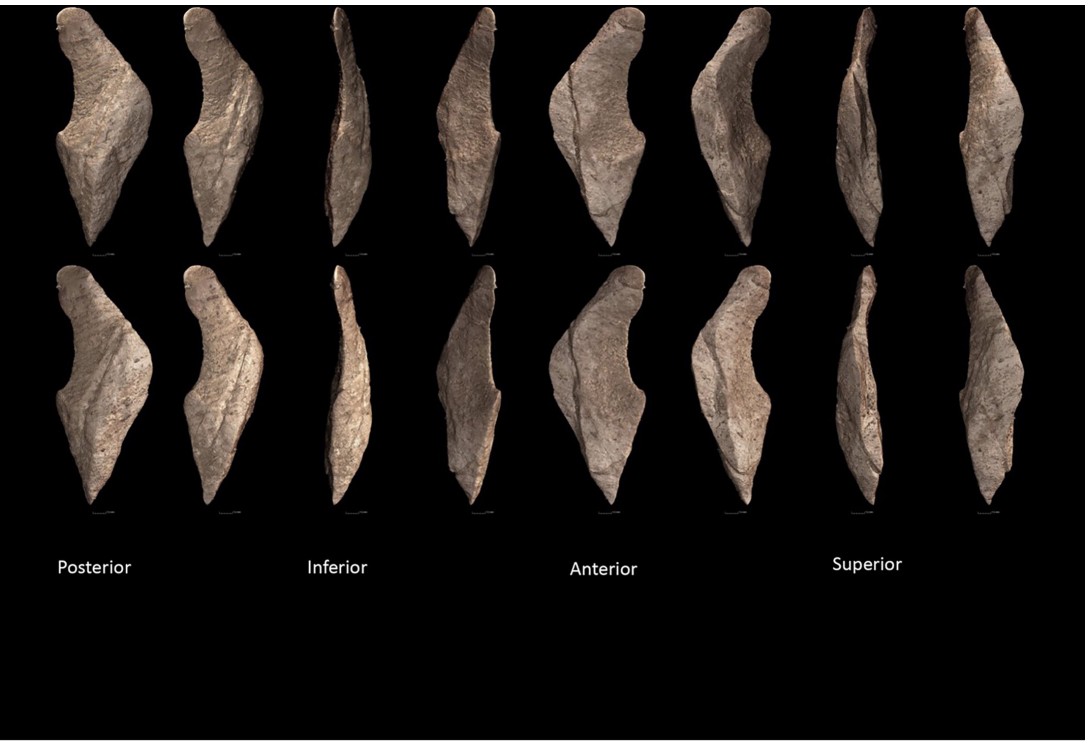

**Appendix 6—figure 21.** Hill Antechamber Artifact 1 (HAA1) showing surface from 8 different angles with 2 different lighting directions. The 3D model results from the segmentation of the synchrotron scan at 16.22 um, the artifact being still in situ in the paster jacket.

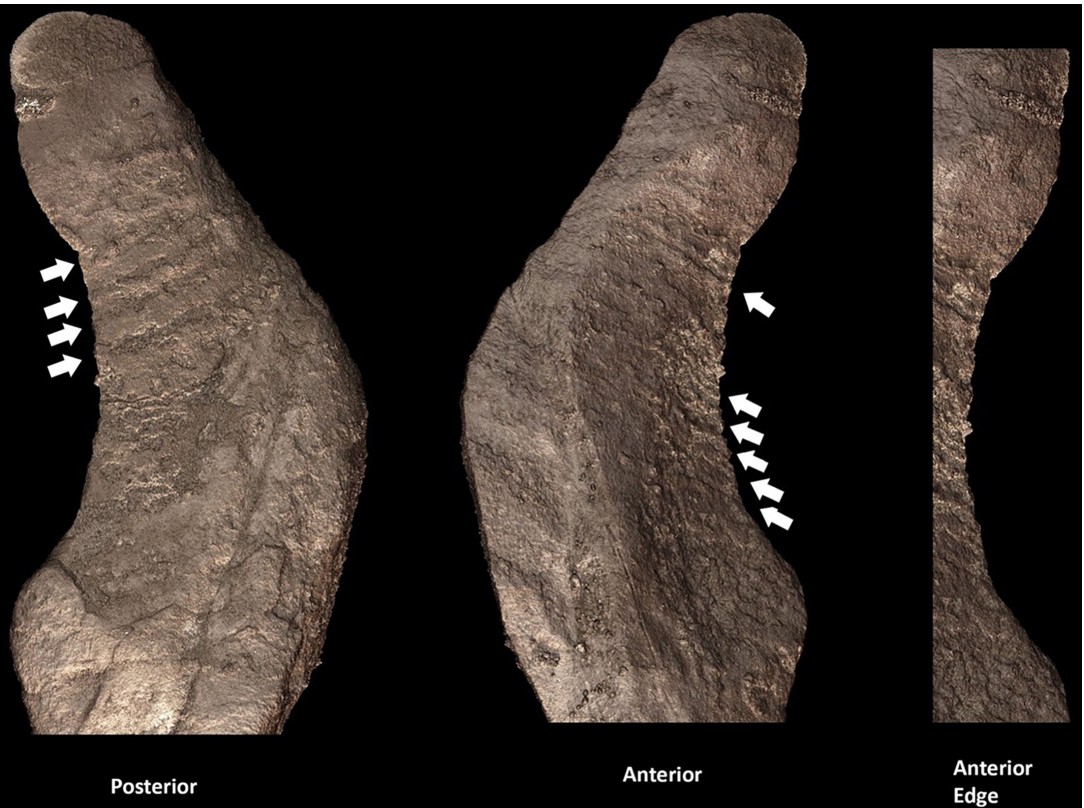

**Appendix 6—figure 22.** Hill Antechamber Artifact 1 (HAA1) close-up from the previous figure with detail showing striations visible on both faces and intersection of these striations with sharp edge of artifact showing appearance of serrations.

