## [Editor Report · eLife Assessment]

The authors study the context of the skeletal remains of three individuals and associated sediment samples to conclude that the hominin species *Homo naledi* intentionally buried their dead. Demonstration of the earliest known instance of intentional funerary practice – with a relatively small-brained hominin engaging in a highly complex behavior that has otherwise been observed from *Homo sapiens* and *Homo neanderthalensis* – would represent a **landmark** finding. The authors have revised their manuscript extensively in light of the reviews of their initial submission, with improved illustration, context, discussion, and theoretical frameworks, leading to an improved case supporting their conclusion that *Homo naledi* intentionally buried their dead. One of the reviewers concludes that the findings **convincingly** demonstrate intentional burial practices, while another considers evidence for such an unambiguous conclusion to be **incomplete** given a lack of definitive knowledge around how the hominins got into the chamber. We look forward to seeing the continued development and assessment of this hypothesis. It is worth noting that the detailed reviews (both rounds) and comprehensive author response are commendable and consequential parts of the scientific record of this study. The editors note that the authors' response repeatedly invokes precedent from previous publications to help justify the conclusions in this paper. While doing so is helpful, the editors also note that scientific norms and knowledge are constantly evolving, and that any study has to rest on its own scientific merit.

---

## [Referee Report · Reviewer #1 (Public review)]

Thank you for allowing me to review the paper "Evidence for deliberate burial of the dead by *Homo naledi*". This remains a very important site for paleoanthropology. I appreciate the work that the crew, especially the junior members of the team, put into this massive project. I appreciate that the authors did revise the paper since that is not a requirement of eLife. Extensive reviews by peer-reviewers have been provided for this paper, as well as professionally published replies (Martinón-Torres et al., 2023; Foecke et al., 2023). The composition, and citations of this version are much improved, though important information, some requested by reviewers, are buried in the supplementary section. It seems important that the authors make these sections more easily accessible to the general reader. The length of the paper is also unnecessary and impedes the readability of the work. Concise clarity is an expectation of most journals. The Netflix documentary was made to appeal to a mass audience, I would hope that the goal of the accompanying publication would be to enable readers to fully comprehend the work behind the claims.

This version of the paper considers at great length many possibilities for how the *H. naledi* skeletal material came to rest in the cave system with some additional figures and data provided. However, quite a lot is still unclear. In my original review I stated, "The authors have repeatedly described how incredibly challenging it is to get into and out of this cave system and all of its chambers." This was a point emphasized in the Netflix documentary. In this version of the paper the authors have included within the supplementary section a brief discussion of other entrances. The work by Robbins et al. 2021 (a peer-reviewed paper in the impact factor rated journal Chemical Geology) is extremely relevant here. In this revision it is noted in the supplementary section that if the Postbox chamber was used as an opening, it would have reduced the length of the access to the system by 80 m. This fact seems important. This section should be moved out of the supplementary material and expanded because the conclusions published by Robbins et al. (2021) indicate a completely different route by which *H. naledi* accessed the cave, but this is hardly mentioned in the revision and deserves attention. To quote the Robbins et al.'s (2021) discussion section 6.3:

"We acknowledge that additional data is required in order to confidently assess the relative timing of the Dragon's Back collapse and entry of *H. naledi*. Nonetheless, the stratigraphic and geochronologic observations presented here, together with those previously published (Dirks et al., 2017) are consistent with the following scenario. Prior to the collapse of the Dragon's Back, sometime before 241 ka (new minimum age for *H. naledi* from RS68), the cave could be entered by *H. naledi* via a shaft in the roof of the Postbox Chamber. From there *H. naledi* could walk along a straight passage that follows a gently descending, SW trending fracture into the Dragon's Back Chamber and, with the Dragon's Back block still attached to the roof, would have only needed to climb over a ~5 m high sill to access the Dinaledi Subsystem behind it. This sill and narrow fracture system behind the Dragon's Back block would have been a major impediment to any flood waters and most other fauna into the Dinaledi Subsystem, but it would have been a more accessible route than that today."

The paper's conclusion continues, "The new dates further constrain the minimum age of *H. naledi* to 241 ka. Thus, *H. naledi* entered the subsystem between 241 ka and 335 ka, during a glacial period, when clastic sediment along the access route into the Dinaledi Subsystem experienced erosion. *H. naledi* would have probably entered the cave in the same way as the clastic sediments did, through an opening in the roof of the Postbox Chamber and may have entered via the Dragon's Back Chamber by climbing a 5 m high sill and passing below the Dragon's Back Block that was then still attached to the roof, to enter the Dinaledi Subsystem. In this context it is important to emphasize that it was not the Dragon's Back Block that prevented high-energy transport of coarse siliciclastic sediment from the Dragon's Back Chamber into the Dinaledi Subsystem, but rather the in situ floor block in the back wall of the Dragon's Back Chamber, against which the Dragon's Back Block slumped after it fell." This conclusion is very different from the complex pathway suggested by Berger et al. Martinón-Torres et al., 2023 also requested elaboration on this point in their reply by stating, "Moreover, recent studies by the Rising Star Cave team also point to a possible different and easier accesses for *H. naledi* into the fossil-bearing cave chambers than the current restricted access chute used by the research team, making clear that the degree of accessibility remains an open question (Robbins et al., 2021). Based on extensive dating studies of speleothem, this research (Robbins et al., 2021) implies that prior to 241 ka and the collapse of the Dragon's Back block hominins and other species could have more easily entered the cave via the Post Box Chamber and beneath the Dragon's Back Block before it fell. This gives access to a series of rifts that allow easier entry to the Dinaledi and other chambers beyond the present-day chute."

Because this paper introduces very different sets of possibilities, it seems impossible to derive an understanding of the processes that occurred 335-241 ka throughout the cave system without going into detail on these other openings, especially openings that are hypothesized to have been used by the hominins in question.

The world cares deeply about the *H. naledi* hominins and their story. I hope that in the coming years these issues are addressed, and perhaps other independent teams are allowed to do a full analysis since science is about replication. In any case, the excavation team has contributed important fossils to paleoanthropology.

Literature cited:

Foecke, Kimberly K., Queffelec, Alain, & Pickering, Robyn. (2023). No Sedimentological Evidence for Deliberate Burial by *Homo naledi* - A Case Study Highlighting the Need for Best Practices in Geochemical Studies Within Archaeology and Paleoanthropology. PaleoAnthropology, 2024.Martinón-Torres, M., Garate, D., Herries, A. I. R., & Petraglia, M. D. (2023). No scientific evidence that *Homo naledi* buried their dead and produced rock art. Journal of Human Evolution, 103464. https://doi.org/10.1016/j.jhevol.2023.103464Robbins, J. L., Dirks, P. H. G. M., Roberts, E. M., Kramers, J. D., Makhubela, T. V., HilbertWolf, H. L., Elliott, M., Wiersma, J. P., Placzek, C. J., Evans, M., & Berger, L. R. (2021). Providing context to the *Homo naledi* fossils: Constraints from flowstones on the age of sediment deposits in Rising Star Cave, South Africa. Chemical Geology, 567, 120108. https://doi.org/10.1016/j.chemgeo.2021.120108

---

## [Referee Report · Reviewer #2 (Public review)]

Before providing my review of the revised version of this study by Berger et al., which explores potential deliberate burials of *Homo naledi* within the Rising Star Cave System, I would like to briefly summarize the key points from my previous review of the earlier version (in 2023). Summarizing my previous review will provide context for assessing how effectively the revised study addresses the concerns I raised previously (in 2023).

In my earlier comments, I highlighted significant methodological and analytical shortcomings that, in my view, undermined the authors' claim of intentional burials by *Homo naledi*. While the study presented detailed geological and fossil data, I found the evidence for intentional burials unconvincing due to insufficient application of archaeothanatological principles and other methodological gaps.

My key concerns included:

(1) The absence of a comprehensive archaeothanatological analysis, particularly with respect to taphonomic changes, bone articulations, and displacement patterns such as the collapse of sediments and bone remains into voids created by decomposition.

(2) Missing or unclear illustrations of bone arrangements, which are critical for interpreting burial positions and processes.

(3) A lack of detailed discussion on the sequence of decomposition, joint disarticulation, sediment infill, and secondary bone displacement.

To convincingly support claims of deliberate burial, I argued that the study must reconstruct the timeline and processes surrounding death and deposition while clearly distinguishing natural taphonomic changes from intentional human actions. I emphasized the importance of integrating established archaeothanatological frameworks, such as those outlined by Duday et al. or Boulestin et al., to provide the necessary analytical rigor.

I will now explain how the revised version of this study has successfully addressed all the concerns raised in my previous review and why I now think that the authors provide sufficient evidence for the presence of "repeated and patterned" deliberate burials (referred to as "cultural burials" by the authors) by *Homo naledi* within the Rising Star Cave System.

In their revised manuscript, the authors have implemented substantial improvements in methodology, analytical depth, and overall presentation, which have effectively resolved the critical issues I previously highlighted. These revisions greatly strengthen their argument for intentional funerary practices. Importantly, the authors remain cautious in their interpretation of the evidence, explicitly refraining from inferring "symbolic" behavior or complex cognitive motivations behind these burials. Instead, they focus on presenting clear evidence for deliberate, patterned practices while leaving the broader implications for *Homo naledi*'s cultural and cognitive capacities open for further investigation. This cautious approach adds to the credibility of their conclusions and avoids overextending the interpretation of the data.

The authors' enhanced application of archaeothanatological principles now offers a more comprehensive and convincing interpretation of the burial features. Key gaps in the earlier version, such as the absence of detailed reconstructions of taphonomic processes, bone articulations, and displacement patterns, have been addressed with thorough analyses and clearer illustrations. The study also now includes a well-structured timeline of events surrounding death and deposition, demonstrating an improved ability to differentiate between natural processes and deliberate human actions. These additions lend greater clarity and rigor to the evidence, making the argument for intentional burials both robust and persuasive.

Furthermore, the revised study presents detailed data on skeletal arrangements, decomposition sequences, and spatial patterns. This information is now relatively well illustrated and contextualized, enabling readers to better understand the complex processes involved in these burial practices. Importantly, the authors provide a stronger theoretical framework, integrating established archaeothanatological methodologies and taphonomic studies that situate their findings within broader archaeological and anthropological discussions of funerary behavior.

That being said, there remain relatively minor issues that could be refined further. Addressing these would help ensure the study is as clear and accessible as possible to the reader. Such adjustments would enhance the overall readability and reinforce the study's impact within the scientific community.

A - Suggested changes:

While the revised version of this study marks a significant improvement, successfully addresses my previous major concerns and provides a convincing argument for deliberate burials by *Homo naledi*, I believe that including both one summary table + one summary figure for each of the three main locations and the-Hill Antechamber, and Dinaledi Chamber (Feature 1 and Puzzle Box)-would further enhance the clarity and accessibility of the findings. Such tables and figures would serve as a valuable reference, allowing readers to more easily follow how the detailed patterns observed at each site fit the criteria for distinguishing intentional from natural processes.

The summary tables should consolidate key information for each location, such as:

(1) Bone articulations: A comprehensive list of articulated skeletal elements, categorized by their anatomical relationships (e.g., labile vs. stable articulations).

(2) Displacement patterns: Documentation of any spatial shifts in bone positions, noting directions and extents of disarticulation.

(3) Sequence of decomposition: Observations regarding the sequence of decomposition, joint disarticulation and associated changes in bone arrangements.

(4) Sediment interaction: Notes on sediment infill and its timing relative to decomposition, including evidence of secondary voids or delayed sediment deposition.

(5) Distinguishing criteria: Clear indications of how each observed pattern supports intentional burial (e.g., structured placement, lack of natural transport mechanisms) versus natural processes (e.g., random dispersal, sediment-driven bone displacement).

Including such tables would not only summarize the complex taphonomic and archaeothanatological data but also allow readers to quickly assess how the evidence supports the authors' conclusions. This approach would bridge the gap between the detailed narrative descriptions and the criteria necessary to differentiate deliberate funerary practices from natural occurrences.

To streamline the main text further, many of the detailed descriptions of individual bones, specific displacement measurements, and other intricate observations could be moved to the supplementary data. This reorganization would maintain the richness of the data for those who wish to explore it in depth, while the summary tables would present the key findings concisely in the main text. This balance between accessibility and detail would ensure that the study appeals to both specialists requiring comprehensive data and readers looking for an overarching understanding of the findings.

In addition to these structural changes, it is crucial to ensure that evidence is consistently illustrated throughout the text.

Importantly the skeletal part representation is provided for Dinaledi Feature 1 in Figure 14, but similar data is not presented for the other burial features, such as those in the Hill Antechamber or Puzzle Box. This inconsistency could make it more challenging for readers to compare the features and fully appreciate the patterns of burial behavior across the different locations. Ensuring that similar types of evidence and analyses are presented uniformly for all features would strengthen the study and make its conclusions more cohesive and compelling.

Adding supplementary figures to represent the skeletal part distribution (as in Figure 14) within each excavated area (i.e., not only for Dinaledi Feature 1 but also for Hill Antechamber and Puzzle Box) would significantly enhance the study's clarity and accessibility. These figures could provide a visual summary of skeletal part representation, allowing readers to easily understand the nature of human remains within each burial context.

Specifically, such figures could:

(1) Illustrate Skeletal Part Representation: By visually mapping the presence and location of various skeletal elements, the figures would make it easier for readers to assess the completeness and arrangement of remains in each feature. This is particularly important for interpreting patterns of bone articulation and disarticulation.

For example, it is quite challenging to determine the exact number and characteristics of the human skeletal remains identified within the Puzzle Box and those recovered through the "subsurface collection" in its surrounding area. The authors state that "at least six individuals" were identified in this area (during "subsurface collection") but provide no further clarification. They simply mention that "most elements" were described previously, without specifying which elements or where this prior description can be found.

(2) Highlight Articulations and Displacements: Figures could indicate which bones are articulated and their relative positions, as well as the spatial distribution of disarticulated elements. This would provide a clear visual context to support interpretations of taphonomic processes.

(3) Facilitate Comparisons Across Locations: By presenting skeletal part representation consistently for each location, the figures would enable readers to directly compare features, reinforcing the argument for "repeated and patterned" behavior.

(4) Simplify Complex Data: Instead of relying solely on textual descriptions, the visual format would allow readers to quickly grasp the key findings, making the study more accessible to a broader audience

By including such figures alongside the proposed summary tables in the main text, the study would achieve a balance between detailed narrative descriptions and concise, visual representation of the data. This approach would strengthen the overall presentation and support the authors' conclusions effectively.

Again, by presenting the data in a structured and comparative format, the new tables + figures could also highlight the differences and similarities between the three locations. This would reinforce the argument for "repeated and patterned" behavior, as the tables would make it easier to observe consistent burial practices across different contexts within the Rising Star Cave System.

Adding these summary tables + figures, ensuring consistent presentation of evidence, and reallocating detailed descriptions to supplementary materials would not require significant new analysis. However, these organizational adjustments would greatly enhance the study's clarity, readability, and overall impact.

B - A few additional changes are needed:

Figure 8: This figure is critical but lacks clarity. Specifically:

Panels 8a-c suffer from low contrast, making details difficult to discern.

Panel 8d (sediment profile) is too small and lacks annotations that would aid interpretation.

Figure S7: While this figure has significantly better contrast than Figures 8a-c, I am unable to identify the "articulated foot ... at right of frame," as mentioned in the caption. Please clarify this by adding annotations directly to the figure.

Page 4, 2nd paragraph: In the sentence "Researchers thus have diverse opinions about how to test whether ...," the word "opinions" should be replaced with a more precise term, such as "approaches."

C - In conclusion, I am impressed by the significant effort and meticulous work that has gone into this revised version of the study. The quality of the new evidence presented is commendable, and the findings now convincingly demonstrate not only clear evidence of intentional burial practices by *Homo naledi* but also compelling indications of post-depositional reworking. These advancements reflect a major improvement in the study's analytical rigor and the robustness of its conclusions, making it a valuable contribution to the understanding of early hominin funerary behavior.

---

## [Author Response]

[The following is the authors’ response to the original reviews.]

We extend our sincere thanks to the editor, referees for eLife, and other commentators who have written evaluations of this manuscript, either in whole or in part. Sources of these comments were highly varied, including within the bioRxiv preprint server, social media (including many comments received on X/Twitter and some YouTube presentations and interviews), comments made by colleagues to journalists, and also some reviews of the work published in other academic journals. Some of these are formal and referenced with citations. Others were informal but nonetheless expressed perspectives that helped enable us to revise the manuscript with the inclusion of broader perspectives than the formal review process. It is beyond the scope of this summary to list every one of these, which have often been brought to the attention of different coauthors, but we begin by acknowledging the very wide array of peer and public commentary that have contributed to this work. The reaction speaks to a broad interest in open discussion and review of preprints.

As we compiled this summary of changes to the manuscript, we recognized that many colleagues made comments about the process of preprint dissemination and evaluation rather than the data or analyses in the manuscript. Addressing such comments is outside the scope of this revised manuscript, but we do feel that a broader discussion of these comments would be valuable in another venue. Many commentators have expressed confusion about the eLife system of evaluation of preprints, which differs from the editorial acceptance or rejection practiced in most academic journals. As authors in many different nations, in varied fields, and in varied career stages, we ourselves are still working to understand how the academic publication landscape is changing, and how best to prepare work for new models of evaluation and dissemination.

The manuscript and coauthor list reflect an interdisciplinary collaboration. Analyses presented in the manuscript come from a wide range of scientific disciplines. These range from skeletal inventory, morphology, and description, spatial taphonomy, analysis of bone fracture patterns and bone surface modifications, sedimentology, geochemistry, and traditional survey and mapping. The manuscript additionally draws upon a large number of previous studies of the Rising Star cave system and the Dinaledi Subsystem, which have shaped our current work. No analysis within any one area of research stands alone within this body of work: all are interpreted in conjunction with the outcomes of other analyses and data from other areas of research. Any single analysis in isolation might be consistent with many different hypotheses for the formation of sediments and disposition of the skeletal remains. But testing a hypothesis requires considering all data in combination and not leaving out data that do not fit the hypothesis. We highlight this general principle at the outset because a number of the comments from referees and outside specialists have presented alternative hypotheses that may arguably be consistent with one kind of analysis that we have presented, while seeming to overlook other analyses, data, or previous work that exclude these alternatives. In our revision, we have expanded all sections describing results to consider not only the results of each analysis, but how the combination of data from different kinds of analysis relate to hypotheses for the deposition and subsequent history of the *Homo naledi* remains. We address some specific examples and how we have responded to these in our summary of changes below.

General organization:

The referee and editor comments are mostly general and not line-by-line questions, and we have compiled them and treated them as a group in this summary of changes, except where specifically noted.

The editorial comments on the previous version included the suggestion that the manuscript should be reorganized to test “natural” (i.e. noncultural) hypotheses for the situations that we examine. The editorial comment suggested this as a “null hypothesis” testing approach. Some outside comments also viewed noncultural deposition as a null hypothesis to be rejected. We do not concur that noncultural processes should be construed as a null hypothesis, as we discuss further below. However, because of the clear editorial opinion we elected to revise the manuscript to make more explicit how the data and analyses test noncultural depositional hypotheses first, followed by testing of cultural hypotheses. This reorganization means that the revised manuscript now examines each hypothesis separately in turn.

Taking this approach resulted in a substantial reorganization of the “Results” section of the manuscript. The “Results” section now begins with summaries of analyses and data conducted on material from each excavation area. After the presentation of data and analyses from each area, we then present a separate section for each of several hypotheses for the disposition and sedimentary context of the remains. These hypotheses include deposition of bodies upon a talus (as hypothesized in some previous work), slow sedimentary burial on a cave floor or within a natural depression, rapid burial by gravity-driven slumping, and burial of naturally mummified remains. We then include sections to test the hypothesis of primary cultural burial and secondary cultural burial. This approach adds substantial length to the Results. While some elements may be repeated across sections, we do consider the new version to be easier to take piece by piece for a reader trying to understand how each hypothesis relates to the evidence.

The Results section includes analyses on several different excavation areas within the Dinaledi Subsystem. Each of these presents somewhat different patterns of data. We conceived of this manuscript combining these distinct areas because each of them provides information about the formation history of the *Homo naledi*-associated sediments and the deposition of the *Homo naledi* remains. Together they speak more strongly than separately. In the previous version of the manuscript, two areas of excavation were considered in detail (Dinaledi Feature 1 and the Hill Antechamber Feature), with a third area (the Puzzle Box area) included only in the Discussion and with reference to prior work. We now describe the new work undertaken after the 2013-2014 excavations in more detail. This includes an overview of areas in the Hill Antechamber and Dinaledi Chamber that have not yielded substantial *H. naledi* remains and that thereby help contextualize the spatial concentration of *H. naledi* skeletal material. The most substantial change in the data presented is a much expanded reanalysis of the Puzzle Box area. This reanalysis provides greater clarity on how previously published descriptions relate to the new evidence. The reanalysis also provides the data to integrate the detailed information on bone identification fragmentation, and spatial taphonomy from this area with the new excavation results from the other areas.

In addition to Results, the reorganization also affected the manuscript’s Introduction section. Where the previous version led directly from a brief review of Pleistocene burial into the description of the results, this revised manuscript now includes a review of previous studies of the Rising Star cave system. This review directly addresses referee comments that express some hesitation to accept previous results concerning the structure and formation of sediments, the accessibility of the Dinaledi Subsystem, the geochronological setting of the *H. naledi* remains, and the relation of the Dinaledi Subsystem to nearby cave areas. Some parts of this overview are further expanded in the Supplementary Information to enable readers to dive more deeply into the previous literature on the site formation and geological configuration of the Rising Star cave system without needing to digest the entirety of the cited sources.

The Discussion section of the revised manuscript is differentiated from Results and focuses on several areas where the evidence presented in this study may benefit from greater context. One new section addresses hypothesis testing and parsimony for Pleistocene burial evidence, which we address at greater length in this summary below. The majority of the Discussion concerns the criteria for recognizing evidence for burial as applied in other studies. In this research we employ a minimal definition but other researchers have applied varied criteria. We consider whether these other criteria have relevance in light of our observations and whether they are essential to the recognition of burial evidence more broadly.

Vocabulary:

We introduce the term “cultural burial” in this revised manuscript to refer to the burial of dead bodies as a mortuary practice. “Burial” as an unmodified term may refer to the passive covering of remains by sedimentary processes. Use of the term “intentional burial” would raise the question of interpreting intent, which we do not presume based on the evidence presented in this research. The relevant question in this case is whether the process of burial reflects repeated behavior by a group. As we received input from various colleagues it became clear that burial itself is a highly loaded term. In particular there is a common assumption within the literature and among professionals that burial must by definition be symbolic. We do not take any position on that question in this manuscript, and it is our hope that the term “cultural burial” may focus the conversation around the extent that the behavioral evidence is repeated and patterned.

Sedimentology and geochemistry of Dinaledi Feature 1:

Reviewer 4 provided detailed comments on the sedimentological and geochemical context that we report in the manuscript. One outside review (Foecke et al. 2024) included some of the points raised by reviewer 4, and additionally addressed the reporting of geochemical and sedimentological data in previous work that we cite.

To address these comments we have revised the sedimentary context and micromorphology of sediments associated with Dinaledi Feature 1. In the new text we demonstrate the lack of microstratigraphy (supported by grain size analysis) in the unlithified mud clast breccia (UMCB), while such a microstratigraphy is observed in the laminated orange-red mudstones (LORM) that contribute clasts to the UMCB. Thus, we emphasize the presence and importance of a laterally continuous layer of LORM nature occurring at a level that appears to be the maximum depth of fossil occurrence. This layer is severely broken under extensive accumulation of fossils such as Feature 1 and only evidenced by abundant LORM clasts within and around the fossils.

We have completely reworked the geochemical context associated with Feature 1 following the comments of reviewer 4. We described the variations and trends observed in the major oxides separate from trace and rare-earth elements. We used Harker variations plots to assess relationships between these element groups with CaO and Zn, followed by principal component analysis of all elements analyzed. The new geochemical analysis clearly shows that Feature 1 is associated with localized trace element signatures that exist in the sediments only in association with the fossil bones, which suggests lack of postdepositional mobilization of the fossils and sediments. We additionally have included a fuller description of XRF methods.

To clarify the relation of all results to the features described in this study, we removed the geochemical and sedimentological samples from other sites within the Dinaledi Subsystem. These localities within the fissure network represent only surface collection of sediment, as no excavation results are available from those sites to allow for comparison in the context of assessing evidence of burial. These were initially included for comparison, but have now been removed to avoid confusion.

Micromorphology of sediments:

Some referees (1, 3, and 4) and other commentators (including Martinón-Torres *et al.* 2024) have suggested that the previous manuscript was deficient due to an insufficient inclusion of micromorphological analysis of sediments. Because these commentators have emphasized this kind of evidence as particularly important, we review here what we have included and how our revision has addressed this comment. Previous work in the Dinaledi Chamber (Dirks et al., 2015; 2017) included thin section illustrations and analysis of sediment facies, including sediments in direct association with *H. naledi* remains within the Puzzle Box area. The previous work by Wiersma and coworkers (2020) used micromorphological analysis as one of several approaches to test the formation history of Unit 3 sediments in the Dinaledi Subsystem, leading to the interpretation of autobrecciation of earlier Unit 1 sediment. In the previous version of this manuscript we provided citations to this earlier work. The previous manuscript also provided new thin section illustrations of Unit 3 sediment near Dinaledi Feature 1 to place the disrupted layer of orange sediment (now designated the laminated orange silty mudstone unit) into context.

In the new revised manuscript we have added to this information in three ways. First, as noted above in response to reviewer 4, we have revised and added to our discussion of micromorphology within and adjacent to the Dinaledi Feature 1. Second, we have included more discussion in the Supplementary Information of previous descriptions of sediment facies and associated thin section analysis, with illustrations from prior work (CC-BY licensed) brought into this paper as supplementary figures, so that readers can examine these without following the citations. Third, we have included Figure 10 in the manuscript which includes six panels with microtomographic sections from the Hill Antechamber Feature. This figure illustrates the consistency of sub-unit 3b sediment in direct contact with *H. naledi* skeletal material, including anatomically associated skeletal elements, with previous analyses that demonstrate the angular outlines and chaotic orientations of LORM clasts. It also shows density contrasts of sediment in immediate contact with some skeletal elements, the loose texture of this sediment with air-filled voids, and apparent invertebrate burrowing activity. To our knowledge this is the first application of microtomography to sediment structure in association with a Pleistocene burial feature.

To forestall possible comments that the revised manuscript does not sufficiently employ micromorphological observations, or that any one particular approach to micromorphology is the standard, we present here some context from related studies of evidence from other research groups working at varied sites in Africa, Europe, and Asia. Hodgkins *et al.* (2021) noted: “Only a handful of micromorphological studies have been conducted on human burials and even fewer have been conducted on suspected burials from Paleolithic or hunter-gatherer contexts.” In that study, one supplementary figure with four photomicrographs of thin sections of sediments was presented. Interpretation of the evidence for a burial pit by Hodgkins *et al.* (2021) noted the more open microstructure of sediment but otherwise did not rely upon the thin section data in characterizing the sediments associated with grave fill. Martinón-Torres *et al.* (2021) included one Extended Data figure illustrating thin sections of sediments and bone, with two panels showing sediments (the remainder showing bone histology). The micromorphological analysis presented in the supplementary information of that paper was restricted to description of two microfacies associated with the proposed “pit” in that study. That study did carry out microCT scanning of the partially-prepared skeletal remains but did not report any sediment analysis from the microtomographic results. Maloney *et al.* (2022) reported no micromorphological or thin section analysis. Pomeroy *et al.* (2020a) included one illustration of a thin section; this study may be regarded as a preliminary account rather than a full description of the work undertaken. Goldberg *et al.* (2017) analyzed the geoarchaeology of the Roc de Marsal deposits in which possible burial-associated sediments had been fully excavated in the 1960s, providing new morphological assessments of sediment facies; the supplementary information to this work included five scans (not microscans) of sediment thin sections and no microphotographs. Fewlass *et al.* (2023) presented no thin section or micromorphological illustrations or methods. In summary of this research, we note that in one case micromorphological study provided observations that contributed to the evidence for a pit, in other cases micromorphological data did not test this hypothesis, and many researchers do not apply micromorphological techniques in their particular contexts.

Sediment micromorphology is a growing area of research and may have much to provide to the understanding of ancient burial evidence as its standards continue to develop (Pomeroy et al. 2020b). In particular microtomographic analysis of sediments, as we have initiated in this study, may open new horizons that are not possible with more destructive thin-section preparation. In this manuscript, the thin section data reveals valuable evidence about the disruption of sediment structure by features within the Dinaledi Chamber, and microtomographic analysis further documents that the Hill Antechamber Feature reflects similar processes, in addition to possible post-burial diagenesis and invertebrate activity. Following up in detail on these processes will require further analysis outside the scope of this manuscript.

Access into the Dinaledi Subsystem:

Reviewer 1 emphasizes the difficulty of access into the Dinaledi Subsystem as a reason why the burial hypothesis is not parsimonious. Similar comments have been made by several outside commentators who question whether past accessibility into the Dinaledi Subsystem may at one time have been substantially different from the situation documented in previous work. Several pieces of evidence are relevant to these questions and we have included some discussion of them in the Introduction, and additionally include a section in the Supplementary Information (“Entrances to the cave system”) to provide additional context for these questions. *Homo naledi* remains are found not only within the Dinaledi Subsystem but also in other parts of the cave system including the Lesedi Chamber, which is similarly difficult for non-expert cavers to access. The body plan, mass, and specific morphology of *H. naledi* suggest that this species would be vastly more suited to moving and climbing within narrow underground passages than living people. On this basis it is not unparsimonious to suggest that the evidence resulted from *H. naledi* activity within these spaces. We note that the accessibility of the subsystem is not strictly relevant to the hypothesis of cultural burial, although the location of the remains does inform the overall context which may reflect a selection of a location perceived as special in some way.

Stuffing bodies down the entry to the subsystem:

Reviewer 3 suggests that one explanation for the emplacement of articulated remains at the top of the sloping floor of the Hill Antechamber is that bodies were “stuffed” into the chute that comprises the entry point of the subsystem and passively buried by additional accumulation of remains. This was one hypothesis presented in earlier work (Dirks et al. 2015) and considered there as a minimal explanation because it did not entail the entry of *H. naledi* individuals into the subsystem. The further exploration (Elliott et al. 2021) and ongoing survey work, as well as this manuscript, all have resulted in data that rejects this hypothesis. The revised manuscript includes a section in the results “Deposition upon a talus with passive burial” that examines this hypothesis in light of the data.

Recognition of pits:

Referee 3 and 4 and several additional commentators have emphasized that the recognition of pit features is necessary to the hypothesis of burial, and questioned whether the data presented in the manuscript were sufficient to demonstrate that pits were present. We have revised the manuscript in several ways to clarify how all the different kinds of evidence from the subsystem test the hypothesis that pits were present. This includes the presentation of a minimal definition of burial to include a pit dug by hominins, criteria for recognizing that a pit was present, and an evaluation of the evidence in each case to make clear how the evidence relates to the presence of a pit and subsequent infill. As referee 3 notes, it can be challenging to recognize a pit when sediment is relatively homogeneous. This point was emphasized in the review by Pomeroy and coworkers (2020b), who reflected on the difficulty seeing evidence for shallow pits constructed by hominins, and we have cited this in the main text. As a result, the evidence for pits has been a recurrent topic of debate for most Pleistocene burial sites. However in addition to the sedimentological and contextual evidence in the cases we describe, the current version also reflects upon other possible mechanisms for the accumulation of bones or bodies. The data show that the sedimentary fill associated with the *H. naledi* remains in the cases we examine could not have passively accumulated slowly and is not indicative of mass movement by slumping or other high-energy flow. To further put these results into context, we added a section to the Discussion that briefly reviews prior work on distinguishing pits in Pleistocene burial contexts, including the substantial number of sites with accepted burial evidence for which no evidence of a pit is present.

Extent of articulation and anatomical association:

We have added significantly greater detail to the descriptions of articulated remains and orientation of remains in order to describe more specifically the configuration of the skeletal material. We also provide 14 figures in main text (13 of them new) to illustrate the configuration of skeletal remains in our data. For the Puzzle Box area, this now includes substantial evidence on the individuation of skeletal fragments, which enables us to illustrate the spatial configuration of remains associated with the DH7 partial skeleton, as well as the spatial position of fragments refitted as part of the DH1, DH2, DH3, and DH4 crania. For Dinaledi Feature 1 and the Hill Antechamber Feature we now provide figures that key skeletal parts as identified, including material that is unexcavated where possible, and a skeletal part representation figure for elements excavated from Dinaledi Feature 1.

Archaeothanatology:

Reviewer 2 suggests that a greater focus on the archaeothanatology literature would be helpful to the analysis, with specific reference to the sequence of joint disarticulation, the collapse of sediment and remains into voids created by decomposition, and associated fragmentation of the remains. In the revised manuscript we have provided additional analysis of the Hill Antechamber Feature with this approach in mind. This includes greater detail and illustration of our current hypothesis for individuation of elements. We now discuss a hypothesis of body disposition, describe the persistent joints and articulation of elements, and examine likely decomposition scenarios associated with these remains. Additionally, we expand our description and illustration of the orientation of remains and degree of anatomical association and articulation within Dinaledi Feature 1. For this feature and for the Hill Antechamber Feature we have revised the text to describe how fracturing and crushing patterns are consistent with downward pressure from overlying sediment and material. In these features, postdepositional fracturing occurred subsequent to the decomposition of soft tissue and partial loss of organic integrity of the bone. We also indicate that the loss by postdepositional processes of most long bone epiphyses, vertebral bodies, and other portions of the skeleton less rich in cortical bone, poses a challenge for testing the anatomical associations of the remaining elements. This is a primary reason why we have taken a conservative approach to identification of elements and possible associations.

A further aspect of the site revealed by our analysis is the selective reworking of sediments within the Puzzle Box area subsequent to the primary deposition of some bodies. The skeletal evidence from this area includes body parts with elements in anatomical association or articulation, juxtaposed closely with bone fragments at varied pitch and orientation. This complexity of events evidenced within this area is a challenge for approaches that have been developed primarily based on comparative data from single-burial situations. In these discussions we deepen our use of references as suggested by the referee.

Burial positions:

Reviewer 2 further suggests that illustrations of hypothesized burial positions would be valuable. We recognize that a hypothesized burial position may be an appealing illustration, and that some recent studies have created such illustrations in the context of their scientific articles. However such illustrations generally include a great deal of speculation and artist imagination, and tend to have an emotive character. We have added more discussion to the manuscript of possible primary disposition in the case of the Hill Antechamber Feature as discussed above. We have not created new illustrations of hypothesized burial positions for this revision.

Carnivore involvement:

Referee 1 suggests that the manuscript should provide further consideration of whether carnivore activity may have introduced bones or bodies into the cave system. The reorganized Introduction now includes a review of previous work, and an expanded discussion within the Supplementary Information (“Hypotheses tested in previous work”). This includes a review of literature on the topic of carnivore accumulation and the evidence from the Dinaledi and Lesedi Chamber that rejects this hypothesis.

Water transport and mud:

The eLife referees broadly accepted previous work showing that water inundation or mass flow of water-saturated sediment did not occur within the history of Unit 2 and 3 sediments, including those associated with *H. naledi* remains. However several outside commentators did refer specifically to water flow or mud flow as a mechanism for slumping of deposits and possible sedimentary covering of the remains. To address these comments we have added a section to the

Supplementary Information (“Description of the sedimentary deposits of the Dinaledi Subsystem”) that reviews previous work on the sedimentary units and formation processes documented in this area. We also include a subsection specifically discussing the term “mud” as used in the description of the sedimentology within the system, as this term has clearly been confusing for nonspecialists who have read and commented on the work. We appreciate the referees’ attention to the previous work and its terminology.

Redescription of areas of the cave system:

Reviewer 1 suggests that a detailed reanalysis of all portions of the cave system in and around the Dinaledi Subsystem is warranted to reject the hypothesis that bodies entered the space passively and were scattered from the floor by natural (i.e. noncultural) processes. The referee suggests that National Geographic could help us with these efforts. To address this comment we have made several changes to the manuscript. As noted above, we have added material in Supplementary Information to review the geochronology of the Dinaledi Subsystem and nearby Dragon’s Back Chamber, together with a discussion of the connections between these spaces.

Most directly in response to this comment we provide additional documentation of the possibility of movement of bodies or body parts by gravity within the subsystem itself. This includes detailed floor maps based on photogrammetry and LIDAR measurement, where these are physically possible, presented in Figures 2 and 3. In some parts of the subsystem the necessary equipment cannot be used due to the extremely confined spaces, and for these areas our maps are based on traditional survey methods. In addition to plan maps we have included a figure showing the elevation of the subsystem floor in a cross-section that includes key excavation areas, showing their relative elevation. All figures that illustrate excavation areas are now keyed to their location with reference to a subsystem plan. These data have been provided in previous publications but the visualization in the revised manuscript should make the relationship of areas clear for readers. The Introduction now includes text that discusses the configuration of the Hill Antechamber, Dinaledi Chamber, and nearby areas, and also discusses the instances in which gravity-driven movement may be possible, at the same time reviewing that gravity-driven movement from the entry point of the subsystem to most of the localities with hominin skeletal remains is not possible.

Within the Results, we have added a section on the relationship of features to their surroundings in order to assist readers in understanding the context of these bone-bearing areas and the evidence this context brings to the hypothesis in question. We have also included within this new section a discussion of the discrete nature of these features, a question that has been raised by outside commentators.

Passive sedimentation upon a cave floor or within a natural depression:

Reviewer 3 suggests that the situation in the Dinaledi Subsystem may be similar to a European cave where a cave bear skeleton might remain articulated on a cave floor (or we can add, within a hollow for hibernation), later to be covered in sediment. The reviewer suggests that articulation is therefore no evidence of burial, and suggests that further documentation of disarticulation processes is essential to demonstrating the processes that buried the remains. We concur that articulation by itself is not sufficient evidence of cultural burial. To address this comment we have included a section in the Results that tests the hypothesis that bodies were exposed upon the cave floor or within a natural depression. To a considerable degree, additional data about disarticulation processes subsequent to deposition are provided in our reanalysis of the Puzzle Box area, including evidence for selective reworking of material after burial.

Postdepositional movement and floor drains:

Reviewer 3 notes that previous work has suggested that subsurface floor drains may have caused some postdepositional movement of skeletal remains. The hypothesis of postdepositional slumping or downslope movement has also been discussed by some external commentators (including Martinón-Torres et al. 2024). We have addressed this question in several places within the revised manuscript. As we now review, previous discussion of floor drains attempted to explain the subvertical orientation of many skeletal elements excavated from the Puzzle Box area. The arrangement of these bones reflects reworking as described in our previous work, and without considering the possibility of reworking by hominins, one mechanism that conceivably might cause reworking was downward movement of sediments into subsurface drains. Further exploration and mapping, combined with additional excavation into the sediments beneath the Puzzle Box area provided more information relevant to this hypothesis. In particular this evidence shows that subsurface drains cannot explain the arrangement of skeletal material observed within the Puzzle Box area. As now discussed in the text, the reworking is selective and initiated from above rather than below. This is best explained by hominin activity subsequent to burial.

In a new section of the Results we discuss slumping as a hypothesis for the deposition of the remains. This includes discussion of downslope movement within the Hill Antechamber and the idea that floor drains may have been a mechanism for sediment reworking in and around the Puzzle Box area and Dinaledi Feature 1. As described in this section the evidence does not support these hypotheses.

Hypothesis testing and parsimony:

Referees 1 and 3 and the editorial guidance all suggested that a more appropriate presentation would adopt a null hypothesis and test it. The specific suggestion that the null hypothesis should be a natural sedimentary process of deposition was provided not only by these reviewers but also by some outside commentators. To address this comment, we have edited the manuscript in two ways. The first is the addition of a section to the Discussion that specifically discusses hypothesis testing and parsimony as related to Pleistocene evidence of cultural burial. This includes a brief synopsis of recent disciplinary conversations and citation of work by other groups of authors, none of whom adopted this “null hypothesis” approach in their published work.

As we now describe in the manuscript, previous work on the Dinaledi evidence never assumed any role for *H. naledi* in the burial of remains. Reading the reviewer reports caused us to realize that this previous work had followed exactly the “null hypothesis” approach that some suggested we follow. By following this null hypothesis approach, we neglected a valuable avenue of investigation. In retrospect, we see how this approach impeded us from understanding the pattern of evidence within the Puzzle Box area. Thus in the revised manuscript we have mentioned this history within the Discussion and also presented more of the background to our previous work in the Introduction. Hopefully by including this discussion of these issues, the manuscript will broaden conversation about the relation of parsimony to these issues.

Language and presentation style:

Reviewer 4 criticizes our presentation, suggesting that the text “gives the impression that a hypothesis was formulated before data were collected.” Other outside commentators have mentioned this notion also, including Martinón-Torres *et al.* (2024) who suggest that the study began from a preferred hypothesis and gathered data to support it. The accurate communication of results and hypotheses in a scientific article is a broader issue than this one study. Preferences about presentation style vary across fields of study as well as across languages. We do not regret using plain language where possible. In any study that combines data and methods from different scientific disciplines, the use of plain language is particularly important to avoid misunderstandings where terms may mean different things in different fields.

The essential question raised by these comments is whether it is appropriate to present the results of a study in terms of the hypothesis that is best supported. As noted above, we read carefully many recent studies of Pleistocene burial evidence. We note that in each of these studies that concluded that burial is the best hypothesis, the authors framed their results in the same way as our previous manuscript: an introduction that briefly reviews background evidence for treatment of the dead, a presentation of results focused on how each analysis supports the hypothesis of burial for the case, and then in some (but not all) cases discussion of why some alternative hypotheses could be rejected. We do not infer from this that these other studies started from a presupposition and collected data only to confirm it. Rather, this is a simple matter of presentation style.

The alternative to this approach is to present an exhaustive list of possible hypotheses and to describe how the data relate to each of them, at the end selecting the best. This is the approach that we have followed in the revised manuscript, as described above under the direction of the reviewer and editorial guidance. This approach has the advantage of bringing together evidence in different combinations to show how each data point rejects some hypotheses while supporting others. It has the disadvantage of length and repetition.

Possible artifact:

We have chosen to keep the description of the possible artifact associated with the Hill Antechamber Feature in the Supplementary Information. We do this while acknowledging that this is against the opinion of reviewer 4, who felt the description should be removed unless the object in question is fully excavated and physically analyzed. The previous version of the manuscript did not rely upon the stone as positive evidence of grave goods or symbolic content, and it noted that the data do not test whether the possible artifact was placed or was intentionally modified. However this did not satisfy reviewer 4, and some outside commentators likewise asserted that the object must be a “geofact” and that it should be removed.

We have three arguments against this line of thinking. First, we do not omit data from our reporting. Whether *Homo naledi* shaped the rock or not, used it as a tool or not, whether the rock was placed with the body or not, it is unquestionably there. Omitting this one object from the report would be simply dishonest. Second, the data on this rock are at 16 micron resolution. While physical inspection of its surface may eventually reveal trace evidence and will enable better characterization of the raw material, no mode of surface scanning will produce better evidence about the object’s shape. Third, the position of this possible artifact within the feature provides significant information about the deposition of the skeletal material and associated sediments. The pitch, orientation, and position of the stone is not consistent with slow deposition but are consistent with the hypothesis that the surrounding sediment was rapidly emplaced at the same time as the articulated elements less than 2 cm away.

In the current version, we have redoubled our efforts to provide information about the position and shape of this stone while not presupposing the intentionality of its shape or placement. We add here that the attitude expressed by referee 4 and other commentators, if followed at other sites, would certainly lead to the loss or underreporting of evidence, which we are trying to avoid.

Consistency versus variability of behavior:

As described in the revised manuscript, different features within the Dinaledi Subsystem exhibit some shared characteristics. At the same time, they vary in positioning, representation of individuals and extent of commingling. Other localities within the subsystem and broader cave system present different evidence. Some commentators have questioned whether the patterning is consistent with a single common explanation, or whether multiple explanations are necessary. To address this line of questioning, we have added several elements to the manuscript. We created a new section on secondary cultural burial, discussing whether any of the situations may reflect this practice. In the Discussion, we briefly review the ways in which the different features support the involvement of *H. naledi* without interpreting anything about the intentionality or meaning of the behavior. We further added a section to the Discussion to consider whether variation among the features reflects variation in mortuary practices by *H. naledi*. One aspect of this section briefly cites variation in the location and treatment of skeletal remains at other sites with evidence of burial.

Grave goods:

Some commentators have argued that grave goods are a necessary criterion for recognizing evidence of ancient burial. We added a section to the Discussion to review evidence of grave goods at other Pleistocene sites where burial is accepted.

References:

Dirks, P. H., Berger, L. R., Roberts, E. M., Kramers, J. D., Hawks, J., Randolph-Quinney, P. S., Elliott, M., Musiba, C. M., Churchill, S. E., de Ruiter, D. J., Schmid, P., Backwell, L. R., Belyanin, G. A., Boshoff, P., Hunter, K. L., Feuerriegel, E. M., Gurtov, A., Harrison, J. du G., Hunter, R., … Tucker, S. (2015). Geological and taphonomic context for the new hominin species *Homo naledi* from the Dinaledi Chamber, South Africa. *eLife*, *4*, e09561. https://doi.org/10.7554/eLife.09561Dirks, P. H., Roberts, E. M., Hilbert-Wolf, H., Kramers, J. D., Hawks, J., Dosseto, A., Duval, M., Elliott, M., Evans, M., Grün, R., Hellstrom, J., Herries, A. I., Joannes-Boyau, R., Makhubela, T. V., Placzek, C. J., Robbins, J., Spandler, C., Wiersma, J., Woodhead, J., & Berger, L. R. (2017). The age of *Homo naledi* and associated sediments in the Rising Star Cave, South Africa. *eLife*, *6*, e24231. https://doi.org/10.7554/eLife.24231Elliott, M., Makhubela, T., Brophy, J., Churchill, S., Peixotto, B., FEUERRIEGEL, E., Morris, H., Van Rooyen, D., Ramalepa, M., Tsikoane, M., Kruger, A., Spandler, C., Kramers, J., Roberts, E., Dirks, P., Hawks, J., & Berger, L. R. (2021). Expanded Explorations of the Dinaledi Subsystem,Rising Star Cave System, South Africa. *PaleoAnthropology*, *2021*(1), 15–22. https://doi.org/10.48738/2021.iss1.68Fewlass, H., Zavala, E. I., Fagault, Y., Tuna, T., Bard, E., Hublin, J.-J., Hajdinjak, M., & Wilczyński, J. (2023). Chronological and genetic analysis of an Upper Palaeolithic female infant burial from Borsuka Cave, Poland. *iScience*, *26*(12). https://doi.org/10.1016/j.isci.2023.108283Foecke, Kimberly K., Queffelec, Alain, & Pickering, Robyn. (n.d.). No Sedimentological Evidence for Deliberate Burial by *Homo naledi* – A Case Study Highlighting the Need for Best Practices in Geochemical Studies Within Archaeology and Paleoanthropology. *PaleoAnthropology*, *2024*. https://doi.org/10.48738/202x.issx.xxxGoldberg, P., Aldeias, V., Dibble, H., McPherron, S., Sandgathe, D., & Turq, A. (2017). Testing the Roc de Marsal Neandertal “Burial” with Geoarchaeology. *Archaeological and Anthropological Sciences*, *9*(6), 1005–1015. https://doi.org/10.1007/s12520-013-0163-2Maloney, T. R., Dilkes-Hall, I. E., Vlok, M., Oktaviana, A. A., Setiawan, P., Priyatno, A. A. D., Ririmasse, M., Geria, I. M., Effendy, M. A. R., Istiawan, B., Atmoko, F. T., Adhityatama, S., Moffat, I., Joannes-Boyau, R., Brumm, A., & Aubert, M. (2022). Surgical amputation of a limb 31,000 years ago in Borneo. *Nature*, *609*(7927), 547–551. https://doi.org/10.1038/s41586-022-05160-8Martinón-Torres, M., d’Errico, F., Santos, E., Álvaro Gallo, A., Amano, N., Archer, W., Armitage, S. J., Arsuaga, J. L., Bermúdez de Castro, J. M., Blinkhorn, J., Crowther, A., Douka, K., Dubernet, S., Faulkner, P., Fernández-Colón, P., Kourampas, N., González García, J., Larreina, D., Le Bourdonnec, F.-X., … Petraglia, M. D. (2021). Earliest known human burial in Africa. *Nature*, *593*(7857), Article 7857. https://doi.org/10.1038/s41586021-03457-8Martinón-Torres, M., Garate, D., Herries, A. I. R., & Petraglia, M. D. (2023). No scientific evidence that *Homo naledi* buried their dead and produced rock art. *Journal of Human Evolution*, 103464. https://doi.org/10.1016/j.jhevol.2023.103464Pomeroy, E., Bennett, P., Hunt, C. O., Reynolds, T., Farr, L., Frouin, M., Holman, J., Lane, R., French, C., & Barker, G. (2020a). New Neanderthal remains associated with the ‘flower burial’ at Shanidar Cave. *Antiquity*, *94*(373), 11–26. https://doi.org/10.15184/aqy.2019.207Pomeroy, E., Hunt, C. O., Reynolds, T., Abdulmutalb, D., Asouti, E., Bennett, P., Bosch, M., Burke, A., Farr, L., Foley, R., French, C., Frumkin, A., Goldberg, P., Hill, E., Kabukcu, C., Lahr, M. M., Lane, R., Marean, C., Maureille, B., … Barker, G. (2020b). Issues of theory and method in the analysis of Paleolithic mortuary behavior: A view from Shanidar Cave. *Evolutionary Anthropology: Issues, News, and Reviews*, *29*(5), 263–279. https://doi.org/10.1002/evan.21854Robbins, J. L., Dirks, P. H. G. M., Roberts, E. M., Kramers, J. D., Makhubela, T. V., HilbertWolf, H. L., Elliott, M., Wiersma, J. P., Placzek, C. J., Evans, M., & Berger, L. R. (2021). Providing context to the *Homo naledi* fossils: Constraints from flowstones on the age of sediment deposits in Rising Star Cave, South Africa. *Chemical Geology*, *567*, 120108. https://doi.org/10.1016/j.chemgeo.2021.120108Wiersma, J. P., Roberts, E. M., & Dirks, P. H. G. M. (2020). Formation of mud clast breccias and the process of sedimentary autobrecciation in the hominin-bearing (*Homo naledi*) Rising Star Cave system, South Africa. *Sedimentology*, *67*(2), 897–919. https://doi.org/10.1111/sed.12666